palaeontology/ecology/plant science

biome reconstruction, proxy biases, climate reconstruction, plant macrofossils, dispersed pollen, light and scanning electron microscopy

**Author for correspondence:**
Johannes M. Bouchal
e-mail: bouchaljm@gmail.com

# Messinian vegetation and climate of the intermontane Florina–Ptolemais–Servia Basin, NW Greece inferred from palaeobotanical data: how well do plant fossils reflect past environments?

Johannes M. Bouchal[1], Tuncay H. Güner[2],
Dimitrios Velitzelos[3], Evangelos Velitzelos[3]
and Thomas Denk[4]

[1]Research Group Aerobiology and Pollen Information, Department of Oto-Rhino-Laryngology, Medical University Vienna, Vienna, Austria
[2]Faculty of Forestry, Department of Forest Botany, Istanbul University Cerrahpaşa, Istanbul, Turkey
[3]Section of Historical Geology and Palaeontology, National and Kapodistrian University of Athens, Faculty of Geology and Geoenvironment, Athens, Greece
[4]Department of Palaeobiology, Swedish Museum of Natural History, Box 50007, 10405 Stockholm, Sweden

JMB, 0000-0002-4241-9075; THG, 0000-0001-9742-1319; TD, 0000-0001-9535-1206

The late Miocene is marked by pronounced environmental changes and the appearance of strong temperature and precipitation seasonality. Although environmental heterogeneity is to be expected during this time, it is challenging to reconstruct palaeoenvironments using plant fossils. We investigated leaves and dispersed spores/pollen from 6.4 to 6 Ma strata in the intermontane Florina–Ptolemais–Servia Basin (FPS) of northwestern Greece. To assess how well plant fossils reflect the actual vegetation of the FPS, we assigned fossil taxa to biomes providing a measure for environmental heterogeneity. Additionally, the palynological assemblage was compared with pollen spectra from modern lake sediments to assess biases in spore/pollen representation in the pollen record. We found a close match of the Vegora assemblage with modern *Fagus–Abies* forests of Turkey. Using taxonomic affinities of leaf fossils, we further established close similarities of the Vegora assemblage

with modern laurophyllous oak forests of Afghanistan. Finally, using information from sedimentary environment and taphonomy, we distinguished local and distantly growing vegetation types. We then subjected the plant assemblage of Vegora to different methods of climate reconstruction and discussed their potentials and limitations. Leaf and spore/pollen records allow accurate reconstructions of palaeoenvironments in the FPS, whereas extra-regional vegetation from coastal lowlands is probably not captured.

# 1. Introduction

The late Miocene (11.6–5.3 Ma) marks the time in the Neogene (23–2.58 Ma) with the largest shift from equable climate to strong latitudinal temperature gradients in both hemispheres [1]. This is well illustrated by the global rise of $C_4$-dominated ecosystems (grasslands and savannahs in the tropics and subtropics; [2]). In the Mediterranean region, vegetation changes did not happen synchronously with modern steppe and Mediterranean sclerophyllous woodlands replacing humid temperate forest vegetation at different times and places during the middle and late Miocene [3–9]. During the latest Miocene (5.9–5.3 Ma), the desiccation of the Mediterranean Sea was caused by its isolation from the Atlantic Ocean. Based on palynological studies across the Mediterranean region, this event did not have a strong effect on the existing vegetation. Open and dry environments existed in southern parts before, during and after this so-called Messinian salinity crisis (MSC; [10]). By contrast, forested vegetation occurred in northern parts of Spain, Italy and the western Black Sea region [5]. Likewise, a vegetation gradient occurred from north and central Italy and Greece to Turkey, where humid temperate forests had disappeared by the early late Miocene [8].

The Florina–Ptolemais–Servia Basin (FPS; [11]) of northwestern Greece and its extensions to the north (Bitola Basin) and south (Likoudi Basin) is one of the best-understood intermontane basins of late Miocene age in the entire Mediterranean region. A great number of studies investigated the tectonic evolution, depositional history and temporal constraints of basin fills (e.g. [11–13]), plant fossils (e.g. [14–19]) and vertebrate fossils [20–23].

The Messinian flora of Vegora in the northern part of the FPS is dated at 6.4–6 Ma and represents the vegetation in this region just before the onset of the MSC (the pre-evaporitic Messinian). This flora has been investigated since 1969 [18] and represents one of the richest late Miocene leaf floras in the eastern Mediterranean along with two other, slightly older, Messinian plant assemblages from the FPS and its southern extension, Likoudi/Drimos and Prosilio/Lava (4–6 in figure 1; [19]). The focus of previous palaeobotanical studies in the FPS has been on macrofossils. By contrast, no comprehensive study of dispersed pollen and spores has been carried out in the FPS. While fruit and seed floras, to a great extent, and leaf floras, to a lesser extent, reflect local vegetation in an area, dispersed pollen and spores provide additional information about the regional vegetation. Therefore, a main focus of the present study is on spores and pollen of the Messinian plant assemblage of Vegora.

We (i) investigated dispersed spores and pollen using a combined light and scanning electron microscopy approach [26–28] that allows a more accurate determination of pollen and hence higher taxonomic resolution. We (ii) then compiled a complete list of plant taxa recorded for the site of Vegora including fruits and seeds, foliage, and spores and pollen. Based on the ecological properties of their modern analogue taxa, we assigned the fossil taxa to functional types (vegetation units) and inferred palaeoenvironments of the FPS during the Messinian. Using leaf physiognomic characteristics, we (iii) conducted a climate leaf analysis multivariate program (CLAMP) analysis [29,30] to infer several climate parameters for the late Miocene of the FPS. We also (iv) used a modified 'coexistence approach' [31,32] based on climatic requirements of modern analogue plant taxa to infer two climate parameters, and (v) a Köppen signature analysis [7,33] based on the Köppen–Geiger climate types in which modern analogue taxa of the fossil taxa occur. Finally, we (vi) discuss how well the translation of fossil plant assemblages into functional types (vegetation units, biomes) works for reconstructing past environments at local and regional scales.

# 2. Material and methods

## 2.1. Geological setting

The old open-pit lignite quarry of Vegora is located in western Macedonia, northwestern Greece, *ca* 2 km east of the town of Amyntaio and is part of the Neogene Florina–Ptolemais–Servia intermontane basin

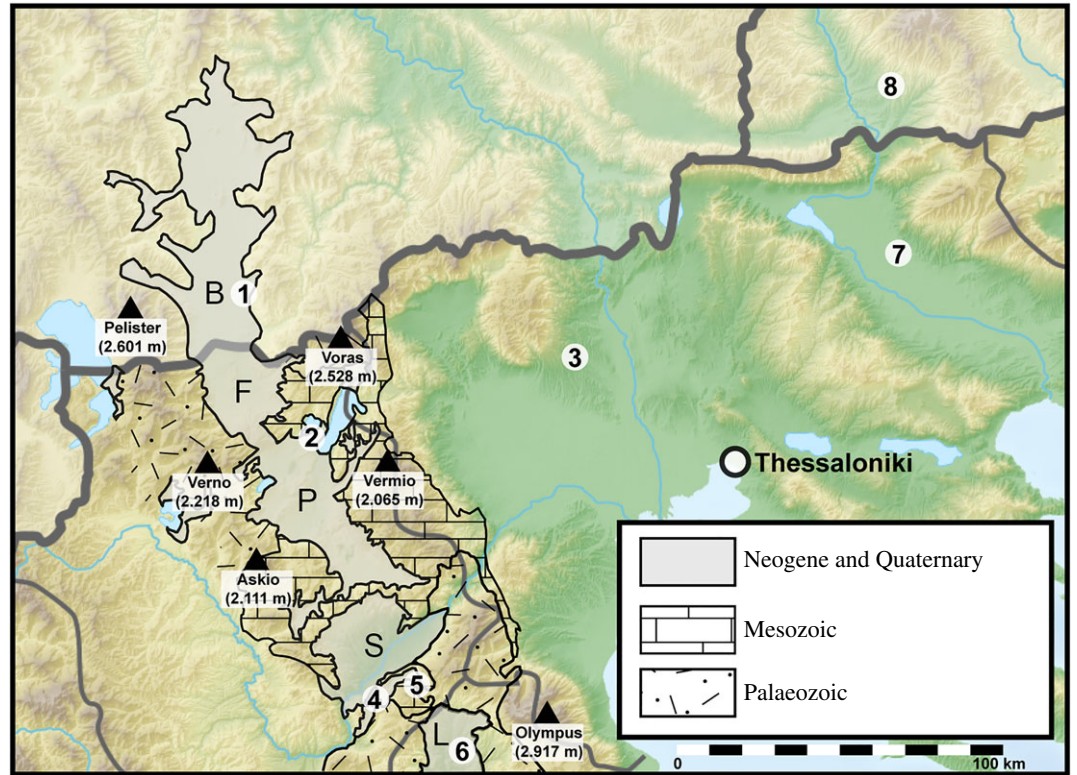

**Figure 1.** Fossil localities and lithological map of the FPS Basin and its extensions to the north (Bitola Basin) and south (Likoudi Basin). Map redrawn after Steenbrink *et al*. [11,12], Ognjanova-Rumenova [24], Ivanov [25] and Koufos [22]. Fossil localities: (**1**) Bitola Basin, Republic of North Macedonia, PF; (**2**) Vegora Basin, MF and PF; (**3**) Dytiko, VF; (**4**) Prosilio, MF; (**5**) Lava, MF; (**6**) Likoudi, MF; (**7**) Serres Basin. (**2–7**) Greece. (**8**) Sandanski Graben, Bulgaria, PF. Bitola Basin (B), Florina sub-Basin (F), Ptolemais sub-Basin (P), Servia sub-Basin (S), Likoudi Basin (L). Plant macrofossils (MF), palynoflora (PF), vertebrate fossils (VF).

(FPS) and its northern (Bitola Basin) and southern extensions (Likoudi Basin; figure 1). The NNW–SSE trending FPS is *ca* 120 km long and presently at elevations between 400 and 700 m.a.s.l. and is flanked by mountain ranges to the east and the west. Main ranges include Baba Planina (Pelister, 2601 m), Verno (2128 m) and Askio (2111 m) to the west of the basin and Voras (2528 m), Vermio (2065 m) and Olympus (2917 m) to the east (figure 1). These ranges mainly comprise Mesozoic limestones, Upper Carboniferous granites and Palaeozoic schists.

Continuous sedimentation since 8 Ma resulted in the accumulation of *ca* 600 m of late Miocene to early Pleistocene lake sediments with intercalated lignites and alluvial deposits.

The FPS Basin formed in the late Miocene as a result of NE–SW extension in the Pelagonian Zone, the westernmost zone of the Internal Hellenides [34–36]. A subsequent Pleistocene episode of NW–SE extension caused the fragmentation of the basin into several subbasins [11].

Basin fills overlay unconformably Palaeozoic and Mesozoic rocks. Alpine and pre-alpine basement of the area consists of Pelagonian metamorphic rocks (gneisses, amphibolites, mica schists, meta-granites and Permian to Triassic meta-sediments) and crystalline limestone of Triassic–Jurassic age (carbonate cover). Subpelagonian ophiolites and deep-sea sediments of Jurassic age, comprising the Vourinos ophiolitic complex, thrust over the Pelagonian carbonate rocks and are covered by Cretaceous strata [34–38].

The Vegora section belongs to the *ca* 300 m thick Komnina Formation, which unconformably overlies pre-Neogene basement and is predominantly composed of alluvial sands and conglomerates, lacustrine (diatomaceous) marls and palustrine clays, with some intercalated (xylite-type) lignite seams [11]. The detailed description of the sequence at the Vegora quarry follows Kvaček *et al*. [18] and Steenbrink *et al*. [11] (figure 2). Since 2000, the lower part of the sequence was not accessible and the exposed sequence started with a *ca* 10 m thick lignite seam (see fig. 3 in [18] versus fig. 2, unit 1, in [11] corresponding to unit 1 in the present figure 2).

The full Vegora section begins with hard marls (greater than 15 m) followed by 10–15 m of clay sands and a white marl layer of 10 m. Then, a formation of clay sands follows with a total thickness of 15–20 m. This formation starts with lignitic marls followed by marls and clay sand intercalations. The sand is rich in mica.

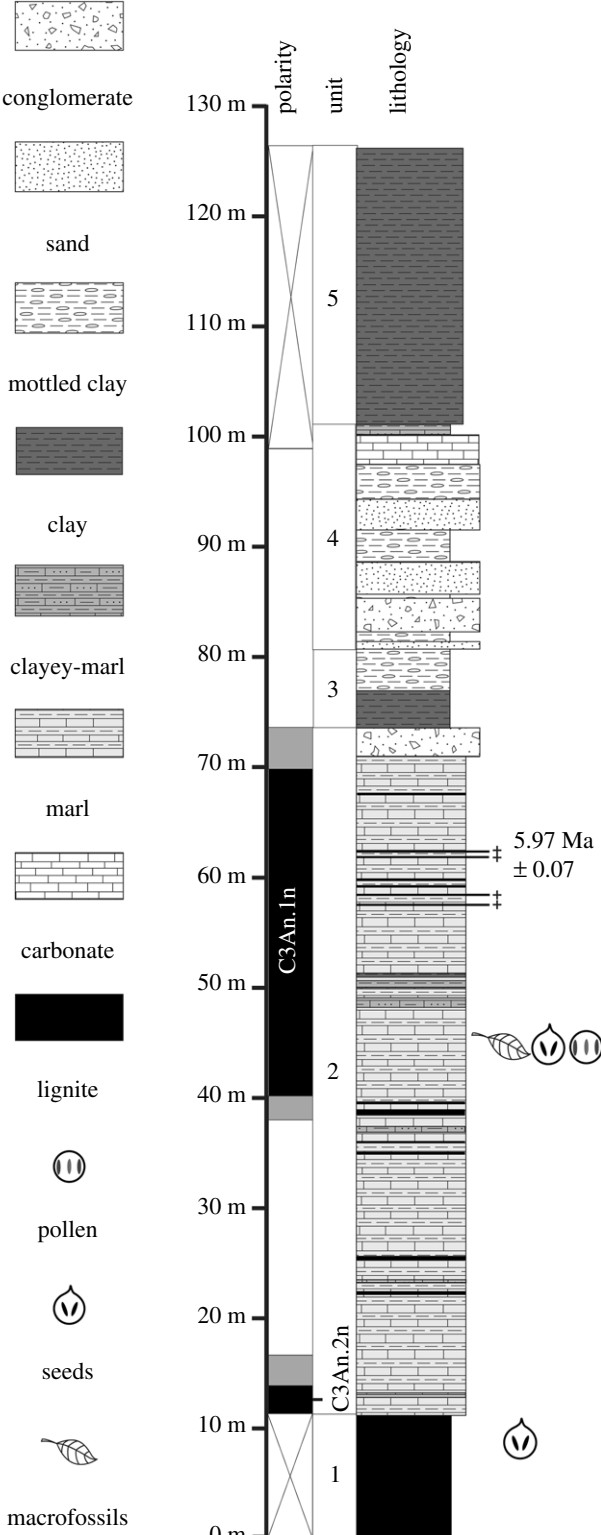

**Figure 2.** Lithology and polarity zones of the Vegora section (redrawn after [11]). In the polarity column, black denotes normal and white reversed polarity; shaded portions indicate undetermined polarity. The lower normal polarity interval corresponds to Subchron C3An.2n and the upper to C3An.1n (after [11]). The position of fossil bearing strata following Velitzelos & Schneider [39] and Kvaček *et al*. [18].

Above this, a lignite seam occurs with a thickness of 12–35 m. Within the seam, three xylitic layers with a total thickness of 10–12 m can be distinguished. The lower xylitic layer is about 3–4 m, the middle 0.5–1.5 m and the upper is 4–9 m in thickness [18].

The upper lignite layer was the first visible layer in 2002 in the section (unit 1 in figure 2). Between the xylitic layers, sand layers of various thickness (0–15 m) occur. In general, the thickness of these sand

layers is smaller towards the N and NE ends of the mine and becomes significantly larger towards the W and SW ends of the mine.

The top of the upper xylitic layer is covered by 3–4 m thick shales, followed by a thick layer of light blue marls, 10–60 m thick (unit 2 in figure 2), and a layer of sandy marls, 10–40 m thick (units 3 and 4 in figure 2). Unit 3 is made up of grey-brown lignitic clay at its base and multi-coloured mottled clays, silts and fine sands higher up. Unit 4 consists of cross-bedded conglomerates and coarse sands at the base overlain by mottled clays, silts and fine sands with calcareous nodules. Finally, the top of the section is made up of dark red, mottled sands, silts and clays (unit 5).

Unit 2 of Steenbrink *et al.* [11] corresponds to the main fossiliferous layers for plant fossils and diatoms [40].

The uppermost layer of the Neogene sediments in the area is a formation of marly limestones, of different thickness, which is not everywhere visible appearing only at the nearby villages of Neapoli and Lakia. All Neogene sediments of the area are inclined by a 10° slope towards NNW.

The rocks on the top of the Neogene section are Quaternary alluvial deposits, conglomerates, sands and gravels. This material, in general, has been supplied from erosion processes of the nearby metamorphic mountains.

## 2.2. Age

In the upper part of unit 2, 4 cm thick layers of tephra rich in biotite were found and used for $^{40}Ar/^{39}Ar$ dating [11]. The calculated age of $5.97 \pm 0.07$ Ma corresponds to pre-evaporitic Messinian and provides an independent age constraint for correlation with the Subchron C3An.1n. In addition, using palaeomagnetic data, the base of unit 2, just above the lignite seam, can be correlated to the astronomical polarity timescale, indicating that its position corresponds to the end of Subchron C3An.2n. This would suggest an age of *ca* 6.4 Ma for the beginning of unit 2. Therefore, the period of deposition of the light blue marls (unit 2) from Vegora can be narrowed down to *ca* 400 ka.

## 2.3. Sample processing

The palynological sample was taken from a slab piece (S115992) from a leaf layer in unit 2 of the Vegora mine. The sample was processed following standard protocols (20% HCl to dissolve carbonate, 40% HF to dissolve silica, 20% HCl to dissolve fluorspar; chlorination, acetolysis; see [28]) and the residue was transferred to glycerol.

## 2.4. Palynological investigation

Light microscopy (LM) micrographs were taken with an Olympus BX51 microscope (Swedish Museum of Natural History [NRM], Stockholm, Sweden) equipped with an Olympus DP71 camera. The same grains were examined using LM and scanning electron microscope (SEM; single grain method; [27,28]). Specimens were sputter-coated with gold for SEM investigation. SEM micrographs were taken with an ESEM FEI Quanta FEG 650 SEM (Stockholm University). Residue and SEM stubs are stored at NRM under specimen numbers S11599201–S11599220. Terminology of palynomorphs follows Punt *et al.* [41] and Halbritter *et al.* [28]. Size categories follow Halbritter *et al.* [28]. Palynomorphs were determined to family, genus or infrageneric level. In cases when no taxonomic affinity could be established, we used fossil form taxa which are not implying a particular systematic affiliation. The systematic palaeobotany section starts with algae, fern and fern allies, gymnosperms and is followed by angiosperms. Angiosperm classification and author names of orders and families follow APG IV [42].

## 2.5. Inferring palaeoclimate estimates

We employed three different (semi)quantitative methods to infer a range of climate parameters for the Messinian of northwestern Greece.

CLAMP (climate leaf analysis multivariate program) is a physiognomy-based, taxon-free method of climate inference and makes use of the relationship between leaf architecture and climate. CLAMP uses calibration datasets of modern vegetation sites across the world to place a fossil leaf assemblage in physiognomic space, which then can be translated into numeric values for several climate parameters [29,30].

The coexistence approach (CA; [43]) is a method of inferring palaeoclimate based on nearest living relatives (NLR) of fossil taxa. CA assumes that for a given climate parameter, the tolerances of all or

**Table 1.** Köppen–Geiger climate categories. Description of Köppen–Geiger climate symbols and defining criteria [49,50]. MAP, mean annual precipitation; MAT, mean annual temperature; $T_{hot}$, temperature of the hottest month; $T_{cold}$, temperature of the coldest month; $T_{mon10}$, number of months where the temperature is above 10℃; $P_{dry}$, precipitation of the driest month; $P_{sdry}$, precipitation of the driest month in summer; $P_{wdry}$, precipitation of the driest month in winter; $P_{swet}$, precipitation of the wettest month in summer; $P_{wwet}$, precipitation of the wettest month in winter; $P_{threshold}$, varies according to the following rules (if 70% of MAP occurs in winter, then $P_{threshold} = 2 \times MAT$, if 70% of MAP occurs in summer then $P_{threshold} = 2 \times MAT + 28$, otherwise $P_{threshold} = 2 \times MAT + 14$). Summer (winter) is defined as the warmer (cooler) six months period of ONDJFM and AMJJAS.

| 1st | 2nd | 3rd | description and criteria |
|---|---|---|---|
| A | | | equatorial/tropical ($T_{cold} \geq 18℃$) |
| | f | | rainforest, fully humid ($P_{dry} \geq 60$ mm) |
| | m | | monsoonal (not Af and $P_{dry} \geq 100 – MAP/25$) |
| | s | | savannah with dry summer ($P_{sdry} < 60$ mm) |
| | w | | savannah with dry winter ($P_{wdry} < 60$ mm) |
| B | | | arid (MAP $< 10 \times P_{threshold}$) |
| | W | | desert (MAP $< 5 \times P_{threshold}$) |
| | S | | steppe (MAP $\geq 5 \times P_{threshold}$) |
| | | h | hot arid (MAT $\geq 18℃$) |
| | | k | cold arid (MAT $< 18℃$) |
| C | | | warm temperate/temperate ($T_{hot} > 10℃$ and $0℃ < T_{cold} < 18℃$ |
| D | | | snow/cold ($T_{hot} > 10℃$ and $T_{cold} \leq 0℃$) |
| | s | | summer dry ($P_{sdry} < 40$ mm and $P_{sdry} < P_{wwet}/3$) |
| | w | | winter dry ($P_{wdry} < P_{swet}/10$) |
| | f | | fully humid/without a dry season (not s or w) |
| | | a | hot summer ($T_{hot} \geq 22℃$) |
| | | b | warm summer (not a and $T_{mon10} \geq 4$) |
| | | c | cool/cold summer (not a or b and $T_{cold} > -38℃$) |
| | | d | extremely continental/very cold winter (not a or b and $T_{cold} \leq -38℃$) |
| E | | | polar ($T_{hot} < 10℃$) |
| | T | | polar tundra ($T_{hot} \leq 10℃$) |
| | F | | polar frost ($T_{hot} \leq 0℃$) |

nearly all taxa in a fossil assemblage will overlap to some degree; this overlap is called the climatic coexistence interval. In a slight modification of this approach [32], the zone of overlap was calculated using the 10th percentile (lower limit) and 90th percentile (upper limit) of the total range for all taxa recorded for a single flora. Following best practices in applying the CA, Utescher *et al.* [43] provided several guidelines to apply the CA in a meaningful way. Among these guidelines, one is to exclude relict taxa (usually monotypic or comprising very few extant species) from the analysis, because of their likely unrepresentative modern distribution. Examples for such taxa are the East and Southeast Asian *Craigia* and *Glyptostrobus*. These taxa had a much wider distribution during parts of the Cenozoic including Arctic regions. For example, Budantsev & Golovneva [44] described *Craigia* from the Eocene Renardodden Formation of Spitsbergen for which they inferred a mean annual temperature (MAT) of 8.4℃ and a coldest month mean temperature (CMMT) of –1℃ based on a CLAMP analysis. By contrast, the two modern species occur in climates with MAT 13.2–21℃ and CMMT 6.3–14.2℃ [45]. Hence, it is assumed that for relict plants, ecological niches may have changed considerably during the Cenozoic (e.g. [46]). For further assumptions of the CA and their critique, see Grimm & Potts [47] and Grimm *et al.* [48]. Climate parameters for the NLR are given in electronic supplementary material, table S1.

Köppen signatures [7,33] is another approach to infer large-scale climatic patterns for the Cenozoic that is based on NLR of fossil taxa. Modern distribution ranges are mapped on Köppen–Geiger climate maps ([49–51]; Global_1986–2010_KG_5 m.kmz; see table 1 for explanations of Köppen–Geiger categories) and the Köppen climate types in which the modern taxa occur are taken as a proxy for the

climate space in which the fossil taxa occurred. It is explicitly stated that climate niche evolution will negatively impact the reliability of the inferred palaeoclimate. To overcome this drawback, subgenera, sections and genera are used as NLR, whereas single species are usually not considered for NLR. The representation of different climate types is first scored for each species within a genus as present (1)/ absent (0) (electronic supplementary material, table S2). To summarize preferences for climate types of all modern analogues, an implicit weighting scheme is used to discriminate between modern analogues that are highly climatically constrained and those that occur in many climate zones. For each modern species, the sum of its Köppen signature is always 1. For example, if a species is present in two Köppen–Geiger climate types, *Cfa* and *Cfb*, both score 0.5. If a species is present in 10 Köppen–Geiger climate types, each of these climate types scores 0.1. The Köppen signature of a genus or section, the preferred NLR of a fossil taxon, is the sum of its species' Köppen signatures for each climate type divided by the total number of scored species for this genus. By this, the percentage representation of each Köppen–Geiger climate type is determined for a genus/section [7]. For pollen taxa of herbaceous and a few woody angiosperm groups that are resolved to family-level only, the distributions of extant members of the family were combined into a general family distribution range and the corresponding Köppen–Geiger climate types determined.

## 2.6. Characterization of terrestrial biomes

For convenience, we use the biome classification of Woodward *et al.* [52] that recognizes five major tree biomes based on the physiognomy of the dominant species: *needleleaf evergreen* (NLE), *needleleaf deciduous* (NLD), *broadleaf evergreen* (BLE), *broadleaf cold deciduous* and *broadleaf drought deciduous* (BLD$_{cold}$, BLD$_{drought}$), and MIXED forests, which consist of tree communities with interspersed mixtures or mosaics of the other four tree biomes. These authors also observed that broadleaf drought deciduous vegetation grades substantially into broadleaf evergreen vegetation. Besides, *shrublands* are defined as lands with woody vegetation less than 2 m tall. *Savannahs* are defined as lands with herbaceous or other understorey systems, where woody savannahs have forest canopy cover between 30 and 60%, and savannah has forest canopy cover between 10 and 30% [52]. This very broad definition of savannah may be strongly oversimplified. Thus, for savannah-like vegetation, we make a distinction between *steppe* and *forest-steppe* of temperate regions with a continuous layer of C$_3$ grasses and *savannah* and *woody savannah* of tropical regions with a continuous layer of C$_4$ grasses [53].

# 3. Results

## 3.1. Pollen and spores: diversity and environmental signal

We determined more than 50 palynomorph taxa from a leaf layer in unit 2 of the Vegora section (figures 3–5). A comprehensive taxonomic account including pollen morphological descriptions and additional LM and SEM micrographs is provided in the electronic supplementary material, S3. The fossil taxa comprise two algae, five ferns, 12 herbaceous plants, one woody liana and more than 30 woody trees and shrubs (table 2). Besides the taxonomic evaluation, 430 palynomorphs were counted to assess the abundance of different taxonomic groups, life forms and pollination syndromes (table 3 and figure 6). Roughly half of the pollen taxa were present in very small amounts (1–3 grains in the counted sample or less than 1%; electronic supplementary material, table S4). The presence and abundance of *Spirogyra* zygospores/aplanospores indicate a lake with shallow lake margins (reed belt with *Typha*) and stagnant, oxygen-rich, open freshwaters. Spores of *Osmunda* (greater than 4%) and *Leavigatosporites haardti* (greater than 1.6%) are of moderate abundance, suggesting that the producing pteridophytes grew close to the sedimentation area, the Messinian Vegora Lake. Among conifers and wind-pollinated trees, strong pollen producers such as *Pinus* (subgenus *Strobus* 11%, subgenus *Pinus* approx. 8%), *Abies* (8.4%), *Cathaya* (7%), the Betulaceae *Alnus* (9.3%) and the Fagaceae *Fagus* (7%) are most abundant. Another group of wind-pollinated trees and shrubs was represented with abundances between 2 and 5%. Among these were both deciduous and evergreen oaks (*Quercus*) and conifers such as *Cedrus* and *Tsuga* and undifferentiated papillate Cupressaceae.

Among the taxa that are represented by single or few pollen grains, a significant number belonged to insect-pollinated plants. Insect-pollinated trees, shrubs and lianas include *Craigia*, *Platycarya*, Castaneoideae and *Parthenocissus*; *Hedera* and *Sassafras* are further insect-pollinated woody taxa,

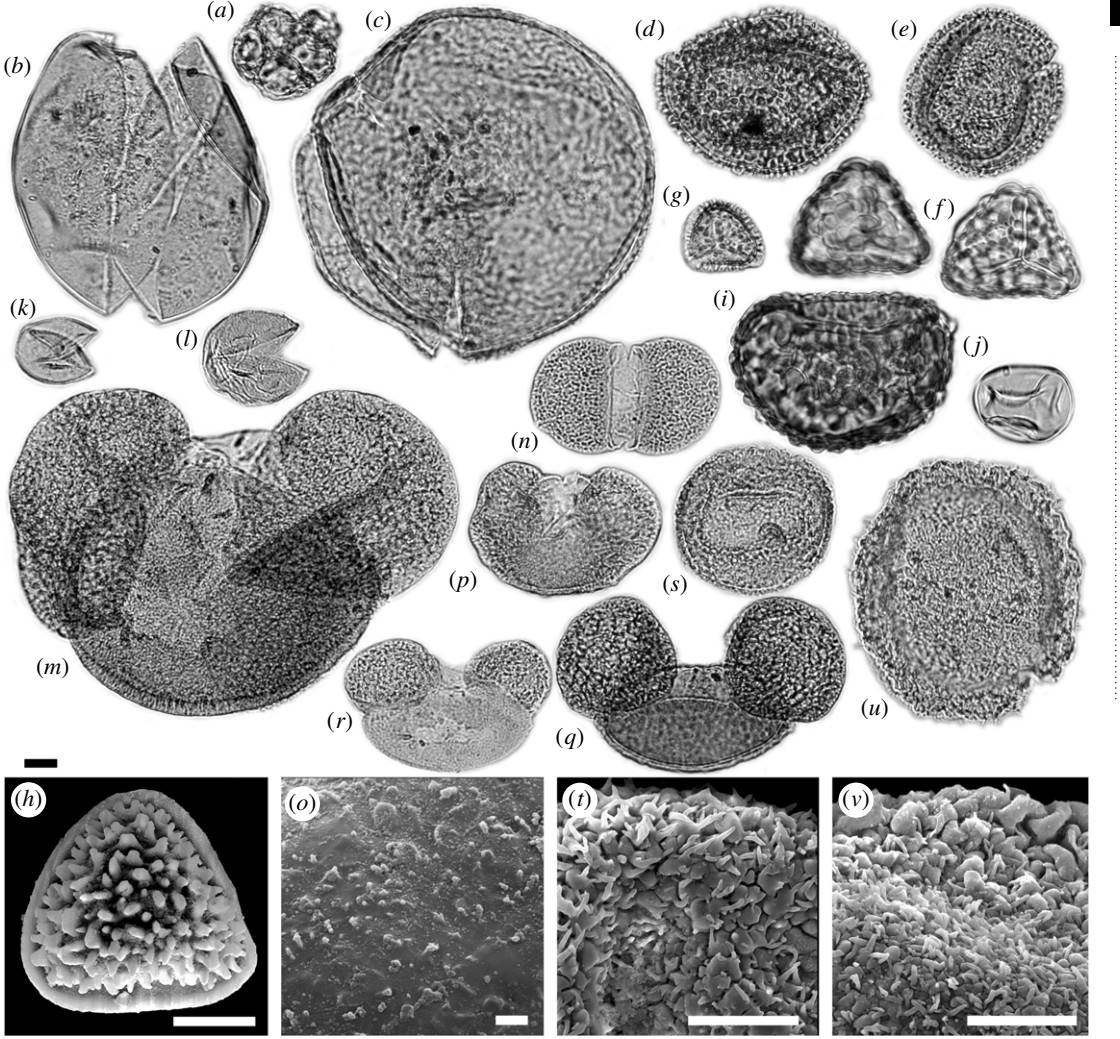

**Figure 3.** LM and SEM micrographs of algae, fern and fern allies, and gymnosperm palynomorphs. (a) *Botryococcus* sp. cf. *B. braunii*, (b) *Spirogyra* sp. 1/*Ovoidites elongatus*, (c) *Spirogyra* sp. 2/*Cycloovoidites cyclus*, (d–e) *Osmunda* sp., (d) EV, (e) PV. (f) *Cryptogramma* vel *Cheilanthes* sp./*Cryptogrammosporis magnoides*, PV. (g–h) *Pteris* sp./*Polypodiaceoisporites corrutoratus*, (g) PV, (h) DV. (i) Davalliaceae vel Polypodiaceae gen. indet./*Verrucatosporites alienus*, EV. (j) Monolete spore fam. indet./ *Leavigatosporites haardti*, EV. (k–l) Papillate Cupressaceae pollen/*Inaperturopollenites hiatus*. (m) *Abies* sp., EV. (n–o) *Cathaya* sp., (n) PV, (o) SEM detail, nanoechinate sculpturing of cappa (PRV). (p) *Cedrus* sp., EV. (q) *Pinus* subgenus *Pinus* sp., EV. (r) *Pinus* subgenus *Strobus* sp., EV. (s–t) *Tsuga* sp. 1, (s) PV, (t) monosaccus and corpus detail, PRV. (u–v) *Tsuga* sp. 2, (u) PV, (v) monosaccus and corpus detail, PRV. Equatorial view (EV), polar view (PV), distal view (DV), proximal view (PRV). Scale bars 10 µm (LM, h,t,v), 1 µm (o).

recorded in the leaf fossil record. Herbaceous taxa comprise Apiaceae, Caryophyllaceae, *Geranium*, *Succisa*, Asteraceae and Cichorioideae.

The ratio arboreal pollen (AP) to non-arboreal pollen (NAP) is 89.5–10.5%, indicating a forest-dominated (tree prevalent) local and regional vegetation according to the threshold values of Favre *et al.* [61]. Forest types (biomes of [52]) represented by the pollen assemblage are needleleaf evergreen and deciduous forests (NLE, NLD), broadleaf deciduous forests (BLD), broadleaf evergreen forests (BLE) and mixed forests (MIXED). In addition, BLD and NLD either thrived on well-drained soils or in temporarily or permanently inundated areas.

A few taxa might also indicate the presence of closed or open shrublands and grasslands (herbaceous taxa including sparse Poaceae with affinity to *Poa/Lolium*, Chenopodioideae, Apiaceae, etc. and woody taxa including palms; see table 2 for other woody taxa). These may have been associated with BLE woodlands (*Quercetum mediterranea*, *Quercus sosnowskyi*) or with mesophytic evergreen forests of *Q. drymeja* (see below). Alternatively, they may have originated from an independent vegetation type (for example, montane grasslands).

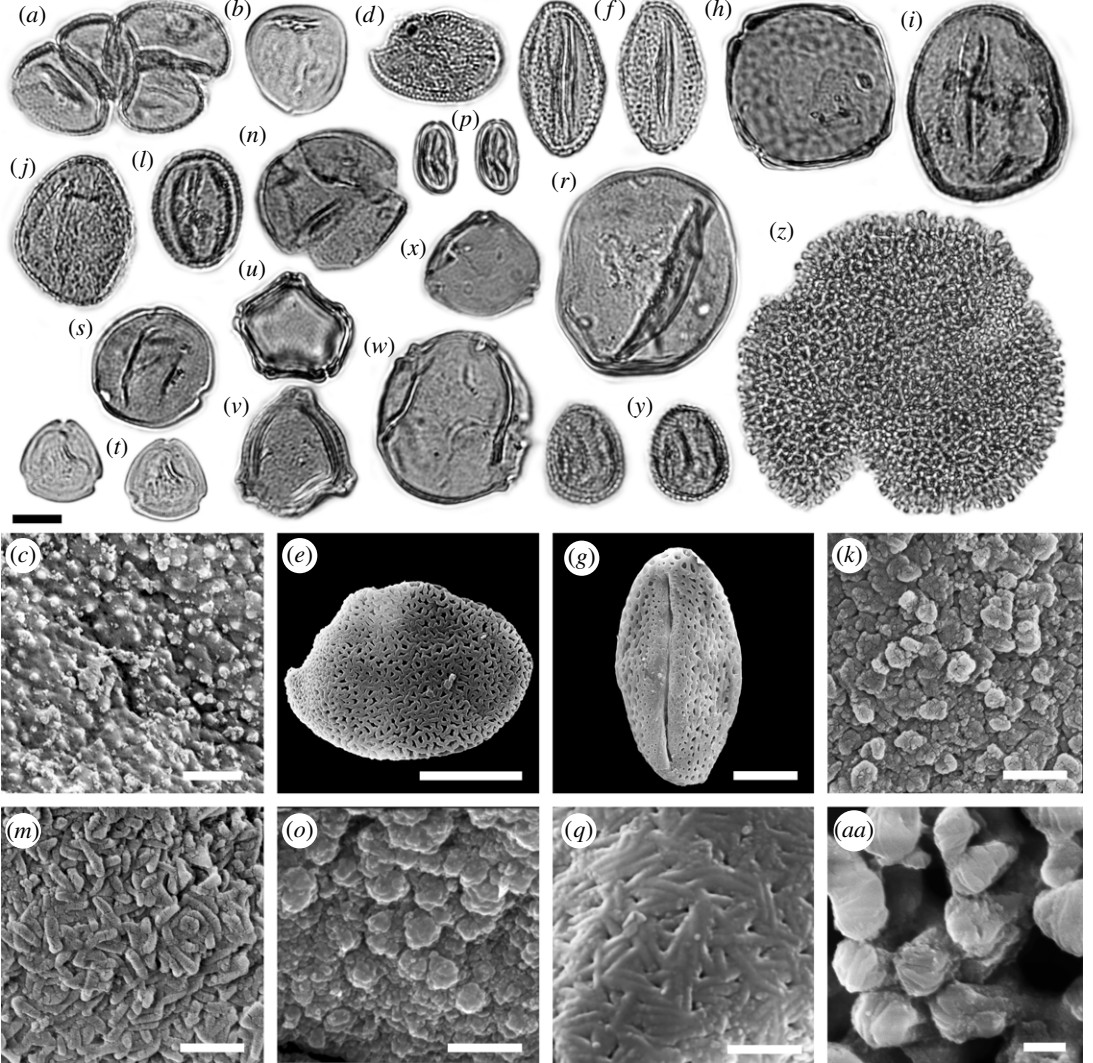

**Figure 4.** LM and SEM micrographs of Poales, Vitales, Rosales, Fagales, Malpighiales and Geraniales. (*a*) *Typha* sp./ *Tetradomonoporites typhoides*, tetrad, PV. (*b,c*) Poaceae gen. indet., EV, (*c*) exine detail, PRV. (*d,e*) Monocotyledonae indet., (*d*) PV, (*e*) PRV. (*f–g*) *Parthenocissus* sp., EV. (*h*) *Ulmus* vel *Zelkova* sp., PV. (*i*) *Fagus* sp., EV. (*j,k*) *Quercus* sect. *Cerris* sp., EV, (*k*) SEM detail, mesocolpium exine sculpturing. (*l–m*) *Quercus* sect. *Ilex* sp., EV, (*m*) SEM detail, mesocolpium exine sculpturing. (*n,o*) *Quercus* sect. *Quercus* sp., PV, (*o*) SEM detail, apocolpium exine sculpturing. (*p,q*) Castaneoideae gen. indet., EV, (*q*) SEM detail, mesocolpium exine sculpturing. (*r*) *Carya* sp., PV. (*s*) *Platycarya* sp., PV. (*t*) Engehardioideae gen. indet., PV. (*u*) *Alnus* sp., PV. (*v*) *Betula* sp., PV. (*w*) *Carpinus* sp., PV. (*x*) *Corylus* sp., PV. (*y*) *Salix* sp., EV. (*z–aa*) *Geranium* sp., (*z*) PV, (*aa*) clavae detail. Equatorial view (EV), polar view (PV), proximal view (PRV). Scale bars 10 µm (LM, *e,g*), 1 µm (*c,k,m,o,q,aa*).

Among needleleaf forest biomes, for some taxa, the attribution to a distinct forest type is not straightforward. For example, conifers such as *Cathaya* may have been part of the montane hinterland vegetation on well-drained soils but may also have been important elements of peat-forming vegetation [62–65].

Based on pollen abundances (electronic supplementary material, table S4), local (close to the lake), regional (occurring in the FPS) and extra-regional (potentially occurring outside the FPS) vegetation can be inferred. Local vegetation consisted of BLD forests subjected to flooding (*Alnus*) and NLD swamp forests (papillate Cupressaceae; [66]). Close to the lake, a mixed forest with *Fagus*, *Abies* and *Cathaya* thrived (using the modern Abant Gölü of northern Turkey as a reference for pollen rain vegetation relationships; [67]). Deciduous oaks (mainly of sect. *Cerris*) also might have been part of local forest vegetation (BLD). *Pinus* and *Cedrus* NLE forests and evergreen oak forests (BLE) grew at some distance from the lake (regional vegetation; using threshold abundances of *Cedrus*, 7%, and evergreen *Quercus*, 20%, as indicators of local source vegetation; [68]).

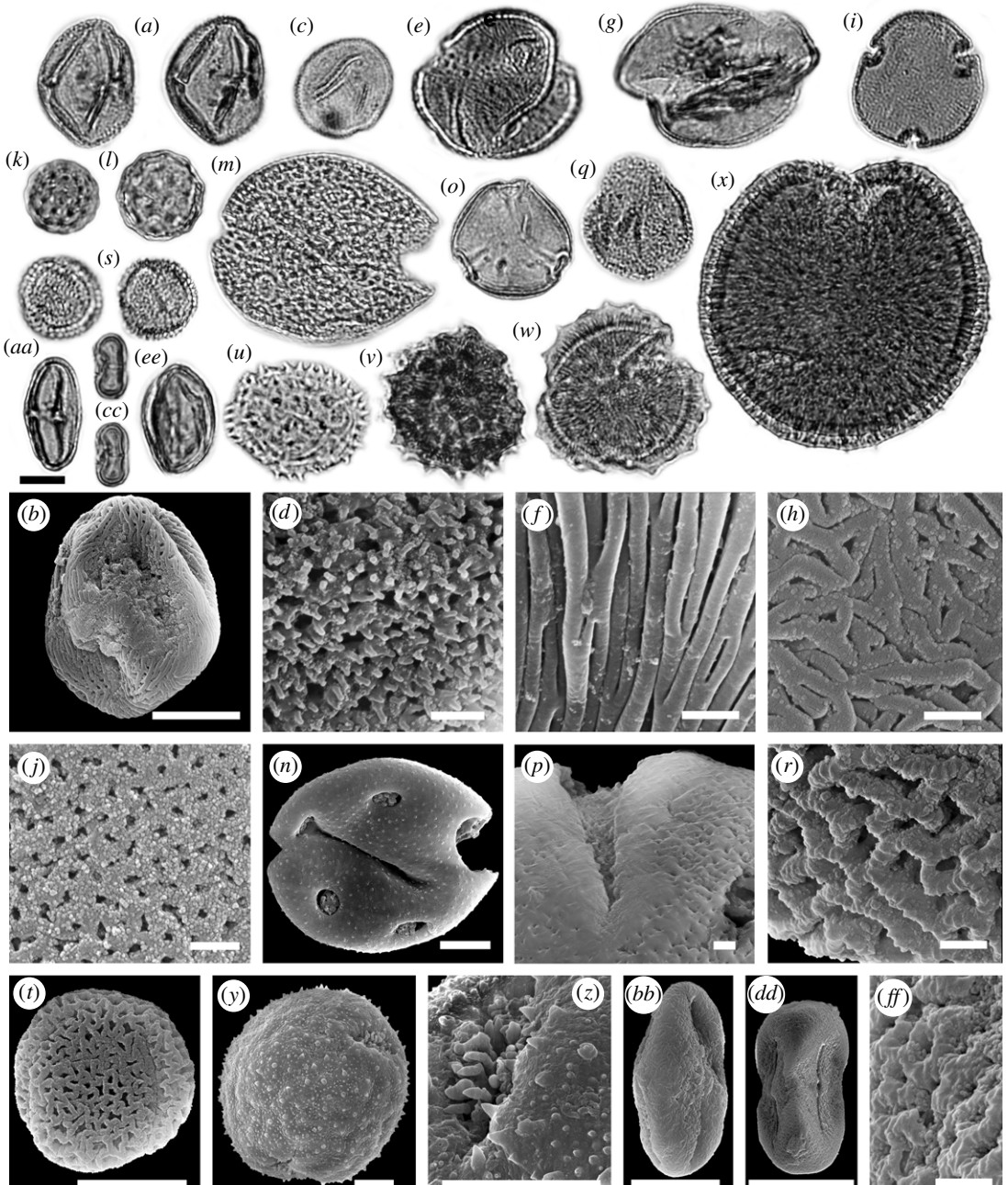

**Figure 5.** LM and SEM micrographs of Sapindales, Malvales, Caryophyllales, Cornales, Asterales, Dipsacales, and Apiales. (*a,b*) *Cotinus* sp., EV. (*c,d*) *Pistacia* sp., (*c*) PV, (*d*) exine SEM detail. (*e,f*) *Acer* sp. 1, (*e*) PV, (*f*) mesocolpium SEM detail. (*g,h*) *Acer* sp. 2, (*g*) PV, (*h*) mesocolpium SEM detail. (*i,j*) *Craigia* sp., (*i*) PV, (*j*) apocolpium SEM detail. (*k*) Amaranthaceae/Chenopodioideae gen. indet. sp. 1. (*l*) Amaranthaceae/Chenopodioideae gen. indet. sp. 2. (*m,n*) Caryophyllaceae gen. indet. (*o,p*) *Nyssa* sp., (*o*) PV, (*p*) exine sculpturing and aperture SEM detail. (*q,r*) *Fraxinus* sp., (*q*) EV, (*r*) mesocolpium SEM detail. (*s,t*) *Olea* sp., EV. (*u*) Cichorioideae gen. indet., PV. (*v*) Asteraceae gen indet. sp. 1, PV. (*w*) Asteraceae gen indet. sp. 2, PV. (*x–z*) *Succisa* sp., (*x,y*) PV, (*z*) aperture SEM detail. (*aa–bb*) Apiaceae gen. indet. sp. 1, EV. (*cc–dd*) Apiaceae gen. indet. sp. 2, EV. (*ee–ff*) Angiosperm pollen fam. et gen. indet., (*ee*) EV, (*ff*) mesocolpium SEM detail. Equatorial view (EV), polar view (PV). Scale bars 10 µm (LM, *b,n,t,y,z,bb,dd*), 1 µm (*d,f,h,j,p,r,ff*).

## 3.2. Fossil leaves, fruits and seeds: diversity and environmental signal

Leaf and fruit remains from the Vegora mine have been collected and described for half a century [16,56,57,69–71]. The most recent reviews are those of Kvaček *et al*. [18] and Velitzelos *et al*. [19]. Table 2 provides an updated taxon list. Fruits and seeds recovered from the lignite seam of the Vegora section represent aquatic and reed vegetation. From the lignite seam also trunks of tall trees

**Table 2.** Plant taxa recorded from unit 1 (lignite seam) and unit 2 (blue marls) of the Vegora section.

| Vegora micro, meso and macro flora | | | | | |
|---|---|---|---|---|---|
| **taxon** | **element** | **reference** | **life form** | **ecology** | **vegetation units [BLD subgroups]**[a] |
| **Algae** | | | | | |
| **Botryococcaceae** | | | | | |
| *Botryococcus* sp. | P | 11 | algae | aquatic | VU1 |
| **Zygnemataceae** | | | | | |
| *Spirogyra* spp. | P | 11 | algae | aquatic | VU1 |
| **Fern and fern allies** | | | | | |
| **Osmundaceae** | | | | | |
| *Osmunda* sp. | P | 11 | fern | swamp, riparian, well-drained lowland forest | VU3, VU4, VU5 |
| *Osmunda parschlugiana* | L | 2, 8 | fern | swamp, riparian, well-drained lowland forest | VU3, VU4, VU5 |
| **Polypodiaceae** | | | | | |
| *Cryptogramma* vel *Cheilanthes* sp. | P | 11 | fern | swamp, riparian, well-drained lowland forest | VU3, VU4, VU5 |
| **Pteridaceae** | | | | | |
| *Pteris* sp. | P | 11 | Fern | swamp, riparian, well-drained lowland forest | VU3, VU4, VU5 |
| **Fam. incerta sedis** | | | | | |
| Davalliaceae vel Polypodiaceae gen. indet. | P | 11 | fern | swamp, riparian, well-drained lowland forest | VU3, VU4, VU5 |
| *Laevigatosporites haardti* | P | 11 | Fern | swamp, riparian, well-drained lowland forest | VU3, VU4, VU5 |
| **Gymnosperms** | | | | | |
| **Ginkgoaceae** | L | | | | |
| *Ginkgo adiantoides* | L | 8, 9 | tree$_{gym}$ | riparian forests | VU4 |
| **Cupressaceae** | | | | | |
| *Cupressus rhenana* | L, R | 6, 8 | tree$_{gym}$ | conifer forest lowland, upland, peat-forming | VU7 |
| *Sequoia abietina* | L | 8, 10 | tree $_{gym}$ | conifer forest lowland, upland, peat-forming | VU7 |

(*Continued.*)

**Table 2.** (Continued.)

| Vegora micro, meso and macro flora | | | | | |
|---|---|---|---|---|---|
| Cryptomeria anglica | L | 10 | Tree$_{gym}$ | conifer forest lowland, upland, peat-forming | VU7 |
| Glyptostrobus europaeus | L, R | 6, 8 | tree$_{gym}$ | swamp forest | VU3 |
| Taxodium dubium | L, R | 6, 8 | tree$_{gym}$ | swamp forest | VU3 |
| Papillate Cupressaceae | P | 11 | tree$_{gym}$ | indifferent | |
| **Pinaceae** | | | | | |
| Abies sp. | P | 11 | tree$_{gym}$ | well-drained lowland and upland forests | VU5, VU6, VU7 |
| Cathaya sp. | P | 11 | tree$_{gym}$ | conifer forest lowland, upland, peat-forming | VU7 |
| Cedrus sp. | P | 11 | tree$_{gym}$ | well-drained lowland and upland forests | VU7 |
| Cedrus vivariensis | R | 4, 8 | tree$_{gym}$ | well-drained lowland and upland forest | VU7 |
| Keteleeria hoehnei | R | 6 | tree$_{gym}$ | conifer forest lowland, upland, peat-forming | VU7 |
| Pinus hampeana (diploxylon) | R | 6, 8 | tree$_{gym}$ | well-drained lowland forest | VU5 |
| Pinus salinarum (diploxylon) | R | 4, 8 | tree$_{gym}$ | well-drained lowland forest | VU5 |
| Pinus spp. | L, R | 8 | tree$_{gym}$ | indifferent | |
| Pinus sp. diploxylon type | P | 11 | tree$_{gym}$ | indifferent | |
| Pinus sp. haploxylon type | P, R | 8, 11 | tree$_{gym}$ | indifferent | |
| Pinus vegorae (haplox.) | R | 4, 6 | tree$_{gym}$ | well-drained lowland forest | VU5 |
| Tsuga spp. | P | 11 | tree$_{gym}$ | conifer forest lowland, upland, peat-forming | VU7 |
| **Angiosperms** | | | | | |
| **Cabombaceae** | | | | | |
| Brasenia sp. | R | 4 | herb | aquatic | VU1 |
| **Lauraceae** | | | | | |
| Daphnogene pannonica | L | 8 | tree | well-drained lowland forest | VU5 [BLD$_{wet}$] |
| Laurophyllum pseudoprinceps | L | 8 | tree | well-drained lowland forest | VU5 [BLD$_{wet}$] |
| Sassafras ferrettianum | L | 7, 8 | tree | riparian, well-drained lowland forest | VU4, VU5 [BLD$_{wet}$] |
| Laurophyllum sp. | L | 8 | tree | indifferent | [BLD$_{wet}$] |

(Continued.)

**Table 2.** (*Continued.*)

| Vegora micro, meso and macro flora | | | | | |
|---|---|---|---|---|---|
| **Potamogetonaceae** | | | | | |
| *Potamogeton* sp. | R | 4 | herb | aquatic | VU1 |
| **Arecaceae** | | | | | |
| *Chamaerops humilis fossilis* | L | 1, 8 | palm | well-drained lowland forest or scrub | VU0, VU4, VU5 [BLD$_{drought}$] |
| **Zingiberaceae** | | | | | |
| *Spirematospermum wetzleri* | R | 4 | herb | bogs, wet meadows | VU2 |
| **Typhaceae** | | | | | |
| *Typha* sp. | P | 11 | herb | aquatic, bogs, swamp forest, riparian forest | VU1, VU2, VU3, VU4 |
| **Poaceae** | | | | | |
| Poaceae gen. indet. | P | 11 | herb | indifferent | |
| **Cyperaceae** | | | | | |
| *Bolboschoenus vegorae* | R | 3, 4 | herb | meadows | VU2 |
| *Cladium* | R | 4 | herb | bogs, wet meadows | VU2 |
| **Ceratophyllaceae** | | | | | |
| *Ceratophyllum* sp. | R | 4 | herb | aquatic | VU1 |
| **Platanaceae** | | | | | |
| *Platanus leucophylla* | L | 8 | tree | riparian, well-drained lowland forest | VU4, VU5 [BLD$_{drought}$] |
| **Vitaceae** | | | | | |
| *Parthenocissus* sp. | P | 11 | liana | swamp forest, riparian, well-drained lowland forest | VU3, VU4, VU6 |
| **Fabaceae** | | | | | |
| *Leguminosites* sp. | L | 8 | tree | indifferent | [BLD] |
| **Ulmaceae** | | | | | |
| *Ulmus* vel *Zelkova* | P | 11 | tree | riparian, well-drained lowland forest | VU0, VU4, VU5 [BLD] |
| *Ulmus plurinervia* | L | 8 | tree | riparian, well-drained lowland forest | VU4, VU5 [BLD$_{cold}$] |
| *Zelkova zelkovifolia* | L | 8 | tree | Mediterranean scrub, riparian, well-drained lowland forest | VU0, VU4, VU5 [BLD$_{drought}$] |

(*Continued.*)

**13**

**Table 2.** (Continued.)

| Vegora micro, meso and macro flora | | | | | |
| --- | --- | --- | --- | --- | --- |
| **Fagaceae** | | | | | |
| Castaneoideae gen indet. | P | 11 | tree | well-drained lowland forest | VU5 [BLD] |
| Castanea sp. | R | 6, 8 | tree | well-drained lowland forest | VU5 [BLD] |
| Fagus sp. | P | 11 | tree | well-drained lowland and upland forest | VU5, VU5 [BLD_cold] |
| Fagus gussonii | L, R | 6, 7, 8 | tree | well-drained lowland and upland forest | VU5, VU6 [BLD_cold] |
| Quercus sect. Cerris | P | 11 | tree | well-drained lowland forest | VU5 [BLD] |
| Quercus cerrisaecarpa | R | 6, 8 | tree | well-drained lowland forest | VU5 [BLD] |
| Quercus gigas | L | 7, 8 | tree | well-drained lowland forest | VU5 [BLD] |
| Quercus kubinyii | L | 8 | tree | well-drained lowland forest | VU5 [BLD drought] |
| Quercus sect. Ilex | P | 11 | tree | Mediterranean scrub, well-drained lowland forest | VU0, VU5 [BLD_drought] |
| Quercus drymeja | L | 7, 8 | tree | well-drained lowland and upland forests | VU5, VU6 [BLD_drought] |
| Quercus mediterranea | L | 7, 8 | tree | Mediterranean scrub, well-drained lowland forest | VU0, VU5 [BLD_drought] |
| Quercus sosnowskyi | L | 7, 8 | tree | well-drained lowland forest | VU5 [BLD_drought] |
| Quercus sect. Quercus | P | 11 | tree | riparian forest, well-drained lowland forest | VU4, VU5 [BLD] |
| Quercus pseudocastanea | L | 8 | tree | well-drained lowland forest | VU5 [BLD_cold] |
| Quercus sp. | R | 8 | tree | indifferent | [BLD] |
| **Juglandaceae** | | | | | |
| Carya sp. | P | 11 | tree | riparian, well-drained lowland forest | VU4, VU5 [BLD] |
| Platycarya sp. | P | 11 | tree | riparian, well-drained lowland forest | VU4, VU5 [BLD] |
| Pterocarya paradisiaca | L | 7, 8 | tree | riparian, well-drained lowland forest | VU4, VU5 [BLD] |
| Engelhardioideae gen. indet. | P | 11 | tree | riparian, well-drained lowland forest | VU4, VU5 [BLD] |
| **Betulaceae** | | | | | |
| Alnus sp. | P | 11 | tree | swamp, riparian forest, well-drained lowland forest | VU3, VU4, VU5 [BLD_cold] |
| Alnus adscendens | L | 8 | tree | well-drained lowland forest | VU5 [BLD_cold] |
| Alnus cecropiifolia | L | 8 | tree | swamp, riparian forest | VU3, VU4 [BLD_cold] |

(Continued.)

**Table 2.** (*Continued.*)

| Vegora micro, meso and macro flora | | | | | [BLD$_{cold}$] |
|---|---|---|---|---|---|
| *Alnus* cf. *kefersteinii* | R | 6, 8 | tree | indifferent | [BLD$_{cold}$] |
| *Alnus ducalis* | L | 7, 8 | tree | well-drained lowland forest | VU5 [BLD$_{cold}$] |
| *Alnus gaudinii* | L | 8 | tree | well-drained lowland forest | VU5 [BLD$_{cold}$] |
| *Alnus julianiformis* | L | 8 | tree | riparian, well-drained lowland forest | VU4, VU5 [BLD$_{cold}$] |
| *Betula* sp. | P | 11 | tree | riparian, well-drained lowland and upland forest | VU4, VU5, VU6 [BLD$_{cold}$] |
| *Betula pseudoluminifera* | L | 8 | tree | well-drained lowland forest | VU5 [BLD$_{cold}$] |
| *Carpinus* sp. | P | 11 | tree | well-drained lowland forest | VU5 [BLD] |
| *Carpinus betulus fossilis* | R | 8 | tree | well-drained lowland forest | VU5 [BLD] |
| *Carpinus grandis* | L | 8 | tree | well-drained lowland forest | VU5 [BLD] |
| *Carpinus kisseri*, group of *C. tschonoskii* | R | 6, 8 | tree | well-drained lowland forest | VU5 [BLD] |
| *Corylus* sp. | P | 11 | shrub | well-drained lowland forest | VU5 [BLD] |
| **Salicaceae** | | | | | |
| *Populus balsamoides* | L | 8 | tree | riparian forest | VU4 [BLD] |
| *Populus populina* | L | 8 | tree | riparian forest | VU4 [BLD] |
| *Populus* spp. | L | 7, 8 | tree | riparian forest | VU4 [BLD] |
| *Salix* sp. | P | 11 | tree | swamp, riparian forest | VU3, VU4 [BLD] |
| **Geraniaceae** | | | | | |
| *Geranium* sp. | P | 11 | herb | steppe, meadows, well-drained lowland forest | VU0, VU2, VU5 |
| **Lythraceae** | | | | | |
| *Decodon globosus* | R | 4 | shrub | swamp | VU3 |
| **Anacardiaceae** | | | | | |
| *Cotinus* sp. (=*Dicotylophyllum* sp. 5) | L, P | 8, 11 | shrub | well-drained lowland forest | VU5 [BLD$_{drought}$] |
| *Pistacia* sp. | P | 11 | tree, shrub | (Mediterranean) scrub, well-drained lowland forest | VU0, VU5 |
| **Sapindaceae** | | | | | |
| *Acer aegopodifolium* | L | 7, 8 | tree | well-drained lowland and upland forests | VU5, VU6 [BLD] |

(*Continued.*)

**Table 2.** (*Continued.*)

| Vegora micro, meso and macro flora | | | | | |
|---|---|---|---|---|---|
| *Acer integrilobum* | L | 8 | tree | well-drained lowland and upland forests | VU5, VU6 [BLD] |
| *Acer limburgense* (sect. *Macrophylla*) | R | 6, 8 | tree | well-drained lowland and upland forests | VU5, VU6 [BLD] |
| *Acer pseudomonspessulanum* | L | 8 | tree | (Mediterranean) scrub, well-drained lowland forest | VU0, VU5 [BLD$_{drought}$] |
| *Acer pyrenaicum* (sect. *Rubra*) | L | 7, 8 | tree | well-drained lowland forest | VU5 [BLD] |
| *Acer subcampestre* | L | 8 | tree | well-drained lowland and upland forest | VU5, VU6 [BLD$_{drought}$] |
| *Acer triaspidatum* (sect. *Rubra*) | L | 8 | tree | swamp, well-drained lowland and upland forest | VU3, VU5, VU6 [BLD$_{cold}$] |
| *Acer* spp. | P, R | 8, 11 | tree | indifferent | [BLD] |
| **Malvaceae** | | | | | |
| *Craigia* sp. | P | 11 | tree | well-drained lowland forest | VU5 [BLD] |
| *Craigia bronnii* | R | 6, 8 | tree | well-drained lowland forest | VU5 [BLD] |
| **Droseraceae** | | | | | |
| *Aldrovandia praevesiculosa* | R | 4 | herb | aquatic | VU1 |
| **Caryophyllaceae** | | | | | |
| Caryophyllaceae gen. indet. | P | 11 | herb | steppe, meadows, well-drained lowland and upland forest | VU0, VU2, VU5, VU6 |
| **Amaranthaceae** | | | | | |
| Amaranthaceae/Chenopodioideae gen. indet. spp. | P | 11 | herb | steppe, meadows | VU0, VU2 |
| **Nyssaceae** | | | | | |
| *Nyssa* sp. | P | 11 | tree | swamp, well-drained lowland and upland forest | VU3, VU5, VU6 [BLD$_{wet}$] |
| **Oleaceae** | | | | | |
| *Fraxinus* sp. | R, P | 8, 11 | tree | riparian forest | VU4 [BLD] |
| *Olea* sp. | P | 11 | tree | Mediterranean scrub, well-drained lowland forest | VU0, VU5 |
| **Asteraceae** | | | | | |
| Asteraceae gen. indet. spp. | P | 11 | herb | steppe, meadows, well-drained lowland and upland forest | VU0, VU2, VU5, VU6, VU7 |
| Cichorioideae gen. indet. | P | 11 | herb | steppe, meadows, well-drained lowland and upland forest | VU0, VU2, VU5, VU6, VU7 |

(*Continued.*)

**Table 2.** (*Continued.*)

| Vegora micro, meso and macro flora | | | | |
|---|---|---|---|---|
| **Caprifoliaceae** | | | | |
| *Succisa* sp. | P | 11 | steppe, meadow, riparian | VU0, VU2, VU4 |
| **Araliaceae** | | | | |
| *Hedera multinervis* | L | 7, 8 | riparian, well-drained lowland forest | VU4, VU5 |
| **Apiaceae** | | | | |
| Apiaceae gen. indet. spp. | P | 11 | steppe, meadows, well-drained lowland and upland forest | VU0, VU2, VU5, VU6, VU7 |
| **Incerta sedis** | | | | |
| Monocotyledone indet. | L | 8 | swamp, riparian, lake margin | VU2, VU3, VU4 |
| Dicotylophyllum sp. 1–4, 6 | L | 8 | | |
| Monocotyledone indet. | P | 11 | | |
| Pollen indet. | P | 11 | | |

aBLD (Broadleaf deciduous forest biome of [52]) was divided in BLD drought, deciduous trees and shrubs that are drought resistant, and BLD cold, deciduous trees and shrubs that are cold tolerant; in addition, we use BLD wet for trees and shrubs that typically occur in humid warm-temperate regions. Vegetation unit (VU) 0: Steppe-like meadows with shrubs and/or small trees scattered or in groups; Mediterranean scrub. VU 1, aquatic; VU 2, bogs, wet meadows; VU 3, swamp forest; VU 4, riparian forest; VU 5, well-drained lowland forest -a 'hot' (Lauraceae, *Chamaerops*, Engelhardioideae, *Olea*); -b 'temperate' (*Castanea, Carpinus, Tilia*) including levee forests; VU 6, well-drained upland forest (-a *Quercus drymeja-mediterranea*; -b *Fagus-Cathaya*); VU 7, well-drained (lowland and) upland conifer forest including hammocks and raised bogs within peat-forming vegetation. Vegetation units from Denk [54]. L, leaves; P, palynomorph; R, reproductive structures. References: 1, [39]; 2, [55]; 3, [56]; 4, [16]; 5, [17]; 6, [57]; 7, [58]; 8, [18]; 9, [59]; 10, [60]; 11, this study.

**Table 3.** Palynomorph abundance of sample S115992.

| taxon | count |
| --- | --- |
| *Pinus* subgen. *Strobus* | 47 |
| *Alnus* sp. | 40 |
| *Abies* sp. | 36 |
| *Cathaya* sp. | 30 |
| *Fagus* sp. | 30 |
| *Pinus* subgen. *Pinus* | 33 |
| *Quercus* sp. (large) | 21 |
| Papillate Cupressaceae | 19 |
| *Spirogyra* sp. | 18 |
| *Osmunda* sp. | 18 |
| *Ulmus* vel *Zelkova* | 18 |
| *Cedrus* sp. | 18 |
| *Carya* sp. | 13 |
| Engelhardioideae gen. indet. | 10 |
| *Quercus* sp. (small) | 9 |
| *Betula* sp. | 8 |
| *Salix* sp. | 8 |
| *Leavigatosporites haardti* | 7 |
| *Tsuga* sp. | 7 |
| Amaranthaceae/Chenopodioideae gen. indet. spp. | 3 |
| Apiaceae spp. | 3 |
| *Carpinus* sp. | 3 |
| *Olea* sp. | 3 |
| Davalliaceae vel Polypodiaceae gen. indet. | 2 |
| Asteraceae gen. indet. spp. | 2 |
| *Typha* sp. | 2 |
| *Corylus* sp. | 2 |
| *Acer* sp. | 2 |
| *Nyssa* sp. | 2 |
| Incertae sedis | 2 |
| *Cryptogramma* vel *Cheilanthes* | 1 |
| *Pteris* sp. | 1 |
| Poaceae gen. indet. | 1 |
| *Geranium* sp. | 1 |
| Caryophyllaceae gen. indet. | 1 |
| Cichorioideae gen. indet. | 1 |
| *Succisa* sp. | 1 |
| *Parthenocissus* sp. | 1 |
| Castaneoideae gen. indet. | 1 |
| *Platycarya* sp. | 1 |
| *Cotinus* sp. | 1 |
| *Pistacia* sp. | 1 |
| *Craigia* sp. | 1 |
| *Fraxinus* sp. | 1 |
| Sum of counted grains and spores | 430 |

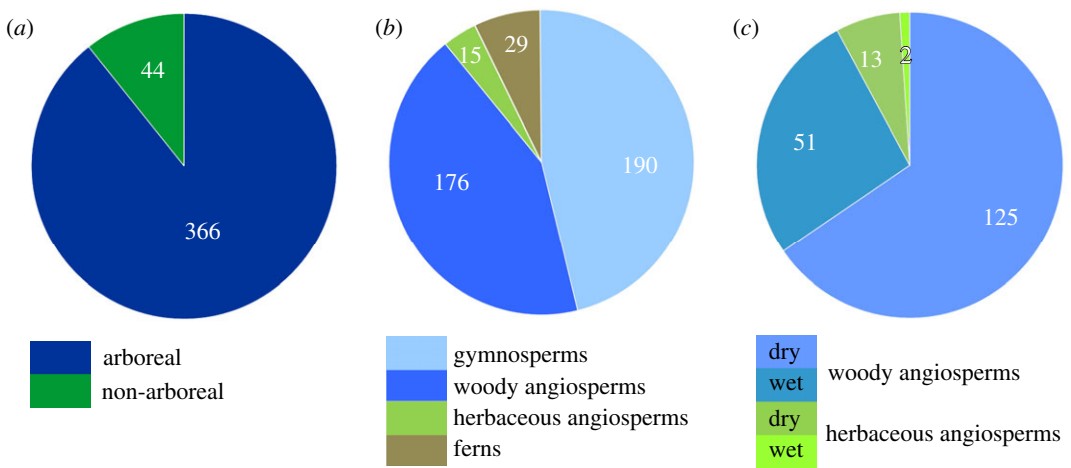

**Figure 6.** Palynomorph abundancies from sample S115992 based on 430 counted pollen and spores. Algae and unidentified angiosperm pollen excluded. (*a*) Ratio between pollen produced from arboreal and non-arboreal taxa. (*b*) Abundancies of woody gymnosperms and angiosperms and of herbaceous angiosperms and ferns. (*c*) Abundancy of woody and herbaceous angiosperms on dry and wet soils.

(as *Sequoioxylon*, [18]) in the upright position were recovered. The plant assemblage of the blue marls represents needleleaf evergreen and deciduous, and broadleaf evergreen and deciduous, as well as mixed forests. Among needleleaf forests, freshwater swamp forests are typically represented by Taxodioideae, while Sequoioideae, *Keteleeria*, *Pinus* and others may have grown on water-saturated peat and on well-drained soils of the hinterland (forest types NLE, NLD). Based on the great abundance of *Alnus* leaves, a local alder swamp can also be inferred (BLD). *Fagus* is among the most abundant taxa based on the number of the recovered leaf remains, suggesting that it was part of the mesic forest vegetation close to the lake. Also, deciduous foliage of *Quercus* sect. *Cerris* (*Q. kubinyi*, possibly *Q. gigas*) might have grown in the vicinity of the lake, either forming mixed stands with *Fagus* or oak-dominated forests. Kvaček *et al.* [18] referred to this vegetation as *Fagetum gussonii*/*Quercetum mixtum*.

Evergreen oaks are abundant in the Vegora leaf assemblage but fairly rare in the pollen record (table 3). This indicates that the leathery leaves of these taxa were transported to the area of sedimentation by slow-flowing streams and that the source vegetation was further away from the lake. Kvaček *et al.* [18] referred to these evergreen forests as sclerophyllous (*Quercetum mediterraneum*). Denk *et al.* [72] distinguished between extant sclerophyllous Mediterranean oak forest and laurophyllous *Q. floribunda* forest from Afghanistan which is a better analogue for the widespread western Eurasian fossil-taxon *Q. drymeja*. Hence, we infer an ecological cline from mesic evergreen oak forests to sclerophyllous forest and shrublands in the Messinian of Vegora (cf. [73,74]).

Well-drained forests dominated by needleleaf taxa occurred in the montane vegetation belt (*Abies*) and on rocky substrates (*Cedrus*, *Pinus*). Only a few taxa are potentially representing open shrubland vegetation (*Acer* spp., *Chamaerops*).

## 3.3. Inferring past climate with CLAMP

In this study, 41 dicot leaf morphotypes were scored for the CLAMP analysis. Given the distinctly temperate appearance of this flora, we used the calibration dataset *Physg3arcAZ_GRIDMet3arAZ*. *Physg3arcAZ* includes 173 sites, among them the 144 *Physg3brcAZ* sites plus 29 sites corresponding to the alpine nest [75]. The alpine locations are the coldest sites known to have a different physiognomic behaviour [75]; they are characterized by a WMMT lower than 16°C and a CMMT lower than 3°C [75]. The reconstructed climate parameters are MAT 10–13.5°C, WMMT 19.2–22.8°C, CMMT 1–5°C, GROWSEAS 6–8 months, GSP 700–1100 mm, MMGSP 110–160 mm, Three_WET 500–780 mm, Three_DRY 180–260 mm and Three_WET to Three_DRY ratio less than 4 (table 4). In terms of the Köppen–Geiger climate classification, this translates into a temperate *Cfb* climate ($T_{cold} > 0$ and less than 18°C; without a dry season; warm summer $T_{hot} < 22$°C).

In addition, we used the calibration dataset *PhysgAsia1_HiResGRIDMetAsia1* that adds 45 sites from China to the *Physg3brcAZ* dataset. Using this dataset, the reconstructed climate parameters are generally cooler and drier than the ones obtained from *Physg3arcAZ*. MAT 8.7–11.5°C, WMMT 19–22.6°C, CMMT

**Table 4.** Estimated climate parameters for the pre-evaporitic Messinian of Vegora from two CLAMP calibration datasets and from CA. MAT, mean annual temperature; CMMT, coldest month mean temperature; WMMT, warmest month mean temperature; GROWSEAS, duration of growing season; MMGSP, mean month growing season precipitation; Three_WET, precipitation of three consecutive wettest months; Three_DRY, precipitation of three consecutive driest months.

| climate parameter | CLAMP Physg3arcAZ | CLAMP PhysgAsia1 | CA modified | CA modified 10–90%iles |
|---|---|---|---|---|
| MAT (°C) | 10–13.5 | 8.7–11.5 | 8.6–21.2 | 9.9–18.4 |
| CMMT (°C) | 1–5 | −2.7–2.3 | ≥1.2 | — |
| WMMT (°C) | 19.2–22.8 | 19–22.6 | — | — |
| GROWSEAS (months) | 6–8 | 5.5–7 | — | — |
| MMGSP (mm) | 110–160 | 100–160 | — | — |
| Three_WET (mm) | 500–780 | 400–750 | — | — |
| Three_DRY (mm) | 180–260 | 80–220 | — | — |
| 3_WET/3_DRY | <4 | <5.5 | — | — |

−2.7–2.3°C, GROWSEAS 5.5–7 months, GSP 460–1100 mm, MMGSP 100–160 mm, Three_WET 400–750 mm, Three_DRY 80–220 mm and Three_WET to Three_DRY ratio less than 5.5. In terms of the Köppen–Geiger climate classification, this translates into a temperate *Cfb* to cold *Dfb* climate ($T_{cold} > 0$ and less than 18°C versus $T_{cold} < 0$°C; both without a dry season and warm summer $T_{hot} <$ 22°C). Score sheets and full documentation of the CLAMP analyses are provided in electronic supplementary material, S5.

## 3.4. Inferring past climate with CA

Using the CA, we estimated CMMT and MAT coexistence intervals to see how CA behaves including and excluding relictual and monotypic taxa. Following Utescher *et al.* [43], relict taxa with very limited modern distribution were excluded from the analysis. Excluded taxa are plotted to the left of the diagram in figure 7. For the monotypic genus *Chamaerops*, the tribus Trachycarpeae was used as NLR. For CMMT, a lower boundary value of 1.2°C is estimated based on the cold tolerance of Trachycarpeae. *Chamaerops* has a slightly warmer CMMT of 4°C. For MAT, the lower boundary is defined by *Zelkova* (8.6°C) and the upper boundary by *Acer* sect. *Acer* (21.2°C). When only the 10–90% percentiles were considered, MAT low was defined by *Zelkova* as 9.9°C and MAT high by *Acer* sect. *Acer* as 18.4°C (table 4 and figure 7).

Inclusion of *a priori* excluded relict species with a limited distribution would greatly change the estimated climate values. CMMT low would be defined by *Craigia* (6.5°C) and CMMT high by *Sequoia* (7.5°C). Likewise, MAT low would be defined by the monotypic conifer *Cathaya* (13.4°C) and MAT high again by *Sequoia* (15.3°C; electronic supplementary material, table S1).

Using only the 10–90% percentiles, MAT low would be defined by *Craigia* (14°C) and MAT high (14.1°C) by *Sequoia*.

## 3.5. Inferring past climate with Köppen signatures

Based on 700 Köppen signatures of modern species (rarely sections and families) genus- to family-specific Köppen signatures were used to generate Köppen signatures for the Vegora assemblage of unit 2. Temperate *C* climates are by far the most common ones represented by modern analogues of the Vegora plant assemblage. *Cfa/b* and *Cwa/b* climates represent 50% of the occurrences of NLR taxa when pollen and spores are considered, and 54% when macrofossils are considered (figure 8). *Csa/b* climates are represented by 11–13%. Snow climates (CMMT < 0°C) are represented by 17% (*Df, Dw*) and 2–3% (*Ds*). Thus, *C* and *D* climates make up more than 80% of all NLR occurrences. By contrast, equatorial climates are represented by 10% (spores and pollen) and 6.5% (macrofossils). Arid *B* climates are represented by less than 10% in the spores/pollen and macrofossil assemblages.

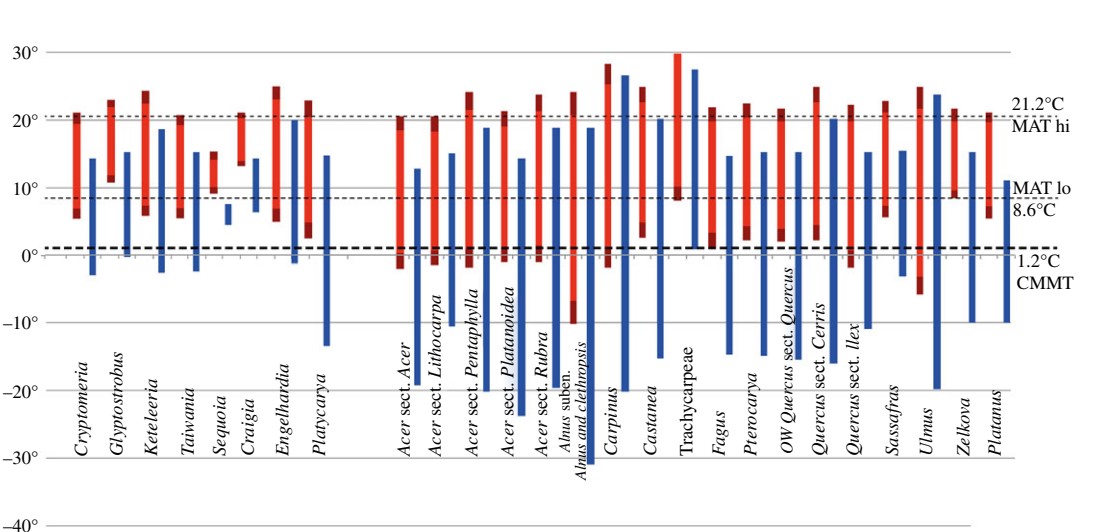

**Figure 7.** Coexistence-approach diagram showing the coexistence intervals for MAT and CMMT. MAT and CMMT climate ranges of relict taxa *a priori* excluded from the analysis are shown on the left side of the diagram. Blue bars, coldest month mean temperature; red bars, 10–90 percentile climatic range (MAT); dark red extensions, full climatic range (MAT). OW, Old World.

## 4. Discussion

### 4.1. How well do plant fossils reflect past environments of the FPS?

It has long been known that there is no exact relationship between fossil (and modern) assemblages of dispersed spores and pollen and the actual vegetation (e.g. [68,76–79]). Marinova *et al.* [79] pointed out several problems when inferring vegetation from pollen diagrams. These included (i) pollen production biases which generally result in the over-representation of woody species and the under-representation of herbaceous species in the pollen assemblage, (ii) transport of tree pollen into non-forested areas resulting in poor delineation of ecotonal boundaries, and (iii) upslope transport of pollen from lowland areas in upland areas resulting in poor delineation of altitudinal vegetation gradients and tree line.

Furthermore, these authors found that samples from small basins (less than 1 km$^2$) are more likely to be reconstructed accurately because they sample an appropriate pollen source area to reflect regional vegetation patterns in relatively heterogeneous landscapes. By contrast, large uncertainties were observed when inferring the local vegetation in large basins, e.g. the Black Sea. Here, large pollen source areas result in strongly mixed signals which do not well discriminate the vegetation belts around a specific site.

We note that this caveat may, in fact, be beneficial when inferring the past vegetation in a larger area. The FPS is a basin that extends *ca* 120 × 30 km and is flanked by high mountains. Hence, a rich pollen assemblage with a strongly mixed signal is expected to reflect the actual vegetation types in the region although it may be challenging to correctly assign particular pollen types to vegetation units. For instance, Ivanov [25] interpreted a pollen diagram from a Tortonian section in the Sandanski Graben (Bulgaria; 8 in figure 1) with a considerable amount of herbaceous pollen (including Poaceae, Amaranthaceae/Chenopodioideae and *Artemisia* making up *ca* 5–20% of the pollen spectrum) to reflect extra-regional open vegetation on an elevated plateau in addition to swamp forests, riparian forests and mixed mesophytic forests developed in a river valley and adjoined slopes. Here, downslope transportation of pollen from open landscapes blurred the local signal of the pollen record, but at the same time added regional and extra-regional vegetation information.

A close relationship between the actual vegetation and the pollen spectrum from recent and Holocene sediment samples has also been reported for northwestern Turkey [67]. Modern surface-sample spectra accurately depicted the regional vegetation, although some taxa were underrepresented in the pollen spectra, while others were overrepresented. For example, pollen spectra of *Fagus–Abies*-dominated

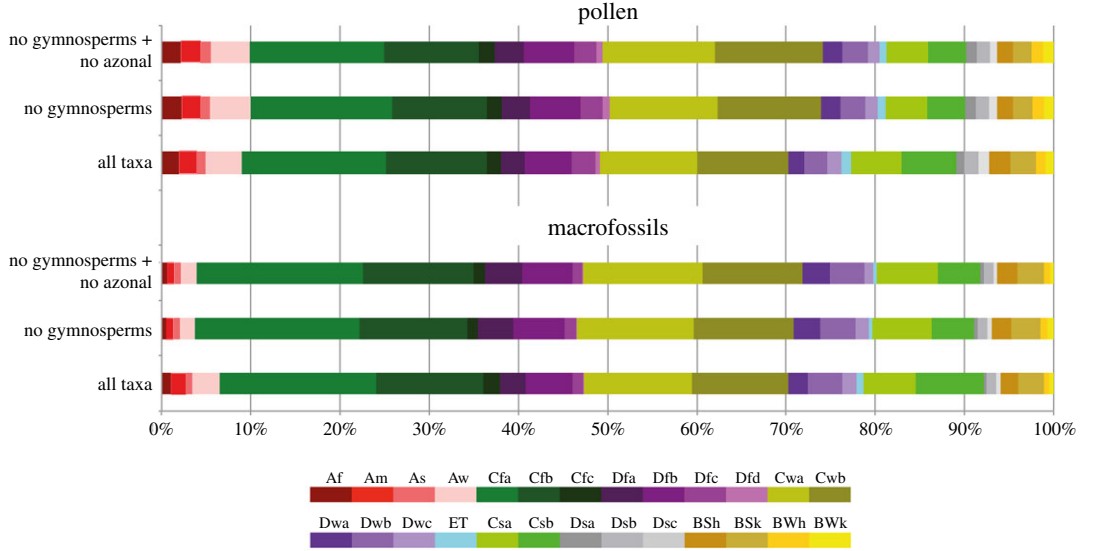

**Figure 8.** Köppen signal diagram for the macrofossil and pollen floras of Vegora. To test and illustrate the stability of the climatic signal, gymnosperms (common alpine elements) and azonal elements (e.g. riparian or swamp vegetation) were excluded in some runs.

areas showed relatively low percentages of these two taxa (10.4 and 7.4%), while high amounts of *Pinus* (*ca* 30%) derived from forests thriving at some distance from the pollen trap. Likewise, comparatively high amounts of *Juniperus*, *Quercus* and *Carpinus* did not reflect the local vegetation but a regional signal. In combination with a weak herbaceous signal (Poaceae less than 5%; Amaranthaceae/ Chenopodioideae, Caryophyllaceae, Apiaceae represented by single pollen grains), the strong arboreal signal provided a fairly accurate picture of the forest communities at a regional scale [67]. In cases of bad pollen preservation (oxidized sedimentary rocks), it should be kept in mind that only pollen with durable exines (high sporopollenin content) will be preserved (e.g. *Pinus*, Amaranthaceae/ Chenopodioideae; [78]) resulting in a biased signal.

In contrast with dispersed spores and pollen, macrofossils (leaves) mainly reflect local and regional vegetation, whereas extra-regional vegetation is usually not reflected. Leaf remains in Vegora are mostly scattered isolated carbonized compression fossils, which are not concentrated abundantly ('Blätterton' layers or paper shales) in distinct fossiliferous layers [18]; both small and larger leaves are usually not fragmented and hence there is no indication for long-distance transport in high-energy depositional settings. At the same time, low pollen abundances of evergreen oaks along with abundant leaf fossils representing evergreen oaks might indicate that these leaves were transported by slow-flowing streams over relatively large distances. Rarely, large fruit bodies are encountered, mostly represented by conifer cones. Therefore, a combined wind and water transport from habitats bordering the lake can be assumed [18]. Among woody plants, Fagaceae (*Fagus*, *Quercus*) are ecologically diverse and niche conserved at the genus/section level. *Fagus* is exclusively found on well-drained soils and hence was not an element of the swamp forest vegetation. However, under humid equable climates, lowland coastal and deltaic forests may contain *Fagus*, and hardwood hammocks with rich broadleaf deciduous and evergreen forests may be present next to aquatic and hydric vegetation [80]. By contrast, white oaks, sect. *Quercus*, may thrive in swamp forests, riparian forests, mesic forests of lowlands and uplands, or may form Mediterranean scrub. These different ecologies are well reflected in leaf morphology, whereas pollen morphology at the sectional level does not discriminate different species/ecologies [81]. Since white oaks are represented by pollen only, no further conclusions can be drawn as to their ecologies. Other sections of *Quercus* (sects *Cerris*, *Ilex*) represented in the Vegora assemblage are highly niche conserved and exclusively found on well-drained soils. Based on differences in leaf morphology (leaf size, deciduousness), fossil taxa such as *Quercus gigas* (leaf lamina up to 22 cm long; sect. *Cerris*) might indicate humid temperate conditions on northern slopes [18], while *Quercus kubinyi* (*Cerris*) might have been part of drier slopes. These fossil-species could have been accessory elements in *Fagus*-dominated or oak-dominated forests (see *Results*). Section *Ilex* comprises evergreen species exclusively growing on well-drained soils. Closest relatives of the Messinian taxa are modern Mediterranean species (the fossil-species *Q. sosnowskyi*

resembles the modern species *Q. alnifolia*, endemic to Cyprus, by leaf shape and leaf epidermal features; [18]) and Himalayan/East Asian species (e.g. *Q. drymeja* resembles the modern *Q. floribunda*, south of the Himalayas; [82]). Inferring the ecological properties of these fossil taxa is not straightforward: morphologically, they either resemble modern East Mediterranean taxa or temperate Himalayan taxa. At the same time, time-calibrated molecular phylogenies suggest that the modern Mediterranean members of sect. *Ilex* diverged from their Himalayan sister species during late Oligocene to early Miocene times, long before the deposition of the plant assemblage of Vegora [83]. Within the Mediterranean clade, the most mesic species *Q. ilex* also occurs in humid temperate forests of the Euxinian region (northern Turkey, western Georgia) and diverged from the remaining species of western Eurasian sect. *Ilex* no later than 9 Ma [83]. Assuming that fully Mediterranean climate conditions, with precipitation minima during the summer, in the Mediterranean region did not establish prior to the early Pliocene [3,19], we speculate that the Messinian members of sect. *Ilex* were chiefly temperate species that went extinct during the Pliocene (cf. [83]). Specifically, *Q. drymeja* might have formed a forest belt above the *Fagetum gussonii*/*Quercetum mixtum* and below the needleleaf evergreen forest belt. Other *Quercus* sect. *Ilex* such as *Quercetum mediterranea* and *Q. sosnowskyi* may have formed woody shrublands or forests on drier sites (edaphically or due to the aspect of the slope).

Concerning the presence of grasslands or open woodlands, the palaeobotanical data at hand cannot discriminate between different scenarios. For taxa that are known from the macrofossil record (*Chamaerops*, evergreen oaks), it is almost certain that they were part of the regional flora of the FPS. The woody genera *Olea*, *Cotinus* and *Pistacia*, known only from the pollen record of unit 2, are typical elements of the present Mediterranean and submediterranean vegetation belt in Southern Europe. Bell & Fletcher [68] found that soil samples in open vegetation plots in northern Morocco recorded 20–35% AP. Main contributors to this regional to extra-regional airborne pollen rain were *Quercus* types and *Olea*. In our sample, AP makes up almost 90% of the total count. *Quercus* certainly was a major component of local to regional vegetation because it is the most prominent component with several deciduous and evergreen species in the leaf flora of Vegora. By contrast, *Olea* makes up less than 1% in the pollen count. No leaf and seed remains reminiscent of *Olea* are recorded from Vegora. This, along with the known ability for long-distance transport [68], might indicate the presence of *Olea* at a greater distance from the Vegora Basin. The same can be assumed for *Cotinus* and *Pistacia*. The latter, however, do also occur in open-canopy pine forests.

In the case of herbaceous taxa represented by single or very few pollen grains in the palynological record, these may also reflect long-distance dispersal (LDD) from high mountain or even from more distant coastal areas to the west of the FPS. They would then provide an extra-regional vegetation signal. Potential elements of open vegetation include Apiaceae, Amaranthaceae/Chenopodioideae, Poaceae, *Geranium*, Caryophyllaceae, Asteraceae and Cichorioideae. Except for Poaceae and Amaranthaceae/Chenopodioideae, these taxa are predominantly insect-pollinated. For wind-pollinated taxa represented with 1–3 grains in the pollen count (Poaceae, Amaranthaceae/Chenopodioideae), we assume that this is indicative of a regional or extra-regional source vegetation. The insect-pollinated taxa, also represented by 1–3 grains in the pollen count, are difficult to assign to either local or regional/extra-regional vegetation. If these groups were local elements, they would have been quite rare, based on the low numbers of their pollen grains. They could have been part of the lakeshore vegetation, of open rocky places, of the understorey of forest vegetation or meadows above the tree line. Alternatively, these elements could have been brought in by LDD from coastal plains to the southeast and east of the FPS.

In sum, the combined macrofossil and microfossil record offers an accurate picture of the different vegetation types present in the FPS during the Messinian. The fossil record suggests that the local and regional vegetation in the FPS comprised a range of ecologically different zonal and azonal forest types, while LDD of several herbaceous taxa may potentially have contributed to an extra-regional pollen signal.

## 4.2. Inferring Messinian pre-evaporitic vegetation of the FPS and adjacent areas

Our multi-proxy palaeobotanical study of the Messinian assemblage of Vegora is based on information from fruits and seeds, leaves, and dispersed pollen and spores. For the main flora in unit 2 (blue marls), we used information from leaf fossils and dispersed spores and pollen.

As discussed above, there is strong evidence for the presence of a wide range of forest and forest/shrubland types in the FPS. Furthermore, a small number of woody and herbaceous taxa could reflect open vegetation. The latter are represented by low numbers of pollen grains, which could be ascribed to LDD from remote areas including dry uplands or coastal plains. In order to evaluate the

pre-evaporitic Messinian vegetation of the FPS, we compared our finds with previously published data on other plant fossil localities in the FPS and surrounding areas. In addition, a vertebrate locality from the Axios valley (Dytiko1, 2, 3; 3 in figure 1) is roughly coeval with the Messinian pre-evaporitic assemblages of the FPS [22]. The hypsodonty index of this fauna of 1.45–1.86 [84] corresponds to the diet types 'mixed-closed habitats', 'regular browsers' and 'selective browsers' according to Janis [85] and hence provides an excellent match with the environments inferred for the FPS.

From Lava (figure 1), Steenbrink et al. [13] investigated two sequences covering two sedimentary cycles each. Based on palaeomagnetic correlation, these sequences are dated as ca 6.8–6.7 Ma and ca 6.3 Ma. The pollen assemblages are comparable to the Vegora assemblage but differ in some respects. First, the Lava sections have a continuous high amount of Pinus pollen (20 to greater than 60%), suggesting that the fossil site was located closer to pine forests than was the Vegora lake. Second, Cedrus pollen is abundant with values between 10 and more than 30%. Third, Steenbrink et al. [13] did not report evergreen oak pollen, although evergreen Quercus is known from Lava based on leaf fossils [19]. Steenbrink et al. [13] inferred a humid temperate climate without dry season for the investigated sedimentary cycles. In addition, they suggested that expansions of Fagus accompanied by a decrease of Abies might reflect subtle increases in montane humidity. Overall, they suggested continuously wet and warm-temperate climate conditions for the investigated period for Lava.

Velitzelos et al. [19] provided revised taxon lists for the roughly coeval macrofossil (leaves and fruits/ seeds) localities Prosilio and Lava (age based on palaeomagnetic correlation, 6.7–6.4 Ma; [11]). The macroflora is very similar to the one from Vegora in terms of composition. However, whereas Q. sosnowskyi is among the most abundant elements in Vegora, only a few leaves represent this species in Prosilio; also Glyptostrobus is much less abundant. Pinus is represented by cones, leaf fascicles and leafy branches; this is in accordance with the high amount of pine pollen documented in the palynological record. Fagus is a frequent element as well, while Abies is not recorded in the macroflora of Prosilio and Lava. From the Prosilio section, no comprehensive palynological study is available. However, Biltekin [86] investigated a 5 cm thick green clay bed dated at 6.6–6.55 Ma (key bed II of [13]) that represents an insolation minimum. This bed, void of leaf fossils, was identified from the Lava and Prosilio sections [13] and is situated between the two palynological sections studied from Lava. The three samples from Prosilio investigated by Biltekin [86] are similar to the Lava pollen assemblages (dominance of conifer pollen, low percentages of herbaceous pollen). However, the samples from Prosilio show peaks of Pinus and Tsuga (samples 1 and 3) and of Cupressaceae (sample 2) that are not known from other pollen assemblages in the FPS. In addition, sample 2 has moderate amounts of Artemisia. The deviating pollen spectrum from this layer was explained by uplifting of the surrounding region [86]. Alternatively, the cooler appearance of this assemblage could reflect the insolation minimum recorded for this interval [13].

Likoudi, 20 km S of Lava, is located in a small basin south of the main FPS (figure 1). The macroflora (leaves, fruits and seeds) is very rich (see revised and updated floral list in [19]). The precise age of the Messinian diatomaceous marls is not clear [14], although it clearly is pre-evaporitic. The flora is characterized by the high diversity of conifers (11 genera of Cupressaceae, Pinaceae and Taxaceae including Torreya—as Egeria sp. in [19]). As in Vegora, Fagus is a dominating element. Other taxa (Cercis, Laria, cf. Nerium) are not known from other FPS floras. Well-preserved cones of Cedrus and cones and leafy twigs of Cathaya and Taiwania suggest that these genera were not growing at high elevations but nearby the area of deposition (lake). If coeval with the Lava deposits, this would explain the relatively high amounts of Cedrus pollen in the palynological section of Lava.

Ivanov & Slavomirova [87] investigated a 70 m succession of lacustrine sediments in the Bitola Basin (northern Macedonia; borehole V-466; 1 in figure 1) about 10 km E of Bitola and 40 km NNW of Vegora. Based on a vertebrate fauna on top of these sediments, the plant-bearing sediments are assigned a late Miocene age [24,88]. From these sediments, abundant leaves of Q. sosnowskyi have been reported [88]. The pollen assemblage is similar to the one from Vegora by its high amounts of woody taxa (both conifers and angiosperms) and the composition and very low abundance of herbaceous taxa. In contrast with the Vegora assemblage, few taxa are represented by high percentages (Pinus 20–40%, Fagus up to 10%, Quercus 10–20%, Taxodioideae and Sequoioideae 10–15% and Alnus up to 20%). In the Vegora sample, only Pinus reached more than 10% in the pollen count.

For north and central Italy, Kovar-Eder et al. [6] and Bertini & Martinetto [89] reported the prevalence of deciduous Fagus and Quercus in the northern parts, whereas in addition sclerophyllous plants (e.g. Quercus mediterranea) were common in Messinian pre-evaporitic assemblages of central and southern Italy (Gabbro I, Senigallia, Palena).

From the Serres Basin (7 in figure 1), Psilovikos & Karistineos [90], Karistineos & Ioakim [91] and Suc et al. [92] reported pollen assemblages that reflect lowland swamp forests (NLD and BLD) and hinterland vegetation, including broadleaf deciduous and needleleaf evergreen forests. The age of these localities is younger than Vegora and based on mammal data [22] and geological data [92] might have been deposited during the post-evaporitic Messinian or the earliest Zanclean.

It is remarkable that post-evaporitic and evaporitic Messinian floras from Italy [89] and Greece [19] mainly reflect moist conditions and persistence of forested environments. This is in strong contrast to conditions from Turkey, where steppe and steppe forest were established by the end of the middle Miocene [8]. Hence, pre-evaporitic sedimentary rocks from the Dardanelles strait contain palynological assemblages that clearly demonstrate the presence of herb-dominated coastal vegetation before the peak of the MSC [93].

By contrast, pre-evaporitic palynological assemblages from northwestern Bulgaria (Drenovets) and from the western Black Sea (site 380-A) show that herbaceous and steppe elements were nearly absent from this region [5,94,95]. Fauquette et al. [5] reconstructed palaeoclimate using the so-called climatic amplitude method, which takes into consideration requirements of modern analogue taxa and pollen abundances of individual taxa. These authors inferred an MAT of (10–) 15–20°C, a CMMT greater than 5°C and MAP of 1000–1500 mm for the pre-evaporitic Messinian of site 380-A.

This is slightly warmer than reconstructed palaeoclimate parameters for the Vegora assemblage (cf. table 4).

Main vegetation types recognized for the pre-evaporitic Messinian of the FPS and their modern analogues are summarized in table 5.

## 4.3. Inferring Messinian pre-evaporitic climate of the FPS and adjacent areas

Fruit/seed and leaf records mainly represent azonal vegetation and may not adequately reflect the zonal vegetation growing at some distance from the area of sedimentation. Zonal vegetation is mainly controlled by large-scale climate (regional, extra-regional). Azonal vegetation is controlled by edaphic conditions rather than large-scale climate. For these reasons, seed/fruit and leaf assemblages either used for physiognomic (CLAMP) or taxonomic (Köppen signatures, CA) methods may not be suitable for a meaningful (representative) climate reconstruction or they may reconstruct a local climate. However, as shown by Ferguson et al. [80], under mild and humid climate conditions, the vegetation close to the area of sedimentation consists of a mosaic of vegetation types, some of which are composed of woody plants that also dominate the hinterland vegetation. For example, natural levee and hardwood hammock vegetation associated with azonal swamp, riparian and bog vegetation in southeastern North America contains taxa such as Fagus, Magnolia, Ilex spp., Carpinus and Symplocos. In this case, the limitations outlined above are not valid. In drier, strongly seasonal climate settings, the azonal vegetation will not be sufficient to produce meaningful (representative) regional climate estimates.

When the palynological record is taken into consideration, both the local and the regional (hinterland, vertical vegetation belts) vegetation is likely to be captured. This is reflected in the much greater diversity of conifers and herbaceous taxa in the pollen record when compared with the macrofossil record in the Vegora assemblage. In addition, LDD may add information from the extra-regional vegetation (e.g. Olea; [67,68]).

The resulting differences in reconstructed climate/vegetation are seen in the Köppen signatures for the leaf fossil and pollen/spore records of Vegora (figure 8). Taxa extending into tropical climates are much better represented in the pollen record than in the leaf record. Therefore, Köppen signatures, although not providing exact values for different climate parameters, offer important qualitative information. This can be illustrated by comparing three floras from the East Mediterranean region, all dominated by Quercus spp. and Fagus and ranging in age from early Burdigalian to Messinian: the early Burdigalian flora of Güvem, Anatolia (MN3, 20–18 Ma; [82,104]), has a tropical signal of 16% in Köppen signatures from pollen and spore assemblages, whereas tropical signal from leaf fossils is 11.9%. The middle Miocene floras of the Yatağan Basin (Langhian/Serravallian, MN6/MN7+8; 14.8–13.8 Ma; [7]) show a trend from the more humid MN6 zone to the more arid MN7+8 zone with tropical signal from pollen and spore assemblages decreasing from 11% in pollen zones 1 and 2 to 7% in the transitional pollen zone 2/3. The tropical signal in Köppen signatures from the macrofossil assemblage from pollen zone 2 is 9%. In the pre-evaporitic assemblages of Vegora, the tropical signal is 9% and 6% for the pollen/spore and leaf assemblages.

By contrast, the CA, in producing exact values for selected climate parameters, will be prone to produce hybrid climates if fossil assemblages represent a range of lowland azonal and hinterland vegetation and vegetation from different vertical belts. Such artificial climates may be randomly expressed as narrow coexistence intervals including different vertical vegetation belts, or in very broad

**Table 5.** Vegetation types recognized for the pre-evaporitic Messinian of the Florina–Ptolemais–Servia Basin. 1, [96]; 2, [64]; 3, [97]; 4, [98]; 5, [99]; 6, [100]; 7, [18]; 8, [101]; 9, [102]; 10, [67]; 11, [73]; 12, [103].

| vegetation type | main (and accessory) taxon/taxa | biome[a] | vegetation unit(s)[b] | modern (Neogene) analogue | references |
|---|---|---|---|---|---|
| swamp forest | Taxodium, Glyptostrobus | NLD | VU3 | Taxodium swamp forests SE USA; (Taxodium/Glyptostrobus swamp forests widespread in N Hemisphere Neogene) | 1, 2 |
| swamp forest | Alnus, (Sassafras) | BLD | VU3 | Alnus swamp forest | 3, 4 |
| riparian forest | Pterocarya, Zelkova, Ulmus, (Sassafras) | BLD | VU4 | riparian and alluvial forest of Georgia and Iran | 1, 3, 4, 5, 6 |
| well-drained forest | Quercus kubinyi, Q. pseudocastanea, (Carpinus, Tilia etc.) | BLD | VU5b | lowland oak–hornbeam forests; ('Quercetum mixtum') | 4, 7 |
| well-drained forest | Fagus, (Quercus pseudocastanea) | BLD | VU5b | lowland beech forests of N Turkey, Georgia, N Iran; ('Fagetum gussonii') | 3, 4, 7 |
| well-drained forest | Fagus, Abies, Cedrus, Cathaya | MIXED | VU6b | Montane Fagus–Abies forest, montane Fagus–Cedrus–Pinus forest; Abant Gölü; Erbaa-Çatalan | 8, 9, 10 |
| well-drained laurophyllous forest | Quercus drymeja, (Q. sosnowsky) | BLE | VU6a | Quercus dilatata association (with Taxus, Pinus, Acer, etc.) | 11 |
| well-drained sclerophyllous forests/shrublands | Quercus mediterranea, Chamaerops, Olea | BLE/SHRUBLAND | VU0 | Mediterranean sclerophyllous forest/shrublands | 10 |
| [?][c] Grassland-steppe forest | Poaceae | GRASSLAND/SHRUBLAND | VU0 | forest-steppe of SE Europe to Afghanistan | 8, 10, 11, 12 |

[a]Biome classification follows the physiognomic approach of [52].

[b]Vegetation units as in table 1.

[c][?] expresses the uncertainty around a possible extra-regional signal in the Vegora pollen record. According to Erdős et al. [103], steppe forest with Stipa and other grasses and different species of Quercus (forest-steppes of the type 'Region A—SE Europe') is characterized by MAP of 420–600 mm; this would be much drier than the inferred MAP for the FPS.

coexistence intervals (cf. table 4). Thus, Köppen signatures are considered a more dynamic way of climate reconstruction as they account for the possibility that different elements may derive from different vertical vegetation belts.

CLAMP (Physg3arcAZ) reconstructed WMMT, which corresponds with a temperate climate with warm summers, *Cfb* Köppen climate type, underscoring the temperate character of the Vegora assemblage. The calibration set Asia 1 reconstructed a cooler climate with CMMT below 0°C. This would appear to be in conflict with the cold tolerance of the palm *Chamaerops* (cf. figure 7) that is recorded from the leaf record and palms in general as documented in the pollen record. However, all three approaches to reconstruct the pre-evaporitic Messinian climate of the FPS suggest a temperate climate with weak seasonality (table 4).

In view of the relatively homogeneous vegetation signal from the FPS (both from dispersed spores/pollen and leaves), the inferred climate is highly plausible.

# 5. Conclusion

The present study used palaeobotanical data to reconstruct palaeoenvironments in an intermontane basin of northwestern Greece shortly before the MSC. For the period 6.4–6 Ma, leaf fossil data and dispersed spores and pollen indicate the presence of various types of forest including riparian and mesic forests, deciduous and evergreen forests, laurophyllous forests and sclerophyllous woodlands. Open landscapes dominated by herbaceous plants were not reconstructed for the FPS Basin. Based on sparse pollen records resulting from potential LDD, it cannot be ruled out that herbaceous plants played a more important role in coastal lowlands to the west of the study area. However, contemporaneous vertebrate faunas from the Axios valley also suggest mixed forested vegetation. We found that the combined leaf and dispersed spore/pollen records allow fairly accurate reconstruction of local and regional vegetation. Leaf fossils offer more species-diagnostic features than pollen and a combination of leaf taphonomy and pollen frequencies allow discriminating local and regional vegetation. Furthermore, specific comparison with modern pollen spectra was made in order to understand biases in pollen abundances. This provided a transfer function for the interpretation of fossil pollen assemblages. The results of this study confirm previous findings of a north to south gradient of temperature and precipitation seasonality in the Mediterranean area during the Messinian. Our results also reinforce the notion that steppe and forest-steppe environments evolved earlier in Turkey with deciduous oaks playing important roles in the woody flora. By contrast, laurophyllous evergreen oak forest persisted in Greece/Italy into the late Miocene and as relict into the Pliocene. Climate reconstructions using three different approaches to climate reconstruction resulted in roughly similar values that translate into a cool temperate *Cfb* climate according to the Köppen–Geiger climate classification. When comparing CLAMP, CA and Köppen signatures, we noted that leaf assemblages may be biased towards autochthonous plant communities and by this, the climate signal will be highly local. However, under humid mild climates plants from the hinterland vegetation may thrive from lowlands to high altitudes and interspersed in riparian landscapes (e.g. on hammocks, gallery forests, etc.). Furthermore, the taphonomy of the fossil plant assemblage of Vegora suggests that slow-flowing streams had transported leaves from different vegetation types of the hinterland into the lake and thus, the leaf taphocoenoses would be representative of both the local and regional vegetation. This causes a further problem for CLAMP and more so for CA: climate signals in the fossil plant assemblage may be highly mixed and may derive from different vertical vegetation belts. Köppen signatures, while not generating exact values for particular climate parameters, overcome the problem of hybrid climates as they collect climatic/environmental signal in a plant assemblage without averaging different signals into a single, possibly artificial, signal.

Data accessibility. This article and all data used in this article are made available in bioRxiv. https://www.biorxiv.org/content/10.1101/848747v1. Electronic supplementary material is available within the figshare repository: https://doi.org/10.6084/m9.figshare.10327646.v1.

Authors' contributions. T.D. designed the study and wrote the first draft. J.M.B. investigated dispersed pollen and spores and made the Köppen signatures and CA analyses. T.H.G. made the CLAMP analysis. T.D., J.M.B., T.H.G., D.V. and E.V. evaluated the fossil data and wrote the final paper.

Competing interests. We declare we have no competing interests.

Funding. This study was financed by a grant of the Swedish Research Council (VR; project no. 2015-03986) to T.D. T.H.G. was supported by The Scientific and Technological Research Council of Turkey (TÜBİTAK), 2219 Post-Doctoral Research Fellowship Program (2017/2), project no. 1059B191700382.

Acknowledgements. We thank the associate editor Emily Lindsey. Valuable suggestions by the reviewers Jean-Pierre Suc and Jakub Sakala helped to improve the final manuscript.

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
