## [Reviewer comments · Royal Society Open Science]

Review History

RSOS-192067.R0 (Original submission)

Review form: Reviewer 1 (Jean-Pierre Suc)

Is the manuscript scientifically sound in its present form?

Yes

Are the interpretations and conclusions justified by the results?

Yes

Is the language acceptable?

Yes

Do you have any ethical concerns with this paper?

No

Have you any concerns about statistical analyses in this paper?

No

Recommendation?

Accept with minor revision (please list in comments)

Comments to the Author(s)

The manuscript submitted by Bouchal et al. is an excellent case-study of parallel investigation of a macroflora and a microflora provided by the same stratigraphic level (there are so few!).

I consider that the text is very well constructed and the figures and tables clear and very useful.

In intermontane small basins such as the Florina-Ptolemais-Servia Basin, it is difficult to determine if the recorded pollen grains come from a short-distance fluvial transport or from a long-distance air transport in contrast to the macroremains which represent the local palaeovegetation. The authors rightly underline the differences in pollen floras from Vegora and Lava although macrofloras are very similar. This is also true for the Prosilio locality, which provided a pollen flora somewhat different from the previous ones (see: Biltekin, 2010 pp. 76–78 – attached) coming from a pre-evaporitic Messinian layer (just underlying the Messinian Erosional Surface shown in Suc et al., 2015).

I have only very few comments on this paper: some minor ones are directly indicated on the annotated manuscript (attached (Appendix A)) and the annotated Supplementary Material Table S4 (attached (Appendix B)).

Maybe, some suggestions expressed below could be followed by the authors:

- to introduce (in lines 495–513 or 515–531) a brief allusion to the pre-evaporitic Messinian pollen flora from Intepe (Dardanelles Strait, Turkey – Melinte-Dobrinescu et al., 2009, *Palaeogeogr. Palaeoclimatol. Palaeoecol.*, 278, 24–39) which displays a very open palaeovegetation (see also for details: Biltekin, 2010 pp. 61–64 – attached (Appendix C));
- to use for comparison (Subsect. 4.2) some of the pollen data yielded by Jiménez-Moreno et al. (2007) – attached (Appendix D);
- to compare the palaeoclimatic estimates provided by the ‘Coexistence Approach’ (Subsect. 4.3) with some palaeoclimatic estimates given by the ‘Climatic Amplitude Method’ on coeval localities from the region s.l. (Site 380: Fauquette et al., 2006);
- to pay some caution to strong differences in palaeoclimate reconstructions between the CLAMP method and transferred pollen records (as evidenced on an early Pleistocene flora by Girard et al., 2019, *Rev. Palaeobot. Palynol.*, 267, 54–61).

As a conclusion, the Bouchal et al.’s manuscript deserves to be rapidly published after a minor revision.

My name can be indicated to the authors.

Dr. Jean-Pierre Suc.

Review form: Reviewer 2 (Jakub Sakala)**Is the manuscript scientifically sound in its present form?**

Yes

Are the interpretations and conclusions justified by the results?

Yes

Is the language acceptable?

Yes

Do you have any ethical concerns with this paper?

No

Have you any concerns about statistical analyses in this paper?

No

Recommendation?

Accept with minor revision (please list in comments)

Comments to the Author(s)

Another very informative paper from Bouchal et al. on the Neogene floras and vegetations of Eastern Mediterranean area. The only pity is that many useful pieces of information are somehow hidden in Supplementary material...

I did some remarks / corrections / suggestions directly in the pdf attached (Appendix E) to this review. But definitely, well done !

Jakub Sakala, March 13, 2020

Review form: Reviewer 3

Is the manuscript scientifically sound in its present form?

No

Are the interpretations and conclusions justified by the results?

No

Is the language acceptable?

Yes

Do you have any ethical concerns with this paper?

No

Have you any concerns about statistical analyses in this paper?

No

Recommendation?

Reject

Comments to the Author(s)

Here are some major concerns.

1. Age control. The manuscript failed to show the details of the chronology data.
2. Pollen sample and treatment. Only one pollen sample was analyzed, which makes it difficult to make climate estimates and biome reconstruction for a certain interval. How much sample was used for treatment? How many pollen grains were counted?
3. Pollen data presentation. A diagram of major pollen taxa should be added in the main text rather than in the supplementary data, as this is the fundamental data.
4. Manuscript structure. The manuscript could be more well-organized. As for now, it is not easy to clearly follow.

Decision letter (RSOS-192067.R0)

03-Apr-2020

Dear Dr Bouchal

On behalf of the Editors, I am pleased to inform you that your Manuscript RSOS-192067 entitled "Messinian vegetation and climate of the intermontane Florina-Ptolemais-Servia Basin, NW Greece: How well do plant fossils reflect past environments?" has been accepted for publication in Royal Society Open Science subject to minor revision in accordance with the referee suggestions. Please find the referees' comments at the end of this email.

A number of attachments have been issued with this message. If you have not received these, please contact the editorial office (openscience@royalsociety.org), and they will be happy to assist.

The reviewers and handling editors have recommended publication, but also suggest some minor revisions to your manuscript. Therefore, I invite you to respond to the comments and revise your manuscript.

- Ethics statement

- Data accessibility

If you wish to submit your supporting data or code to Dryad (<http://datadryad.org/>), or modify your current submission to dryad, please use the following link:
<http://datadryad.org/submit?journalID=RSOS&manu=RSOS-192067>

- Competing interests

- Authors' contributions

- Acknowledgements

- Funding statement

Because the schedule for publication is very tight, it is a condition of publication that you submit the revised version of your manuscript before 12-Apr-2020. Please note that the revision deadline will expire at 00.00am on this date. If you do not think you will be able to meet this date please let me know immediately.

- 1) A text file of the manuscript (tex, txt, rtf, docx or doc), references, tables (including captions) and figure captions. Do not upload a PDF as your "Main Document";
- 2) A separate electronic file of each figure (EPS or print-quality PDF preferred (either format should be produced directly from original creation package), or original software format);
- 3) Included a 100 word media summary of your paper when requested at submission. Please ensure you have entered correct contact details (email, institution and telephone) in your user account;
- 4) Included the raw data to support the claims made in your paper. You can either include your data as electronic supplementary material or upload to a repository and include the relevant doi within your manuscript. Make sure it is clear in your data accessibility statement how the data can be accessed;

5) All supplementary materials accompanying an accepted article will be treated as in their final form. Note that the Royal Society will neither edit nor typeset supplementary material and it will be hosted as provided. Please ensure that the supplementary material includes the paper details where possible (authors, article title, journal name).

If your manuscript is newly submitted and subsequently accepted for publication, you will be asked to pay the article processing charge, unless you request a waiver and this is approved by Royal Society Publishing. You can find out more about the charges at <https://royalsocietypublishing.org/rsos/charges>. Should you have any queries, please contact openscience@royalsociety.org.

on behalf of Dr Emily Lindsey (Associate Editor)
openscience@royalsociety.org

Associate Editor Comments to Author (Dr Emily Lindsey):

This manuscript represents an important contribution to paleobotany, the investigation of the relationship between pollen/spore and plant macrofossil proxies. The article has been reviewed by three reviewers, two of whom recommend accepting it with only minor revisions, and one of whom recommended rejection of the article. However, it seems that the concerns of the third reviewer can be addressed by various revisions that were suggested by the three reviewers -- restructuring the article as necessary for clarity and general editing; making sure all relevant data is clearly presented; moving some figures from the supplemental data into the main manuscript; expanding discussion of/comparisons with other sites/paleoclimate studies; and more explicitly acknowledging potential shortcomings.

Reviewer comments to Author:
Reviewer: 1

Comments to the Author(s)
The manuscript submitted by Bouchal et al. is an excellent case-study of parallel investigation of a macroflora and a microflora provided by the same stratigraphic level (there are so few!).

I consider that the text is very well constructed and the figures and tables clear and very useful. In intermontane small basins such as the Florina-Ptolemais-Servia Basin, it is difficult to determine if the recorded pollen grains come from a short-distance fluvial transport or from a long-distance air transport in contrast to the macroremains which represent the local palaeovegetation. The authors rightly underline the differences in pollen floras from Vegora and Lava although macrofloras are very similar. This is also true for the Prosilio locality, which provided a pollen flora somewhat different from the previous ones (see: Biltekin, 2010 pp. 76–78 – attached) coming from a pre-evaporitic Messinian layer (just underlying the Messinian Erosional Surface shown in Suc et al., 2015).

I have only very few comments on this paper: some minor ones are directly indicated on the annotated manuscript (attached) and the annotated Supplementary Material Table S4 (attached). Maybe, some suggestions expressed below could be followed by the authors:

- to introduce (in lines 495–513 or 515–531) a brief allusion to the pre-evaporitic Messinian pollen flora from Intepe (Dardanelles Strait, Turkey – Melinte-Dobrinescu et al., 2009, *Palaeogeogr. Palaeoclimatol. Palaeoecol.*, 278, 24–39) which displays a very open palaeovegetation (see also for details: Biltekin, 2010 pp. 61–64 – attached);
- to use for comparison (Subsect. 4.2) some of the pollen data yielded by Jiménez-Moreno et al. (2007) – attached;
- to compare the palaeoclimatic estimates provided by the ‘Coexistence Approach’ (Subsect. 4.3) with some palaeoclimatic estimates given by the ‘Climatic Amplitude Method’ on coeval localities from the region s.l. (Site 380: Fauquette et al., 2006);
- to pay some caution to strong differences in palaeoclimate reconstructions between the CLAMP method and transferred pollen records (as evidenced on an early Pleistocene flora by Girard et al., 2019, *Rev. Palaeobot. Palynol.*, 267, 54–61).

As a conclusion, the Bouchal et al.’s manuscript deserves to be rapidly published after a minor revision.

My name can be indicated to the authors.

Dr. Jean-Pierre Suc.

Reviewer: 2

Comments to the Author(s)

Another very informative paper from Bouchal et al. on the Neogene floras and vegetations of Eastern Mediterranean area. The only pity is that many useful pieces of information are somehow hidden in Supplementary material...

I did some remarks / corrections / suggestions directly in the pdf attached to this review. But definitely, well done !

Jakub Sakala, March 13, 2020

Reviewer: 3

Comments to the Author(s)

Here are some major concerns.

1. Age control. The manuscript failed to show the details of the chronology data.
2. Pollen sample and treatment. Only one pollen sample was analyzed, which makes it difficult to make climate estimates and biome reconstruction for a certain interval.
How much sample was used for treatment? How many pollen grains were counted?

3. Pollen data presentation. A diagram of major pollen taxa should be added in the main text rather than in the supplementary data, as this is the fundamental data.

4. Manuscript structure. The manuscript could be more well-organized. As for now, it is not easy to clearly follow.

Author's Response to Decision Letter for (RSOS-192067.R0)

See Appendix F.

Decision letter (RSOS-192067.R1)

04-May-2020

Dear Dr Bouchal,

It is a pleasure to accept your manuscript entitled "Messinian vegetation and climate of the intermontane Florina-Ptolemais-Servia Basin, NW Greece: How well do plant fossils reflect past environments?" in its current form for publication in Royal Society Open Science. The comments of the reviewer(s) who reviewed your manuscript are included at the foot of this letter.

Kind regards,
Andrew Dunn

Royal Society Open Science Editorial Office
Royal Society Open Science
openscience@royalsociety.org

on behalf of Dr Emily Lindsey (Associate Editor)
openscience@royalsociety.org

Appendix A**ROYAL SOCIETY
OPEN SCIENCE****Messinian vegetation and climate of the intermontane
Florina-Ptolemais-Servia Basin, NW Greece: How well do
plant fossils reflect past environments?**

Journal:	Royal Society Open Science
Manuscript ID	RSOS-192067
Article Type:	Research
Date Submitted by the Author:	28-Nov-2019
Complete List of Authors:	Bouchal, Johannes; Swedish Museum of Natural History, Palaeobiology Güner, Tuncay H.; Istanbul University Cerrahpaşa, Faculty of Forestry, Department of Forest Botany Velitzelos, Dimitrios; National and Kapodistrian University of Athens Faculty of Geology and Geoenvironment, Section of Historical Geology and Palaeontology Velitzelos, Evangelos; National and Kapodistrian University of Athens Faculty of Geology and Geoenvironment, Section of Historical Geology and Palaeontology Denk, Thomas; Swedish Museum of Natural History, Dep. of Palaeobotany
Subject:	Palaeontology < EARTH SCIENCES, ecology < BIOLOGY, plant science < BIOLOGY
Keywords:	Biome reconstruction, proxy biases, climate reconstruction, plant macrofossils, dispersed pollen, light and scanning electron microscopy
Subject Category:	Earth science

Author-supplied statements

Relevant information will appear here if provided.

Ethics

Does your article include research that required ethical approval or permits?:

This article does not present research with ethical considerations

Statement (if applicable):

CUST_IF_YES_ETHICS :No data available.

Data

It is a condition of publication that data, code and materials supporting your paper are made publicly available. Does your paper present new data?:

Yes

Statement (if applicable):

This article and all data used in this article are made available in bioRxiv.

<https://www.biorxiv.org/content/10.1101/848747v1>

Supplementary Material are available within the figshare repository:

<https://doi.org/10.6084/m9.figshare.10327646.v1>

Conflict of interest

I/We declare we have no competing interests

Statement (if applicable):

CUST_STATE_CONFLICT :No data available.

Authors' contributions

This paper has multiple authors and our individual contributions were as below

Statement (if applicable):

[revised manuscript text omitted]

L = leaves; P = palynomorph; R = reproductive structures

1) Velitzelos & Schneider, 1979; 2) Velitzelos & Petrescu, 1981; 3) Velitzelos et al., 1983; 4) Velitzelos & Gregor, 1985; 5) Mai & Velitzelos, 1992; 6) Mai & Velitzelos, 1997;

7) Velitzelos & Kvaček, 1999; 8) Kvaček et al., 2002; 9) Denk & Velitzelos, 2002; 10) Velitzelos & Denk, 2002; 11) this study

Climate parameter	CLAMP Physg3arcAZ	CLAMP PhysgAsia1	CA modified	CA modified 10-90%iles
MAT (°C)	10–13.5	8.7–11.5	8.6–21.2	9.9–18.4
CMMT (°C)	1–5	-2.7–2.3	≥ 1.2	-
WMMT (°C)	19.2–22.8	19–22.6	-	-
GROWSEAS (months)	6–8	5.5–7	-	-
MMGSP (mm)	110–160	100–160	-	-
Three_WET (mm)	500–780	400–750	-	-
Three_DRY (mm)	180–260	80–220	-	-
3 WET/3 DRY	< 4	< 5.5	-	-

MAT = mean annual temperature, CMMT = coldest month mean temperature, WMMT = warmest month mean temperature, GROWSEAS = duration of growing season, MMGSP = mean month growing season precipitation, Three_WET = precipitation of three consecutive wettest months, Three_DRY = precipitation of three consecutive driest months.

Table 3. Vegetation types recognised for the pre-evaporitic Messinian of the Florina–Ptolemais–Servia Basin.

Vegetation type	Main (and accessory) taxon/taxa	Biome ^a	Vegetation unit(s) ^b	Modern (Neogene) analogue	References
Swamp forest	Taxodium , Glyptostrobus	NLD	VU3	Taxodium swamp forests SE USA; (Taxodium / Glyptostrobus swamp forests widespread in N Hemisphere Neogene)	1, 2
Swamp forest	Alnus , (Sassafras)	BLD	VU3	Alnus swamp forest	3, 4
Riparian forest	Pterocarya , Zelkova , Ulmus , (Sassafras)	BLD	VU4	Riparian and alluvial forest of Georgia and Iran	1, 3, 4, 5, 6
Well-drained forest	Quercus kubinyi , Q. pseudocastanea , (Carpinus , Tilia etc.)	BLD	VU5b	Lowland oak-hornbeam forests; ("Quercetum mixtum")	4, 7
Well-drained forest	Fagus , (Quercus pseudocastanea)	BLD	VU5b	Lowland beech forests of N Turkey, Georgia, N Iran; ("Fagetum gussonii")	3, 4, 7
Well-drained forest	Fagus , Abies , Cedrus , Cathaya	MIXED	VU6b	Montane Fagus-Abies forest, montane Fagus-Cedrus-Pinus forest; Abant Gölü; Erbaa-Çatalan	8, 9, 10
Well-drained laurophyllous forest	Quercus drymeja , (Q. sosnowsky)	BLE	VU6a	Quercus dilatata association (with Taxus , Pinus , Acer etc.)	11
Well-drained sclerophyllous forests/shrublands	Quercus mediterranea , Chamaerops , Olea	BLE/ SHRUBLAND	VU0	Mediterranean sclerophyllous forest/shrublands	10
[?] ^c Grassland-steppe forest	Poaceae	GRASSLAND/ SHRUBLAND	VU0	Forest-steppe of SE Europa to Afghanistan	8, 10, 11, 12

1) Mai, 1995; 2) Dolezych & Schneider, 2007; 3) Denk et al., 2001; 4) Akhiani et al., 2010; 5) Maharramova, 2015; 6) Kozłowski et al., 2018; 7) Kvaček et al., 2002; 8) Mayer & Aksoy, 1986; 9) Akkemik, 2003; 10) van Zeist & Bottema, 1991; 11) Freitag, 1971; 12) Erdős et al., 2018

^aBiome classification follows the phsiognomic approach of Woodward et al., 2004. ^bVegetation units as in Table 1.

^c[?] expresses the uncertainty around a possible extra-regional signal in the Vegora pollen record. According to Erdős et al. (2018) steppe forest with *Stipa* and other grasses and different species of *Quercus* (forest-steppes of the type 'Region A - SE Europe') is characterized by MAP of 420-600 mm; this would be much drier than the inferred MAP for the FPS.

Figure 1. Fossil localities and lithological map of the Florina–Ptolemais–Servia Basin. Map redrawn after Steenbrink et al. (1999, 2006), Ognjanova-Rumenova (2005), Ivanov (2001) and Koufos (2006). Fossil localities: (1) Bitola Basin, Republic of North Macedonia, PF. (2) Vegora Basin, MF and PF (3) Dytiko, VF. (4) Prosilio, MF. (5) Lava, MF. (6) Likoudi, MF. (7) Serres Basin. (2–7) Greece. (8) Sandanski Graben, Bulgaria, PF. Abbreviations: Plant macrofossils (MF), palynoflora (PF), vertebrate fossils (VF).

71x52mm (300 x 300 DPI)

Figure 2. Lithology and polarity zones of the Vegora section (redrawn after Steenbrink et al., 2006). Position of fossil bearing strata following Velitzelos and Schneider (1979) and Kvaček et al. (2002).

119x284mm (300 x 300 DPI)

Figure 3. Light microscopy (LM) and scanning electron microscopy (SEM) micrographs of algae, fern and fern allies, and gymnosperm palynomorphs.

[revised manuscript text omitted]

¹Swedish Museum of Natural History, Department of Palaeobiology, Box 50007, 10405
Stockholm, Sweden

²Faculty of Forestry, Department of Forest Botany, Istanbul University Cerrahpaşa, Istanbul,
Turkey

³National and Kapodistrian University of Athens, Faculty of Geology and Geoenvironment,
Section of Historical Geology and Palaeontology, Greece

The late Miocene is marked by pronounced environmental changes and the appearance of
strong temperature and precipitation seasonality. Although environmental heterogeneity is to
be expected during this time, it is challenging to reconstruct palaeoenvironments using plant
fossils. We investigated leaves and dispersed spores/pollen from 6.4–6 Ma strata in the
intermontane Florina-Ptolemais-Servia Basin (FPS) of NW Greece. To assess how well plant
fossils reflect the actual vegetation of the FPS, we assigned fossil-taxa to biomes providing a
measure for environmental heterogeneity. Additionally, the palynological assemblage was
compared to pollen spectra from modern lake sediments to assess biases in spore/pollen
representation in the pollen record. We found a close match of the Vegora assemblage with
modern *Fagus–Abies* forests of Turkey. Using taxonomic affinities of leaf fossils, we further
established close similarities of the Vegora assemblage with modern laurophyllous oak forests
of Afghanistan. Finally, using information from sedimentary environment and taphonomy, we
distinguished local and distantly growing vegetation types. We then subjected the plant
assemblage of Vegora to different methods of climate reconstruction and discussed their
potentials and limitations. Leaf and spore/pollen records allow accurate reconstructions of
palaeoenvironments in the FPS, whereas extra-regional vegetation from coastal lowlands is
likely not captured.

25 35 **Keywords**

Biome reconstruction, proxy biases, climate reconstruction, plant macrofossils, dispersed
pollen, light and scanning electron microscopy

29 39 **1. Introduction**

The late Miocene (11.6–5.3 Ma) marks the time in the Neogene (23–2.58 Ma) with the largest
shift from equable climate to strong latitudinal temperature gradients in both hemispheres
(Herbert et al., 2016). This is well illustrated by the global rise of C₄ dominated ecosystems
(grasslands and savannahs in the tropics and subtropics; Cerling et al., 1997). In the
Mediterranean region, vegetation changes did not happen synchronously with modern steppe
and Mediterranean sclerophyllous woodlands replacing humid temperate forest vegetation at
different times and places during the middle and late Miocene (Suc, 1984; Kovar-Eder, 2003;
Fauquette et al., 2006; Kovar-Eder et al., 2014; Bouchal et al., 2018; Denk et al., 2018; Suc et
al., 2018). During the latest Miocene (5.9–5.3 Ma) the desiccation of the Mediterranean Sea
was caused by the isolation of the Mediterranean Sea from the Atlantic Ocean. Based on
palynological studies across the Mediterranean region, this event did not have a strong effect
on the existing vegetation. Open and dry environments existed in southern parts before,
during, and after this so-called Messinian Salinity Crisis (MSC). In contrast, forested
vegetation occurred in northern parts of Spain, Italy, and the Black Sea (Fauquette et al.,
2006). Likewise, a vegetation gradient occurred from N and C Italy and Greece to Turkey,
where humid temperate forests had disappeared by the early late Miocene (Denk et al., 2018).
The Florina-Ptolemais-Servia Basin (FPS) of NW Greece is one of the best-understood
intermontane basins of late Miocene age in the entire Mediterranean region. A great number
of studies investigated the tectonic evolution, depositional history, and temporal constraints of
basin fills (e.g. Steenbrink et al. 1999, 2000, 2006), plant fossils (e.g. Knobloch & Velitzelos,
1986a,b; Velitzelos & Gregor, 1985; Mai & Velitzelos, 1992; Kvaček et al., 2002; Velitzelos
et al., 2014), and vertebrate fossils (van de Weerd, 1979; Koufos 1982, 2006; Koufos et al.,
1991).

The Messinian flora of Vegora in the northern part of the FPS is dated at 6.4–6 Ma and
represents the vegetation in this region just before the onset of the Messinian salinity crisis
(the pre-evaporitic Messinian). This flora has been investigated since 1969 (Kvaček et al.,
2002) and represents one of the richest late Miocene leaf floras in the eastern Mediterranean
along with two other, slightly older, Messinian plant assemblages from the FPS,
Likoudi/Drimos and Prosilio/Lava (4–6 in Fig. 1; Velitzelos et al., 2014). The focus of
previous palaeobotanical studies in the FPS has been on macrofossils. In contrast, no
comprehensive study of dispersed pollen and spores has been carried out in the FPS. While
fruit and seed floras, to a great extent, and leaf floras, to a lesser extent, reflect local
vegetation in an area, dispersed pollen and spores provide additional information about the
regional vegetation. Therefore, a main focus of the present study is on spores and pollen of
the Messinian plant assemblage of Vegora.
We (i) investigated dispersed spores and pollen using a combined light and scanning electron
microscopy approach (Daghlian, 1982; Zetter, 1989; Halbritter et al., 2018) that allows a
more accurate determination of pollen and hence higher taxonomic resolution. We (ii) then
compiled a complete list of plant taxa recorded for the site of Vegora including fruits and
seeds, foliage, and spores and pollen. Based on the ecological properties of their modern
analogue taxa, we assigned the fossil-taxa to functional types (vegetation units) and inferred
palaeoenvironments of the FPS during the Messinian. Using leaf physiognomic
characteristics, we (iii) conducted a Climate Leaf Analysis Multivariate Program (CLAMP)
analysis (Spicer, 2008; Yang et al., 2011) to infer several climate parameters for the late
Miocene of the FPS. We also (iv) used a modified “Co-existence approach” (Mosbrugger &
Utescher, 1997) based on climatic requirements of modern analogue plant taxa to infer two
climate parameters and (v) a Köppen signature analysis (Denk et al., 2013; Bouchal et al.,
2018) based on the Köppen-Geiger climate types in which modern analogue taxa of the fossil-
taxa occur. Finally, we (vi) discuss how well the translation of fossil plant assemblages into
functional types (vegetation units, biomes) works for reconstructing past environments at
local and regional scales.

**2. Material and Methods**

*2.1. Geological setting*

The old open-pit lignite quarry of Vegora is located in western Macedonia, NW Greece, ca. 2
95 km E of the town of Amyntaio and is part of the Neogene Florina-Ptolemais-Servia
intermontane basin (FPS). The FPS is part of the Pelagonian basin that extends to the north
into North Macedonia (Fig. 1). The NNW–SSE trending FPS is ca. 120 km long and presently
at elevations between 400 and 700 m a.s.l. and is flanked by mountain ranges to the east and
the west. Main ranges include Baba Planina (2,601 m), Verno (2,128 m), and Askio (2,111 m)
to the east of the basin and Voras (2,528 m), Vermio (2,065 m), Olympus (2,917 m) to the
west (Fig. 1). These ranges are mainly comprised of Mesozoic limestones, Upper

[revised manuscript text omitted]
 (Koufos, 2006). The hypsodonty index of this fauna of 1.45–1.86 (NOW database, <http://pantodon.science.helsinki.fi/nw/locality.php?p=ecometrics>) corresponds to the diet types “mixed closed habitats”, “regular browsers”, and “selective browsers” according to Janis (1988) and hence provides an excellent match with the environments inferred for the FPS.

From Lava (Fig. 1), Steenbrink et al. (2000) investigated two sequences covering two
sedimentary cycles each. Based on palaeomagnetic correlation these sequences are dated as c.
6.8–6.7 Ma and c. 6.3 Ma. The pollen assemblages are comparable to the Vegora assemblage
but differ in some respects. First, the Lava sections have a continuous high amount of *Pinus*
pollen (20 to >60%) suggesting that the fossil site was located closer to pine forests than was
the Vegora lake. Second, *Cedrus* pollen is abundant with values between 10 and >30%. Third,
Steenbrink et al. (2000) did not report evergreen oak pollen, although evergreen *Quercus* is
known from Lava based on leaf fossils (Velitzelos et al., 2014). Steenbrink et al. (2000)
inferred a humid temperate climate without dry season for the investigated sedimentary
cycles. In addition, they suggested that expansions of *Fagus* accompanied by a decrease of
*Abies* might reflect subtle increases in montane humidity. Overall, they suggested
continuously wet and warm-temperate climate conditions for the investigated period for Lava.
Velitzelos et al. (2014) provided revised taxon lists for the roughly coeval macrofossil (leaves
and fruits/seeds) localities Prosilio and Lava (age based on palaeomagnetic correlation, 6.7–
6.4 Ma; Steenbrink et al., 2006). The macroflora is very similar to the one from Vegora in
terms of composition. However, whereas *Quercus sosnowskyi* is among the most abundant
elements in Vegora, only a few leaves represent this species in Prosilio; also *Glyptostrobus* is
much less abundant. *Pinus* is represented by cones, leaf fascicles, and leafy branches; this is
in accordance with the high amount of pine pollen documented in the palynological record.
*Fagus* is a frequent element as well, while *Abies* is not recorded in the macroflora.
Likoudi, 20 km S of Lava, is located in a small basin south of the main FPS (Fig. 1). The
macroflora (leaves, fruits and seeds) is very rich (see revised and updated floral list in
Velitzelos et al., 2014). The precise age of the Messinian diatomaceous marls is not clear
(Knobloch & Velitzelos, 1986) although it unambiguously is pre-evaporitic. The flora is
characterised by the high diversity of conifers (11 genera of Cupressaceae and Pinaceae,
including *Torreya* – as *Egeria* sp. in Velitzelos et al., 2014). As in Vegora, *Fagus* is a
dominating element. Other taxa (*Cercis*, *Laria*, cf. *Nerium*) are not known from other FBS
floras. Well-preserved cones of *Cedrus* and cones and leafy twigs of *Cathaya* and *Taiwania*
suggest that these genera were not growing at high elevations but nearby the area of
deposition (lake). If coeval with the Lava deposits, this would explain the relatively high
amounts of *Cedrus* pollen in the palynological section of Lava.
Ivanov & Slavomirova (2002) investigated a 70 m succession of lacustrine sediments in the
Bitola Basin (Northern Macedonia; borehole V-466; 1 in Fig. 1) about 10 km E of Bitola and
40 km NNW of Vegora. Based on a vertebrate fauna on top of these sediments the plant-
bearing sediments are assigned a late Miocene age (Dumurdžanov et al., 2002; Ognjanova-
Rumeno, 2005). From these sediments, abundant leaves of *Quercus sosnowskyi* have been

[revised manuscript text omitted]

Erdős L, Ambarli D, Anenkhonov OA, Bátori Z, Cserhalmi D, Kiss M, Kröel-Dulay G, Liu H,
Magnes M, Molnár Z, et al. 2018 The edge of two worlds: A new synthesis on Eurasian
forest-steppes. *Appl. Vegetation Sci.* 21, 345–362.
- Favre E, Escarguel G, Suc J-P, Vidal G, Thévenod L. 2008 A contribution to deciphering the
meaning of AP/NAP with respect to vegetation cover. *Rev. Palaeobot. Palynol.* 148,
13–35.
- Fang J, Wang Z, Tang Z. 2009 Atlas of Woody Plants in China. Volumes 1 to 3 and index.
Beijing: Higher Education Press.
- Fauquette S, Suc J-P, Bertini A, Popescu S-M, Warny S, Bachiri Taoufiq N, Perez Villa, M-J,
Chikhi H, Subally D, Feddi N, Clauzon G, Ferrier J. 2006 How much did climate force
the Messinian salinity crisis? Quantified climatic conditions from pollen records in the
Mediterranean region. *Palaeogeogr., Palaeoclimatol., Palaeoecol.* 238, 281–301.
- Ferguson DK, Hofmann C-C, Denk T. 1999 Taphonomy: field techniques in modern
environments. In: Jones TP, Rowe NP (eds) *Fossil Plants and Spores: modern*
*techniques*. London: Geological Society, pp 210–213.
- Freitag, H. 1971 Die natürliche Vegetation Afghanistans. *Vegetatio* 22, 285–344.
- Gersonde R, Velitzelos E. 1978 Diatomeenpaläoökologie im Neogen-becken von Vegora N-
W Mazedonien (vorläufige Mitteilung). *Ann. Géol. Pays Hellèn.* 30, 373–382.
- Grimm GW, Bouchal JM, Denk T, Potts A. 2016 Fables and foibles: A critical analysis of the
Palaeoflora database and the Coexistence Approach for palaeoclimate reconstruction.
*Rev. Palaeobot. Palyn.* 233, 611–622.
- Grimm GW, Potts A. 2016 Fallacies and fantasies: the theoretical underpinnings of the
Coexistence Approach for palaeoclimate reconstruction. *Clim. Past* 12, 611–622.
- Halbritter H, Ulrich S, Grimsson F, Weber M, Zetter R, Hesse M, Buchner R, Svojtka M,
Frosch-Radivo A. 2018 *Illustrated Pollen Terminology*, 2nd ed. Cham (Switzerland):
Springer Nature.
- Herbert TD, Lawrence KT, Tzanova A, Peterson LC, Caballero-Gill R, Kelly CS. 2016 Late
Miocene global cooling and the rise of modern ecosystems. *Nature Geosci.* 9, 843–847.
- Ivanov DA. 2001 Palaeoecological interpretation of a pollen diagram from the Sandanski
graben (Southwest Bulgaria). *Comptes Rend. Acad. Bulg. Sci.* 54, 65–68.
- Ivanov DA, Slavomirova E. 2002 Preliminary palynological data on Neogene flora from
Bitola Basin (F.Y.R.O.M.). *Comptes Rend. Acad. Bulg. Sci.* 55, 81–86.
- Janis CM. 1986 An estimation of tooth volume and hypsodonty indices in ungulate mammals,
and the correlation of these factors with dietary preference. In: DE Russell, JP Santoro,
D Sigogneau-Russell (eds) *Teeth revisited*. Proc. 7th Int. Symp. Dental Morph. Mém.
*Mus. Nat. Hist. naturelle, Sér. C* 53, 367–387.
- Jiang X-L, Hipp AL, Deng M, Su T, Zhou Z-K, Yan M-X. 2019 East Asian origins of
European holly oaks (*Quercus* section *Ilex* Loudon) via the Tibet-Himalaya. *J.*
*Biogeogr.* 46, 2188–2202. doi.org/10.1111/jbi.13654
- Karistinos N, Ioakim C. 1989 Palaeoenvironmental and palaeoclimatic evolution of the
Serres Basin (N. Greece) during the Miocene. *Palaeogeogr., Palaeoclimatol.,*
*Palaeoecol.* 70, 275–285.
- Knobloch E, Velitzelos E. 1986a Die obermiozäne Flora von Likudi bei Ellassona (Thessalien,
Griechenland). *Doc. Nat.* 29, 5–20.
- Knobloch E, Velitzelos E. 1986b Die obermiozäne Flora von Prosilion bei Kozani (Süd-
Mazedonien, Griechenland). *Doc. Nat.* 29, 29–33.

Kottek M, Grieser J, Beck C, Rudolf B, Rubel F. 2006 World map of the Köppen-Geiger
climate classification updated. *Meteorol. Z.* 15, 259–263.
Koufos GD. 1982 *Hipparion crassum* Gervais, 1859 from the lignites of Ptolemais
(Macedonia-Greece). *Proc. Kon. Ned. Akad. Wetten. B.* 85, 229–239.
Koufos GD. 2006 The Neogene mammal localities of Greece: Faunas, chronology and
biostratigraphy. *Hellen. J. Geosci.* 41, 183–214
Koufos GD, Kostopoulos DS, Koliadimou KK. 1991 Un nouveau gisement de mammifères
dans le Villafranchien de Macédoine occidentale (Grèce). *C. R. Acad. Sci. Paris, ser. II*
313, 831–836.
Kovar-Eder J, Jechorek H, Kvaček Z, Parashiv V. 2008 The Integrated Plant Record: An
essential tool for reconstructing Neogene zonal vegetation in Europe. *Palaios* 23, 97–
111.
Kovar-Eder J, Kvaček Z, Martinetto E, Roiron P. 2006 Late Miocene to Early Pliocene
vegetation of southern Europe (7–4 Ma) as reflected in the megafossil plant record.
*Palaeogeogr., Palaeoclimatol., Palaeoecol.* 238, 321–339.
Kozłowski G, Bétrisey S, Song Y. 2018 Wingnuts (*Pterocarya*) and walnut family. Relict
trees: linking the past, present and future. *Natural History Museum Fribourg:*
*Switzerland.*
Kvaček Z, Velitzelos D, Velitzelos E. 2002 Late Miocene Flora of Vegora Macedonia N.
Greece. Athens: Koralis.
Maharramova E. 2015 Genetic diversity and population structure of the relict forest trees
*Zelkova carpinifolia* (Ulmaceae) and *Pterocarya fraxinifolia* (Juglandaceae) in the
South Caucasus. PhD Dissertation. Freie Universität Berlin.
Mai HD. 1995 Tertiäre Vegetationsgeschichte Europas. Stuttgart: Gustav Fischer.
Mai DH, Velitzelos E. 1992 Über fossile Pinaceen-Reste im Jungtertiär von Griechenland.
*Feddes Repert.* 103, 1–18.
Mai DH, Velitzelos E. 1997 Paläokarpologische Beiträge zur jungtertiären Flora von Vegora
(Nordgriechenland). *Feddes Repert.* 108, 507–526.
Marinova E, Harrison SP, Bragg F, Connor S, de Laet V, Leroy SAG, Mudie P, Atanassova J,
Bozilova E, Caner H, Cordova C, Djamali M, Filipova-Marinova M, Gerasimenko N,
Jahns S, Kouli K, Kotthoff U, Kvavadze E, Lazarova M, Novenko E, Ramezani E,
Röpke A, Shumilovskikh L, Tanțău I, Tonkov S. 2018 Pollen-derived biomes in the
Eastern Mediterranean–Black Sea–Caspian–Corridor. *J. Biogeogr.* 45, 484–499.
Mayer H, Aksoy, H. 1986 Wälder der Türkei. Stuttgart: Gustav Fischer.
Mosbrugger V, Utescher T. 1997 The coexistence approach — a method for quantitative
reconstructions of Tertiary terrestrial palaeoclimate data using plant fossils.
*Palaeogeogr., Palaeoclimatol., Palaeoecol.* 134, 61–86
Ognjanova-Rumenova N. 2005 Upper Neogene siliceous microfossils from Pelagonia Basin
(Balkan Peninsula). *Geol. Carpathica* 56 (4), 347–358.
Pavlides SB, Mountrakis DM. 1986 Neotectonics of the Florina–Vegorit–Ptolemais
Neogene Basin (NW Greece): an example of extensional tectonics of the greater Aegean
area. *Ann. Géol. Pays Hellen.* 33, 311–327.
Peel MC, Finlayson BL, McMahon TA. 2007 Updated world map of the Köppen-Geiger
climate classification. *Hydrol. Earth Syst. Sci.*, 11, 1633–1644.
Psilovikos A, Karistinos N. 1986 A depositional sedimentary model for the Neogene
uraniferous lignites of the Serres graben, Greece. *Palaeogeogr., Palaeoclimatol.,*
*Palaeoecol.* 56, 1–16.

Punt W, Hoen PP, Blackmore S, Nilsson RH, Le Thomas A. 2007 Glossary of pollen and
spore terminology. *Rev. Palaeobot. Palynol.* 143, 1–81.
- Ratnam J, Bond WJ, Fensham RJ, Hoffmann WA, Archibald S, Lehmann CER, Anderson
MT, Higgins SI, Sankaran M. 2011 When is a ‘forest’ a savanna, and why does it
matter? *Glob. Ecol. Biogeogr.* 20, 653–660.
- Roberts S, Rassios A, Wright L, Vacondios I, Vrachatis G, Grivas E, Nesbitt RW, Neary CR,
Moat T, Kostantopoulou L. 1988 Structural controls on the location and form of the
Vourinos chromite deposits. In: Boissonnas J, Omenetto P (eds) *Mineral deposits within*
*the European Community*. Springer, Berlin Heidelberg New York, pp 249–266.
- Rubel F, Brugger K, Haslinger K, Auer I. 2017 The climate of the European Alps: Shift of
very high resolution Köppen-Geiger climate zones 1800–2100. *Meteorol. Z.*, 26, 115–
125. <https://doi.org/10.1127/metz/2016/0816>.
- Schneider W. 1992 Floral successions in the Miocene swamps and bogs of Central. Europe.
*Z. Geol. Wiss.* 20, 555–570.
- Spicer RA. 2008 CLAMP. In: Gornitz V. Ed. *Encyclopedia of Paleoclimatology and Ancient*
*Environments*. Dordrecht: Springer.
- Steenbrink J, Van Vugt N, Hilgen FJ, Wijbrans JR, Meulenkamp JE. 1999 Sedimentary
cycles and volcanic ash beds in the lower Pliocene lacustrine succession of Ptolemais
(NW Greece): discrepancy between $^{40}\text{Ar}/^{39}\text{Ar}$ and astronomical ages. *Palaeogeogr.*,
*Palaeoclimatol.*, *Palaeoecol.* 152, 283–303.
- Steenbrink J, Van Vugt N, Kloosterboer-van Hoeve ML, Hilgen FJ. 2000 Refinement of the
Messinian APTS from sedimentary cycle patterns in the lacustrine Lava section (Serbia
Basin, NW Greece). *Earth Planet. Sci. Lett.* 181, 161–173.
- Steenbrink J, Hilgen FJ, Krijgsman W, Wijbrans JR, Meulenkamp JE. 2006 Late Miocene to
Early Pliocene depositional history of the intramontane Florina-Ptolemais-Serbia Basin,
NW Greece: Interplay between orbital forcing and tectonics. *Palaeogeogr.*,
*Palaeoclimatol.*, *Palaeoecol.* 238, 151–178.
- Suc J-P. 1984 Origin and evolution of the Mediterranean vegetation and climate in Europe.
*Nature* 307, 429–432.
- Suc J-P, Popescu S-M, Do Couto D, Clauzon G, Rubino J-L, Melinte-Dobrinescu MC,
Quillévéré F, Brun J-P, Dumurdžanov N, Zagorchev I, Lesdić V, Tomić D, Sokoutis D,
Meyer B, Macaleț R, Jelen B, Rihelj H. 2015 Marine gateway vs. fluvial stream within
the Balkans from 6 to 5 Ma. *Mar. Pet. Geol.* 66, 231–245.
- Suc J-P, Popescu SM, Fauquette S, Bessedik M, Jiménez-Moreno G, Taoufiq B, Zheng Z,
Medail F, Klotz S. 2018 Reconstruction of Mediterranean flora, vegetation and climate
for the last 23 million years based on an extensive pollen dataset. *Ecol. mediterr.* 44,
53–85.
- Traverse A. 2007 *Paleopalynology*. Topics in Geobiology 28. Dordrecht, Springer.
- Utescher T, Bruch AA, Erdei B, François I, Ivanov D, Jacques FMB, Kern AK, Liu Y-SC,
Mosbrugger V, Spicer RA. 2014 The Coexistence Approach—Theoretical background
and practical considerations of using plant fossils for climate quantification.
*Palaeogeogr.*, *Palaeoclimatol.*, *Palaeoecol.* 410, 58–73.
- Van de Weerd A. 1979 Early Ruscinian rodents and lagomorphs (Mammalia) from the
lignites near Ptolemais (Macedonia, Greece). *Proc. Kon. Nederl. Akad. Wet.*, B. 82,
127–170.
- Van Zeist W, Bottema S. 1991 Late Quaternary vegetation of the Near East. *Beih. Tübinger*
*Atlas Vord. Orient, Reihe A (Naturwiss.)* 18, 1–156.

Van Zeist W, Woldring H, Stapert, D. 1975 Late Quaternary vegetation and climate of
southwestern Turkey. *Palaeohist.* 14, 35–143.
- Velitzelos D, Denk T. 2002 Leaf epidermal characteristics of late Tertiary conifers from
Greece: taxonomic significance and limitations. 6th Europ. Paleobot. Palynol. Conf.
Athens, Greece, Abstracts, 183–184.
- Velitzelos D, Bouchal JM, Denk T. 2014 Review of the Cenozoic floras of Greece. *Rev.*
*Palaeobot. Palynol.* 204, 1–15.
- Velitzelos E, Gregor H-J. 1985 Neue paläofloristische Befunde im Neogen Griechenlands.
*Doc. Nat.* 25, 1–4.
- Velitzelos E, Krach JE, Gregor H-J, Geissert F. 1983 *Bolboschoenus vegorae* – ein Vergleich
fossiler und rezenter Rhizomknollen der Strandbinse. *Doc. Nat.* 5, 1–57.
- Velitzelos E, Kvaček Z. 1999 Review of the late Miocene flora of Vegora western
Macedonia, Greece. *Acta Palaeobotanica, Suppl.* 2 (Proceed. 5th EPPC) 419–427.
- Velitzelos E, Petrescu I. 1981 Seltene pflanzliche Fossilien aus dem Braunkohlebecken von
Vegora. *Ann. géol. Pays hellén.* 30, 767–777.
- Velitzelos E, Schneider HE. 1979 Jungtertiäre Pflanzenfunde aus dem Becken von Vegora in
West-Mazedonien. 3. Mitteilung: Eine Fächerpalme (*Chamaerops humulis* L.). *Ann.*
*géol. Pays hellén.* 29, 796–799.
- Woodward FI, Lomas MR, Kelly CK. 2004 Global climate and the distribution of plant
biomes. *Phil. Trans. R. Soc. Lond. B.* 359, 1465–1476.
- Yang J, Spicer RA, Spicer TEV, Li C-S. 2011 'CLAMP Online': a new web-based
palaeoclimate tool and its application to the terrestrial Paleogene and Neogene of North
America. *Palaeobiodiv. Palaeoenviro.* 91, 163–183.
- Zetter R. 1989 Methodik und Bedeutung einer routinemäßigen kombinierten
lichtmikroskopischen und rasterelektronenmikroskopischen Untersuchung fossiler
Mikrofloren. *Cour. Forschungsinst. Senck.* 109, 41–50.

**Table and Figure Captions**

**Table 1.** Plant taxa recorded from unit 1 (lignite seam) and unit 2 (blue marls) of the Vegora
section.

**Table 2.** Estimated climate parameters for the pre-evaporitic Messinian of Vegora from two
CLAMP calibration datasets and from CA.

**Table 3.** Vegetation types recognised for the pre-evaporitic Messinian of the Florina–
Ptolemais–Servia Basin.

**Figure 1.** Fossil localities and lithological map of the Florina–Ptolemais–Servia Basin.
Map redrawn after Steenbrink et al. (1999, 2006), Ognjanova-Rumenova (2005), Ivanov
(2001) and Koufos (2006). Fossil localities: (1) Bitola Basin, Republic of North Macedonia,
PF. (2) Vegora Basin, MF and PF (3) Dytiko, VF. (4) Prosilio, MF. (5) Lava, MF. (6)
Likoudi, MF. (7) Serres Basin. (2–7) Greece. (8) Sandanski Graben, Bulgaria, PF.
Abbreviations: Plant macrofossils (MF), palynoflora (PF), vertebrate fossils (VF).

**Figure 2.** Lithology and polarity zones of the Vegora section (redrawn after Steenbrink et al.,
2006). Position of fossil bearing strata following Velitzelos and Schneider (1979) and Kvaček
et al. (2002).

**Figure 3.** Light microscopy (LM) and scanning electron microscopy (SEM) micrographs of
algae, fern and fern allies, and gymnosperm palynomorphs.

(a) *Botryococcus* sp. cf. *B. kurzii*. (b) *Spirogyra* sp. 1/ *Ovoidites elongatus*. (c) *Spirogyra* sp.
2/*Cycloovoidites cyclus*. (d–e) *Osmunda* sp., (d) EV, (e) PV. (f) *Cryptogramma* vel
*Cheilanthes* sp, PV. (g–h) *Pteris* sp., (g) PV, (h) DV. (i) Davalliaceae vel Polypodiaceae sp./
*Verrucatosporites alienus* (R.Potonié) P.W.Thomson et Pflug, 1953, EV. (j)
*Leavigatosporites haardti*, EV. (k–l) *Inaperturopollenites hiatus*. (m) *Abies* sp., EV. (n–o)
*Cathaya* sp., (n) PV, (o) SEM detail, nanoechinolate sculpturing of cappa (PRV). (p) *Cedrus*
sp., EV. (q) *Pinus* subgenus *Pinus* sp., EV. (r) *Pinus* subgenus *Strobilus* sp., EV. (s–t) *Tsuga*
sp. 1, (s) PV, (t) monosaccus and corpus detail, PRV. (u–v) *Tsuga* sp. 2, (u) PV, (v)
monosaccus and corpus detail, PRV.
Abbreviations: equatorial view (EV), polar view (PV), distal view (DV), proximal view
(PRV). Scale bars 10 µm (LM, h, t, v), 1 µm (o).

**Figure 4.** LM and SEM micrographs of Poales, Vitales, Rosales, Fagales, Malpighiales, and
Geraniales.

(a) *Typha* sp, tetrad, PV. (b–c) Poaceae gen. indet., EV, (c) exine detail, PRV. (d–e)
Monocotyledone indet., (d) PV, (e) PRV. (f–g) *Parthenocissus* sp., EV. (h) *Ulmus* vel *Zelkova*
sp., PV. (I) *Fagus* sp., EV. (j–k) *Quercus* sect. *Cerris* sp., EV, (k) SEM detail, mesocolpium
exine sculpturing. (l–m) *Quercus* sect. *Ilex* sp., EV, (m) SEM detail, mesocolpium exine
sculpturing. (n–o) *Quercus* sect. *Quercus* sp., PV, (o) SEM detail, apocolpium exine
sculpturing. (p–q) Castanoideae gen. indet. sp., EV, (q) SEM detail, mesocolpium exine
sculpturing. (r) *Carya* sp., PV. (s) *Platycarya* sp., PV. (t) Engehardioideae gen. indet., PV. (u)
*Alnus* sp., PV. (v) *Betula* sp., PV. (w) *Carpinus* sp., PV. (x) *Corylus* sp., PV. (y) *Salix* sp.,
EV. (z–aa) *Geranium* sp., (z) PV, (aa) clavae detail.
Abbreviations: equatorial view (EV), polar view (PV), proximal view (PRV). Scale bars 10
973 µm (LM, e, g), 1 µm (c, k, m, o, q, aa).

**Figure 5.** LM and SEM micrographs of Sapindales, Malvales, Caryophyllales, Cornales,
Asterales, Dipsacales, and Apiales.

(a–b) *Cotinus* sp, EV. (c–d) *Pistacia* sp., (c) PV, (d) exine SEM detail. (e–f) *Acer* sp. 1, (e)
PV, (f) mesocolpium SEM detail. (g–h) *Acer* sp. 2, (g) PV, (h) mesocolpium SEM detail. (i–j)
*Craigia* sp., (i) PV, (j) apocolpium SEM detail. (k) Amaranthaceae gen. indet. sp. 1. (l)
Amaranthaceae gen. indet. sp. 2. (m–n) Caryophyllaceae gen. indet. sp. (o–p) *Nyssa* sp., (o)
PV, (p) exine sculpturing and aperture SEM detail. (q–r) *Fraxinus* sp., (q) EV, (r)
mesocolpium SEM detail. (s–t) *Olea* sp., EV. (u) Cichorioideae gen. indet. sp., PV. (v)
Asteroideae gen indet. sp. 1, PV. (w) Asteroideae gen indet. sp. 2, PV. (x–z) *Valeria* sp., (x–
y) PV, (z) aperture SEM detail. (aa–bb) Apiaceae gen. indet. sp. 1, EV. (cc–dd) Apiaceae gen.
indet. sp. 2, EV. (ee–ff) Angiosperm pollen fam. et gen. indet. sp., (ee) EV, (ff) mesocolpium
SEM detail.

Abbreviations: equatorial view (EV), polar view (PV). Scale bars 10 µm (LM, b, n, t, y, z, bb,
dd), 1 µm (d, f, h, j, p, r, ff).

**Figure 6.** Coexistence-Approach diagram showing coexistence intervals for MAT and
CMMT. MAT and CMMT climate ranges of relict taxa *a priori* excluded from the analysis
are shown on the left side of the diagram.

Blue bars, coldest month mean temperature; red bars, 10–90 percentile climatic range; dark
red extensions, full climatic range.

**Figure 7.** Köppen signal diagram for the macrofossil and pollen floras of Vegora.

To test and illustrate the stability of the climatic signal, gymnosperms (common alpine
elements) and azonal elements (e.g. riparian or swamp vegetation) were excluded in some
runs.

Supplementary Material

Supplementary Material Tables S1 - Climatic parameters of NLR

Table S1 (1). Fossil species and climatic parameters of the corresponding NLR (depending on the fossil-species and their botanical affinities, climate parameters of species, sections, subgenera, genera, or subfamilies are used as NLR).

Table S1 (2). Climatic parameters of NLR.

Supplementary Material Tables S2 - Köppen-Geiger climate type signatures.

Table S2 (1). Scored Köppen-Geiger signatures of all NLR species of the macrofossil and pollen flora of Vegora.

Tables S2 (2). Köppen-Geiger signature values and diagram of the macrofossil and pollen flora of Vegora.

Supplementary Material S3 - Systematic palaeobotany and descriptions of palynomorphs from the plant fossil bearing strata of Vegora (sample S115992).

Supplementary Material Tables S4 - Palynomorph abundance of sample S115992.

Supplementary Material S5 - Coding of leaf physiognomic characters for morphotypes from the Vegora lignite mine macroflora. Output PDF files from online CLAMP analysis

(<http://clamp.ibcas.ac.cn>).

Supplementary Material S6 - Köppen-Geiger categories

Appendix B

Supplementary Material Tables S4 - Palynomorph abundance of sample S115992

Sorted by abundance

Taxon	Count	%	Type
Pinus subgen. Strobus	47	11.0%	G
Alnus sp.	40	9.3%	Tw
Abies sp.	36	8.4%	G
Cathaya sp.	30	7.0%	G
Fagus sp.	30	7.0%	T
Pinus subgen. Pinus	33	7.7%	G
Quercus sp. (L)	21	4.9%	T
Papillate Cupressaceae	19	4.4%	G
Spirogyra	18	4.2%	A
Osmunda sp.	18	4.2%	F
Ulmus vel Zelkova sp.	18	4.2%	T
Cedrus	18	4.2%	G
Carya sp.	13	3.0%	T
Engelhardioideae gen. indet. sp.	10	2.3%	T
Quercus sp. (S)	9	2.1%	T
Betula sp.	8	1.9%	T
Salix sp.	8	1.9%	Tw
Leavigatosporites haardti	7	1.6%	F
Tsuga sp.	7	1.6%	G
Amaranthaceae gen. indet. sp.	3	0.7%	H
Apiaceae spp.	3	0.7%	H
Carpinus sp.	3	0.7%	T
Olea sp.	3	0.7%	T
Davalliaceae vel Polypodiaceae gen. sp.	2	0.5%	F
Asteroidaeae gen. indet. sp.	2	0.5%	H
Typha sp.	2	0.5%	Hw
Corylus sp.	2	0.5%	T
Acer sp.	2	0.5%	T
Nyssa sp.	2	0.5%	Tw
Incerta sedis	2	0.5%	U
Cryptogramma vel Cheilanthes sp.	1	0.2%	F
Pteris sp.	1	0.2%	F
Poaceae gen. indet.	1	0.2%	H
Geranium sp.	1	0.2%	H
Caryophyllaceae gen. indet. sp.	1	0.2%	H
Cichorioideae gen. indet. sp.	1	0.2%	H
Succisa sp.	1	0.2%	H
Parthenocissus sp.	1	0.2%	T
Castanoideae gen. indet. sp.	1	0.2%	T
Platycarya sp.	1	0.2%	T
Cotinus sp.	1	0.2%	T
Pistacia sp.	1	0.2%	T
Craigia sp.	1	0.2%	T
Fraxinus sp.	1	0.2%	Tw

Sum of counted grains and spores 430

Sorted by Taxonomy

Taxon	Count	%	Type
Spirogyra	18	4.2%	A
Osmunda sp.	18	4.2%	F
Leavigatosporites haardti	7	1.6%	F
Davalliaceae vel Polypodiaceae gen. sp.	2	0.5%	F
Cryptogramma vel Cheilanthes sp.	1	0.2%	F
Pteris sp.	1	0.2%	F
Pinus subgen. Strobus	47	11.0%	G
Abies	36	8.4%	G
Cathaya	30	7.0%	G
Pinus subgen. Pinus	33	7.7%	G
Papillate Cupressaceae	19	4.4%	G
Cedrus	18	4.2%	G
Tsuga	7	1.6%	G
Amaranthaceae gen. indet. sp.	3	0.7%	H
Apiaceae spp.	3	0.7%	H
Asteroidaeae gen. indet. sp.	2	0.5%	H
Poaceae gen. indet.	1	0.2%	H
Geranium sp.	1	0.2%	H
Caryophyllaceae gen. indet. sp.	1	0.2%	H
Cichorioideae gen. indet. sp.	1	0.2%	H
Succisa sp.	1	0.2%	H
Typha sp.	2	0.5%	Hw
Fagus sp.	30	7.0%	T
Quercus sp. (L)	21	4.9%	T
Ulmus vel Zelkova sp.	18	4.2%	T
Carya sp.	13	3.0%	T
Engelhardioideae gen. indet. sp.	10	2.3%	T
Quercus sp. (S)	9	2.1%	T
Betula sp.	8	1.9%	T
Carpinus sp.	3	0.7%	T
Olea sp.	3	0.7%	T
Corylus sp.	2	0.5%	T
Acer sp.	2	0.5%	T
Parthenocissus sp.	1	0.2%	T
Castanoideae gen. indet. sp.	1	0.2%	T
Platycarya sp.	1	0.2%	T
Cotinus sp.	1	0.2%	T
Pistacia sp.	1	0.2%	T
Craigia sp.	1	0.2%	T
Alnus sp.	40	9.3%	Tw
Salix sp.	8	1.9%	Tw
Nyssa sp.	2	0.5%	Tw
Fraxinus sp.	1	0.2%	Tw
Incerta sedis	2	0.5%	U

Algae (A)

Fern (F)	29	7.1%
Gymnosperm (G)	190	46.2%
Angiosperm tree (T)	125	30.4%
Angiosperm herb (H)	13	3.2%
Angiosperm tree wet (Tw)	51	12.4%
Angiosperm herb wet (Hw)	2	0.5%
Uncertain Angiosperm	2	0.5%
Sum	412	

Total arboreal vs non arboreal

Arboreal (all)	366	89.5%
Non arboreal (all)	44	10.8%

Total Fern, Gymnosperm, Angio tree and herbs

Fern	29	7.1%
Gymnosperm	190	46.5%
Angiosperm T	176	43.0%
Angiosperm H	15	3.7%

Only Angiosperms

Angiosperm T	125	65.4%
Angiosperm Tw	51	26.7%
Angiosperm H	13	6.8%
Angiosperm Hw	2	1.0%
Sum	191	

Some suggested changes (italics) are highlighted in yellow or written in red

THESE

en cotutelle

présentée

devant l'UNIVERSITE CLAUDE BERNARD – LYON 1

et l'UNIVERSITE TECHNIQUE D'ISTANBUL

pour l'obtention

du **DIPLOME DE DOCTORAT**

(arrêté du 7 Août 2006)

présentée et soutenue publiquement à Istanbul le 21 Décembre 2010

par

Demet BİLTEKİN

**Vegetation and climate of North Anatolian and North Aegean region
since 7 Ma according to pollen analysis**

**Directeurs de thèse: Jean-Pierre SUC
Namık ÇAĞATAY**

Jury :

**M. Namık ÇAĞATAY, Professeur, Université Technique d'Istanbul, Directeur de thèse
Naci GÖRÜR, Professeur, Université Technique d'Istanbul, Examineur
Speranta-Maria POPESCU, Chercheur, Institut de Physique du Globe Paris, Examineur
Jean-Pierre SUC, Directeur de Recherche émérite du CNRS, Directeur de thèse
Frédéric THEVENARD, Professeur, Université C. Bernard-Lyon1, Examineur
M. Namık YALÇIN, Professeur, Université d'Istanbul, Rapporteur**

ABSTRACT

Anatolia is an area inhabited today by relict thermophilous plants: *Liquidambar orientalis*, *Parrotia persica*, *Pterocarya fraxinifolia*, *Zelkova crenata* (Angiosperms) and *Cedrus* (Gymnosperm). These trees constitute forests relatively close to *Artemisia* steppes, being the two types of vegetation in competition during the climatic cycles along the last 2.6 million years. Thus, this makes the greatest interest for palynological investigations in the region. This study concerns a long marine section (DSDP Site 380 from the southwestern deep Black Sea: Late Miocene to Present) and onshore exposed sections (marine and lacustrine sediments) from the Late Miocene and/or Early Pliocene. The study area corresponds to the surroundings of the Marmara Sea (Enez, İntepe, Eceabat, Burhanlı, West Seddülbahir), southwestern Black Sea (DSDP Site 380), and northern Greece (Ptolemais Notio, Ptolemais Base, Prosilio, Trilophos and Lion of Amphipoli). The main target of this study is to reconstruct vegetation and climate during this time-interval in the region. The high-resolution pollen analysis of the 1,073.50 m long Black Sea Site 380 (to which I directly contributed for the interval 702.40 – 319.03 m) documents in great detail the evolution of vegetation and climate from the Late Miocene up to Present. Two vegetation types were alternately dominant for the last 7 million years: thermophilous forests and open vegetations including *Artemisia* steppes. At the early Messinian (before the Messinian Salinity Crisis), herbs prevailed in the Dardanelles area while mid- (*Tsuga*) and high-altitude (*Abies* and *Picea*) conifers were abundant with Cupressaceae close to the Olympe Mount (Prosilio). After the Messinian Salinity Crisis, North Aegean vegetation was mainly characterized by open plant ecosystems nearby forest assemblages with mesothermic trees (deciduous *Quercus*, *Carya*, *Zelkova*, etc.). In addition, strengthening of altitudinal conifers (*Cedrus*, *Tsuga*, *Abies* and *Picea*) may signify some uplift of the regional massifs. During the Late Miocene, most of the megathermic (tropical) and mega-mesothermic (subtropical) plants declined because of the climatic deterioration. However, some of them survived during the Late Pliocene, such as those which constituted coastal swamp forests (*Glyptostrobus*, *Engelhardia*, Sapotaceae, *Nyssa*) or composed deciduous mixed forests with mesothermic trees. Simultaneously, herbaceous assemblages (with Amaranthaceae-Chenopodiaceae, Poaceae, Asteraceae Asteroideae, Asteraceae Cichorioideae, etc.) became a prevalent vegetation component despite steppe elements (*Artemisia*, *Ephedra*, *Hippophae rhamnoides*) did not significantly develop. This suggests cooler and chiefly drier conditions during the Late Pliocene. At the Early Pleistocene (2.6 Ma), as a response to the onset of Arctic glaciations, mega-mesothermic elements rarefied despite some taxa persisted (Taxodiaceae: probably *Glyptostrobus*, *Engelhardia*, Sapotaceae, and *Nyssa*). In parallel, deciduous mixed forest assemblages composed of mesothermic trees (deciduous *Quercus*, *Betula*, *Alnus*, *Liquidambar*, *Fagus*, *Carpinus orientalis*, *Carpinus betulus*, *Tilia*, *Acer*, *Ulmus*, *Zelkova*, *Carya*, *Pterocarya*, etc.) almost disappeared too while steppe environments strongly enlarged. Then, *Artemisia* steppic phases developed during longer temporal intervals than mesophilous tree phases all along the glacial-interglacial cycles (first with a period of 41 kyrs, then 100 kyrs). This suggests shorter interglacials (warm and humid climate) than glacials (cool to cold and dry climate). From the beginning of the Ioanian Stage (1.8 Ma), herbaceous ecosystems (with Amaranthaceae-Chenopodiaceae, Poaceae, Asteraceae Asteroideae, Asteraceae Cichorioideae, etc.) and *Artemisia* steppes still continuously enlarged up today. Such an expansion of *Artemisia* steppes in the Ponto-Euxinian region was observed at the earliest Pliocene (DSDP Site 380) but their earliest settlement in Anatolia seems to have occurred in the Early Miocene (Aquitanian). The development of the *Artemisia* steppes in Anatolia might result from the uplift of the Tibetan Plateau. At last, relictuous plants such as *Carya*, *Carpinus orientalis*, *Pterocarya*, *Liquidambar orientalis*, *Zelkova* persisted up today for most of them. This story can be explained by some influence of the Asian monsoon which reinforced as a result from the uplifted Tibetan Plateau.

RESUME

L'Anatolie est un secteur aujourd'hui habité par des plantes thermophiles en situation de refuges. Sont concernées des Angiospermes (*Liquidambar orientalis*, *Parrotia persica*, *Pterocarya fraxinifolia*, *Zelkova crenata*) et une Gymnosperme (*Cedrus*), arbres de forêts contrastant avec la steppe à *Artemisia*, les deux types de végétation en forte compétition pendant les cycles climatiques des 2,6 derniers millions d'années. Ainsi, ceci confère-t-il le plus grand intérêt aux investigations palynologiques dans la région. Cette étude concerne un long enregistrement sédimentaire marin (Site profond DSDP 380 en Mer Noire sud-occidentale) et des affleurements à terre de dépôts marins ou lacustres du Miocène supérieur et(ou) du Pliocène inférieur. Le secteur étudié à terre porte sur les environs de la Mer de Marmara (Enez, Eceabat, Seddülbahir) et la Grèce septentrionale (Ptolemais Notio et Ptolemais Base, Prosilio, Trilophos et Lion d'Amphipoli). L'objectif principal de cette recherche est de reconstruire la végétation et le climat régionaux pendant ces intervalles de temps. L'enregistrement pollinique à haute résolution des 1.073,50 m du Site 380 (auquel j'ai contribué de façon significative à travers l'intervalle 702,4 – 319 m) documente en détail l'évolution de la végétation et du climat de la fin du Miocène à l'Actuel. Deux types de végétation y furent alternativement dominants au cours des 7 derniers millions d'années : les forêts de plantes thermophiles et les steppes à *Artemisia*. Au début du Messinien (avant la Crise de salinité messinienne), les herbes étaient dominantes dans la région des Dardanelles tandis que les conifères de moyenne (*Tsuga*) et haute altitude (*Abies* et *Picea*) abondaient avec les Cupressaceae près du Mont Olympe (Prosilio). Dans cette même région nord-égéenne, la végétation était après la Crise de salinité messinienne caractérisée principalement par des formations ouvertes à côté de groupements forestiers à arbres mésothermes (*Quercus* décidus, *Carya*, *Zelkova*, etc.). Par ailleurs, l'expansion des conifères altitudinaux (*Cedrus*, *Tsuga*, *Abies* et *Picea*) y est documentée et semble traduire un soulèvement des massifs environnants. A la fin du Miocène, la plupart des éléments mégathermes (tropicaux) et méga-mésothermes (subtropicaux) avaient régressé en raison des détériorations climatiques. Cependant, certains d'entre eux ont survécu pendant le Pliocène supérieur, notamment ceux qui constituaient des forêts littorales marécageuses (*Glyptostrobus*, *Engelhardia*, Sapotaceae, *Nyssa*) ou participaient à des forêts mixtes avec des arbres décidus mésothermes. Pendant ce temps, les formations ouvertes à herbes (Amaranthaceae-Chenopodiaceae, Poaceae, Asteraceae Asteroideae, Asteraceae Cichorioideae, etc.) sont devenues prédominantes dans la végétation sans que les éléments steppiques (*Artemisia*, *Ephedra*, *Hippophae rhamnoides*) soient très abondants. Ceci suggère un refroidissement au Pliocène supérieur et surtout l'installation de conditions plus sèches. Au début du Pléistocène (2,6 Ma), sous l'effet des premières glaciations arctiques, les éléments méga-mésothermes se sont très raréfiés malgré la persistance de quelques reliques (Taxodiaceae : probablement *Glyptostrobus*, *Engelhardia*, Sapotaceae, *Nyssa*). Simultanément, les forêts mixtes à éléments mésothermes (*Quercus* décidus, *Betula*, *Alnus*, *Liquidambar*, *Fagus*, *Carpinus orientalis*, *Carpinus betulus*, *Tilia*, *Acer*, *Ulmus*, *Zelkova*, *Carya*, *Pterocarya*, etc) ont aussi quasiment disparu tandis que les environnements steppiques se développaient fortement. Désormais, tout au long des cycles glaciaire-interglaciaire (d'abord de 41 ka de périodicité puis de 100 ka), les steppes à *Artemisia* occuperont plus d'espace temporel que les phases arborées. Ceci suggère des interglaciaires (chauds et humides) plus courts que les glaciaires (frais à froids et secs). Depuis le début de l'étage Ionien (1,8 Ma), les environnements à herbes (Amaranthaceae-Chenopodiaceae, Poaceae, Asteraceae Asteroideae, Asteraceae Cichorioideae, etc.) et les steppes à *Artemisia* n'ont cessé de s'étendre jusqu'à aujourd'hui. Cette expansion des steppes à *Artemisia* dans la région du Pont-Euxin a été observée au tout début du Pliocène (Site DSDP 380) mais leur premier enregistrement en Anatolie date de l'Aquitaniien (Miocène inférieur). Le développement de la steppe à *Artemisia* en Anatolie pourrait résulter du soulèvement du Plateau tibétain. Enfin le maintien dans cette région de plantes thermophiles reliques en situation de refuges (*Carya*, *Carpinus orientalis*, *Pterocarya*, *Liquidambar orientalis*, *Zelkova*), dont certaines jusqu'à nos jours, peut être expliqué par l'influence grandissante de la mousson asiatique dont le renforcement aurait aussi résulté du soulèvement du Plateau tibétain.

ACKNOWLEDGEMENTS

This PhD thesis study is the French-Turkish (Co-tutelle de thèse) enabled me chance for studying between the University of Claude Bernard-Lyon1 and the İstanbul Technical University. I would like to thank all people helped me during my PhD thesis study.

Firstly, I would like to thank my thesis directors: Prof. Dr. Jean-Pierre SUC (University of Claude Bernard-Lyon 1) and Prof. Dr. Namik ÇAĞATAY (İstanbul Technical University). They were of great help. Warm thanks to Prof. Dr. Jean-Pierre SUC who supported me in Palynology, thanks to him for his great experience on pollen grain taxonomy, on identifying pollen grains and for his endless patience during my thesis. He encouraged and helped me everytime during my stay in the University of Claude Bernard-Lyon 1. I also would like to thank to him for his efforts, advices and guidance during my thesis study. Warm thanks to Prof. Dr. Namik ÇAĞATAY who gave me opportunity to study Palynology and He introduced me to Prof. Dr. Jean-Pierre SUC. He always encouraged and helped me during my PhD thesis. I also would like to thank for his precious advices, his enthusiasm and his support during my thesis.

Many thanks to the members of my Committee of Pilotage in the University of Claude Bernard-Lyon1: Speranta-Maria POPESCU, Marc PHILIPPE, Serge LEGENDRE and Gilles ESCARGUEL for their advices and collaboration.

Many thanks to member of my thesis Committee in İstanbul Technical University: Prof. Dr. Naci GÖRÜR, Prof. Dr. Mehmet SAKINÇ, Prof. Dr. Ercan ÖZCAN and Prof. Dr. Namik YALÇIN of İstanbul University for their advices and contribution.

I would like to thank Dr. Speranta-Maria POPESCU for her helping and advising during my thesis and staying in the University of Claude Bernard-Lyon1.

I am very grateful towards the members of the Examination Board of my thesis who accepted to report on my manuscript (Prof. Dimiter IVANOV from Sofia, Prof. Paul ROIRON from Montpellier and Prof. Namik YALÇIN from Istanbul) and/or to discuss it at my oral defense (Prof. Frédéric THEVENARD, Dr. Speranta-Maria POPESCU, Prof. Naci GÖRÜR and Prof. Namik YALÇIN).

Thanks to the technical support in the Laboratory PEPS of the University C. Bernard – Lyon 1 and thanks to the French Embassy by the financial support obtained (thesis in cotutelle) in Lyon.

Thanks to TÜBİTAK and EMCOL (Eastern Mediterranean Oceanography Center) for financial support in İstanbul during my PhD thesis.

Many thanks to Lysiane THENEVOD, Mathieu DALİBARD, Anissa SAFRA, and Simona BOROI, I shared with them microcope work in the lobaratory and also thanks to Philippe SORREL, Eric FAVRE, Sébastien JOANNIN, Florent DALESME and Gwénael JOUANNIC for all their friendship.

Many thanks to EMCOL staff: Ümmühan SANCAR, Umut Barış ÜLGEN, Emre DAMCI, Sena AKÇER, Dursun ACAR, Zeynep ERDEM and Ayşe KAPLAN for all their friendship during my PhD thesis.

Finally, I thank my family, they supported me through my education life. Especially my mother, she always believed in me and encouraged me to begin my PhD studies and supported every step in my thesis study.

TABLE OF CONTENTS

1. INTRODUCTION	1
1.1 Aims of the study.....	8
2. PHYSIOGRAPHY, STRATIGRAPHY AND PALEO GEOGRAPHY OF THE STUDY AREAS	9
2.1 TURKEY.....	9
2.1.1 Present-day vegetation.....	9
2.1.2 Climate.....	14
2.1.2.1 Turkey.....	14
2.1.2.2 Greece and Macedonia.....	18
2.1.3 Stratigraphy of the study areas.....	20
2.1.4 Paleogeography.....	25
3. METHOD	32
3.1 Sampling and chemical processing.....	33
3.2 Identification of pollen grains.....	34
4. CHRONOLOGY OF THE STUDIED SECTIONS	42
5. RESULTS	46
5.1 DSDP Site 380.....	46
5.1.1 Lithology.....	46
5.1.2 High-resolution pollen record of DSDP Site 380.....	51
5.2 Gulf of Saros.....	59
5.2.1 Enez.....	59
5.3 Dardanelles Strait.....	62
5.3.1 İntepe.....	62
5.3.2 West of Seddülbahir.....	66
5.3.3 Eceabat.....	68
5.3.4 Burhanlı.....	69
5.4 Western Macedonia.....	71
5.4.1 Ptolemais Notio.....	71
5.4.2 Ptolemais Base.....	74
5.5 Northern Greece.....	77
5.5.1 Trilophos.....	77
5.5.2 Prosilio.....	78
5.5.3 Lion of Amphipoli.....	80
6. DISCUSSION	82
6.1 Flora and floristic refuges.....	82
6.2 Vegetation.....	90
6.2.1 The development of Artemisia steppes.....	96
6.3 Climate.....	100
6.3.1 Global climate context during the Miocene and Pliocene.....	100
6.3.2 Climatic evolution of the studied areas.....	103
7. CONCLUSIONS	108
REFERENCES	112

ABBREVIATIONS

MSC	: Messinian Salinity Crisis
NAF	: North Anatolian Fault
Ma	: Million years
cP	: Continental Polar Air Mass
mP	: Marine Polar Air Mass
cT	: Continental Tropical Air Mass
mT	: Marine Tropical Air Mass
PJF	: Polar Front Jet
STJ	: Subtropical Jet
ITCZ	: Intertropical Convergence Zone

1. INTRODUCTION

Present flora and vegetation of the North Aegean region and Anatolia show peculiar characteristics that find their origin in the past (Zohary, 1973; Quézel and Médail, 2003). In fact, this area counts today a lot of relictous plants (such as *Platanus orientalis*, *Liquidambar orientalis*, *Pterocarya fraxinifolia*, *Zelkova crenata*, *Cedrus libani*) inhabiting separated places within more or less thermophilous residual forests (Fig. 1a, 1b). In addition, vegetation shows high contrasts between forest (Pontus Euxinus forests, mid- to high-altitude forests) and open landscapes (mediterranean assemblages, pre-steppic to steppic ecosystems). This plant assemblage constitutes the alone current testimony of the flora and primary vegetation which inhabited the Northern Mediterranean region during the last millions years. The geographical situation of the studied area makes the greatest interest of this region for palynological studies. Pollen analyses developed on a botanical background are very rare in Turkey if it is almost inexistent. There are some studies on Miocene and Pliocene in Anatolia (Nakoman, 1967; Benda, 1971; Akgün & Akyol, 1999), but (1) their very poor botanical interest because of a very limited pollen morphological approach, and (2) the highly questionable quantitative information that they are supposed to provide make them almost completely unusable. This study is the first investigation to have been developed in the region on a fine pollen morphology investigation resulting in reliable botanical comprehensive information. Pollen identification was performed after their accurate morphology examination by comparing Neogene pollen grains with their living relatives using databanks of modern pollen grains and modern-past pollen grains photographs (atlases, databases) with respected to botanical nomenclature. At present, the history of the flora, vegetation and climate of the Mediterranean region are very well-documented for the last 23 Myrs after the thesis of Jean-Pierre Suc (1980) and the about twenty theses that he supervised (e.g.: Bessedik, 1985; Zheng, 1987; Combourieu-Nebout, 1987; Drivaliari, 1993; Fauquette, 1997; Bachiri-Taoufic, 2000; Popescu, 2001; Jiménez-Moreno, 2005; Joannin, 2007; Favre, 2007).

Figure 1a. Distribution of *Liquidambar*, *Pterocarya* and *Zelkova*.

Figure 1b. Distribution of *Cedrus libani*.

Suc, J-P. (1980) studied different areas from the North-western Mediterranean region. He demonstrated that the modern Mediterranean vegetation took root in the Late Pliocene¹. Two important events occurred at 3.4 and 2.6 Ma. The former was the establishment of the modern Mediterranean vegetation and the latter the first evidence of the vegetation response in the Mediterranean region to the earliest glacial in the Northern Hemisphere. The previous vegetation was impacted by glacial-interglacial fluctuations and finally, by human activity.

Jiménez-Moreno, G. (2005) documented Early to Late Miocene vegetation and climate dynamics from the South-eastern Europe to the North-eastern Mediterranean. There was a progressive rarefaction of the most thermophilous trees and shortening of the broad-leaved evergreen forest. On the contrary, there were a development in mesothermic (mainly deciduous) elements, altitudinal trees and herbs during the Middle and Late Miocene. This can be related to enlargement of the East Antarctic Ice Sheet (EAIS) and regional uplift.

Popescu, S. (2001) carried out high resolution pollen analysis on the Lupoia section (SW Romania). Pollen records enabled the reconstruction of the early Pliocene vegetation of southwestern Romania. Repeated changes in vegetation occurred with clay-lignites alternation. While altitudinal trees corresponded a decrease in temperature, thermophilous trees developed under humid conditions.

Popescu *et al.*, (2006) studied pollen records of the western Dacic Basin. The Early Zanclean sediments of Dacic Basin was provided by pollen records and eccentricity curve. According to this, thermophilous plants increased during the lowest eccentricity minima (in 400 kyrs cycles). On the contrary, altitudinal elements are enriched during the highest eccentricity maxima.

Popescu, S. (2006) studied to investigate paleovegetation during the Late Miocene-Early Pliocene from high-resolution pollen analysis in DSDP Site 380. According to this study, the Late Miocene vegetation was characterized by delta environment. During Early Pliocene, two vegetation types were defined by thermophilous plants and dry steppes.

¹ Here, we follow the chronostratigraphic nomenclature recently adopted by IUGS (Gibbard *et al.*, 2009) where Pliocene is constituted by two stages, Zanclean (5.332 – 3.6 Ma) and Piacenzian (3.6 – 2.588 Ma), Gelasian becoming the first stage of Pleistocene (i.e. Quaternary).

In order to be sure to well appreciate the contrast between forest and steppe vegetation, the investigated region has been defined a little larger than the Anatolia region and includes also the surroundings of the Marmara Sea, a part of the southwestern Black Sea shorelines and of northern Greece (Fig. 1.2).

The region was subject to intense paleogeographic changes controlled by regional tectonics extremely active since 6 Ma and by the partly coexisting desiccation of the Mediterranean and Black seas (5.6-5.33 Ma) (Armijo *et al.*, 1999; Görür *et al.*, 1997, 2000; Gillet, 2004; Clauzon *et al.*, 2005; Melinte *et al.*, 2009). The North Anatolian Fault (NAF) extends from Karlıova to the Gulf of Saros along the Black Sea mountains of North Anatolia (Fig. 1.3). It seems to have originated during the Late-Middle Miocene when the Anatolian plate separated. The westward motion of the Anatolia plate with respect to Eurasia and African plates induced great geodynamic changes in the Eastern Mediterranean. This gave rise to the Aegean extensional regime and deformation of Anatolia (Şengör, 1979).

Today, relictous plants are distributed in the eastern Mediterranean region. Such an evolution was forced by the successive coolings in the Antarctic area first (at 14 Ma then 5.8 Ma), then especially the repeated Arctic coolings (since 3.6 Ma) that controlled the glacial-interglacial cycles since 2.6 Ma. Simultaneously, the environments in the South Mediterranean, already characterized by open vegetations since the earliest Miocene (probably because of the neighborhood of the pre-existing Sahara Desert), were enriched in *Artemisia* steppe element probably originating from the Anatolian Plateau (*Artemisia*) (Popescu, 2006) that repeatedly invaded the entire Mediterranean realm at each Arctic glaciation. Some testimonies of these relictous floras and thermophilous vegetations exist both to the West (mountains of South Morocco, Canary Islands) and to the East (Anatolia, southern Caucasus) (Quézel and Médail, 2003).

Figure 1.2. Map showing the studied pollen localities (black dots).

Figure 1.3. Map showing the main tectonic elements of eastern Mediterranean regions (modified from McKenzie, 1972; Şengör *et al.*, 1985; Okay *et al.*, 1999).

1.1. Aims of the study

Anatolia is one of the most important refuge area. The geographic position of Turkey also makes concerned area significant for palynological investigations. There are some pollen studies both Miocene and Pliocene in Turkey. However, they have lack of accurate botanical identification and therefore they are questionable. Hence, this thesis work is the first study based on the pollen botanical nomenclature which improve significantly floristic and vegetation interpretations. In this study, 436 samples from both marine (in the western Black Sea, DSDP Site 380) and outcrops sediments in the NW Turkey (Enez, İntepe, Eceabat, Burhanlı and West Seddülbahir), Northern Greece (Prosilio, Trilophos and Lion of Amphipoli) and Western Macedonia (Ptolemais Notio and Ptolemais Base) which covers time-intervals between the Late Miocene-Early Pleistocene are analysed palynologically. On the whole, samples are rich in terms of pollen grains. Some samples are barren in pollen grains. Because of this, these samples are not taken into account in the synthetic and detailed pollen diagrams. The targets of this thesis using the pollen analysis of sediments, their identification being botanically driven are:

- (1) to document the history of the flora and vegetation of Anatolia, then to compare with the Western Mediterranean region which is already well-known,
- (2) the reconstruction of the vegetation through the studied time-window using pollen records,
- (3) to follow-up on the tropical and subtropical thermophilous plants (distribution, abundance, etc.) during the Late Cenozoic,
- (4) to determine global climatic changes (coolings, warmings, glacial-interglacial cycles),
- (5) to assess the role possibly played by the African-Asian monsoon effect in the persistence of floral refuges in the studied regions, and
- (6) to determine the relative influence of regional geodynamics.

2. PHYSIOGRAPHY, STRATIGRAPHY AND PALEO GEOGRAPHY OF THE STUDY AREA

2.1. TURKEY

2.1.1. PRESENT-DAY VEGETATION

In spite of its diversity and complexity, the present-day vegetation of Anatolia may be summarized as follows, relating both to the Mediterranean, Irano-Touranian, European and Euxino-Hyrcanian phytogeographic regions (Fig. 2.1).

On the whole, the Mediterranean realm concerns the West and South coastal areas, with a variable width (100 to 300 km), but it sporadically appears along the North shoreline and also in some encased valleys within the Pontic Ranges. It is mainly controlled by climate (Akman & Ketenoglu, 1986). Several bioclimates and altitudinal vegetation belts have been defined according to rainfall and altitude (Quézel & Médail, 2003). The thermo-Mediterranean belt is constituted by assemblages with *Olea europea* and *Pistacia lentiscus* where *Ceratonia* is scarce, and also by coniferous forests with *Pinus brutia* occupying large areas. Some riparian forests are noteworthy, as they show *Alnus* associated with *Liquidambar orientalis* and *Platanus orientalis*. Basically, the meso-Mediterranean belt should be characterized by sclerophyllous oaks, but *Quercus ilex* is actually very rare being only present from Samsun to Trabzon. *Quercus calliprinos* is obviously the most frequent sclerophyllous oak while *Q. aucheri* is recorded along the Lycian shoreline. Here, deciduous oaks (*Quercus cerris*, *Q. trojana*, *Q. ithaburensis*, etc.) have been almost everywhere replaced by cultivations. On contrary, *Pinus brutia* occupies an important place which is not yet completely understood (Boydak, 2006). The supra-Mediterranean belt is theoretically inhabited by deciduous associations. This is right in the northern region where *Quercus pubescens*, *Q. cerris*, *Q. petraea* subsp. *iberica* take up a significant place, often with *Carpinus orientalis* and *Ostrya carpinifolia*. Contrarily, westward and southward, as also on the Amanos Mountains, this vegetation belt is practically invaded by *Pinus brutia*, whereas deciduous trees are very scarce and restricted to residual localities. *Pinus nigra* subsp. *pallasiana* already appears in its uppermost part. The Mediterranean montane belt is actually the altitudinal coniferous belt with abundant specimens of *Pinus pallasiana* and also

Cedrus libani occupying significant areas on the Taurus and Anti-Taurus massifs, in association with *Abies cilicica* westward Antalya in spite of distinct ecological requirements. These forests, generally very deteriorated, are often replaced by pre-steppic associations with arborescent *Juniperus* (*J. excelsa*, *J. foetidissima*). The oro-Mediterranean belt (Quézel, 1973) is invaded by meadows and steppes where prickly cushion-shaped xerophytes are abundant, a belt already influenced by the Irano-Touranian conditions. The Irano-Touranian phytogeographic region encompasses the Anatolian Plateau, mostly eastward the Centro-Anatolian Ridge, and westward areas characterized by annual precipitations lower than 200 mm. Man greatly disturbed this region because of repeated attempts in development since antiquity (Akman & Quézel, 1996). The area is occupied by a very rich steppe vegetation where *Artemisia* is relatively subsidiary, at least to the West. Pre-steppic structures with trees appear only over reliefs where precipitations are higher and man activity less apparent. *Quercus pubescens* subsp. *anatolica* is widely present, very often in a state of grazed shoots in the northwestern part of the region. Westward, *Pinus pallasiana* and *Juniperus excelsa* are prevalent while Irano-Touranian oaks (*Q. libani*, *Q. brantii*, *Q. infectoria* subsp. *boissieri*) grow to the East.

The European phytogeographic region is secondary in Anatolia. Only some deciduous hilly structures with *Quercus* and *Carpinus betulus* may belong to it. Some other European associations are more obvious within the montane vegetation belt, from the Kaz Mount to the area of Kastamonu, where beautiful forests develop including *Fagus orientalis* (often difficult to distinguish from *F. sylvatica*) and firs (*Abies equi-troyani*, *A. bornmuelleriana*).

The Euxino-Hyrcanian phytogeographic region, characterized by high precipitations and the lack of any summer drought, develops all along the Black Sea shoreline. Here, wonderful hilly forests still exist, dominated by deciduous elements (*Quercus hartwissiana*, *Q. macranthera*, *Carpinus betulus*, *Castanea sativa*, with *Fagus orientalis* and *Rhodendrum ponticum* in some places, even *Rh. flavum*). Some alluvial associations and riparian forests show *Alnus*, *Fraxinus*, and *Pterocarya* in some localities (see below).

Figure 2.1. The present-day vegetation map of Turkey (Quézel & Médail, 2003).

The montane belt is mainly occupied by *Fagus orientalis* and *Rhododendron ponticum*, with locally *Abies nordmannian*, and Eastward *Picea orientalis*. *Pinus sylvestris* is present from place to place in marginal areas, especially to the South. The subalpine and alpine belts are mainly developed to the East where Caucasus influences infer within a very diversified flora (*Juniperus communis* and *J. sabina* coexist with several Ericaceae). *Buxus sempervirens (colchica)*, often associated with *Taxus baccata*, abound on the rare calcareous spaces in the region. Back to the Euxinian zone, a transition area has been identified between the Mediterranean and Irano-Touranian phytogeographic regions, the so-called Pre-Pontic region (Quézel *et al.*, 1980), the vegetation of which is dominated by *Abies* spp., *Pinus nigra* subsp. *pallasiana*, and *P. sylvestris*.

May Anatolia be considered as a present-day refuge area of a thermophilous flora?

Some warm-temperate Eurasian taxa (such as *Liquidambar*, *Pterocarya*, *Cedrus*) have already emerged from this brief overview of the Anatolian vegetation (Quézel, 1995), currently recorded in the European Late Cenozoic pollen records as it will be emphasized below. According also to Browicz (1982-1994), few taxa are still present in Anatolia (*Liquidambar*, *Pterocarya*, *Zelkova*) and in the Hyrcanian zone (*Parrotia*). *Zelkova crenata* is today recorded only in two very restricted riparian localities in easternmost Anatolia close to the Van Lake, although it is still well-developed in Abkhazia, Small Caucasus, and mainly in the Hyrcanian region. In addition, this genus is still present in residual stations of Crete (*Z. abelicea*) and Sicily (*Z. sicula*) (Quézel 1995).

Liquidambar orientalis is concentrated in some more or less important areas (Fig. 2.1): the vastest of which concerns the alluvial and riparian forests of the southwesternmost part of Anatolia (mainly the area of Köyceğiz – Marmaris), another one of significantly less extent locates northeastward Antalya (Köseler area; Akman *et al.*, 1993), the third one along the Oronte River close to Hatay is today questionable because it seems that it was not recently re-visited. The strong reducing of the two last localities is attested by ancient documents indicating that *Liquidambar* was abundant during the Hellenic time and intensely used for producing styrax (Amigues, 2007).

Pterocarya is still present in alluvial forests along the Black and Marmara seas, being relatively abundant in the latter (Fig. 2.1). It is also recorded in some localities near the Iskenderun Gulf where precise information is missing (Fig. 2.1). However, the tree is frequent out of Anatolia in the above-mentioned regions where *Zelkova* is living. *Cedrus* benefits from better conditions (Quézel & Médail, 2003): it abounds on the Taurus and Anti-Taurus massifs, although it is declining. Few reduced localities persist on the back slope of the Pontic Ranges (Erbaa region; Fig. 4), the indigenous status of which is supported by pollen data (Bottema, 1986). Some other plants should be added to the above discussed Anatolian relicts, such as *Diospyros lotus*, *Ilex colchica*, *Rhododendron* spp., even *Quercus pontica* and *Osmanthus decorus* which grows today along the Black Sea shoreline (Quézel, 1986). Anatolia can undoubtedly be considered as a present-day refuge area of warm-temperate plants, however with less importance than the Hyrcanian region.

2.1.2. CLIMATE

2.1.2.1. TURKEY

The location and geographical characteristics of Turkey give a variety of climates, landscapes and plant diversity. Turkey is located in large Mediterranean geographical area. The climate is characterized by Mediterranean macro climate. Eastern Mediterranean region is influenced by three main atmospheric systems (Fig. 2.2): the main middle to high latitude westerlies to the north and northwest, the mid-latitude subtropical high-pressure systems extending from the Atlantic across the Sahara and the monsoon climates of Indian subcontinent and East Africa (Akcar and Schlüchter, 2005). Marine tropical air masses (mT) bring hot and humid air from the tropical north Atlantic. Continental tropical airstreams (cT) convey from the northern African and Arabian deserts. It passes over the Mediterranean Sea, and they can obtain moisture and then condensate over the southern coasts of Anatolia. Marine polar air masses (mP) carry the humid and cold air from the polar north Atlantic. They have significant influence when they progress over the Mediterranean Sea. Continental polar air masses (cT) bring the dry and cold air from Siberia. They can acquire moisture and condensate on the northern coasts of Turkey (over the Black Sea) (Fig. 2.2). The climatic conditions are warm-temperate in Turkey (Erinç, 1959). It is now usually known that the climate variability in the middle- and high-latitude continental Northern Hemisphere mainly controlled by the Arctic Oscillation and North Atlantic Oscillation (AO/NAO) at interannual and interdecadal timescales (Thompson and Wallace, 2001). This changing patterns also affect the climate of Turkey and its surrounding fields (Cullen and deMenocal, 2000; Karaca et al., 2000; Türkeş and Erlat, 2003; Karabök et al., 2005; Kahya and Cengiz, 2007). The secondary cyclogenesis in eastern Mediterranean enables a physical linkage between the NAO (known as a key provider of precipitation to the Middle East region) (Cullen and deMenocal, 2000) and climatic surface variables in Turkey (Kahya and Cengiz, 2007). Turkey's climate is modified by its topographic relief that result in great regional differences in the amount of mean annual precipitation and by rapid transitions from rainy areas to dry ones. Most abundant precipitation (>1000 mm) occurs in Black Sea coast in the north and on the western Taurus Mountains in the southwest.

The Eastern Black Sea and the Western Mediterranean coasts are the wettest areas of the country in winter, with a mean rainfall total of more than 650 mm (Türkeş, 1996). Approximately half of the country has less than 50 mm mean rainfall in summer, with a minimum of less than 5 mm along the Turkey–Syria border. Mean annual rainfall total is about 300 mm over continental central Anatolia. Besides, along the Western Black Sea, Eastern Black Sea, and Western Mediterranean coasts are more than 1,000 mm. The highest mean annual rainfall total was recorded on the Eastern Black Sea coast (2,304 mm). Over the continental Mediterranean region, mean annual rainfall increases from south (with about 400 mm) to north (with about 800 mm).

Figure 2.2. Atmospheric air masses affecting the Eastern Mediterranean region (cP: Continental Polar Air Mass; mP: Marine Polar Air Mass; cT: Continental Tropical Air Mass; mT: Marine Tropical Air Mass, PJF: Polar Front Jet; STJ: Subtropical Jet; ITCZ: Intertropical Convergence Zone (modified from Wigley&Farmer, 1982).

The annual rainfall is more than 500 mm over a considerable part of the continental eastern Anatolia region, and it increases over mountains. According to climatic differences of the regions in Turkey due to the existence of irregular topography, four macroclimate types are determined (Erinç, 1996). These macroclimate types are as follows hereafter (Fig. 2.3):

1) I - Steppe Climate:

In this climate type, semi-arid conditions dominate. Rainfall pattern resembles the coasts of Mediterranean. It is divided into two types:

- a) **Ia – Anatolian steppe climate:** The summers are hot (20-25°C) and the winters are cold (0-3°C).
- b) **Ib – Southeastern Anatolian steppe climate:** While the summers are considerably hot (>30°C), the winters are cold (0-5°C). High evaporation is observed (annually 1000-2000 mm).

2) **II – Black Sea Climate:** All seasons are rainy. It is composed of three types according to rainfall and temperature:

- a) **IIa – Eastern Black Sea climate:** it has high rainfall. Winters are temperate.
- b) **IIb – Central Black Sea climate:** with an average rainfall.
- c) **IIc – Western Black Sea climate:** less amount of rainfall, winters and summers have less temperature.

3) **III – Mediterranean Climate:** Although high annual precipitation, it is observed a severe summer aridity. This climate type is divided into two types according to temperature: IIIa () and IIIb

- a) **IIIa – Mediterranean climate:** very high summer temperature. In the winters, small amount of snow.
- b) **IIIb – Marmara region climate:** very cold winters, low evaporation.

4) **Eastern Anatolian Climate:** very cold winters, it is divided into two types:

- a) **IVa – All seasons with precipitation:** it represents a continental climate regime.
- b) **IVb – Arid summer type:** high precipitation in winter and spring; little precipitation and high evaporation during summer and autumn.

Figure 2.3. Macroclimate types of Turkey. Ia: Anatolian steppe climate; Ib: Southeastern Anatolian steppe climate; IIb: Central Black Sea climate; IIc: Western Black Sea climate; IIIa: Mediterranean climate; IIIb: Marmara climate; IVa: All seasons with precipitation type; IVb: Arid summer type (modified from Erinç, S., 1996).

2.1.2.2. GREECE AND MACEDONIA

The climate in Greece is a Mediterranean type climate with dry and hot summers (Mariolopoulos, 1938). Between October and March exist cold and rainy period, from April to September; warm and dry period exist. The coldest months are January and February with average minimum temperature ranging between 5-10 degrees Celsius. Rainfall is high on the west coast, about 1000 mm (Mariolopoulos, 1925). The main factors controlling the climatic conditions in Greece are the atmospheric circulation, the latitude, the altitude and, generally, the orography, the Mediterranean sea surface temperature (SST) distribution, the land-sea interactions (distance from the sea) and smaller-scale processes (Lolis *et al.* 1999). In Greece, there are five climatic regions. These regions are :

- 1) in the western coast of Greece and the islands of Ionian Sea which take high amounts of precipitation. The maximum precipitation observed in autumn and a minimum precipitation occurs during the summer. The annual temperature range is small in this region,
- 2) in the Aegean region (the islands of the Aegean Sea and the west coast of southern Greece) exist low winter temperatures, high summer temperatures and low precipitation. The annual precipitation is on average,
- 3) in the northern and the central part of Greece is characterized by long duration storms, short drought periods, low temperatures during winter and large annual temperature range,
- 4) in Crete and the southern Greece characterize the Mediterranean desert type climate with low annual precipitation and droughts of long duration,
- 5) in the Pindous Mountain range which divides into Greece as western and eastern regions, and the mountains of northern, central and southern Greece. The climate in this region is the typical climate of mountain areas with high annual precipitation and strong gradients of precipitation and temperature with elevation (Loukas *et al.*, 2001).

The frontal depressions approaching Greece (from January to April) are rain producers along the western coast of Greece and in the central Aegean Sea. During winter, in the Atlantic near the Gibraltar Strait depressions originate. High rainfall is limited to the islands and the coastal areas of Ionian Sea with some influence in Thessaly (central Greece). The frontal depressions approaching Greece from the west they cause southwest winds over the Ionian and Aegean Seas forcing the maritime air eastwards (Xoplaki *et al.*, 2000). During summer, the high pressure belts of the subtropics drifts northwards.

In Macedonia, the climate is characterized by the submediterranean to a continental and mountainous climate. However, The Mediterranean climate basically influences. An Average daily temperature exist in Skopje, ranging from 32° to -3° (Kendrovski, 2006). Continental climate occurs in the central part of Macedonia. The average annual temperature is 12°C and summer temperature is 19°C (April-September). During winter, an average temperature is 5°C between October and March (Hristovski *et al.*, 2007). In addition, in winter, there is little wind and rain. In addition during this season, a slight lowering of pressure over central eastern Europe brings low-pressure zone over the Mediterranean. The cyclonic depressions in the winter months follow one another from west to east over the Mediterranean. Spring and Autumn are signed by heavy thunderstorm which cause rainfall. The dry season lasts two months in Macedonia (Ogilvie, 1920).

2.1.3. STRATIGRAPHY OF THE STUDY AREAS

The Marmara region mainly consists of the İstanbul and the Strandja zones to the North, the Sakarya zone and İzmir-Ankara Zone to the south. These zones are overlain by fore-arc Thrace Basin rocks formed during Eocene-Oligocene time. Today, these zones are separated from each other by the major structural elements (suture zones/transform faults). The northern shorelines of Marmara Sea are generally cliffy and shore type includes pocket beaches. Neogene rocks are widely distributed in the NE Aegean, around of the Sea of Marmara and Greece (Fig. 2.4). In the north-west Marmara shorelines, south of the North Anatolian Fault, from Gaziköy westward, along the Çanakkale shorelines, there exist Miocene micaceous quartz sandstones (Kirazlı and Gazhanedere Formations) (Türkecan and Yurtsever, 2002).

Kirazlı Formation conformably overlies the Gazhanedere Formation. The Kirazlı Formation consists of cross-bedded, yellow sandstones with rare mudstone and conglomeratic intercalations in the northern Gulf of Saros. The Alçıtepe Formation is widely distributed in the Sea of Marmara and the Gulf of Saros regions (Sayar, 1987; Sümengen *et al.*, 1987; Siyako *et al.*, 1989; Görür *et al.*, 1997; Çağatay *et al.*, 1999; Sakınç *et al.*, 1999; Görür *et al.*, 2000).

Alçıtepe Formation lies conformably over the Kirazlı Formation (Sakınç *et al.*, 1999; Yaltrak *et al.*, 2000). However, other workers claims unconformably relationship between the Kirazlı and Alçıtepe Formations (Armijo *et al.*, 1999; Melinte *et al.*, 2009). The nannofossil data show that the age of the Alçıtepe Formation is younger than the Messinian Salinity Crisis (Melinte *et al.*, 2009). The Alçıtepe Formation demonstrates different facies characteristics in the northern coast of the Gulf of Saros and Gelibolu and Biga peninsulas. In Enez, (NW of the Gulf of Saros), Alçıtepe Formation (Mactra-bearing limestone section) is 23-m thick and includes a rich and sandstone intercalations in the upper part .

Figure 2.4. Distribution of Neogene rocks in the Marmara regions and North Aegean (simplified from Türkecan and Yurtsever, 2002; Okay *et al.*, 1996 and Aldanmaz, 2002; Bornovas *et al.*, 1983).

The formation is overlain unconformably by the alluvial fan deposits of the Conkbayırı Formation in the Gelibolu Peninsula. The Alçıtepe Formation contains mudstone and marl in the lower part, bioclastic and oolitic limestones with marl in the upper part in the Gelibolu and Biga peninsulas. The Alçıtepe Formation is overlain with an erosional unconformity by the Göztepe Formation (NN12 zone), composed of shallow marine siltstone and sandstone with ostrea banks and mollusc-rich sandy interbeds towards the upper part.

The DSDP 380 Black Sea core includes five stratigraphic units and fourteen sub-units identified by Ross (1978). Unit 1 consists of terrigenous sediments, including muds, sandy silts. Unit 2 includes aragonite, sideritic and calcitic siltstone, interbedded in muds. Unit 3 compose of seekride, including calcitic oozes and marls. Unit 4 includes calcitic, sideritic, aragonitic and dolomitic, interbedded in muds and Unit 5 consists of Black shales with dolomite and zeolitic silt intercalations (Fig. 2.5).

Neogene rocks in the Florina-Ptolemais-Servia (FPS) Basin (Upper Miocene-Lower Pliocene) is located in Greece. The lacustrine sediments in this basin are appeared in a series of open-pit lignite quarries. The age of the Ptolemais section is between 5.3 and 3.9 Ma (Van Vugt *et al.*, 1998; Steenbrink *et al.*, 2000). The Florina, Ptolemais and Servia sub-basins are located between 300 and 700m above the sea level. These sub-basins are surrounded by mountains (~2000 m) that consist of Mesozoic limestones, Upper Carboniferous granites and Paleozoic schists. The Late Miocene-Early Pleistocene lake sediments contain lignites and alluvial deposits.

The studied stratigraphic sections include four lithostratigraphic units: Komnina Formation, Ptolemais Formation, Proastio Formation and Perdika Formation. The Ptolemais Formation is ~110 m thick and includes alternation of lignites and lacustrine marls with fluvial sand, silts and some volcanic ash intercalations. The age of the Ptolemais Formation is Early Pliocene (MN 14 and 15) based on paleontological data, magneto- and cyclo-stratigraphy and $^{40}\text{Ar}/^{39}\text{Ar}$ dating (Van Vugt *et al.*, 1998; Steenbrink *et al.*, 1999). The Komnina Formation is approximately 300 m thick and overlays unconformably the pre-Neogene basement. The formation contains alluvial sands, conglomerates, lacustrine marls and clays with some intercalated lignite seams (Steenbrink *et al.*, 2006).

Figure 2.5. Stratigraphic of studied sedimentary sections in DSDP Site 380, Intepe, Enez and Ptolemais.

The middle part of the Komnina Formation is dated as the Late Miocene based on the small mammals (de Bruijn *et al.*, 1999), magneto and cyclo-stratigraphy (Steenbrink *et al.*, 2000). The Prosilio section is located in 10 km SW of Servia. This section includes ~200 m of lacustrine and alluvial sediments (marls, lignites, clays, sands and conglomerates) (Steenbrink *et al.*, 2006).

2.1.4. PALEOGEOGRAPHY

Paleogeography affects the climate, and thus also influence vegetation and fauna. During the Neogene, convergence between the Eurasian plate and African plate caused Tethys to close and form in its place the Mediterranean Sea and Paratethys (Meulenkamp and Sissingh, 2003). Paratethys realm includes a part of the Alps, Carpathians, Pannonian, Dacic and the Euxinian basin (Black Sea, Caspian Sea and Aral Sea today). In the Late Tortonian (Pannonian), larger portions of northern Peri-Tethys were emerged and extensive sedimentation started to break up in the western and central domains. During this time, alluvial deposits and lacustrine carbonates accumulated in the Ebro Basin (NE Iberia) (Meulenkamp *et al.*, 2000b). In the central Europe, brackish to fluviolacustrine conditions existed in central Paratethys in the Late Tortonian. Mediterranean marine connection with intra-arc domains no longer existed.

Ephemeral marine incursions in the outer Carpathian were restricted to Dacic basin. The sediments of the Late Tortonian (Middle Maeotian) includes nannoplankton assemblages indicating the lower part of NN11 zone (Fornaciari *et al.*, 1997; Marunteanu and Papaianopol, 1998). In addition, in the Late Tortonian Dacic basin became a part of the Eastern Paratethys. In the latest Early to earliest Middle Miocene, marine invasion occurred in the central part of Arabian Platform. However, sea regressed during the late Middle Miocene (Meulenkamp *et al.*, 2000b) (Fig. 2.6).

In the late Miocene (Late Messinian; Late Pannonian-Early Pontian), the Messinian Salinity Crisis affected the Mediterranean basins (Fig. 2.7). Evaporites were deposited in different depths (Popov *et al.*, 2006). According to the largely accepted hypothesis (CIESM, 2007; Clauzon *et al.*, 2001), sea level drop occurred in two steps separated by a flooding event. The first step (5.8 Ma), Mediterranean margins were impacted (sea-level fall of ca. 150 m). The second step occurred in an outstanding sea level fall of about 1500 m at 5.6 Ma and effected the whole basin (Clauzon *et al.*, 1996). The Paratethys had a strong influence on the Mediterranean region during the Messinian Salinity Crisis. Between the two low-stand phases, the Lago Mare event took place probably originating from the Paratethys (Cita *et al.*, 1978a).

Figure 2.6. Paleogeographic map of the Late Tortonian (8-7 Ma), indicating position of continental, shallow and deep basins. Thick black lines show fault zones (modified from Meulenkamp *et al.*, 2000b; Meulenkamp and Sissingh, 2003).

The Lago Mare facies is characterised by common brackish shallow water fauna: *Congeria*, *Dreissena*, *Melanopsis* among molluscs; *Cyprideis pannonica* gr., *Loxoconcha*, *Tyrrhenocythere*, etc., among ostracods (Ruggieri, 1967; Cita and Colombo, 1979) and endemic Paratethyan dinocysts *Galeacysta etrusca* (Müller *et al.*, 1999; Bertini *et al.*, 1995; Bertini, 2002). In the Late Messinian (Early Pontian), Eastern Paratethys reached its maximum areal extent (Popov *et al.*, 2006) (Fig. 2.8). The eastern and northern margins of the early Pontian basin was caused by the transgression (Popov *et al.*, 2004). At the same time, paleogeographic changes were controlled by regional tectonics, and the desiccation of the Mediterranean and Black seas that were partly coeval during the Messinian took place (Armijo *et al.*, 1999; Görür *et al.*, 1997, 2000; Gillet, 2004; Clauzon *et al.*, 2005; Melinte *et al.*, 2009).

Figure 2.7. Palinspastic paleogeographic map for Late Miocene (Late Messinian, Late Pannonian-Early Pontian) showing shallow and deep basins (modified from Popov *et al.*, 2006; Olteanu and Jipa, 2006).

Significant paleogeographic changes occurred during the Late Miocene in the eastern Mediterranean. During the Messinian, the MSC effected on terrestrial and marine ecosystems (i.e., planktonic foraminifers, calcareous nannoplankton and dinoflagellates). The MSC has been observed all over the Mediterranean region including Aegean Sea. The rocks of Messinian-Early Zanclean age are widely distributed in northwestern Turkey. The Messinian erosional surface has been observed in the Mediterranean area and Eastern part of Black Sea (Clauzon *et al.*, 1996). Some localities indicate discontinuity occurred by weak erosion in İtepe (Çağatay *et al.*, 2007, Melinte *et al.*, 2009). During the Late Miocene, continental and marine sedimentation existed in northern Anatolia (Görür *et al.* 1997). Marine sedimentation existed in the Black Sea area. Continental sedimentation developed in basins formed by the North Anatolian Fault which initiated during the Early-Late Miocene (Barka and Hancock, 1984; Barka, 1985, 1992). Mediterranean extensively dried up (Hsü, 1972, 1974; Adams *et al.*, 1977; Ryan and Cita, 1978) and also Black Sea desiccated (Gillet, 2004; Popescu, 2006) during the Messinian.

Paleogeography of the studied region covering time-window from the Early Messinian to the Latest Messinian-Earliest Zanclean is shown in Figs. 2.8-2.9. Pollen records from the studied areas provide information about palaeoenvironments just before and just after the Messinian Salinity Crisis (Melinte *et al.*, 2009). Herbs were abundant before the Messinian Salinity Crisis in Burhanlı, Eceabat and İtepe sections. Black Sea witnessed desiccation during the Messinian Salinity Crisis (Hsü and Giovanoli, 1979), as indicated by the 19 m thick “Pebbly Breccia” (containing blocks of stromatolitic dolomite) in DSDP Drill Hole 380 (Ross *et al.*, 1978). This presumably produced a shorter break in the pollen record. In addition, diatom data of the Black Sea hole suggests that Black Sea was very shallow at that time (Schrader, 1978). Deep desiccated basin evaporates were deposited during that time. At this time, subtropical and warm-temperate trees were abundant in the southwestern Black Sea (Popescu, 2006). Before the MSC, in northern Greece (Prosilio), meso-microthermic (mainly *Tsuga*) and microthermic trees (*Abies* and *Picea*) are abundant. This could be explained as by uplifting of the surrounding region.

Figure 2.8. The Paleogeographic map of Marmara region and Greece and Macedonia before the Messinian Salinity Crisis (Early Messinian) (modified from Görür *et al.*, 1997; Sakıncı *et al.*, 1999; Vasiliev *et al.*, 2004; Çağatay *et al.*, 2006; Krijgsman *et al.*, 2010).

Figure 2.9. The Paleogeographic map of Marmara region and Greece and Macedonia after the Messinian Salinity Crisis (Latest Messinian-Earliest Zanclean) (modified from Sakıncı and Yalıtırak, 2005; Rögl and Steininger, 1983; Meulenkamp and Sissingh, 2003).

The connections between the Mediterranean and Paratethys were enabled after the Messinian Salinity Crisis. During the Pliocene time interval, the Mediterranean Sea was inundated by marine waters due to the connection with Atlantic Ocean after the Messinian Salinity Crisis (Hsü and Bernoulli, 1978). Early Zanclean reflooding occurred within two steps: collapse (at 5.480 Ma) and widening (at 5.330 Ma) of the Gibraltar Strait (Clauzon *et al.*, 2007). During the Early Pliocene, northern margin of the Sea of Marmara Basin was uplifted and eroded, while the southern margin turned into continental areas (Görür, *et al.*, 1997). At the Latest Messinian-Earliest Zanclean (After MSC), altitudinal conifers (*Cedrus*, *Abies*, *Picea* and *Pinus*) indicate an augmentation in the north-western Aegean (i.e. İntepe and west Seddülbahir). This could indicate uplifting of the region. Indeed, uplifting occurred during the Messinian due to propagation of the North Anatolian Fault (NAF) (Armijo *et al.*, 1999; Melinte *et al.*, 2009).

During the Middle Pliocene (Piacenzian)-Early Pleistocene (Gelasian), Iberian domain emerged. In these basins which located in south-eastern Iberia (Aguirre, 1998) and Atlantic coast, alluvial and shallow marine sediments deposited (Fig. 2.10). In the central Paratethys (intra-Carpathian domains) continental clastic accumulated in the Middle Pliocene-Early Pleistocene. Back-arc basin in the south-western part were filled with Pliocene deposits (reaching thickness about 1000metres) (Meulenkamp *et al.*, 1996). In addition, widespread volcanism occurred in the Styrian and Danube basins, Great Hungarian plain, South Slovakian-North Hungarina volcanic domain and south-eastern Transylvania (Szabo *et al.*, 1992). Nevertheless, the faunal composition of Dacic basin changed significantly (extinction of Limnocyprids and appearance of Unionids, Viviparids and Melanopsids) during the Pliocene (Meulenkamp *et al.*, 2000b). Eastern Paratethys contained two major basins (Dacic-Euxinian basin system and the Caspian basin) since latest Miocene (Late Pontian). In the Caspian basin occurred a major regression with reduction of salinity in the Early Pliocene. The Akchagylian Sea was characterised by low salinity and euryhaline biotas (Meulenkamp *et al.*, 2000b). Pollen assemblages and macroplant fossils (leaf remains) show that existence of a forested hinterland with similar to those of present-day taiga.

Figure 2.10. Paleogeographic map of the Piacenzian-Gelasian (3.4-1.8 Ma), indicating position of continental, shallow and deep basins. Thick black lines show fault zones (modified from Meulenkamp *et al.*, 2000b; Meulenkamp and Sissingh, 2003).

They show climatic conditions changing from cool and dry towards relatively warm and wet (broad-leaved forest zone; Neogene System, 1986). The connection between the Mediterranean and Paratethys during the Late Pliocene is also supported by its faunal assemblages in the sediments. The faunal distribution of the sediments shows that the Marmara Basin was firstly invaded by Paratethys and then by the Mediterranean during the Late Pliocene. At the beginning of the Pleistocene (2.6 my ago), the climate got cooler and glacial-interglacial cycles appeared in the Northern Hemisphere. This is also well recorded in the pollen spectra. For instance, *Artemisia* steppe became important during the glacial periods (DSDP Site 380). At the interglacial periods, forest formations developed. Nevertheless, during glacial periods, the Sea of Marmara was isolated from the Mediterranean and became a brackish water environment and reconnected during the Quaternary interglacials including the Early Holocene (Stanley and Blanpied, 1980; Smith *et al.*, 1995; Aksu *et al.*, 1999; Çağatay *et al.*, 2009).

3. METHOD

Palynology is the science of the present and fossil palynomorphs such as pollen, spores, dinoflagellate cysts and acritarchs. Because of the strong wall (exine) of the pollen grains, they can be well-preserved for a longtime in the sediments. Pollen provide a high resolution and continuous record of climate. Palynology is especially a very good tool for assessing paleovegetation and paleoclimate history. In addition, palynological studies are used for biodiversity, biostratigraphy and characterisation of the past environmental changes. Samples used in this study are located in the western Black Sea (DSDP 380 Site), NW Turkey and Northern Greece and Macedonia (Fig. 1.2). Pollen grains are generally well preserved in the sediments. In this study, a total of 436 samples have been analysed (Table 1.1). 378 of these samples (Late Miocene-recent) are from the Black Sea DSDP borehole and the remaining 58 samples come from outcrops. The four samples (374.5, 376, 413 and 593 m) from Black Sea core, three samples (samples 1, 2 and 3) from Trilophos, three samples (samples 1, 3, 4, and 5) from Eceabat, five samples (samples 1, 2, 4, 5 and 6) from Burhanlı, two samples (samples 1 and 2) from Ptolemais Notio are barren (containing no or very low number of pollen grains). The high-resolution long-term pollen record of DSDP Site 380 completely covers the last 7 million years. The top 0-308.46 m was analysed by S. Boroi, and the lower part 704.34-1019.85 m by S.-M. Popescu. In this thesis, the studied interval of the DSDP borehole covers the interval between 319.030 and 702.4 meters the middle part of the hole.

Studied locations	Number of analysed samples
DSDP 380 A borehole	378
Enez	8
İntepe	8
Eceabat	5
Burhanlı	6
Seddülbahir	2
Prosilio	6
Ptolemais (Notio&Base)	16
Trilophos	5
Lion Amphipoli	2
Total: 436 samples	

Table 1.1. Study locations and number of samples.

3.1. Sampling and chemical processing

Sampling intervals were taken differently. In DSDP 380 Site, samples were taken approximately at 0.5 m intervals. The outcrop samples have a one meter intervals. The sampling were done always with maximum precaution to avoid the contamination of samples. For the chemical treatment ca. 20 grams sediment was used. The samples were processed using the classical method (Cour, 1974).

The analysis was processed as indicated below:

1. Weighted ca. 20 grams sediment (depending on the sort of sediment),
2. Remove carbonate content of sediment using HCl acid (35%) for 12 hours,
3. Add water twice,
4. Eliminate silicates in the sediment using HF acid (70%) for 24 hours,
5. Add water twice,
6. $ZnCl_2$ (density>2) is used to separate palynomorps in the sediments, and then samples are centrifuged at 1000 r.p.m. for 10 minutes,
7. Add HCl acid (35%) to dissolve minerals which are left during $ZnCl_2$ reaction,
8. Centrifuge at 2500 r. p. m. for 5 minutes,
9. Wash deposited samples 2 times to eliminate $ZnCl_2$ and HCl acid at 2500 r. p. m. for 10 minutes,
10. Sieve the remaining residue using 10 μ m nylon sieve,
11. Centrifuge again to remove remaining water at 2000 r. p. m. for 10 minutes,
12. Add glycerol. The glycerol is added as much as final residue,
13. Calculate the volume of residue sediment with glycerol,
14. Mount samples on slide by placing the residue 50 ml, adding glycerol, covering it with the thin slide cover and sticking it with glue (histolaque).

3.2. Identification of pollen grains

The analysis on the microscope has been performed using two light-transmitted microscopes (alternately in İstanbul and Lyon), Zeiss and Leica with different oil-immersion objectives (x25, x40 and x100). The analysis consisted in identifying and counting pollen grains along several lines. Spores were not considered due to their poor presence in the sediments. The identification was done from end to end parallel to the longest edge of the slide. So, the same pollen grain could never be encountered twice in this way. The pollen grains were counted until a minimum of 150 pollen grains, excluding *Pinus*. Because *Pinus* is generally overrepresented owing to their prolific production and having the ability of transportation in air and water. The botanical identification is made by the study of morphological characters of pollen grains, which are compared with the living relatives. Pollen identification benefited from many pollen photographs, atlases, and also Photopal website (<http://134.214.206.5/photopal>). All pollen data are available on the web from the “Cenozoic Pollen and Climatic values” database (CPC) (<http://134.214.206.5/cpc>). In this study, 107 different taxa were identified. During the analysis, several pollen species were photographed (In Figs. 3.1-3.2). However, some pollen grains could not be identified because of their poor preservation, and so they were defined as indeterminate in the pollen diagrams. All identified taxa and species are shown in Table 2.1. Complete pollen data are presented in the form of synthetic pollen diagrams (Suc, 1984a) and detailed pollen diagrams. In this kind of detailed pollen diagrams, taxa are individually indicated with their percentage. In synthetic pollen diagrams, different taxa are grouped into 12 different groups according to the ecological significance of their living relatives (Table 3.1). Thus, such diagrams allow comparison with the other pollen records obtained from the other localities, such as the European and Mediterranean regions. Moreover, they are also convenient for comparison with oxygen isotope curves in order to contribute to reconstruction of paleoclimate evolution. The groups used in the synthetic diagrams are from left to right:

Megathermic trees	Mesothermic trees		Herbs		Steppe
Avicennia alba Euphorbiaceae Rubiaceae Rutaceae	deciduous Quercus Corylus Betula Alnus	Salix Tamarix Fraxinus Ilex	Poaceae Caryophyllaceae Amaranthaceae-Chenopodiaceae Asteraceae Asteroideae Asteraceae Cichorioideae	Plumbaginaceae Centaurea Mercurialis Alismataceae Scabiosa Viola Saxifragaceae Euphorbia Thalictrum Solanaceae Cannabis Urticaceae Clematis Linaceae Campanulaceae Boraginaceae Oenotheraceae Cistus Galium Knautia	Artemisia Ephedra Hippophae rhamnoides
Mega-mesothermic trees	Carpinus orientalis Juglans Populus Juglans cf. cathayensis Rhus Fagus Pterocarya Ulmus Buxus sempevirens Liquidambar orientalis Carpinus betulus Platanus Hedera Nyssa Carya Oleaceae Vitis	Tilia Celtis Eucommia Lonicera	Plantago Liliaceae Geranium Sparganium Ericaceae Cyperaceae Potamogeton Lamiaceae Thymillaceae Typha Brassicaceae Fabaceae Rumex Polygonum Apiaceae Myriophyllum Convolvulus Erodium Papavareceae	Viola Saxifragaceae Euphorbia Thalictrum Solanaceae Cannabis Urticaceae Clematis Linaceae Campanulaceae Boraginaceae Oenotheraceae Cistus Galium Knautia	Mediterranean xerophytes
Taxodiaceae Engelhardia Taxodium type Platycarya Sapotaceae Distylium Microtropis fallax Ginkgo Loropetalum Arecaceae					Quercus ilex type Olea Ligustrum Myrtaceae Phillyrea
Microthermic trees					Non-significant
Abies Picea					Ranunculaceae Rosaceae
Meso-microthermic trees	Castanea type Acer Zelkova				Cathaya
Cedrus Tsuga					Pinus
					Cupressaceae

Table 2.1. Taxa identified in the study.

PLATE 1

- 1**, *Abies* (Ptolemais Base);
- 2-4**, *Acer* (Ptolemais Base);
- 5-7**, *Galium* (Ptolemais Base);
- 7-11**, *Cistus* (Ptolemais Base);
- 12-14**, *Fagus* (Ptolemais Base);
- 15-16**, Lamiaceae (Ptolemais Base);
- 17-19**, *Myrica* (Ptolemais Base);
- 20**, *Typha* (Ptolemais Base);
- 21**, Rosaceae (Ptolemais Base);
- 22**, *Zelkova* (Ptolemais Base);
- 23**, *Alnus* (Ptolemais Notio);
- 24**, Apiaceae (Ptolemais Notio);
- 25**, Asteraceae Asteroideae (Ptolemais Notio);
- 26**, *Carya* (Ptolemais Notio);
- 27**, Caryophyllaceae (Ptolemais Notio);
- 28**, *Engelhardia* (Enez);
- 29**, deciduous-*Quercus* (Ptolemais Base);

Figure 3.1. Some pollen photos from the studied regions.

PLATE 2

- 30, *Cedrus* (Ptolemais Notio);
- 31, *Tsuga* (Ptolemais Notio);
- 32, *Lonicera* (Ptolemais Notio);
- 33, *Avicennia alba* (DSDP Site 380);
- 34, *Polygonum* (Ptolemais Notio);
- 35, *Tilia* (Ptolemais Notio);
- 36, *Pterocarya* (Ptolemais Base);
- 37, *Carpinus orientalis* (Enez);
- 38, Amaranthaceae-Chenopodiaceae (Enez);
- 39, Taxodiaceae: probably *Glyptostrobus* (DSDP Site 380);
- 40, *Sparganium* (Enez);
- 41, *Artemisia* (DSDP Site 380);
- 42, *Corylus* (Trilophos);
- 43, Poaceae (DSDP Site 380);
- 44, Asteraceae Cichorioideae (Ptolemais Notio).

Figure 3.2. Continued.

-  Megathermic elements (Tropical trees)
-  Mega-mesothermic elements (Subtropical trees)
-  Cathaya
-  Mesothermic elements (Warm-temperate trees)
-  Pinus
-  Meso-microthermic elements (Mid-altitude trees)
-  Microthermic elements (High-altitude trees)
-  Non-significant elements
-  Cupressaceae
-  Mediterranean xerophytes
-  Herbs
-  Steppe elements

Table 3.1. Groups used in synthetic pollen diagrams according to classification of Nix (1982).

- Megathermic (= tropical) elements: *Avicennia alba*, a mangrove tree; Euphorbiaceae, Rubiaceae, Rutaceae, Arecaceae, etc.;
- Mega-mesothermic (= subtropical) elements: mainly Taxodiaceae (including *Taxodium* type and *Glyptostrobus*), *Engelhardia*, Sapotaceae, *Microtropis fallax*, *Distylium*;
- *Cathaya*, a conifer living today at mid-altitude in subtropical China;
- Mesothermic (=warm-temperate) elements: deciduous *Quercus*, *Carya*, *Pterocarya*, *Carpinus orientalis*, *Juglans*, *Juglans* cf. *cathayensis*, *Celtis*, *Zelkova*, *Ulmus*, *Tilia*, *Acer*, *Liquidambar* cf. *orientalis*, *Alnus*, *Salix*, *Populus*, *Fraxinus*, *Buxus sempervirens* type, *Betula*, *Fagus*, *Hedera*, *Lonicera*, *Ilex*, *Tilia*, etc.;
- *Pinus*;
- Meso-microthermic (=mid-altitude) elements: *Cedrus* and *Tsuga*;
- Microthermic (= high-altitude) trees: *Abies* and *Picea*;
- Non-significant elements: some cosmopolitan or widely distributed elements such as Rosaceae and Ranunculaceae;
- Cupressaceae;

- Mediterranean xerophytes: *Quercus ilex* type, *Olea*, *Phillyrea*, *Ligustrum*, etc.;
- Herbs: Poaceae, Amaranthaceae-Chenopodiaceae, Asteraceae Asteroideae, Asteraceae Cichorioideae, *Geranium*, *Convolvulus*, *Erodium*, Lamiaceae, *Plantago*, *Euphorbia*, Brassicaceae, Apiaceae, *Rumex*, *Polygonum*, Cyperaceae Campanulaceae, Ericaceae, Solanaceae, etc.; some halophytes such as Caryophyllaceae, Plumbaginaceae are included within the herbs; some herbs contain water plants such as; *Potamogeton*, *Sparganium* and Typhaeae,
- Steppe elements: *Artemisia*, *Ephedra* and *Hippophae rhamnoides*.

4. CHRONOLOGY OF THE STUDIED SECTIONS

A total of 10 sections and 436 samples have been studied for pollen analysis (in Table 1). The chronology of studied sections are mainly supported by calcareous nannoplankton data and other biostratigraphic data (Table 4.1). The chronostratigraphy of the studied locations are given in Figure 4.1. In the DSDP Black Sea core, firstly, seventeen samples were selected corresponding to warm phases in the pollen diagram for nannofossils. These depths are 219, 223.02, 326.14, 334.50, 368.43, 461.53, 471.50, 476.46, 504.35, 509.35, 518, 548.50, 586.49, 682.95, 708.20, 748.45, and 840.07 mbsf. The eight of the seventeen samples yielded nannofossils. The corresponding depths are 219, 223.02, 326.14, 368.43, 476.46, 504.35, 748.45, and 840.07 mbsf. These chronological limitations are assigned in the sediments using ages of the lowest occurrence (LO), highest occurrence (HO), lower consistent occurrence (LCO) and highest consistent occurrence (HCO) of the species as determined by Raffi *et al.* (2006):

- at 840.07 m depth, *Triquetrorhabdulus rugosus* and *Ceratolithus acutus* are observed. The age is between 5.345 Ma (*C. acutus* LO) and 5.279 Ma (*T. rugosus* HO) (early Zanclean);
- at 748.45 m depth, *Reticulofenestra pseudoumbilicus* displays an age older than 3.839-3.79 Ma (*R. pseudoumbilicus* HO) (late Zanclean);
- at 476.46 m and 504.35 m depth, *Discoaster brouweri* displays an age older than 2.06 – 1.926 Ma (*D. brouweri* HO) (late Gelasian);,
- at 368.43 m depth, medium-sized *Gephyrocapsa* shows an age younger than 1.73 – 1.67 Ma (medium-sized *Gephyrocapsa* spp. LO) (Calabrian),
- at 326.14 m depth, *Helicosphaera sellii* exhibits an age older than 1.34-1.256 Ma (*H. sellii* HO) (Calabrian).
- at depths 223.02 and 219 m, the presence of *Reticulofenestra asanoi* displays that these samples are between 1.136 Ma (*R. asanoi* LCO) and 0.901 Ma (*R. asanoi* HCO) (Calabrian).

Study Location	Country	Age determination	Age	Pollen Analysis
DSDP Site 380	Turkey, near the Bosphorus	Melinte, MC. (personal information)	Pliocene-Lower Pleistocene	S. Boroi, D. Biltekin, S.-M. Popescu
İntepe	Turkey	Melinte et al., 2009	Early-Late Messinian	J.-P. Suc, D. Biltekin
Eceabat	Turkey	Melinte et al., 2009	Early Messinian	D. Biltekin
Burhanlı	Turkey	Melinte et al., 2009	Latest Tortonian-Early Messinian	D. Biltekin
West Seddülbahir	Turkey	Melinte et al., 2009	Latest Messinian-Earliest Zanclean	J.-P. Suc
Enez	Turkey	Melinte et al., 2009	Messinian (after MSC)-Earliest Zanclean	D. Biltekin
Ptolemais Notio and Ptolemais Base	Macedonia	van Vugt et al., 1998, Steenbrink et al., 1999	Early Pliocene	D. Biltekin
Prosilio	Greece	Steenbrink et al., 2000,2006	Late Miocene	D. Biltekin
Lion of Amphipoli	Greece	Melinte, M.C. (personal information)	Early Zanclean	J.-P. Suc
Trilophos	Greece	Melinte, M.C. (personal information)	Latest Messinian-Earliest Zanclean	D. Biltekin

Table 4.1. Age control of the study areas.

Figure 4.1. The chronostratigraphic position of the studied sections from the Late Miocene to the Early Pleistocene.

Samples from the studied sections in the Gulf of Saros (Enez) and Dardanelles regions (İntepe, west Seddülbahir, Burhanlı, Eceabat) were analysed for nanoplankton. The Enez section contain poor to moderate nannoflora in 7 samples. They indicate the co-occurrence of *Triquetrorhabdulus rugosus* and *Ceratolithus acutus*. They represents NN12b nannofossil subzone indicating the extreme end of Messinian (after the MSC) to earliest Zanclean (Melinte *et al.*, 2009). The studied sections in the Dardanelles are correspond with the Messinian erosional surface (Melinte *et al.*, 2009). The nannoflora content of Eceabat indicates poor to moderate preservation and few reworked specimens. They contain *Amaurolithus primus*, *Reticulofenestra pseudumbilicus*, *R. Rotaria* (samples 1, 2, 4, 5), *Nicklithus amplificus* (samples 1 and 3) and *Triquetrorhabdulus rugosus*. This nannoflora indicates NN11c nannofossil subzone. It belongs to an early Messinian age (Melinte *et al.*, 2009). The nannofloral community of west Seddülbahir includes *Triquetrorhabdulus rugosus*, *Reticulofenestra pseudumbilicus*, *Ceratolithus acutus*.

The age of this section is from NN12a to NN12b subzones, from the latest Messinian to the earliest Zanclean (Melinte *et al.*, 2009). In the Burhanlı section, nannoflora community; *Reticulofenestra pseudoumbilicus*, *Reticulofenestra rotaria*, *Triquetrorhabdulus rugosus* and *Nicklithus amplificus* were recorded in the samples. According to this nannoflora assemblages, age of section is the latest Tortonian to early Messinian (upper part : NN11b subzone; lower part: NN 11c subzone). In the İntepe section (samples from 18 to 24), the nannofossil content of sediments are *Amaurolithus primus*, *Reticulofenestra rotaria* (samples 14-18), *Nicklithus amplificus* (samples 1-7), *Triquetrorhabdulus rugosus* (samples 1-31) and *Ceratolithus acutus* corresponding to NN11c and maybe NN12a subzone. The age of samples from Western Macedonia and Northern Greece (Ptolemais Notio and Base, Prosilio, Trilophos and Lion of Amphipoli) were also determined. The age of the Ptolemais Formation is based on paleontological data (small mammals) and magneto- and cyclostratigraphy and $^{40}\text{Ar}/^{39}\text{Ar}$ dating, according to which it is the Early Pliocene (MN 14 and 15) (van de Weerd, 1979; van Vugt *et al.*, 1998; Steenbrink *et al.*, 1999). In Trilophos and Lion of Amphipoli, datation is based on nannofossils. According to this, Lion of Amphipoli is in the Early Zanclean age (Melinte, M.C., personal correspondence). The nannoflora of Trilophos was analysed in five samples.

Only two samples (samples 4 and 5) have nanoplankton content. The nanoplankton assemblages in Trilophos contain *Triquetrorhabdulus rugosus*, *Reticulofenestra pseudoumbilicus*, *Ceratolithus acutus*, which belongs to NN12 a, b subzones (the latest Messinian to the earliest Zanclean) (Melinte, M.C., personal correspondence). The indirect age determination for Prosilio section is based on correlation of thick green clay bed with abundant fish teeth and vertebrate. This bed has been found in the 5 km easterly Lava quarry dated as 6.57 Ma years (Steenbrink *et al.*, 2000). The correlation of polarity sequence in Prosilio section (Late Miocene) with geomagnetic polarity time scale gives the lower normal polarity; subchron C3An.2n, middle normal interval with C3An.1n and the upper interval with Thvera (Steenbrink *et al.*, 2006).

5. RESULTS

5.1. DSDP SITE 380

The Black Sea borehole was taken from near the Bosphorus on the basin apron in a water depth of 2107 meters (southwestern Black Sea). The length of the core is 1073.5 meters (42°05.94'N and 29°36.82'E). In this thesis, 319.03 to 702.4 mbsf interval was studied which correspond to Units (Unit 2 and Unit 4(4a) in Figure 5.1.

5.1.2. LITHOLOGY

The sedimentary sequence of the core is divided into five main units and 13 subunits (Fig. 5.1) (Ross, 1978). These units and subunits are described below:

UNIT 1 (0-332.5 m)

In this unit includes mainly terrigenous sediments. The sediments are silty clay, sandy silts, and rare sands as thin laminae. The silts, sands and muds comprise feldspars, quartz, clay minerals, detrital carbonates, pyrite, organic matter, heavy minerals and diatoms. The clay minerals consists mainly of illite and smectite, with smaller amounts of kaolinite and chlorite. In addition, detrital carbonates exist in a small amounts. Unit 1 contains sediments which were deposited during the marine incursions and the intervening lacustrine periods. This unit is subdivided into five subunits, the first two of which were not recovered by the DSDP coring:

Subunit 1a:

This subunit is generally represented by a 30 cm-thick laminated in Black Sea cores. In general, this subsunit is a typical sediment of the present Black Sea deposited in the last 3000 a. It consists of nannofossil (coccolithophore) ooze (Ross and Degens, 1974).

Subunit 1b (0-2 m):

The Subunit 1b is a dark gray sapropel and rich in organic matter and diatoms. Its age spans from 8 ka to 3 ka.

Subunit 1c (2-42 m):

The subunit 1b includes muds and sandy silts. This subunit is the one of the sandy intervals at this hole. Silty sands compose of micas, detrital carbonates, quartz,

feldspars, heavy minerals, clays, opaque minerals and shell components. The terrigenous sediments are micaceous.

These sediments were deposited under lacustrine, fresh to brackish water environment. This is supported by the presence of fresh-water diatoms, such as; *Stephanodiscus* and *Melosira*.

Subunit 1d (45-76 m):

This subunit constitutes chiefly diatomaceous muds. During the deposition of this unit, marine influence was dominant. Diatoms are generally marine. However, The scarcity of foraminifers and nannofossil communities show that the environment was not under fully marine conditions. The presence of *Gephyrocapsa sp.* indicates that salinity of Black Sea was more than 18‰ (Percival, 1978 and Bukry, 1973). This subunit is the first brackish-marine sequence below the Holocene (Marine Isotope Stage 5e).

Subunit 1e (76-142.5 m):

This subunit includes terrigenous sediments. The silts and muds consist of quartz, feldspars, clay minerals and detrital carbonates. The one interval contains diatoms of some brackish-water species.

Subunit 1f (142.4-171 m):

The sediments of this subunits are greenish gray to dark greenish gray. The Diatoms are abundant. Thin sandy silts layers are common. They are in terms of quartz and feldspar. The content of diatom species indicate brackish-marine condition.

Subunit 1g (171-266 m):

Subunit 1g chiefly contains muds. The thin silt and clay intervals are common. One interval is diatomaceous with fresh-water species. The lithology of this subunit is alike subunits 1c and 1e.

Subunit 1h (266-332.5 m):

This subunit includes muds and turbidite intervals. It includes two diatom-rich levels, containing fresh-water species in 294.5-304 m. This subunit also contains *Ammonia beccarii*, that is abundant at the bottommost part of this subunit. Thus, this indicates that this subunit were deposited in a brackish-marine environment.

Figure 5.1. Lithology of DSDP 380 Site A Black Sea core (Ross, 1978). Climate zones in the diagram: γ (Glacial), B (Interglacial), β (Glacial), A (Interglacial), α (Glacial). Studied interval covers 319.03 to 702.4.

UNIT 2 (332.5-446.5 m)

The Unit 2 is represented by several interbedded of carbonate-rich layers. The upper part of the unit includes aragonitic sediments. The bottom part of the unit contains the lowest occurrence of siderite-rich sediments. Siderite-rich marls are observed in thin layers or laminae. In addition, calcareous oozes are present in calcite.

The other sediment is aragonite. Also the carbonates, sapropelic and diatomaceous muds, laminated and varve-like clays and sandy silts are observed. The dominant lithology of this unit usually is mud. It consists of quartz, feldspars, clay minerals and a little amount of detrital carbonates.

The color varies from greenish gray to olive gray or to dark greenish gray. The darker sediments are rich in terms of pyrite. However, the olive/light olive gray sediments include diatoms and carbonates. The chemical sediments contain aragonite, siderite and calcite. Siderite mostly is in marls. The siderite-rich layers are light olive gray in color and from a few to numerous centimeters thickness.

Calcite-rich marls and oozes or seekride are also represented in the core. The upper section of this unit was deposited under marine-brackish conditions. The presence of the *Braarudosphaera* flora indicates that Black Sea was effected by strong marine influence, the salinity may have reached 22‰ (Percival, 1978 and Bukry, 1973). The oldest sediments of this unit may have been formed in a fresh or brackish lake.

UNIT 3 (446.5-644.5 m)

This unit is featured by the existence of seekreide. The dominant content of this unit are muds, marls and seekreide. The upper section of the unit still includes siderite. The bottom part of this unit is placed just above the diatomaceous ooze. Also this unit consists of an ostracode fauna such as; *Candona-Loxoconcha* assemblages that show the deposition in fresh-water lakes (Benson, 1978 and Olteanu, 1978).

UNIT 4 (644.6-969 m)

This unit contains the various sediment types including also chemical sediments. The upper part of the unit includes siderite layer in 664.6 meters. The bottom of the unit is transitional to an underlying black shale. In this unit, diatoms and carbonate-rich sediments are also present. Besides, it is noteworthy to mark the presence of pebbly mudstones and breccias. This unit is divided into five subunits:

Subunit 4a (644.6-718 m):

This subunit is represented by the existence of several manganosiderite intercalations in diatomaceous clay. In the upper and the bottom levels are characterized by the siderite layers. The main lithology of this subunit are mud, diatomaceous clay and the terrigenous sediments. In the upper section, clays are usually structureless. However, at the lower part, they have some distinct laminations. The structureless clay levels contain chiefly clay minerals and numerous amounts of diatoms. The main clay mineral is smectite with illite, and a small amount of kaolinite and chlorite. The laminated layers contain diatom-rich and clay-rich lamina. At the same time, siderite is common in this subunit. The siderite-rich sediments are pale olive color, thin layers or as nodules. The diatom content of the subunit suggests that depositional environment was possibly from fresh to brackish (Schrader, 1978). Besides, dinoflagellates are present abundantly. These suggest the marine flux period.

Subunit 4b (718-850.3 m):

This subunit constitute seekreide. Diatomaceous clays and marls are also presented. The upper boundary this subunit is represented by the base of the siderite. The main sediments in this subunit are clays, calcite (chemical) and diatoms. The seekreide varves contain calcite-rich light greenish gray and clay-rich dark greenish gray lamina. The diatomite varves comprise light olive green calcite-bearing diatomaceous marl and darker olive green diatomaceous clay. At the base, dinoflagellates and acritarchs are common. The toward to upper part, they are presented as a less amount.

Subunit 4c (850.3-864.5 m):

This subunit is defined by the existence of aragonite and magnesian calcite. The main lithology of the subunit is diatomaceous shales in olive-black color. The sediments in this subunit were formed under brackish-marine condition.

The abundance of *Braarudosphaera* flora shows that the salinity may have reached 22‰ (Percival, 1978 and Bukry, 1973). The existence of *Bolivina* indicates stenohaline conditions (Gheorghian, 1978). Dinoflagellates and acritarchs are common that indicate marine influence.

Subunit 4d (864.5-883.5 m):

The subunit is characterized by pebbly mudstones, stromatolitic dolomites and conglomerates. The dolomite was deposited such in an intertidal to supratidal environment (Stoffers, 1978). The sea level in the Black Sea was possibly very shallow. This is also supported by shallow habitat diatoms (Schrader, 1978). The major lithology is conglomerate, slump breccia or pebbly mudstone.

Subunit 4e (838.5-969 m):

This subunit contains dolomite which is in the form of laminated seekreide and marls. The upper part of this subunit is above the base of the pebbly mudstone. The main lithology is calcareous mud or marl. It has dark greenish color. Interbedded in the marl sequence are calcitic, aragonitic and dolomitic sediments. These chemical sediments form in three distinct sediments content: laminated marl, carbonate varves and dolomite.

UNIT 5 (969-1073.5 m)

This unit is characterized by the presence of black shales with zeolitic sandstones and dolomite. The black shales are in greenish black and fissile. They contain clays, organic matter, quartz, feldspars and pyrite. The content of small benthic foraminifers in the sequence indicate that the black shales were deposited in a brackish-marine environment.

5.1.3. High-resolution pollen record of DSDP 380

In DSDP Site 380 Black Sea hole, studied interval is from 319,03 m to 702,40 meters below sea floor (mbsf) corresponding to 2.15-4.6 Ma time interval. High-resolution pollen analysis of the core provides significant data for paleovegetation and paleoclimate during early Pliocene to early Pleistocene (from Zanclean to Gelasian). The vegetation is characterized by different plants groups. Most of them are inherited from Miocene. Along the study intervals, the flora is dominated by mostly two vegetation type (Fig. 5.3). They are thermophilous plants and herbs characterizing steppe. Thermophilous vegetation is characterized by megathermic (tropical), megamesothermic (subtropical) and mesothermic (warm-temperate) elements.

Megathermic elements are Euphorbiaceae, Rubiaceae, Arecaceae, Rutaceae and *Avicennia alba*. However, they are in small amount in the studied intervals. Mega-mesothermic elements are characterized by Taxodiaceae (chiefly *Glyptostrobus*), type *Taxodium*, *Engelhardia*, *Platycarya* and Sapotaceae. Among them, particularly Taxodiaceae swamp forests show high abundances. Percentage of Taxodiaceae reaches up around 80% (Figs. 5.2-5.3). Mesothermic elements contain deciduous *Quercus*, *Betula*, *Corylus*, *Juglans*, *Pterocarya*, *Buxus sempervirens*, *Liquidambar orientalis*, *Nyssa*, *Acer*, *Castanea*, etc.

This an evergreen and deciduous mixed forest include a riparian vegetation composed of *Salix*, *Alnus*, *Carpinus orientalis*, *Zelkova*, *Carya*, *Ulmus*, etc. The mesothermic plants are presented frequently along studied intervals. *Quercus* reaches up 10% in some levels and *Zelkova* reaches up 11%. Herbs are involved by mainly Amaranthaceae-chenopodiaceae, Poaceae, Asteraceae-astroideae, Asteraceae-cichorioideae, Brassicaceae, Caryophyllaceae, Lamiaceae, *Rumex*, Apiaceae, *Centaurea*, etc. At the same time, herbs include some fresh-water plants such as; *Sparganium*, *Potamogeton*, *Typha*, *Myriophyllum*. Among herbs, Amaranthaceae-chenopodiaceae has higher amount. Its percentage approaches 88%. Steppe elements comprise mainly *Artemisia*, *Ephedra* and *Hippophae rhamnoides*. *Artemisia* is represented abundantly with a percentage of around 84% (Figs. 5.2-5.3). The rest of steppe elements (*Ephedra* and *Hippophae rhamnoides*) do not vary significantly.

Figure 5.2. Detailed pollen diagram of DSDP Site 380 between 319.03-460m.

Figure 5.3. Detailed pollen diagram of DSDP Site 380 between 460.54-702.4 m.

In addition, altitudinal coniferous trees such as; microthermic elements (*Abies* and *Piceae*) and meso-microthermic elements (*Cedrus* and *Tsuga*) appear in a small number. Also, *Cathaya* has a less amount. *Cathaya* is a gymnosperm, living today in the subtropical mid-altitude forest of southern China.

Pollen groups in the synthetic pollen diagram are: 1; Megathermic elements (*Avicennia alba*, Rutaceae, Arecaceae, Rubiaceae), 2; Mega-mesothermic elements (Taxodiaceae, *Engelhardia*, *Taxodium* type, *Platycarya*, Sapotaceae), 3; *Cathaya*, 4; Mesothermic elements (deciduous *Quercus*, *Betula*, *Alnus*, *Carya*, *Pterocarya*, *Zelkova*, *Ulmus*, *Fagus*, etc.), 5; *Pinus*, 6; Meso-microthermic elements (*Cedrus* and *Tsuga*), 7; Microthermic elements (*Abies* and *Piceae*), 8; Cupressaceae, 9; Herbs (Amaranthaceae-chenopodiaceae, Poaceae, Asteraceae Astroideae, Asteraceae Cichorioideae, Brassicaceae, Caryophyllaceae, Lamiaceae, *Rumex*, Apiaceae, *Polygonum*, etc., 10; Steppe elements (*Artemisia*, *Ephedra* and *Hippophae rhamnoides*). Mediterranean xerophytes and non-significant elements have small amounts. Therefore, they were excluded from the synthetic pollen diagram. *Pinus* is represented abundantly. Its percentage gets at 53%. The pollen record of the core enables the identification of the different vegetation stages with subdivisions. Additionally, these stages are correlated with oxygen isotope curve (with MIS). The stages are described below (Popescu *et al.*, 2010):

Pollen zone 2 (624-702,40 meters): This zone is divided into three subzones (2a, 2b, 2c) (Fig. 5.4). These pollen subzones are defined by Zagwijn, 1960; Zagwijn and Suc, 1984. Pollen zone 2 is characterized by the abundance of herbs (mainly Amaranthaceae-chenopodiaceae). This show that drier climate conditions existed during this time. On the contrary, thermophilous trees are not very much. Nevertheless, *Artemisia* steppes also have a decrease in this zone. At the top of zone 2, *Artemisia* displays an increase.

Pollen zone 3 (603-624 meters): In this zone, herbs are abundant with *Artemisia*, reaching up 59%. On the other hand, thermophilous plants have a decreased amounts. This time intervals corresponds the earliest glacials in the Northern Hemisphere. This zone matches Marine Isotope Stages (MIS 104-96).

Figure 5.4. The synthetic pollen diagram of DSDP Site 380 obtained in this study. Pollen groups in the diagram: 1; Megathermic elements, 2; Mega-mesothermic elements, 3; *Cathaya*, 4; Mesothermic elements, 5; *Pinus*, 6; Meso-microthermic elements, 7; Microthermic elements, 8; Cupressaceae, 9; Herbs, 10; Steppe elements (see for explanation in the next page). The synthetic pollen diagram with oxygen isotope curve showing Marine Isotope Stages (Shackleton *et al.*, 1990, 1995), pollen zones, NW European climatostratigraphy (Zagwijn, 1960, 1998) and nannofossil biohorizons (Raffi *et al.*, 2006). Chronostratigraphy, Lourens *et al.* (2004).

Pollen zone 4 (603-461 meters): Pollen zone 4 is divided into three subzones. These are 4a, 4b and 4c. This zone is dominated by chiefly thermophilous forests. Among them, subtropical trees are particularly abundant. Thermophilous trees quickly are sequenced by herbs. In this zone, herb elements are represented abundantly. *Artemisia* is not abundant. Its percentage is around 11% in average. This zone corresponds the Tiglian warm period (Zagwijn, 1960; 1963; Zagwijn and Suc, 1984).

Pollen zone 5 (395-461 meters): In this zone, herbs are prominent (mostly Amaranthaceae-chenopodiaceae). With herbs, *Artemisia* shows higher frequency. Subtropical trees do not change very much. Besides, mesothermic elements have an increase. This zone corresponds to the Eburonian stage (MIS 62-50) (Zagwijn, 1975).

Pollen zone 6 (319,030-392,010 meters): In this zone, Herbs are abundant with *Artemisia* steppes. Nevertheless, mega-mesothermic elements and mesothermic elements are prominent. This zone corresponds Waalian phase. The subdivisions are 6a and 6b in the study intervals.

The other important result is *Avicennia alba* (Mangrove) and some tropical Euphorbiaceae plants. *Avicennia* was observed between 781.63 meters and 1018.85 meters in the lower part of the section (Popescu, 2006). In this study, it was observed at 412.53 meters. *Avicennia* disappeared from North Mediterranean 14 Ma ago (Serravalian) and from North Africa at 5.3 Ma (Chikhi, 1992; Bachiri Taoufiq et al., 2000). According to these results, thermophilous plants persisted up in the region than in the other regions of Mediterranean. Thermophilous plants were well recorded during the Pliocene time and the Pleistocene (Fig. 5.5). Relict plants such as *Carya*, *Carpinus orientalis*, *Pterocarya*, *Liquidambar orientalis*, *Zelkova* are still living in the Anatolia. This situation can be explained by the Asian monsoon climate effect.

Figure 5.5. The distribution of thermophilous trees during the Pliocene-Pleistocene in the DSDP Site 380.

5.2. Gulf of Saros

5.2.1. Enez

Enez is located in the eastern shoreline of the Enez lagoon (40°46'24" N, 26°04' E) near the border of Turkey with Greece. The study area includes the Pliocene and the Late Quaternary deltaic deposits of Meriç River overlying the Kirazlı Formation (Çağatay *et al.*, 1998, 2006). The seven samples of eight brownish clay sediments (thickness ~8 m) from the Pliocene bottomset deposits in the Enez section are rich in pollen grains. The flora are characterized by the herbaceous vegetation and warm-temperate trees in all the samples (mesothermic elements) (Fig. 5.6). Herbs reach up more than 75% in the lowest part of the section. Inside this group; Poaceae, Amaranthaceae-chenopodiaceae, Asteraceae-asteroideae, Asteraceae-cichorioideae are abundant in the samples. The other herbs elements are: Caryophyllaceae, Plumbaginaceae, Solanaceae, *Scabiosa*, Papavareceae, *Centaurea*, etc. Herbs elements also contain some water plants such as; *Potamogeton* and *Sparganium*.

Figure 5.6. The synthetic pollen diagram of Enez section. Note that only the samples with statistically significant pollen (minimum 150) numbers were analysed. The numbers in the diagram show the pollen groups: 1; megathermic elements (*Arecaceae*, *Sapotaceae*), 2; mega-mesothermic elements (*Taxodiaceae*, *Engelhardia*, *Ginkgoaceae*, *Loropetalum* and *Distylium*), 3; *Cathaya*, 4; mesothermic elements (*Quercus*, *Carya*, *Pterocarya*, *Zelkova*, *Carpinus orientalis*, *Alnus*, *Ulmus*, *Corylus*, etc.), 5; *Pinus*, 6; meso-microthermic elements (*Cedrus*, *Tsuga*), 7; microthermic elements (*Abies*, *Piceae*), 8; *Cupressaceae*, 9; herbs (*Asteraceae-Asteroidae*, *Asteraceae-Cichorioideae*, *Poaceae*, *Amaranthaceae-chenopodiaceae*, *Brassicaceae*, *Plumbaginaceae*, etc.) and include some water plants (*Sparganium*, *Potamogeton*), 10; steppe elements (*Artemisia*).

Mesothermic elements (warm-temperate trees) are characterized by the abundance of deciduous *Quercus* (reaches up 8%). The other mesothermic elements in the samples are presented by *Alnus*, *Carya*, *Pterocarya*, *Liquidambar orientalis*, *Corylus*, *Zelkova*, *Ulmus*, *Carpinus orientalis*, *Juglans*, *Buxus sempervirens*, etc. In addition, alitudinal trees (mainly *Tsuga* and *Abies*, *Cedrus*, *Picea*) are also presented frequently in the samples.

Figure 5.7. The distribution of thermophilous trees in Enez section during the end of the Messinian (after MSC)-the earliest Zanclean.

Megathermic trees are rare in the sediments. Mega-mesothermic elements are not abundant in the samples. They are frequent in samples 4 and 6. Among them; Taxodiaceae, *Engelhardia*, Ginkgoaceae, *Loropetalum* and *Distylium* were recorded. *Ginkgo* (gymnosperm) was observed rarely in sample 7. This plant disappeared from Europe 1.7-2.7 Ma ago. They are living in subtropical China today (Gong *et al.*, 2008). *Cathaya* is recorded frequently in samples 1 and 6. *Pinus* is also abundant in the samples and its abundance increases towards the top of the section. Cupressaceae is frequent in samples 1, 3 and 6. The augmentation in herbs groups in the lower part of the Enez section indicates more open and drier conditions. Thermophilous plants dominated during the end of the Messinian and the earliest Zanclean in Enez section (Fig. 5.7). All taxa in the samples are shown in the detailed pollen diagram (Fig. 5.8).

Figure 5.8. The detailed pollen diagram of Enez section.

5.3. Dardanelles Strait

5.3.1. İntepe

İntepe is located in the south of Dardanelles Strait (40°1'27" N, 26°20'33" E, Fig. 1.2). İntepe section possesses approximately 77 m thickness (Gillet *et al.*, 1978; Sakıncı and Yaltrak, 2005). The 36 m of İntepe section was studied in this study. Studied section includes clays, sands, calcareous sandstones and thin limestones (Fig. 5.9). Upper part of İntepe section comprises yellowish sands and pebbly sandstones. *Maetra* shells abundant in the section. In addition, *Melanopsis* shells are observed. Middle part of the section contains a 5 cm thick lignite which correspond to the unconformity related to the MSC (Gillet *et al.*, 1978). The unconformity is overlain by a 2 cm thick sand and *Maetra*. The sediments were dated by nannofossil. The nannofossil content of sediments are *Amaurolithus primus*, *Reticulofenestra rotaria* (samples 14-18), *Nicklithus amplificus* (samples 1-7), *Triquetrorhabdulus rugosus* (samples 1-31), *Ceratolithus acutus* (Melinte *et al.*, 2009). According to nannoflora, the samples (between 18 and 24) belong to NN11c and probably NN12a subzone (Melinte *et al.*, 2009). Among the samples, only 8 samples are rich in terms of pollen grains. Pollen analysis of İntepe section shows that changes in vegetation before and after the Messinian Salinity Crisis (MSC). While the vegetation is characterised by herbs and arboreal trees (chiefly warm-temperate trees) before the MSC, besides herbs and arboreal trees, subtropical elements, mid- and high altitude trees, *Pinus* and *Cathaya* also display an increase after the MSC. Herbs are dominated by Poaceae, Asteraceae-Asteroideae, Asteraceae-Cichorioideae, and Amaranthaceae-chenopodiaceae. Among them, Poaceae reaches up 20% and Amaranthaceae-chenopodiaceae reaches up 10% (Fig. 5.10). Arboreal vegetation is dominated by warm-temperate trees such as; *Quercus*, *Zelkova*, *Carpinus orientalis*, *Carya*, *Pterocarya*, *Acer*, *Ulmus*, *Juglans*, etc. Nevertheless, subtropical elements (Taxodiaceae, *Engelhardia*) have low amount before the MSC. After the MSC, increase in the mega-mesothermic trees, *Pinus* and altitudinal trees such as, *Cedrus*, *Abies* and *Picea* may demonstrate more distal location with respect to the paleoshoreline. On the other hand, some aquatic plants (*Sparganium*, *Potamogeton*, *Typha*, *Myriophyllum* and Alismataceae) are well represented after the MSC. Hence, presence of Amaranthaceae-chenopodiaceae and aquatic plants demonstrate nearby coastal environments in the region after the MSC (Fig. 5.9).

Figure 5.9. The synthetic pollen diagram of İntepe section with a lithological log. Note that only the samples with statistically significant pollen (minimum 150) numbers were analysed. The numbers in the diagram show the pollen groups: 1; mega-mesothermic elements (*Taxodiaceae*, *Engelhardia*), 2; *Cathaya*, 3; mesothermic elements (*Quercus*, *Carya*, *Pterocarya*, *Zelkova*, *Carpinus orientalis*, *Alnus*, etc.), 4; *Pinus*, 5; meso-microthermic elements (*Cedrus*), 6; microthermic elements (*Abies* and *Picea*), 7; mediterranean xerophytes 8; herbs (*Asteraceae-Asterioideae*, *Asteraceae-Cichorioideae*, *Poaceae*, *Amaranthaceae-chenopodiaceae*, *Apiaceae*, etc.), and include some fresh water plants (*Sparganium*, *Potamogeton*, *Typha*, etc), 9; steppe elements (*Artemisia*, *Ephedra*).

Figure 5.10. The detailed pollen diagram of İntepe section.

The increasing of altitudinal trees (mainly *Cedrus*, *Abies* and *Picea*) could indicate uplift of the area during the Late Messinian (Melinte *et al.*, 2009). Calculation of mean annual temperature from paleoclimatic transfer function based on pollen assemblages (Fauquette *et al.*, 1998, 1999) in order to estimate minimum palaeoaltitude of the nearby massif. The result indicate that in the İntepe samples (from 22 to 26), mean annual temperature is 16.5°C (range: 15-18.5°C). Thermophilous plants are well observed before and after the MSC (Fig. 5.11).

Figure 5.11. The distribution of thermophilous plants in İntepe section before and after the Messinian Salinity Crisis (MSC).

Moreover, dinoflagellate cyst flora of İntepe section was analysed for reconstruction of coastal marine environment. They contain 12 taxa. *Pediastrum* (fresh water algae) and *Botryococcus* can be added to dinoflagellate cyst. They indicate a fresh water input. Dinoflagellate community include oceanic species, neritic species : *Spiniferites mirabilis*, *Spiniferites membranaceus* and marine autotrophic cosmopolitan species: *Lingulodinium machaerophorum*, *Operculodinium centrocarpum* sensu Wall&Dale, *Spiniferites bentorii*, *Spiniferites bentorii* subsp. *truncates*, *Spiniferites hyperacanthus*, *Spiniferites ramosus*, *Spiniferites bulloideus*, *Spiniferites* spp (Melinte *et al.*, 2009).

Besides, dinoflagellate assemblages contain Paratethyan brackish species such as; *Galeacysta etrusca* and *Impagidinium globosum*. Samples from 15 to 19, marine euryhaline species dominated at that time (*Operculodinium centrocarpum*, *Lingulodinium machaerophorum*, *Spiniferites bulloideus*, *Spiniferites ramosus*, *Spiniferites bentorii*, *Spiniferites bentorii* subsp. *truncates*, *Spiniferites hyperacanthus*). The relative abundance of *Spiniferites* spp. reaches up above 50%. The presence of this species indicates marine conditions. Sample 15 contains only Paratethyan brackish species. Samples from 20 to 23, *Pediastrum* indicates an increase. This suggests that fresh-brackish water conditions dominated during that time with less than 4.6 ‰ salinity, sometimes the marine interrupting conditions. Samples 24 and 26 indicate an increase of salinity once again (Melinte *et al.*, 2009).

5.3.2. West of Seddülbahir

West of Seddülbahir (40°02'38" N, 26°10'55" E, Fig.1.2), the studied section consists of ~30 m thick clays. They are rich in molluscs. The nannofloral community of west Seddülbahir includes *Triquetrorhabdulus rugosus*, *Reticulofenestra pseudoumbilicus*, *Ceratolithus acutus*. This section belongs to NN12a and NN12b subzones, corresponding to the Latest Messinian to the Earliest Zanclean (Melinte *et al.*, 2009). Only sample 5 and 3 are rich in pollen grains. The flora of west of Seddülbahir are characterized by herbs, mesothermic elements, meso-microthermic elements and *Cathaya* (Fig. 5.12). Non-boreal flora includes herbs with smaller amount of steppe elements (only *Ephedra*). Herbs are Amaranthaceae-chenopodiaceae, Poaceae, Asteraceae Asteroideae, Asteraceae Cichorioideae, Apiaceae, Caryophyllaceae, Ericaceae, *Rumex*, Saxifragaceae, etc., and include some fresh water plant such as *Typha*. Mid-altitudinal trees (mainly *Cedrus* and *Tsuga*) are abundant in the region after the MSC. This suggests that some uplift events occurred in the region after the Messinian Salinity Crisis. *Pinus* conifer pollen is also abundant during that time. They are overspread due to its prolific character. Nevertheless, *Cathaya* is abundant with subtropical plants (Taxodiaceae, *Distylium*, *Microtropis fallax* and *Engelhardia*) in west of Seddülbahir.

Figure 5.12. The synthetic pollen diagram of west of Seddülbahir section with a lithological log. Note that only the samples with statistically significant pollen (minimum 150) numbers were analysed. The numbers in the diagram show the pollen groups: 1; mega-mesothermic elements (*Taxodiaceae*, *Distylium*, *Microtropis fallax* and *Engelhardia*), 2; *Cathaya*, 3; mesothermic elements (*Quercus*, *Carya*, *Pterocarya*, *Zelkova*, *Carpinus orientalis*, *Alnus*, etc.), 4; *Pinus*, 5; meso-microthermic elements (*Cedrus* and *Tsuga*), 6; microthermic elements (*Abies* and *Picea*), 7; Cupressaceae, 8; mediterranean xerophytes 9; herbs (Asteraceae-Asteroidae, Asteraceae-Cichorioideae, Poaceae, Amaranthaceae-chenopodiaceae, Apiaceae, etc., and include some fresh water plant (*Typha*)).

Among mesothermic elements, *Quercus*, *Carpinus orientalis*, *Pterocarya*, *Juglans*, *Betula*, *Zelkova*, *Carpinus betulus*, *Ulmus*, *Buxus sempervirens*, etc. were recorded. High altitudinal trees *Abies* and *Picea* were observed rarely. Mediterranean xerophytes such as *Quercus ilex*, *Ligustrum*, *Olea* and *Phillyrea* are presented frequently. Cupressaceae does not vary significantly, but it is presented frequently. All pollen taxa in west of Seddülbahir are shown in the detailed pollen diagram (Fig. 5.13).

Figure 5.13. The detailed pollen diagram of west of Seddülbahir.

5.3.3. Eceabat

At Eceabat (40°11'30" N, 26°21'18" E, Fig.1.2), four samples are the whitish clayey base (20 m thick) belonging to Kirazlı Formation and one sample (10 m higher) is a clayey intercalation with calcareous tabular deposits belonging to Alçitepe Formation (Sakinç *et al.*, 1999). Eceabat nannoflora exhibits poor to moderate preservation and few reworked specimens. They include *Amaurolithus primus*, *Reticulofenestra pseudoumbilicus*, *R. Rotaria* (samples 1, 2, 4, 5), *Nicklithus amplificus* (samples 1 and 3) and *Triquetrorhabdulus rugosus*. These nannoflora belong to the NN11c nannofossil subzone, corresponding to an Early Messinian age (Melinte *et al.*, 2009). Five samples were analysed palynologically. However, only one sample (sample 2) was rich in pollen grains (Fig. 5.14). Pollen flora in Eceabat is characterized by mainly herbs elements. In this group, they are dominated by mainly Amaranthaceae, Chenopodiaceae, Asteraceae-Asteroidae, Asteraceae-Cichorioideae, Poaceae, etc. with fresh water plant *Sparganium*. Steppe elements are represented by *Artemisia* and *Ephedra*. Subtropical trees are indicated by Taxodiaceae and *Engelhardia*. Taxodiaceae is presented frequently. *Engelhardia* has less amount. Among mesothermic elements (warm-temperate trees) *Quercus*, *Carya*, *Zelkova*, *Alnus*, *Hedera*, *Salix* are observed. Mid-altitudinal tree *Tsuga* is rare as well as Mediterranean xerophytes and Cupressaceae in the sample 2. *Pinus* is also presented abundantly.

Figure 5.14. The synthetic pollen diagram of Eceabat section with a lithological log. Note that only the samples with statistically significant pollen (minimum 150) numbers were analysed. The numbers in the diagram show the pollen groups: 1; mega-mesothermic elements (*Taxodiaceae* and *Engelhardia*), 2; mesothermic elements (*Quercus*, *Carya*, *Zelkova*, *Alnus*, etc.), 3; *Pinus*, 4; meso-microthermic elements (*Tsuga*), 5; herbs (*Asteraceae* *Asteroideae*, *Asteraceae* *Cichorioideae*, *Poaceae*, *Amaranthaceae* *Chenopodiaceae* etc., and include fresh water plant *Sparganium*.

5.3.4. Burhanlı

Burhanlı is located in the Gelibolu Peninsula (40°18'17" N, 26°33'08" E, Fig.1.2). The nannoflora community, *Reticulofenestra pseudoumbilicus*, *Reticulofenestra rotaria*, *Triquetrorhabdulus rugosus* and *Nicklithus amplificus*, were found in the samples. According to this nannoflora assemblages, age of the samples ranges from the Latest Tortonian to early Messinian (upper part : NN11b subzone; lower part: NN 11c subzone). Six clay samples with alternating sands belonging to Kirazlı Formation were analysed palynologically. However, only one sample (sample 3) has an enough pollen grains. The vegetation is dominated by herbs (Fig. 5.15). They are recorded by chiefly *Asteraceae* *Cichorioideae*, *Asteraceae* *Asteroideae*, *Amaranthaceae* *Chenopodiaceae*, *Poaceae*, etc. with fresh water plant *Potamogeton*. Warm-temperate trees (*Carpinus orientalis*, *Zelkova*, *Liquidambar orientalis*, *Carpinus betulus*, *Ulmus* and *Carya* are represented frequently. In addition, subtropical trees (*Taxodiaceae* and *Engelhardia*) and mid-altitudinal trees (*Cedrus*) are observed frequently in the sample 3.

Figure 5.15. The synthetic pollen diagram of Burhanlı section with a lithological log. Note that only the samples with statistically significant pollen (minimum 150) numbers were analysed. The numbers in the diagram show the pollen groups: 1; mega-mesothermic elements (*Taxodiaceae* and *Engelhardia*), 2; mesothermic elements (*Carya*, *Zelkova*, *Carpinus orientalis*, *Liquidambar orientalis*, etc.), 3; *Pinus*, 4; meso-microthermic elements (*Cedrus*), 5; herbs (*Asteraceae-Asterioideae*, *Asteraceae-Cichorioideae*, *Poaceae*, *Amaranthaceae-chenopodiaceae* etc., and include fresh water plant *Potamogeton*, 6; steppe elements (*Artemisia* and *Hippophae rhamnoides*).

Pinus is abundantly recorded, percentage of *Pinus* reaches up 30%. Steppe elements (*Artemisia* and *Hippophae rhamnoides*) are also represented frequently. High altitudinal trees (*Picea*) and Rosaceae (non-significant elements) are rare in sample 3.

5.4. Western Macedonia

5.4.1. Ptolemais Notio

The sediments of Ptolemais Notio region cover the early Pliocene age. The vegetation in the region is characterized by the herbs and mesothermic elements (Fig. 5.16). Herbs are dominated chiefly by Poaceae, Asteraceae Asteroideae, Asteraceae Chichorioideae, Amaranthaceae Chenopodiaceae and *Sparganium*. The mesothermic elements are characterized by the abundances of *Alnus*, more than 10% (Fig. 5.18).

Figure 5.16. The synthetic pollen diagram of Ptolemais Notio. Note that only the samples with statistically significant pollen (minimum 150) numbers were analysed. The numbers in the diagram show the pollen groups: 1; mega-mesothermic elements (*Taxodiaceae*, *Engelhardia*), 2; mesothermic elements (*Quercus*, *Carya*, *Pterocarya*, *Zelkova*, *Carpinus orientalis*, *Fraxinus*, *Alnus*, etc.), 3; *Pinus*, 4; meso-microthermic elements (*Cedrus* and *Tsuga*), 5; microthermic elements (*Abies* and *Picea*), 6; non-significant (*Ranunculaceae*), 7; Cupressaceae, 8; herbs (*Asteraceae Asteroideae*, *Asteraceae Chichorioideae*, *Poaceae*, *Amaranthaceae Chenopodiaceae*, *Apiaceae*, *Polygonum*, etc. and include some water plants (*Sparganium*, *Potamogeton*, *Typha*), 9; steppe elements (*Artemisia*, *Hippophae rhamnoides*).

The other evergreen-deciduous mixed forest, such as deciduous *Quercus*, *Betula*, *Carpinus orientalis*, *Zelkova*, *Tilia*, etc., are present in the detailed pollen diagram (Fig. 5.18). The deciduous *Quercus* is present in abundance with percentage of deciduous *Quercus* reaching up to 10% in the samples. Megathermic trees such as *Taxodiaceae* and *Engelhardia* are commonly presented frequently. *Poaceae* is more than 20%, and the other herbs such as, *Asteraceae Asteroideae*, *Asteraceae Chichorioideae* have been observed abundantly.

Mid-altitude and high altitude trees (such as; *Cedrus*, *Tsuga*, *Abies* and *Picea*) are present commonly. Nevertheless, *Cathaya* do not vary significantly during this time. Besides, *Pinus* is abundant and reaches up 20%. Cupressaceae and non-significant

elements (Ranunculaceae) are not much abundant. Mediterranean xerophytes are very scarce in the samples. Because of this, it is not included in the synthetic pollen diagram. Steppe elements are common and represented in the synthetic pollen diagram. *Artemisa* is frequent. However, *Hippophae rhamnoides* is rare in the samples. Thermophilous plants were well recorded in the Ptolemais Notio section. The distribution of them during the Early Pliocene are shown in Figure 5.17.

Figure 5.17. The distribution of thermophilous plants during the Early Pliocene in

Figure 5.18. Detailed pollen diagram of Ptolemais Notio.

5.4.2. Ptolemais Base

The vegetation of Ptolemais Base region is characterized by mainly warm-temperate trees, herbs and steppes (Fig. 5.19). The evergreen-deciduous mixed forest includes deciduous *Quercus*, *Alnus*, *Betula*, *Carpinus orientalis*, *Zelkova*, *Liquidambar*, *Oleaceae*, *Fagus* etc. Some of them, especially, deciduous *Quercus* has high abundance during this time. Percentage of deciduous *Quercus* reaches up 36% in the samples. Also, *Fagus* is commonly present with its percentage reaching up to 6%.

Figure 5.19. The synthetic pollen diagram of Ptolemais Base. The numbers in the diagram show the pollen groups: 1; mega-mesothermic elements (*Taxodiaceae*, *Engelhardia*), 2; mesothermic elements (*Quercus*, *Carya*, *Pterocarya*, *Zelkova*, *Carpinus orientalis*, *Liquidambar*, *Fraxinus*, *Alnus*, etc.), 3; *Pinus*, 4; meso-microthermic elements (*Cedrus* and *Tsuga*), 5; microthermic elements (*Abies* and *Picea*), 6; non-significant (*Ranunculaceae* and *Rosaceae*), 7; mediterranean xerophytes (*Olea*, *Quercus ilex* type), 8; herbs (*Asteraceae-Asterioideae*, *Asteraceae-Cichorioideae*, *Poaceae*, *Amaranthaceae-chenopodiaceae*, *Cistus*, etc.) and include some water plants (*Sparganium*, *Potamogeton*, *Typha*), 9; steppe elements (*Artemisia*, *Ephedra*).

Herbs also became more abundant with steppes (mainly *Artemisa* and *Ephedra*). *Poaceae* is abundant with an average of 17%, and *Amaranthaceae* *Chenopodiaceae* has been observed frequently. Mega-mesothermic (*Taxodiaceae* and *Engelhardia*) elements are not abundant. Mid-altitude and high altitude trees (i.e., *Cedrus*, *Tsuga*, *Abies* and *Picea*) are also frequently present.

The average percentages of *Cedrus* and *Abies* reach up 4% and 3% respectively. *Cathaya* and *Cupressaceae* do not vary significantly during this time. *Ranunculaceae* and *Rosaceae* (non-significant elements) are frequent.

Mediterranean xerophytes (*Olea* and *Quercus ilex*) are present in small amounts in the samples. Thermophilous trees were observed in the samples during the Early

Pliocene in Ptolemais Base. They are exhibited in Figure 5.20. The high abundance of mesothermic plants and herbs with steppes may be due to warm-temperate climate conditions in the region. *Pinus*, a conifer tree, is very abundant, probably due to the capacity of its saccate pollen for long distance transport (Heusser 1988; Suc & Drivaliari, 1991; Cambon *et al.*, 1997; Beaudouin, 2003). Moreover, all taxa with relatively high percentages are represented in the detailed pollen diagram (Fig. 5.21).

Figure 5.20. The distribution of thermophilous trees during the Early Pliocene in Ptolemais Base section.

Figure 5.21. Detailed pollen diagram of Ptolemais Base.

5.5. Northern Greece

5.5.1 Trilophos

Trilophos is located (40°45'90" N, 22°98'57" E, Fig.1.2) The section belongs to NN12 biozone corresponding to the Latest Messinian-Earliest Zanclean. The vegetation is represented by the abundance of warm-temperate trees, herbs and altitudinal elements (Fig. 5.22). The evergreen-deciduous mixed forest, such as deciduous *Quercus*, *Alnus*, *Betula*, *Carpinus orientalis*, *Zelkova*, etc., are presented in the detailed pollen diagram (Fig. 5.23). Especially, *Alnus* and deciduous *Quercus* have shown high abundance during this time. Percentage of deciduous *Quercus* reaches up to 7%. The amount of *Alnus* is 10% percent. Herbs also are represented highly. *Poaceae* has an average abundance of approximately 10%. Other herbs, Amaranthaceae Chenopodiaceae and Asteraceae Cichorioideae have been observed frequently. Mid-altitude and high altitude trees (such as, *Cedrus*, *Tsuga*, *Abies*) also show abundance. *Tsuga* is abundant and *Abies* is 6%. Nevertheless, *Cedrus* is present commonly. Subtropical trees (Taxodiaceae and *Engelhardia*) have less amount in the samples.

Figure 5.22. The synthetic pollen diagram of Trilophos. Note that only the two samples of five with statistically significant pollen (minimum 150) numbers were analysed. The numbers in the diagram show the pollen groups: 1; mega-mesothermic elements (*Taxodiaceae*, *Engelhardia*), 2; mesothermic elements (*Quercus*, *Carya*, *Pterocarya*, *Zelkova*, *Carpinus orientalis*, *Betula*, *Alnus*, etc.), 3; *Pinus*, 4; meso-microthermic elements (*Cedrus* and *Tsuga*), 5; microthermic elements (*Abies*), 6; herbs (Asteraceae-Asteroideae, Asteraceae-Cichorioideae, *Poaceae*, Amaranthaceae-chenopodiaceae, *Geranium*, etc. and include some water plants (*Sparganium*, *Potamogeton*, *Typha*), 7; steppe element (*Artemisia*).

Figure 5.23. Detailed pollen diagram of Trilophos.

Pinus also is abundant during this time. All taxa with percentages are represented in the detailed pollen diagram (Fig. 5.23).

5.5.2. Prosilio

The Prosilio section is located in 10 km SW of Servia. The vegetation in Prosilio section is characterized by the altitudinal trees (mid- and high altitude trees) and Cupressaceae (Fig. 5.24). The altitudinal trees are dominated by *Tsuga*, *Abies* and *Picea*. The percentage of *Tsuga* reaches up to 39%. *Abies* is about 17% and *Picea* is about 6%. *Cathaya* is abundant in sample 3. Herbs are presented frequently in the samples. They are Asteraceae Asteroideae, Asteraceae Cichorioideae, Poaceae, Amaranthaceae Chenopodiaceae and fresh water plant *Sparganium*. Steppe elements (*Artemisia* and *Hippophae rhamnoides*) are the same as herbs. They reach up to 5% in sample 2. Mesothermic trees (*Quercus*, *Ulmus* and *Alnus*) have low abundance in the samples. All taxa in the samples are shown in the detailed pollen diagram (Fig. 5.25). *Pinus* is also abundant with about 20% on average. In addition, Prosilio sediments contain *Botryococcus* colonies. *Botryococcus* is a green algae, generally live in freshwater (swamps, ponds, and lakes) (Gray, 1960; Tappan, 1980, Guy-Ohlson, 1992). Some forms of them also tolerates brackish environments (Wake and Hillen, 1980; DeDeckker, 1988). *Botryococcus* is commonly accepted that fossil colonies of *Botryococcus* indicate freshwater input and depositional settings affected

by freshwater (Batten and Grenfell, 1996). The abundance of *Botryococcus* show lake-level fluctuations during that time in Prosilio. Besides, Cupressaceae is very abundant in sample 2. The percentage of Cupressaceae reaches up to 61% in sample 2. Distribution of Cupressaceae shows changes from warm-humid conditions to dry and warm to cold climatic conditions. Using their morphology, it is difficult to determine them. If Cupressaceae profile is parallel to Taxodiaceae profile, which is thought to be a subtropical tree *Chamaecyparis* (Popescu, 2001).

However, Cupressaceae profile in Figure 5.24 is fit with high altitudinal conifer trees (*Abies* and *Picea*). Therefore, the high presence of Cupressaceae may indicate an existing dry and cold climatic conditions corresponding to a decrease in temperature. Moreover, the mid- and high altitudinal trees could indicate some uplift of the surrounding area.

Figure 5.24. The synthetic pollen diagram of Prosilio. Note that only the three samples of six with statistically significant pollen (minimum 150) numbers were analysed. The numbers in the diagram show the pollen groups: 1; *Cathaya*, 2; mesothermic elements (*Quercus*, *Alnus* and *Ulmus*), 3; *Pinus*, 4; meso-microthermic elements (mainly *Tsuga*), 5; microthermic elements (*Abies* and *Picea*), 6; Cupressaceae, 7; herbs (Asteraceae-Asteroideae, Asteraceae-Cichorioideae, Poaceae, Amaranthaceae-chenopodiaceae and include water plant *Sparganium*, 8; steppe element (*Artemisia* and *Hippophae rhamnoides*).

Figure 5.25. The detailed pollen diagram of Prosilio.

5.5.3 Lion of Amphipoli

The Lion of Amphipoli section represents the Early Zanclean. During this time vegetation is characterized by mega-mesothermic, mesothermic, mid- and high altitude trees, herbs and *Cathaya* (Fig. 5.26). Mega-mesothermic trees, belonging to these plant assemblages, such as Taxodiaceae, *Engelhardia* and *Taxodium* type are observed. Taxodiaceae has a frequency, its percentage reaches up to 3%. *Engelhardia* and *Taxodium* type are rare. The deciduous forest are represented by deciduous *Quercus*, *Populus*, *Buxus sempervirens*, *Carya*, *Zelkova*, *Ulmus*, *Pterocarya*, *Juglans*, *Fagus*, *Betula*, *Acer* and *Tilia*. Among this plant associations, deciduous *Quercus* is abundant with 7%. The mid-altitude trees such as, *Cedrus* are less abundant (4%). High altitude trees such as *Abies* and *Piceae* are also observed. *Abies* is rare. However, *Picea* is frequent as well as *Cedrus* (4.7%). *Cathaya* does not change significantly (1.6%). The group of the herbs (mainly Poaceae, Amaranthaceae Chenopodiaceae, Asteraceae Asteroideae, Brassicaceae, *Potamogeton*, Cyperaceae, *Plantago*, Caryophyllaceae, and Asteraceae Cichorioideae) are abundant, the percentage of herbs reaches up 10%. In addition, it should be mentioned that conifer pollen, mainly *Pinus* and indeterminate Pinaceae are particularly abundant. This is due to the resistance of saccate pollen to long distance transportation.

Figure 5.26. The synthetic pollen diagram of Lion of Amphipoli. Note that only the samples with statistically significant pollen (minimum 150) numbers were analysed. The numbers in the diagram show the pollen groups: 1; mega-mesothermic elements (*Taxodiaceae*, *Engelhardia*, *Taxodium* type), 2; *Cathaya*, 3; mesothermic elements (*Quercus*, *Carya*, *Pterocarya*, *Zelkova*, *Ulmus*, etc.), 4; *Pinus*, 5; meso-microthermic elements (*Cedrus*), 6; microthermic elements (*Abies* and *Picea*), 7; herbs (*Asteraceae* Asteroideae, *Asteraceae* Cichorioideae, *Poaceae*, *Amaranthaceae* *Chenopodiaceae*, etc. and include water plant *Potamogeton*).

6. DISCUSSION

6.1. Flora and floristic refuges

A total of 11 sections and 436 samples have been studied for pollen analysis in this study. Totally 107 different taxa were identified. The list of identified taxa are indicated in Table 2.1. Flora of the study areas is diversified in terms of thermophilous elements which are characterized by a peculiar story; they are arranged according to the Nix's (1982) classification:

- (1) megathermic elements (tropical): *Avicennia alba*, Euphorbiaceae, Rubiaceae and Rutaceae;
- (2) mega-mesothermic elements (subtropical): Taxodiaceae including *Taxodium*-type *Engelhardia*, *Platycarya*, Sapotaceae, *Distylium*, *Microtropis fallax*, *Ginkgo*, *Loropetalum*, Arecaceae and *Cathaya*;
- (3) mesothermic elements (warm-temperate): *Carpinus orientalis*, *Juglans*, *Juglans* cf. *cathayensis*, *Carya*, *Pterocarya*, *Liquidambar orientalis*, *Platanus*, *Nyssa*, *Ulmus*, *Zelkova*, *Celtis* and *Eucommia*;
- (4) meso-microthermic elements (cool-temperate): *Cedrus* and *Tsuga*.
- (5) Microthermic elements (boreal): *Abies* and *Picea*.

During the Late Miocene (early-late Messinian) and earliest Zanclean, floras are generally characterized by mesothermic trees such as *Quercus*, *Alnus*, *Carya*, *Buxus sempervirens*, *Ulmus*, *Juglans*, *Betula*, *Carpinus orientalis*, *Celtis*, *Carpinus betulus*, *Eucommia*, *Pterocarya*, *Liquidambar orientalis* etc. in the North Aegean and Northern Greece [i.e. İntepe, West Seddülbahir, Enez, Eceabat, Burhanlı (the latest Tortonian - early Messinian) and Prosilio]. Subtropical trees are Taxodiaceae (including *Taxodium*-type), *Engelhardia*, *Ginkgo*, *Loropetalum*, *Distylium* and *Microtropis fallax*, Arecaceae and Sapotaceae. They are common during this time-interval in the North Aegean. Subtropical elements are not abundant in İntepe, West Seddülbahir, Enez, Eceabat, and Burhanlı.

Among subtropical trees, *Ginkgo* is only observed in the Enez section. Today, *Ginkgo biloba* is the only species of Ginkgoaceae; it is living in China (Gong, 2008). The earlier record of Ginkgoaceae in Anatolia was found in Seyitömer Basin during the Early-Middle Miocene (Yavuz Işık, 2007). In addition, macrofossil (leaf) of

Ginkgoaceae was found at the Early-Late Pliocene boundary in western Hungary (Hably, 1998). Among the herbs, Poaceae, Amaranthaceae-Chenopodiaceae are common. *Pinus* is overrepresented in most of the samples.

Meso-microthermic and microthermic trees (*Cedrus* and *Tsuga*, *Abies* and *Picea*) are recorded in the study areas. *Tsuga* pollen is abundant at Enez and Prosilio. *Cedrus* is abundant at İntepe (after the MSC) and West Seddülbahir. *Cathaya* is observed at İntepe, West Seddülbahir, Enez and Prosilio.

Subtropical elements are frequent in DSDP Site 380 and mainly represented by Taxodiaceae (including *Taxodium*- and *Sequoia*-types, *Sciadopitys*), *Engelhardia*, *Myrica*, Sapotaceae, *Microtropis fallax*, *Distylium* cf. *sinensis*, Araliaceae, *Nyssa*, etc. during the Late Miocene (Popescu, 2006). Tropical trees are few and they are represented by *Amanoa*, *Fothergilla*, *Exbucklandia*, *Avicennia*, Euphorbiaceae, Sapindaceae, Loranthaceae and Acanthaceae in the lower part of the 380 pollen record (Popescu, 2006).

Pollen records from the other time-spans of the Miocene have also shown the existence of thermophilous trees in Anatolia. The pollen results from Ermenek (central Taurus, Turkey) pointed out thermophilous plants during the Aquitanian: Euphorbiaceae, Rubiaceae, Sapotaceae, Taxodiaceae, *Engelhardia*, *Liquidambar orientalis*, *Carya*, *Ulmus*, *Pterocarya*, *Zelkova*, *Carpinus betulus*, Anacardiaceae, *Carpinus orientalis*, *Juglans*, *Platanus*, *Rhus*, etc. (Biltekin, unpublished). During that time, especially *Cedrus* is very abundant within sediments from the region. Today, *Cedrus libani* is mostly found in the Taurus (Turkey), but also in Syria and Lebanon (Hajar, 2010). Cedar belt is located between 1200 and 1400 meters in Eastern Mediterranean (rainfall: 1000-2000 mm/year) (Quézel, 1998). For this region, *Cedrus libani* might be an endemic species in the Taurus. In future studies, after detailed morphological analysis, it could be correlated with *Cedrus* of Ermenek. In addition, *Cedrus* is still existed as relict in Morocco, Algeria, Cyprus and Lebanon (Quézel, 1998; Quézel and Médail, 2003). *Cathaya* is also found in the region during the early Miocene.

Pollen results of Seyitömer Basin (Kütahya, Western Anatolia) have shown that flora was rich in terms of thermophilous plants (Taxodiaceae, *Engelhardia*, *Carya*, *Zelkova*, *Liquidambar*, *Cedrus*, etc. during the Early-Middle Miocene (Yavuz Işık, 2007). During the same time-interval, Taxodiaceae, *Engelhardia*, Hamamelidaceae, *Nyssa*, *Myrica*, Sapotaceae, Araliaceae, *Carpinus orientalis*, *Liquidambar*, *Parrotia*

persica, *Ulmus*, *Carya*, *Zelkova*, *Pterocarya*, *Juglans*, *Cedrus* are presented in the central Turkey (the Pelitçik Basin) (Yavuz Işık, 2009). During the Burdigalian, Euphorbiaceae, Taxodiaceae, Cyrillaceae-Clethraceae, Sapotaceae, Hamamelidaceae, *Engelhardia*, *Pterocarya*, *Carya*, *Juglans*, *Celtis*, *Ulmus*, *Zelkova*, *Carpinus*, *Liquidambar*, Araliaceae, *Cathaya*, *Cedrus* existed in the Güvem Basin (NW Central Anatolia) (Yavuz Işık, 2008). In Çatakbağyaka (west-south Turkey), thermophilous trees such as Euphorbiaceae, *Mussaenda*-type, Rubiaceae, *Alchornea*, Passifloraceae, Taxodiaceae, Arecaceae, *Myrica*, Sapotaceae, *Distylium*, Hamamelidaceae, *Engelhardia*, Celastraceae, *Carya*, *Juglans*, *Pterocarya*, *Liquidambar*, *Parrotia persica*, Anacardiaceae, *Eucommia*, *Zelkova*, *Carpinus*, *Carpinus orientalis*, *Celtis*, *Cathaya* and *Cedrus* were found during the Langhian (~14.8-15.0 Ma) (Jiménez-Moreno, 2005).

Pollen records are also available from the nearby areas and other areas of the Mediterranean region for the Miocene. Thermophilous trees such as Sapotaceae, Araliaceae, Theaceae, *Reevesia*, *Pandanus*, Schizaeaceae, Gleicheniaceae, Taxodiaceae, *Engelhardia*, *Alangium*, *Symplocos*, *Itea*, *Chloranthus*, *Myrica*, *Liquidambar*, *Celtis*, *Nyssa*, *Planera*, *Zelkova*, *Ulmus*, *Platanus*, *Carpinus*, *Juglans*, *Carya*, *Pterocarya*, *Tsuga*, etc. lived in NW Bulgaria during the Badenian (Middle Miocene) (Ivanov, 2002). In the west Bulgaria (Beli Breg Coal), subtropical plants are in low quantity (*Engelhardia*, *Platycarya*, *Symplocos*, Sapotaceae and Arecaceae). Mid-high altitude trees (*Tsuga*, *Cedrus* and *Cathaya*) are also important in the pollen flora. *Carya*, *Pterocarya*, *Ulmus*, *Zelkova*, *Eucommia*, *Juglans* are common during the Late Miocene (Ivanov, 2007).

During the Late Tortonian-Messinian, thermophilous elements are Mimosaceae, Euphorbiaceae, Rubiaceae, Taxodiaceae (*Sequoia* type in Samos, *Taxodium* type in the Pikermi area), *Engelhardia*, Sapotaceae, Arecaceae, *Symplocos*, *Nyssa*, *Myrica*, *Carya*, *Juglans* are recorded in the southern-central and northern Greece (Ioakim, 2005).

The main difference with these regions is the high abundance of Taxodiaceae in the Late Miocene localities in this study. Other differences are abundance of *Tsuga* (in Enez and Prosilio) and *Cedrus* (at İntepe after the MSC). In addition, Cupressaceae (Juniper family) are more abundant (reaching up to 61%) at Prosilio during this time. Cupressaceae contain species living in a warm-humid conditions and some others in dry-cold conditions (Suc and Popescu, 2005; Popescu, 2006).

It is difficult to assess a finer identification than the family level because of the very poor variability of morphological characters of Cupressaceae. In this situations, their interpretation is made according to their behavior in the pollen records. Here, Cupressaceae show a parallel curve to those of mid- and high-altitude elements, steppes (*Artemisia* and *Hippophae rhamnoides*) and herbs. This could indicate lowering in temperature. Indeed, studies on modern and fossil species of Cupressaceae (*Thuja*) show that this taxon can survive under the cold to freezing conditions (LePage, 2003).

There are studies on macroflora data in Turkey for the Miocene and Pliocene (Kasaplıgil, 1977; Sakıncı, 2007) and neighbouring areas (Kovar-Eder, 2006; Kovar-Eder, 2008; Velitzelos, 1990). Sakıncı (2007) examined silicated trees in Thrace from Late Miocene-Pliocene age. Within silicated tree assemblages, Podocarpaceae, Anacardiaceae, Fagaceae, Juglandaceae, *Engelhardia*, Lauraceae, Fabaceae Caesalpinoideae, Fabaceae Mimosoideae, Rosaceae Prunoideae (*Prunus*), Asteraceae have been observed. These silicated tree assemblages are comparable and convenient with pollen results from this study. *Engelhardia*, *Fagus*, *Juglans*, Asteraceae are also recorded in Enez and DSDP Site 380.

Kovar-Eder *et al.* (2006) reconstructed and mapped the vegetation using macroplant data from southern Europe (Greece and adjacent areas, Italy, southern France and Spain) from the Late Miocene to Early Pliocene. The results show that sclerophyllous oaks developed during the Late Miocene and humid subtropical conditions dominated in Italy during the Early Pliocene. Southern France and Spain recorded a decrease in thermophilous plants during that time-interval.

Pollen flora for the Pliocene is dominated by swampy component (with mainly *Glyptostrobus*), and an evergreen-deciduous mixed assemblage. Within them, Taxodiaceae (*Glyptostrobus*) is very abundant and. *Glyptostrobus* is almost observed in the most of samples of DSDP core between 702.4 and 319.03 m depth with changing quantity. A detailed scanning electronic microscope analysis of pollen grains indicates that Taxodiaceae pollen grains have the same morphology with *Glyptostrobus pensilis* (i.e. *Glyptostrobus lineatus*). The last occurrence of *Glyptostrobus* in the Black Sea region was expected during the Late Pliocene to Early Pleistocene. On contrary, analyses of DSDP Site 380 document its persistence up to Recent. *Glyptostrobus* was occupying the deltaic coastal areas.

Taxodiaceae disappeared earlier from the northern Mediterranean region (3.6 Ma), later from the southern Mediterranean (2.6 Ma), from Italy and Crete (1.3 Ma) and disappeared from Black Sea and Lake Baikal very recently (Suc *et al.*, 2004; Popescu *et al.*, 2010) (Fig. 6.1).

Figure 6.1. Latest records of Taxodiaceae swamps in the Mediterranean region.

Quercus is also abundant in the pollen flora. *Alnus* is generally seen in most of the sediments as well as *Carpinus orientalis*, *Zelkova*, *Abies*. *Cathaya* is not very frequent and *Cedrus* and *Tsuga* are not abundant but they are usually recorded. Microthermic trees such as *Abies* and *Picea* are not frequent.

Other thermophilous trees are Euphorbiaceae, Rubiaceae, Arecaceae, Rutaceae, *Avicennia alba*, *Engelhardia*, *Platycarya*, Sapotaceae, *Juglans*, *Juglans* cf. *cathayensis*, *Pterocarya*, *Ulmus*, *Liquidambar orientalis*, *Carpinus betulus*, *Platanus*, *Nyssa*, *Carya*, and *Zelkova*. *Avicennia* is a mangrove element, it was scarcely recorded in the previous work of DSDP Site 380 (at 1018.85 and 781.63 m depth; Popescu, 2006). In the pollen records of this study, *Avicennia alba* is very rare and has been only observed at 412.53 m depth. *Avicennia* disappeared from the northern Mediterranean at 14 Ma (Serravalian), from southern Mediterranean and Sicily at ca. 5.3 Ma (Bessedik, 1985; Suc and Bessais, 1990), and at ca. 1.6 Ma from the southern Black Sea (Popescu *et al.*, 2010) (Fig. 6.2). This suggests that tropical trees persisted in Anatolia longer than in the other regions areas of the Mediterranean region. Among the other flora components, herbs are dominated by Poaceae, Amaranthaceae-Chenopodiaceae, Asteraceae Asteroideae, Asteraceae Cichorioideae.

Steppe assemblages include *Artemisia*, *Hippophae rhamnoides* and *Ephedra*. Within them, *Artemisia* is very abundant.

Figure 6.2. Latest records of *Avicennia* mangrove in the Mediterranean region.

Pinus and Mediterranean xerophytes (*Quercus ilex*-type, *Olea*, *Ligustrum*) are not frequent. Pollen records have similar floristic patterns to the Early Pliocene recorded at Garraf1 (NW Mediterranean), Susteren 752.72 (Netherlands), Rio Maior F16 (Portugal), Wolka Ligezawska (Poland) (Suc, 1984; Zagwijn, 1960; Diniz, 1984; Popescu *et al.*, 2010).

Kasaplıgil (1977) recorded plant macrofossils (cones, fruits, seeds, leaves and branches) in Güvem Basin which is located at 125 km from the Black Sea coastline (NW Central Anatolia). In the Pliocene Güvem macroflora, *Glyptostrobus europaeus* is abundant. This situation agrees with the pollen results of DSDP Site 380. Other common species is *Sequoia langsdorfii*. Among other species, there are *Carpinus miocenica*, *Magnolia sprengeri*, *Menispermum*, *Myrica banksiaefolia*, *Persea indica fossilis*, *Platanus*, *Platycarya miocenica*, *Pterocarya pterocarpa fossilis*, *Zelkova ungeri*, *Liquidambar europaeum*, *Acer angustilobum*, several species of *Quercus*, *Castanopsis*, *Ulmus*, *Alnus*, *Betula aff. Luminifera*, *Populus tremula fossilis*, *Fagus*, *Castanea*, *Castanopsis*, *Tsuga*, *Salvinia*, *Potamogeton*, *Typha*, *Pinus*, *Picea*, etc. Kasaplıgil (1977) also examined pollen grains. Pollen grains include *Alnus*, *Betula*, *Menispermum*, *Pterocarya*, *Nyssa*, *Ulmus*, *Zelkova*, *Quercus*, *Salix*, *Cedrus*, *Picea*, etc.

In NW Macedonia and Western Greece (Ptolemais Notio, Ptolemais Base, Trilophos and Lion of Amphipoli), flora is depicted by mesothermic trees (*Quercus*, *Carya*, *Pterocarya*, *Zelkova*, *Ulmus*, *Liquidambar*, *Carpinus orientalis*, *Juglans*, etc.). Among them, *Quercus*, *Pterocarya*, *Zelkova*, *Alnus*, *Betula* are abundant. Tropical trees are scarcely found. Subtropical plants (i.e. Taxodiaceae, *Engelhardia* and *Taxodium* type) are presented in small amounts. *Cathaya* is not very important in the regions except at Lion of Amphipoli. *Cedrus* and *Tsuga* are frequent. Among herbs, Poaceae, Amaranthaceae-Chenopodiaceae, Asteraceae Asteroideae, Asteraceae Cichorioideae are recorded. Within them, Poaceae is very abundant in Ptolemais Notio section. Mid- and high-altitude conifers are represented by *Cedrus*, *Tsuga*, *Abies* and *Picea*. The previous palynological studies from the Ptolemais Basin show that the flora is dominated by Taxodiaceae, Cyperaceae, deciduous *Quercus* and Poaceae (Kloosterboer-van Hove *et al.*, 2001, 2006). In this study, more taxa have been identified when comparing with the previous microfloristic studies in the Ptolemais Basin.

In addition, Cyperaceae are not abundant in the samples. Other differences from previous studies are the abundance of mid- and high-altitude conifers (*Cedrus*, *Tsuga*, *Abies* and *Picea*) during the Early Pliocene. This could display the presence of elevated nearby reliefs.

Macroflora (leaves, fruits, seeds) from the Ptolemais area evidences mixed-mesophytic forest assemblages during the Pliocene (Velitzelos, 1990). This macrofloristic results are comparable with pollen data from Ptolemais Notio and Ptolemais Base.

Pollen records of the studied areas indicate that prominent changes occurred in the flora. Results show how Anatolia recorded the floral extinctions. The Neogene successive coolings began at 14 Ma and resulted in the vanishing of thermophilous plants in the Northern Hemisphere mid-latitudes. As a consequence, two residual refuge areas developed in the Mediterranean region. One of them locates in the Northeastern Mediterranean region as documented by the persisting taxa *Zelkova*, *Pterocarya*, *Liquidambar* and *Cedrus*. The other refuge area is in the Southwestern Mediterranean region. *Laurus*, *Argania* (Sapotaceae) and *Cedrus* are relict plants in this area (Quézel & Médail, 2003). On the Anatolia coastlines, *Pterocarya fraxinifolia* and *Liquidambar orientalis* are the only thermophilous elements to persist near-slopes of coastal ranges (Fig. 6.3).

Figure 6.3. Latest records of some thermophilous warm-temperate trees in the Mediterranean region. Dark blue circles indicates refuge areas.

In the pollen records of this study, they have been observed since the Late Miocene. This makes the study area the most important refuge domain in the region (Zohary, 1973; Quézel & Médail, 2003). Relict plants were well-defined in the pollen records. They were living in Anatolia during Pliocene-Pleistocene, during the Early Pliocene and latest Miocene.

Most of them, such as *Carya*, *Parrotia persica*, *Cathaya*, *Cedrus* and *Tsuga* persisted in the region up today. Their disappearance seems to have happened during the Middle Pleistocene. Taxodiaceae were still living on the Rhodes Island 500 kyrs ago (Suc and Popescu, 2005). However, Taxodiaceae disappeared from the Euxinian-Hyrcanian region sub-recently (Fig. 5.30). The persistence of relict plants in Anatolia could be explained by the effective influence of the Asiatic monsoon which almost continuously provided water masses along a longitudinal gradient allowing the maintain of plants requiring warm and humid climatic conditions despite the successive Quaternary glaciations.

6.2. Vegetation

Observed changes in vegetation according to this study will be described since the Late Miocene and discussed in this section. Abundance and diversity of identified pollen grains enable a reliable comparison with the present-day plant ecosystems. The most crucial parameters controlling the vegetation organization in altitude are both temperature and precipitation. Therefore, vegetation reconstructed from this study must be compared with the organization in altitudinal belts of present-day forest in southeastern China (Wang, 1961) (Fig. 6.4), as it is the closest living example for the Miocene European flora (Suc, 1984; Axelrod *et al.*, 1996; Jiménez-Moreno, 2005; Jiménez-Moreno *et al.*, 2005; Jiménez-Moreno, 2006; Jiménez-Moreno *et al.*, 2007a,b; 2008a,b; Jiménez-Moreno *et al.*, 2009). Thus, the North-Aegean Late Neogene vegetation could be split into ecologically different environments:

- (1) open lowlands were characterized by open vegetation with steppe elements such as Poaceae, Amaranthaceae-Chenopodiaceae, *Convolvulus*, *Ephedra*, *Artemisia*, etc., and both some halophytes such as Caryophyllaceae, Plumbaginaceae and in some places an aquatic ecosystem constituted by *Potamogeton*, *Sparganium*, *Typha*, *Thalictrum*, Liliaceae, etc.; and the Mediterranean xerophytes such as *Quercus ilex* type, *Olea*, *Ligustrum*, etc.;
- (2) a broad-leaved evergreen forest, from sea level to around 700 m altitude composed by *Glyptostrobus* or *Taxodium*-type, *Rhus*, Euphorbiaceae, Rutaceae, *Engelhardia*, Sapotaceae, *Distylium*, *Ilex*, etc.;
- (3) an evergreen and deciduous mixed forest, above 700 m in altitude characterized by deciduous *Quercus*, *Engelhardia*, *Platycarya*, *Carya*, *Pterocarya*, *Carpinus orientalis*, *Celtis*, *Fagus*, *Acer*, *Hedera*, *Liquidambar* cf. *orientalis*, etc. Within this vegetation belt, riparian vegetation is identified, composed of *Alnus*, *Salix*, *Carya*, *Carpinus*, *Zelkova*, *Ulmus*, *Liquidambar*, etc. The shrub level is dominated by *Ilex*, Caprifoliaceae, Ericaceae, etc.;
- (4) a mid-altitude deciduous and coniferous mixed forest, above 1000 m with *Betula*, *Fagus*, *Cathaya*, *Pinus*, *Cedrus* and *Tsuga*;

- (5) a high-altitude coniferous forest, above 1800 m in altitude with *Abies* and *Picea*;
- (6) herbaceous meadows in high altitude, above 2800 m, develop in more humid areas.

Figure 6.4. The vegetation organization in altitude in the southeast China (ca. 25-30° of latitude) (from Wang, 1961).

The vegetation is mainly characterized by herbs in İntepe, Burhanlı and Eceabat areas (Fig. 6.5). Among them Poaceae, Asteraceae Asteroideae, Asteraceae Cichorioideae, and Amaranthaceae-Chenopodiaceae are dominating in the regions during the early Messinian. Arboreal elements are not abundant and constituted of warm-temperate trees such as deciduous *Quercus*, *Zelkova*, *Carpinus orientalis*, *Carya*, *Pterocarya*, etc. *Pinus* was abundant in the Eceabat and Burhanlı areas. The other elements are subtropical trees, mid- and high-altitude trees, mediterranean xerophytes, steppe elements, non-significant trees and Cupressaceae. They are not prevalent in the pollen floras from that time. During the Late Miocene, the vegetation was mainly dominated by altitudinal coniferous trees (mid- and high-altitude elements) such as chiefly *Tsuga*, *Abies* and *Picea* and Cupressaceae in Prosilio (northern Greece).

Herbs (Asteraceae Asteroideae, Asteraceae Cichorioideae, Poaceae, Amaranthaceae-Chenopodiaceae) and steppe elements (*Artemisia* and *Hippophae rhamnoides*) are in small amounts. Also mesothermic trees (*Quercus*, *Ulmus* and *Alnus*) display low abundance in the pollen floras. *Pinus* is also abundant. In addition, sediments include *Botryococcus* colonies. The presence and abundance of *Botryococcus* indicate freshwater input and lake-level fluctuations during that time. Besides, Cupressaceae is very common in the area.

The vegetation is depicted by generally warm-temperate elements and herbs during the Late Miocene (latest Messinian)-Early Pliocene (earliest Zanclean) in the region (Enez, İntepe, West Seddülbahir, Ptolemais Notio, Ptolemais Base, Lion of Amphipoli and Trilophos). In open vegetation, mainly Poaceae, Asteraceae Asteroideae, Asteraceae Cichorioideae, and Amaranthaceae-Chenopodiaceae, etc. (with water plants: *Sparganium*, *Typha*, *Potamogeton*) are found. This agrees with outcomes of the CARAIB vegetation model (Favre, 2007). Potential vegetation maps constructed from interpolation of pollen data obtained for most of them before this study confirm that herbs and mesothermic plants were largely developed in the region during the Zanclean (Figs. 6.6-6.7).

An evergreen and deciduous mixed forest was composed of mesothermic elements such as deciduous *Quercus*, *Carya*, *Pterocarya*, *Fagus*, *Acer*, *Carpinus orientalis*, *Liquidambar orientalis*, *Ilex*, also *Engelhardia*, etc. Within this vegetation belt, a riparian vegetation is identified with *Salix*, *Alnus*, *Carya*, *Carpinus orientalis*, *Zelkova*, *Ulmus*, *Liquidambar*, etc.

In addition, mid- and high-altitude trees (*Cedrus*, *Tsuga*, *Abies* and *Picea*) increased in the region during the Late Miocene-Early Pliocene. This could be caused by some uplift of the surrounding massifs. *Cathaya*, a mid-altitude conifer, does not vary significantly in the region during that time. Mediterranean xerophytes, Cupressaceae and steppe (*Artemisia*) elements are in small amounts. Nevertheless, *Pinus* seems to be over-represented.

During the Late Miocene-Early Pliocene, vegetation on the Southwest Black Sea shorelines (as illustrated by the DSDP Site 380 pollen content) is characterized by thermophilous plants and herbs with steppe assemblages. Among thermophilous trees, subtropical (Taxodiaceae including *Taxodium*-type, *Sequoia*-type, *Sciadopitys*, *Engelhardia*, *Myrica*, etc.) and warm-temperate trees (i.e., deciduous *Quercus*, *Carya*, *Pterocarya*, *Juglans*, *Zelkova*, *Ulmus*, etc.) are recorded (Popescu, 2006).

Within herb and steppe elements, Poaceae, Amaranthaceae-Chenopodiaceae and *Artemisia* are abundant during that time. The similar vegetation trends were observed in southwestern Romania during the Early Pliocene where developed swampy forests competing with herbaceous vegetation (Popescu, 2006). The significant changes occurred in the vegetation during the Late Pliocene (early Piacenzian)-Early Pleistocene (Gelasian). The most of the plants were inherited from the Miocene. Most of the thermophilous plants disappeared from the region in contrast to a continuous increase in mesothermic trees.

In DSDP Site 380, it is seen that the competition between two vegetation types prevailed during the studied time-interval, opposing forest assemblages and herbs with *Artemisia* steppes. At the end of the Zanclean, herbaceous environments (with mainly Amaranthaceae-Chenopodiaceae, Poaceae, Asteraceae Asteroideae, Asteraceae Cichorioideae) dominated in the region with *Artemisia* steppes. The percentage of Amaranthaceae-Chenopodiaceae reaches up around 88%. During this time, subtropical trees were not abundant with mesothermic plants.

At the early Piacenzian, corresponding to Pollen zone 4 (603-460.540 m) (see for detail Popescu *et al.*, 2010), swamp forests were well developed in contrast to herb landscapes with steppe elements. Within swamp forests, Taxodiaceae (mainly probably *Glyptostrobus*) prevailed with *Engelhardia*, Sapotaceae, *Nyssa*, etc. Deciduous mixed forests mainly composed of mesothermic elements such as deciduous *Quercus*, *Betula*, *Alnus*, *Liquidambar*, *Fagus*, *Carpinus orientalis*, *Carpinus betulus*, *Tilia*, *Acer*, *Ulmus*, *Zelkova*, *Carya*, *Pterocarya*, etc. were largely developed during this time, but situated at higher altitude.

Figure 6.5. Late Neogene synthetic pollen diagrams in the study region. Numbers show the plants groups in synthetic pollen diagrams: 1, Megathermic elements; 2, Mega-mesothermic elements; 3, *Cathaya*; 4, Mesothermic elements; 5, *Pinus*; 6, Meso-microthermic elements; 7, Microthermic elements; 8, Non-significant elements; 9, Cupressaceae; 10, Mediterranean xerophytes; 11, Herbs; 12, Steppe elements.

Figure 6.6. Interpolated vegetation map for herbs at the Zanclean (Favre, 2007).

Figure 6.7. Interpolated vegetation map for mesothermic trees at the Zanclean (Favre, 2007).

Within this vegetation belt, riparian forest also developed with *Salix*, *Alnus*, *Carya*, *Carpinus orientalis*, *Zelkova*, *Ulmus*, *Liquidambar*, etc. The BIOME4 and paleodata reconstructions indicate that temperate forest dominated in northeastern Europe during the Middle Pliocene (Salzmann *et al.*, 2008). Herb assemblages were mainly composed of Amaranthaceae-Chenopodiaceae, Poaceae, Asteraceae Asteroideae, Asteraceae Cichorioideae, Caryophyllaceae, Brassicaceae, *Polygonum*, *Rumex*, etc. Steppe elements (i.e., *Artemisia*, *Ephedra*, *Hippophae rhamnoides*) were frequent during that time. At the beginning of the Pleistocene, glacial-interglacial cycles occurred with a major decrease in temperature.

This caused strong alternations in vegetation. Glacial-interglacial changes in vegetation are marked by alternations of *Artemisia* steppes with herbs and forest assemblages (composed of mesothermic trees). This situation is well observed in the studied interval of DSDP Site 380. At the Early-Middle Pleistocene (early-middle Gelasian-early Calabrian), herbs are common with steppes (*Artemisa*, *Ephedra* and *Hippophae rhamnoides*), including mostly Amaranthaceae-Chenopodiaceae, Poaceae, Asteraceae Asteroideae, Asteraceae Cichorioideae, Brassicaceae, etc. Mega-mesothermic trees are not abundant, but mesothermic plants (i.e., mainly deciduous *Quercus*, *Alnus*, *Betula*, *Carpinus orientalis*, etc.) show somewhat increasing trends.

Nevertheless, the same vegetation trend is observed in the southern Apennines (Italy). Here the vegetation is characterized by the competition between steppe taxa and forest assemblages during the Lower-Middle Pleistocene (Sabato *et al.*, 2005).

The other observed vegetation types through study intervals are altitudinal forests made of microthermic and meso-microthermic trees, mediterranean xerophytes, non-significant elements and Cupressaceae. Microthermic (*Abies* and *Picea*) and meso-microthermic (*Cedrus* and *Tsuga*) conifers are not frequent as well as *Pinus*, mediterranean xerophytes (*Olea*, *Quercus ilex* type, *Ligustrum*), non-significant elements (Rosaceae and Ranunculaceae) and Cupressaceae.

6.2.1. The development of *Artemisia* steppes

Artemisia is the well-known cosmopolitan wind-pollinated sagebrush. Past evidences of *Artemisia* steppes have probably existed since the Middle Tertiary from arid or subarid areas of temperate Asia (Wang, 2003; Yunfa *et al.*, 2010). Development of *Artemisia* in mid-altitudes of central Asia was strongly encouraged by uplift of the Tibetan Plateau during the Miocene (in Fig. 6.8). Moreover, the other effects on diversification and distribution of *Artemisia* are global cooling and Asian monsoon in that area. The presence of open vegetation without a significant development of *Artemisia* steppes in other regions of Mediterranean (i.e., in the southern Mediterranean) is established since the earliest Miocene (Suc *et al.*, 1995a, b; Bachiri Taoufiq *et al.*, 2001; Jiménez-Moreno and Suc, 2007). Pollen results from the studied area enable information on the earliest development of *Artemisa* steppes in Anatolia in time and space.

Figure 6.8. Diagram indicating the origin and development of *Artemisia* (from Yunfa *et al.*, 2010), with global climate (Zachos *et al.*, 2008), capital letters in the diagram: A, B, C, D are adapted from Li, 1991; Li and Fang, 1999; An *et al.*, 2006; Rowley and Currie, 2006; d. Wan *et al.*, 2007) and the Asian monsoon intensity (Wan *et al.*, 2007).

The occurrence of *Artemisia* in Anatolia could be divided into three parts as the Miocene, Pliocene and Pleistocene. The earliest records of *Artemisia* steppes in Anatolia concern the Early Miocene (Aquitanian~23.0-20.4 Ma) from the central Taurus, more precisely the Ermenek region (Biltekin, unpublished). In this area, *Artemisia* is observed in almost all the samples and reaches up to around 10%. In addition, *Artemisia* steppes were found in Çatakbağyaka (west-southern Aegean region) during the Middle Miocene (Langhian). Also *Artemisia* was found in western Anatolia (Seyitömer Basin, Kütahya) at the Late Early-Middle Miocene (Yavuz Işık, 2007).

In other parts of the world, it was common in the western part of the Tibetan Plateau during the Miocene (Yunfa *et al.*, 2010) and in the Snake River Plain (America) during the Miocene (~12 Ma) (Davis and Ellis, 2010), in the western north America at the Early Miocene, in the Northeastern America at the Middle Miocene and in the central Europe during the Late Oligocene (Graham, 1996).

The main development of the *Artemisia* steppes with open herbaceous vegetation in Anatolia began during the Late Miocene-Early Pliocene (Zanclean) and in the Ponto-Euxinian region during the earliest glacial phases. In the other studied regions i.e., Eceabat, Burhanlı, İntepe, *Artemisia* is present but in very small amounts.

The marked changes occurred between the Miocene and the Pliocene. Because of this, high resolution pollen records of the Black Sea DSDP Site 380 are very informative about the development of *Artemisia* steppes in Anatolia (Fig. 6.9).

In the studied intervals, the increase of *Artemisia* continued through the Pliocene. It reached up to ca. 62% during the Pliocene. When the glacial-interglacial cycles began at the beginning of the Pleistocene, *Artemisia* was very abundant as along the whole Pleistocene until present. Maximum abundance of *Artemisia* is about 85% at 401.44 m in DSDP Site 380. According to the pollen results, the earliest settlement of sagebrush (*Artemisia*) steppes seems to be extended to the Early Miocene (Aquitanian) in the central Taurus (Biltekin, unpublished). Overall the development of *Artemisia* in Anatolia could result from the combined effects of uplift of the Tibetan Plateau, onset of global cooling and reinforcement of the Asian monsoon (Zhisheng *et al.*, 2001, Yunfa *et al.*, 2010).

Figure 6.9. Chronological distribution of *Artemisia* steppes since the Early Miocene until today in the studied region.

6.3. Climate

6.3.1. Global Climate Context during the Miocene and Pliocene

Neogene climate constitutes the transition from greenhouse conditions of Paleogene to the icehouse conditions of Quaternary. East Antarctic Ice Sheet (EAIS) expanded at the beginning of the Neogene (Pagani *et al.*, 1999; Zachos *et al.*, 2001) (Fig. 6.10). This situation is well-documented at worldwide-scale with decrease in temperature and positive oxygen isotope excursion (Miller *et al.*, 1991; Zachos *et al.*, 2001). The benthic foraminiferal oxygen isotope values give the evidence for at least nine glacial events during the Miocene, four of them occurred during the Early Miocene (Miller *et al.*, 1991; Pagani *et al.*, 1999).

Until the Middle Miocene (~15 Ma) global ice volume stayed low (with slightly bottom water temperatures) with several brief glaciations (i.e., Mi-events) (Zachos *et al.*, 2001). The low CO₂ values are in correspondence with major glaciations (Kürschner *et al.*, 2008). At the Late-Middle Miocene (Upper Burdigalian-Lower Langhian, ~17-15 Ma) a warm phase occurred known as the Miocene Climatic Optimum (Zachos *et al.*, 2001, 2008) (in Fig. 6.10). During this warm phase, CO₂ concentrations were 500 ppmv (Kürschner *et al.*, 2008). After this warm period, the Monterey cooling event occurred at 14 Ma ago. This event coincides with ice sheet expansion in Antarctica (Flower and Kennett, 1993, 1994; Miller *et al.*, 1991; Zachos *et al.*, 2001).

During the early Late Miocene, low-mid altitude surface waters of world oceans warmed up. This global climate variability was induced by two events: the closure of the Indonesian Seaway at 8-5.2 Ma and the onset of the Tibetan Plateau uplift (Zhisheng *et al.*, 2001; Zhang *et al.*, 2009; Yunfa *et al.*, 2010). During the Late Miocene (Tortonian), high seasonality existed in the Eastern Mediterranean (Eronen *et al.*, 2009). Summer drought increased from Tortonian to the Messinian.

Figure 6.10. Global deep-sea oxygen records with main events (taken from Zachos *et al.*, 2001).

High evaporation and low rainfall occurred with lower seasonality due to the increased duration of summer aridity in the Eastern Mediterranean during the Messinian (Eronen *et al.*, 2009). The Pliocene constitutes the transition from relatively warm episodes to the cooler climates of the Pleistocene (Suc, 1984; Dowsett and Poore 1991; Lisiecki and Raymo 2007; Haywood *et al.*, 2009). The Pliocene epoch could be split into three phases: (1) the Early Pliocene warm period including three inner subdivisions, (2) a relatively short-lived ‘warm interval’ at ca. 3 Ma, known as the mid-Pliocene warm interval and (3) a climatic deterioration during the Late Pliocene leading to the high-magnitude climate variability associated with glacial/interglacial cycles of Pleistocene.

Although a progressive cooling existed during the Tertiary, the Pliocene seems to have been warmer than today (Jansen *et al.* 2007). The Early Pliocene was an interval of global warmth characterized by high CO₂ levels (Van der Burgh *et al.*, 1993; Raymo *et al.*, 1996; Kuerschner *et al.*, 1996; Billups *et al.*, 2008) and warm sea-surface temperatures (in upwelling regions and at high latitudes) (e.g., Dowsett *et al.*, 1992; Herbert and Schuffert, 1998; Dowsett *et al.*, 2005; Wara *et al.*, 2005; Lawrence *et al.*, 2006; Ravelo *et al.*, 2007).

Figure 6.11. Distribution of modern and mid-Pliocene land and sea ice in the Northern Hemisphere (from Dowsett *et al.*, 1994).

Figure 6.12. Distribution of modern and mid-Pliocene land and sea ice in the Southern Hemisphere (from Dowsett *et al.*, 1994).

During the middle Pliocene (~3 Ma), the paleontological data, sea level, vegetation, land-ice distribution, sea-ice distribution and sea surface temperatures (SST) were reconstructed. The middle Pliocene sea level was at least 25 m higher than present because of the reduction in size of the East Antarctic Ice Sheet (Dowsett *et al.*, 1994). The Pliocene winter reconstructions (Dowsett *et al.*, 1994) indicate that sea ice covered the north coast of Siberia and Greenland in the Northern Hemisphere, regions which are today completely covered by ice during the winter (Fig. 6.11). In the Southern Hemisphere, sea ice was then in the Weddell Sea, the coast of the Queen Maun Land, Wilkes Land and Marie Byrd Land (Fig. 6.12) (Dowsett *et al.*, 1994).

At the end of the Pliocene, with the onset of major Northern Hemisphere Glaciations at approximately 2.6 Ma, climate got cooler and glacial–interglacial cycles appeared in the Northern Hemisphere (Lisiecki and Raymo, 2007). The mid-Pleistocene transition (MPT) was a crucial event when the dominant periodicity of glacial response changed from 41 to 100 kyrs. The “saw-tooth” asymmetry of glacial cycles first appears shortly after the onset of major Northern Hemisphere Glaciations and duration of interglacial phases decreased at 1.4 Ma (Lisiecki and Raymo, 2007).

6.3.2. Climatic evolution of the studied areas

Results of pollen data provide a climate synthesis of the studied areas during the Late Miocene-Early Pleistocene. Before the Messinian Salinity Crisis, at the early Messinian, abundance of mega-mesothermic and mesothermic trees in the Ponto-Euxinian region indicate that subtropical, i.e. warm climate conditions, existed during that time in the region (Popescu, 2006). Climate also was humid according to the existence of thermophilous elements which require very humid conditions during all the year (Wang, 1961). In the Northern Aegean area, in the İntepe, Burhanlı and Eceabat regions (Melinte *et al.*, 2009), the high abundance of open vegetation elements (Poaceae, Asteraceae Asteroideae, Asteraceae Cichorioideae, and Amaranthaceae-Chenopodiaceae, etc.) indicate drier climate conditions. However, abundance of mesothermic trees (deciduous *Quercus*, *Carya*, *Zelkova*, etc.) with few subtropical taxa (Taxodiaceae, *Engelhardia*) also suggests a coastal freshwater marsh (Fig. 6.13).

At the Late Miocene (before the MSC), Prosilio pollen flora shows that the presence and high abundance of mid- (mainly *Tsuga*) and high-altitude (*Abies* and *Picea*) conifers and high abundance of Cupressaceae in the northern Greece during this time-interval. The existence of mid- and high-altitude trees with Cupressaceae suggests that some elevated massifs existed in the region with a cool-temperate climate.

During the Late Miocene (latest Messinian)-Early Pliocene (earliest Zanclean), after the MSC, in Enez, West Seddülbahir, İntepe, Ptolemais Notio, Ptolemais Base, Lion of Amphipoli and Trilophos, the vegetation mainly characterized by herbs (Poaceae, Asteraceae Asteroideae, Asteraceae Cichorioideae, and Amaranthaceae-Chenopodiaceae, etc.) while forest assemblages were composed of mesothermic elements such as deciduous *Quercus*, *Carya*, *Zelkova*, etc. (in Fig. 6.13). This points out warm and dry climate conditions at low altitude in the region.

The studies based on CO₂ trend during the Miocene indicates that the presence of C4 plants (grasslands) at the Middle-Late Miocene (Kürschner *et al.*, 2008). Another significant change observed in the pollen floras is the increase in conifer (mainly *Cedrus*, *Abies* and *Picea*) (Melinte *et al.*, 2009). Calculation of the mean annual temperature based on pollen records have been performed using the “climatic amplitude” transfer function (Fauquette *et al.*, 1998). For İntepe samples 22-26, calculated mean annual temperature is 16.5°C (range: 15-18°C) (Melinte *et al.*, 2009). The same trend, i.e. the increase in conifers, is also observed in areas (Ptolemais Notio, Ptolemais Base, Lion of Amphipoli and Trilophos). This could be caused by some uplift of the massifs of the region as considered in the İntepe region.

During the Late Miocene-Early-middle Pliocene, after the MSC, mega-mesothermic and mesothermic elements were abundant in the Ponto-Euxinian region, in contrast to herbs with some development of *Artemisia* steppes during cooler phases of Pollen zone 1 (Popescu, 2006; Popescu *et al.*, 2010). Warm and humid climate existed in the region during interglacials. While mesothermic plants increased during the Late Miocene-Early Pliocene, the impoverishment in thermophilous trees since the Miocene is regarded as the result of a continuous and progressive decrease in temperature since 14 Ma (Zachos *et al.*, 2001; Darby, 2008).

Figure 6.13. Synthetic pollen diagrams of the studied localities. DSDP Site 380: S. Boroi, D. Biltekin and S.-M. Popescu, Grouping of plants follows Suc (1984): 1, Megathermic elements; 2, Mega-mesothermic elements; 3, *Cathaya*; 4, Mesothermic elements; 5, *Pinus*; 6, Meso-microthermic elements; 7, Microthermic elements; 8, Non-significant elements; 9, Cupressaceae; 10, Mediterranean xerophytes; 11, Herbs; 12, Steppe elements. The reference oxygen isotope curve is from Shackleton *et al.* (1990, 1995).

During the middle-late Zanclean, according to pollen record from DSDP Site 380, the vegetation was characterized by dominant herbs with weakly developed *Artemisia* steppes. Subtropical trees are not abundant. Mesothermic trees are frequent. Cupressaceae and *Cathaya* were rare during that time which corresponds to Pollen zone 2 (702.80-624 m) of Popescu *et al.* (2010). During that time climate was not very dry but cooler conditions existed. At the latest Zanclean, vegetation is depicted by herbs with higher amounts of *Artemisia*. Mega-mesothermic and mesothermic trees are then not very important. The high presence of herbs and *Artemisia* steppes during this time suggests that climate was cooler and drier.

Swamp forests (with *Glyptostrobus*, *Engelhardia*, Sapotaceae, *Nyssa*) seriously reduced at ca. 3.4 Ma but persisted during the Piacenzian (Late Pliocene) in contrast to significantly extending herbs with *Artemisia* steppes. Reduced deciduous mixed forests with mesothermic trees (deciduous *Quercus*, *Betula*, *Alnus*, *Liquidambar*, *Fagus*, *Carpinus orientalis*, *Carpinus betulus*, *Tilia*, *Acer*, *Ulmus*, *Zelkova*, *Carya*, *Pterocarya*) persisted too. During this time, herbs composed of Amaranthaceae-Chenopodiaceae, Poaceae, Asteraceae Asteroideae, Asteraceae Cichorioideae, Caryophyllaceae, Brassicaceae, *Polygonum*, *Rumex*, etc. strongly strengthened. However, steppe elements are low during that time. This documents that cooler and drier climatic conditions existed since 3.4 Ma.

There is some starting competition between moister-warmer phases to cooler-drier ones announcing the forthcoming interglacial-glacial phases. At ~3 Ma, paleoclimatic studies show that a warm event existed (Draut *et al.*, 2003). Climate was warmer than today in the Arctic regions of North America, Iceland, Russia and western-central Europe (Dowsett *et al.*, 1994). In addition, diatom studies in the deep sea cores (DSDP 266, ODP 699A, ODP 747A and Eltanin Core 50-28) Southern Ocean also support that climate was warmer during that time, summer surface temperatures are more than 3-4°C warmer than present at latitudes between 55° and 60°S (Barron, 1996).

This warmer period is also documented by pollen data of DSDP Site 380. During this time, forest communities were somewhat more developed, especially represented by mega-mesothermic trees such as mainly Taxodiaceae, in contrast to lower representation of herbs and steppe elements. Results of the GISS General Circulation Model (GCM) indicate the following temperatures in southern Europe: 6°C warmer over the Iberian Peninsula, 2-4°C warmer over the rest of southern Europe. Precipitation and winter soil moisture were close to modern levels (Chandler *et al.*, 1994).

At the beginning of Pleistocene (~2.6 Ma), when climate got cooler, glacial-interglacial cycles turned into a strong and rapid competition between forests and open vegetations. At the earliest Gelasian (Early Pleistocene), mega-mesothermic elements continued to rarefy. Nevertheless within them, Taxodiaceae (probably *Glyptostrobus*), *Engelhardia*, Sapotaceae, and *Nyssa* survived. Mesothermic elements composed of deciduous *Quercus*, *Betula*, *Alnus*, *Liquidambar*, *Fagus*, *Carpinus orientalis*, *Carpinus betulus*, *Tilia*, *Acer*, *Ulmus*, *Zelkova*, *Carya*, *Pterocarya*, etc. also almost disappeared. Herb phases were long with a prevalence of glacials over interglacials.

During the Earliest Ioanian, mega-mesothermic elements were in lower amounts (except several short peaks in the diagram). From time to time, mesothermic trees were abundant despite a longer and intense development of herbs with high *Artemisia* at the end of the interval. This demonstrates shorter and cooler interglacials in opposition with longer, colder and drier glacials. In the early-middle Ioanian, thermophilous forest elements enlarged again with strong repeated fluctuations between forest and open environments. After this time, herbs (with *Artemisia* steppes) have continued to increase up to the present-day.

7. CONCLUSIONS

This study has been carried out on 11 sections (in the Black Sea: DSDP 380 Site; in Thrace: Enez; in the Northern Aegean: İntepe, Seddülbahir, Burhanlı, Eceabat; in Macedonia: Ptolemais Notio and Ptolemais Base, in Greece: Prosilio, Lion of Amphipoli, and Trilophos. Most of the samples are well-dated by nannofossils. Pollen grains are generally well preserved in these sediments. Totally, 436 samples have been analysed, 378 of these samples (Early Pliocene-Early Plesitocene) are from the Site 380 borehole and the remaining 58 samples come from outcrops in different areas. In this study, 107 different taxa have been identified representative of various ecological environments. Pollen floras from these localities are rich and diversified and taxa have been arranged from the temperature requirement viewpoint according to the Nix's classification (1982):

- (1) megathermic elements (tropical): *Avicennia alba*, Euphorbiaceae, Rubiaceae and Rutaceae;
- (2) mega-mesothermic elements (subtropical): Taxodiaceae including *Taxodium*-type, *Engelhardia*, *Platycarya*, Sapotaceae, *Distylium*, *Microtropis fallax*, *Ginkgo*, *Loropetalum*, Arecaceae and *Cathaya*;
- (3) mesothermic elements (warm-temperate): *Carpinus orientalis*, *Juglans*, *Juglans* cf. *cathayensis*, *Carya*, *Pterocarya*, *Liquidambar orientalis*, *Platanus*, *Nyssa*, *Ulmus*, *Zelkova*, *Celtis* and *Eucommia*;
- (4) meso-microthermic elements (cool-temperate): *Cedrus* and *Tsuga*.
- (5) microthermic elements (boreal): *Abies* and *Picea*.

Nevertheless, an important factor could be as today the altitude which controls temperature and precipitation. Hence, vegetation should be organized in altitudinal belts after comparison with the current plant ecosystems (see Discussion chapter). Pollen records of the studied localities indicate changes in vegetation and climate in time and space. The vegetation is depicted mainly by herbs in İntepe, Burhanlı and Eceabat during the early Messinian. Before the Messinian Salinity Crisis (MSC), trees are relatively abundant in the İntepe area and mainly composed of mesothermic elements (deciduous *Quercus*, *Carya*, *Zelkova*, etc.) while vegetation is dominated by herbs (Poaceae, Asteraceae Asteroideae, Amaranthaceae-Chenopodiaceae, Caryophyllaceae, etc.) with *Artemisia*.

At the Late Miocene, mid- (mainly *Tsuga*) and high-altitude (*Abies* and *Picea*) coniferous trees were abundant in Prosilio (northern Greece). Another significant result in the pollen spectra of Prosilio is the abundance of Cupressaceae, the curve of which is consistent with those of mid- and high-altitude trees. This suggests that cooler climate existed in the region. Moreover, the abundance of altitudinal conifers displays that some uplift of the surrounding massifs occurred before that time.

Just after the MSC, at the latest Messinian-earliest Zanclean, herbs are still abundant in İntepe, Enez, West Seddülbahir, Trilophos, Lion of Amphipoli, Ptolemais Notio and Ptolemais Base (Poaceae, Amaranthaceae-Chenopodiaceae, Asteraceae, etc.). Among arboreal trees, mesothermic elements are common at this time. Within subtropical trees, *Ginkgo* is recorded at Enez. Today, *Ginkgo biloba* is the only species of Ginkgoaceae, living in China. The other prominent result is a significant increase in altitudinal trees (*Cedrus*, *Tsuga*, *Abies*, *Picea* and *Pinus*) which denotes some uplift in the Dardanelles area during the MSC.

Vegetation and climate of the Pliocene time is accurately documented by the high-resolution pollen record from DSDP Site 380 (SW Black Sea) characterized by various plant assemblages. Most of them were inherited from the Miocene. Several megathermic (tropical) trees suffered because of the progressive decrease in temperature since 14 Ma. Within megathermic (tropical) elements, it is appreciable to mention the last evidence for the Mediterranean region *s.l.* of *Avicennia alba*, a mangrove tree, in the Early Pleistocene from DSDP Site 380, at about 1.6 Ma. As several records of *Avicennia* are known from the underlying sediments of DSDP Site 380, one may postulate that a residual *Avicennia* mangrove persisted on the coastal areas in the Ponto-Euxinian region.

On the whole according to Site 380 pollen record, it is established that a strong competition between arboreal trees and herbs with *Artemisia* steppes started in the Black Sea region at the earliest Pliocene which became more pronounced at 2.6 Ma. During the Early Pliocene (at the end of the Zanclean), herbs composed of mainly Amaranthaceae-Chenopodiaceae, Poaceae, Asteraceae Asteroideae, Asteraceae Cichorioideae developed with *Artemisia* steppes. During this time, Amaranthaceae-Chenopodiaceae reaches up ca. 88%. In addition, subtropical trees were not abundant as mesothermic plants. This suggests cooler and probably drier climatic conditions in the region.

Swamp forests (with *Glyptostrobus*, *Engelhardia*, Sapotaceae, *Nyssa*) persisted during the Piacenzian (Late Pliocene) in contrast to extending herbs with *Artemisia* steppes. Reduced deciduous mixed forests with mesothermic trees (deciduous *Quercus*, *Betula*, *Alnus*,

Liquidambar, *Fagus*, *Carpinus orientalis*, *Carpinus betulus*, *Tilia*, *Acer*, *Ulmus*, *Zelkova*, *Carya*, *Pterocarya*) persisted too. During this time, vegetation was depicted by herbs such as mainly Amaranthaceae-Chenopodiaceae, Poaceae, Asteraceae Asteroideae, Asteraceae Cichorioideae, Caryophyllaceae, Brassicaceae, *Polygonum*, *Rumex*, etc., containing some aquatic plants e.g., *Sparganium*, *Potamogeton*, *Typha*, etc. Steppe elements are composed of *Artemisia*, *Ephedra*, *Hippophae rhamnoides*). They are in small amounts during that time. This indicates that cooler and drier climatic conditions existed.

At the earliest Gelasian (Early Pleistocene), mega-mesothermic elements strongly rarefied. Among them, Taxodiaceae (probably *Glyptostrobus*), *Engelhardia*, Sapotaceae, and *Nyssa* persisted. Deciduous mixed forest assemblages composed of mesothermic trees (warm-temperate) such as deciduous *Quercus*, *Betula*, *Alnus*, *Liquidambar*, *Fagus*, *Carpinus orientalis*, *Carpinus betulus*, *Tilia*, *Acer*, *Ulmus*, *Zelkova*, *Carya*, *Pterocarya*, etc. almost disappeared too. Within this vegetation belt, a riparian forest also perpetuated with *Salix*, *Alnus*, *Carya*, *Carpinus orientalis*, *Zelkova*, *Ulmus*, *Liquidambar*, etc. Very weak fluctuations are recorded opposing thermophilous trees and the highly dominant herbs. Herb phases were long with a prevalence of glacials over interglacials.

During the Earliest Ionian (Middle Pleistocene), vegetation is characterized by mega-mesothermic elements with low amounts (except some peaks in the diagram), high amounts of mesothermic trees and longer temporal development of herbs with high *Artemisia* at the end of the interval. This demonstrates shorter and cooler interglacials in opposition with longer, colder and drier glacials. At the early-middle Ionian (394.50-302.40 m), is observed increase in thermophilous forest elements with strong repeated fluctuations between forest and open (with higher *Artemisia* steppes) environments. This indicates prominent climatic fluctuations between interglacials and glacials. After this period, herbs mostly Amaranthaceae-Chenopodiaceae, Poaceae, Asteraceae Asteroideae, Asteraceae Cichorioideae, Brassicaceae, etc., with *Artemisia* steppes (*Artemisia*, *Ephedra* and *Hippophae rhamnoides*) have continued to increase until today.

An outstanding result is the development of *Artemisia* steppes in Anatolia. The earliest settlement of *Artemisia* steppes in Anatolia belongs to the Early Miocene (Aquitanian~23.0-20.4 Ma) in the Ermenek region, central Taurus (Biltekin, unpublished). *Artemisia* is common in this region during the Early Miocene (ca. 10%). The noticeable change is located between the Miocene and the Pliocene. The high-resolution pollen record of DSDP Site 380 clearly documents the development of *Artemisia* steppes in Anatolia. The main development of *Artemisia* with non-boreal (herbaceous) vegetation in Anatolia started during the Early

Pliocene in the Ponto-Euxinian region. Indeed, at Eceabat, Burhanlı and İntepe localities, *Artemisia* is found in small amounts. The increase of *Artemisia* continued through the Pliocene (reaching up about 62%).

At the beginning of the Pleistocene (~2.6 Ma), when started glacial-interglacial cycles, *Artemisia* steppes continued to develop through the Pleistocene until today. The development of *Artemisia* in Anatolia could both relate with the uplifting of the Tibetan Plateau, global cooling and reinforcement of the Asian monsoon.

At last, thermophilous elements such as *Pterocarya* and *Liquidambar* have persisted on coastal slopes of Anatolia. They have been observed in the pollen floras since the Early Miocene (Aquitanian). Some others, such as *Carya*, *Parrotia persica*, *Cathaya* and *Tsuga* persisted in the region. Their disappearance seems to have happened during the Middle Pleistocene. However, Taxodiaceae disappeared from the Euxinian-Hyrcanian region sub-recently. The persistence of relict plants in Anatolia can be explained by the significant influence of the Asian monsoon which takes place along a longitudinal gradient. This brings about the preservation of relict plants in Anatolia.

REFERENCES

- Adams, C. C., Benson, R. H., Kidd, R. B., Ryan, W. B. F. and Wright, R. C., 1977.** The Messinian salinity crisis and evidence of Late Miocene eustatic changes in the world ocean. *Nature*, 269, 383–6.
- Akcar, N and Schlüchter, C., 2005.** Paleoglaciations in Anatolia: A schematic review and first results, *Eiszeitalter und Gegenwart* 55, 102-121, 8 Abb., Hannover.
- Akgün, F., Akyol, E., 1999.** Palynostratigraphy of the coal-bearing Neogene deposits graben in Büyük Menderes Western Anatolia. *Geobios*, 32, 3, 367–383.
- Akman, Y., Ketenoglu, O., 1986.** The climate and vegetation of Turkey. In “Plant Life of SW Asia”, *Proceedings of the Royal Society of Edimburgh*, Section B, 113-122.
- Akman, Y., Quézel, P., Ketenoglu, O., Kurt, F., 1993.** Analyse syntaxonomique des forêts de *Liquidambar orientalis* en Turquie. *Ecologia Mediterranea*, 19, 49–57.
- Akman, Y., and Quézel, P., 1996.** La steppe centro-anatolienne: interpretation.phytoécologique. Actes des 7èmes Rencontres de l’A.R.P.E. Provence Alpes-Côte d’Azur. Colloque scientifique international Bio’Mes, Digne, 127–131.
- Aksu, A.E., Hiscott, R.N. and Yafiar, D., 1999.** Oscillating Quaternary water levels of the Marmara Sea and vigorous outflow into the Aegean Sea from the Marmara Sea-Black Sea drainage corridor. *Marine Geology* 153, 275–302.
- Aldanmaz E., 2002.** Mantle source characteristics of alkali basalts and basanites in an extensional intracontinental plate setting, western Anatolia, Turkey: Implications for multi-stage melting. *Int. Geol. Rev.*, 44:440-457.
- Altner, D., Koçyiğit, A., Farinacci, A., Nicosia, U. and Conti, M. A., 1991.** Jurassic, Lower Cretaceous stratigraphy and paleogeographic evolution of the southern part of north-western Anatolia. *Geologica Romana*, 28, 13-80.
- Amigues, S., 2007.** Le styrax et ses usages antiques. *Journal des Savants*, Juillet-Décembre, 2, 263–318.
- An, Z.S., Song, Y.G., Zhang, P.Z., Wang, E.Q., Wang, S. M., Qiang, X.K., Li, L., Chang, H., Liu, X.D., Zhou, W.J., Liu, W.G., Cao, J.J., Li, X.Q., Shen, J., Liu, Y., Ai, L., 2006.** Changes of the monsoon-arid environment in China and growth of the Tibetan Plateau since the Miocene. *Quaternary Science* 26(5), 678–693.
- Armijo, R., Meyer, B., Hubert, A., Barka, A., 1999.** Westward propagation of the North Anatolian fault into the northern Aegean: timing and kinematics. *Geology* 27, 267–270.
- Axelrod, D. I., Al-Shehbaz, I., Raven, P., 1996.** History of the modern flora of China. In: Zhang, Aoluo, Wu, Sugong (Eds.), *Floristic Characteristics and Diversity of East Asian Plants*. InSpringer-Verlag, Berlin, pp. 43–55.

- Bachiri Taoufiq, N., 2000.** Les environnements marins et continentaux du corridor rifain au Miocène supérieur d'après la palynologie. Thesis, Univ. Casablanca, 206 p.
- Bachiri Taoufiq, N., Barhoun, N., Suc, J.-P., Méon, H., Elaouad, Z., Benbouziane, A., 2000.** Environnement, végétation et climat du Messinien au Maroc. *Paleontologia I Evolució*, 32-33, 127-145.
- Bachiri Taoufiq, N., Barhoun, N., Suc, J.-P., Meon, H., Elaouad, Z. and Benbouziane, A., Jiménez-Moreno, G. et al., 2001.** Environment, végétation et climat du Messinien au Maroc. *Paleontologia i Evolucio*, 32-33, 127-138.
- Barka, A. and Hancock, P.L., 1984.** Neotectonic deformation patterns in the convex-northward arc of the North Anatolian fault zone: Dixon, J. E, and Robertson, A.H.F., eds. *The Geological Evolution of the Eastern Mediterranean. Geological Society of London, Special Publication*, 17, 285-296.
- Barka, A., 1985.** Kuzey Anadolu Faz Zonundaki bazı Neojen-Kuvaterner havzaların jeolojisi ve tektonik evrimi. *Ketin Simpozyumu Kitabı*, 209-227.
- Barka, A.A., 1992.** The North Anatolian fault Zone. *Ann. Tectonicae*, 6, 164-195.
- Barron, J. A., 1996.** Diatom constraints on the position of the Antarctic Polar Front in the middle part of the Pliocene. *Marine Micropaleontology* 27 (1996) 195-213.
- Batten, D.J., and Grenfell, H.R., 1996.** Botryococcus. In Jansonius, J., and McGregori, D.C. (Eds), *Palynology: principles and applications*. Am. Assoc. Stratigr. Palynol. Found., 1:205-214.
- Beaudouin, C., 2003.** Effets du dernier cycle climatique sur la végétation et la sédimentation de la plate-forme du golfe du Lion d'après la palynologie. *PhD thesis*, Université Claude Bernard Lyon-1, France, 403 p.
- Benda, L., 1971.** Grundzüge einer pollenanalytischen Gliederung des Türkischen Jungtertiärs. *Beihefte zum Geologischen Jahrbuch*, 113, 1-45.
- Benson, W. E., Sheridan, R. E., and Shipboard Scientific Party, 1978.** Site 391: Blake-Bahama Basin. In Benson, W. E., Sheridan, R. E., et al., *Init. Repts. DSDP*, 44: Washington (U.S. Govt. Printing Office), 153-336.
- Bertini, A., Corradini, D., Suc, J.-P., 1995.** On *Galeacysta etrusca* and the connections between the Mediterranean and the Paratethys. *Rom. J. Stratigr.* 76 (suppl. 7), 141-142.
- Bertini, A., 2002.** Palynological evidence of Upper Neogene environments in Italy. *Acta Univ. Carolinae, Geol.*, 46(4), 15-25.
- Bessedik M., 1985.** *Reconstitution des environnements miocènes des régions nord-ouest méditerranéennes à partir de la palynologie*. Thèse, Univ. Montpellier 2, 162 p.

- Billups, K., Kelly, C. and Pierce, E., 2008.** The late Miocene to early Pliocene climate transition in the Southern Ocean. *Palaeogeography, Palaeoclimatology, Palaeoecology* 267 (2008) 31–40.
- Bornovas, J., Tsiambaou, Th., 1983.** Geological map of Greece. Institute of geology and mineral exploration 1:500.000 scale.
- Bottema, S., 1986.** Late Quaternary and modern distribution of forest and some taxa in Turkey. *Proceedings of the Royal Society of Edinburgh*, **89B**, 103–111.
- Boydak, M., 2006.** Biology and silviculture of Turkish Red Pine (*Pinus brutia* Ten.). Lazer Ofset Matbaa Tesisleri San, Ankara, 253 p.
- Browicz, K., 1982-1994.** Chorology of Trees and Shrubs in the S-W Asia and Adjacent regions. Polish Academy of Sciences, Institute of Dendrology, 10 volumes.
- Bukry, D., 1973.** Low Latitude Cocolith Biostratigraphic Zonation. In: Edgar, N.T., Saunders, J.B., et al. (Eds.), *Initial Reports DSDP*, vol. 15. US Govt. Printing Office, Washington, pp. 685–703.
- Cambon G. et al., 1997.** Modern pollen deposition in the Rhône delta area (lagoon and marine sediments) France. *Grana*, **36**, 105-113.
- Chandler, M., Rind, D. and Thompson, R., 1994.** Joint investigations of the Middle Pliocene climate II: GISS GCM Northern Hemisphere results. *Glob. Planet. Change* 9, 197–219.
- Chikhi, H., 1992.** Une palynoflore méditerranéenne à subtropicale au Messinien préévaporitique en Algérie. *Géol. Médit.* 19 (1), 19-30.
- CIESM, 2007.** The Messinian Salinity Crisis from mega-deposits to microbiology-Consensus Report. CIESM Workshop Monographs, Almeria, 7-10 November, 2007.
- Cita, M.B., Wright, R.C., Ryan, W.B.F., Longinelli, A., 1978.** Messinian palaeoenvironments. In: Hsu, K.J., Montadert, L. (Eds.), *Initial Reports of the Deep Sea Drilling Project*, vol. 42 part 1. US Government Printing Office, Washington, pp. 1003–1035.
- Cita, M.B. and Colombo, L., 1979.** Sedimentation in the latest Messinian at Capo Rossello (Sicily). *Sedimentology* 26, 497–522.
- Clauzon, G., 1996.** Limites de sequence et evolution geodynamique. *Geomorphologie*, 1, 3-22.
- Clauzon, G., Suc, J.-P., Gautier, F., Berger, A., Loutre, M.-F., 1996.** Alternate interpretation of the Messinian salinity crisis: Controversy resolved? *Geology* 24 (4), 363-366.
- Clauzon, G., Rubino, J.-L., Casero, P., 2001.** Regional modalities of the Messinian salinity crisis in the framework of a two phases model. Late Miocene to Early Pliocene

environments and ecosystems. 2nd EEDEN Workshop. Sabadell, Spain, pp. 17–18.

- Clauzon G., Suc J.-P., Popescu S.-M., Marunteanu M., Rubino J.-L., Marinescu F. et Melinte M.C., 2005.** Influence of the Mediterranean sea-level changes over the Dacic Basin (Eastern Paratethys) in the Late Neogene. The Mediterranean Lago Mare facies deciphered. *Basin Research*, 17 : 437-562.
- Combourieu-Nebout, N., 1987.** Les premiers cycles glaciaires– interglaciaires en région méditerranéenne d'après l'analyse palynologique de la série Plio–Pleistocène de Crotona (Italie méridionale). Thesis, Univ. Montpellier II, 161 pp. (unpublished).
- Cour, P., 1974.** Nouvelles techniques de détection des flux et de retombées polliniques: étude de la sédimentation des pollens et des spores à la surface du sol. *Pollen et Spores* 16 (1), 103-141.
- Cullen, H.M. and deMenocal, P.B., 2000.** North Atlantic influence on Tigris-Euphrates streamflow. *Int. J. Climatology*, 20: 853-863.
- Çağatay, N.M., Görür, N., Alpar, B., Saatçılar, R., Akkök, R., Sakıncı, M., Yüce, H., Yalıtırak, C., Kuşçu, I., 1998.** Geological evolution of the Gulf of Saros, NE Aegean Sea. *Geo Marine Letter* 18, 1–9.
- Çağatay, M.N., Görür, N., Alpar, B., Saatçılar, R., Akkök, R., Sakıncı, M., Yüce, H., Yalıtırak, C., Kuşçu, İ., 1999.** Geological evolution of the Gulf of Saros, NE Aegean Sea. *Geo Mar. Lett.* 18, 1–9.
- Çağatay, N.M., Görür, N., Flecker, R., Sakıncı, M., Tünoğlu, C., Ellam, R., Krijgsman, W., Vincent, S., Dikbaş, A., 2006.** Paratethyan–Mediterranean connectivity in the Sea of Marmara region (NW Turkey) during the Messinian. *Sedimentary Geology* 188–189, 171–187.
- Çağatay, M. N., Suc, J.-P., Clauzon, G. and Melinte, M. C., 2007.** Messinian in Northwest Turkey: implications for paleogeographic evolution and water mass exchange between Paratethys and Mediterranean. The Messinian Salinity Crisis from mega-deposits to microbiology-Consensus Report. CIESM Workshop Monographs, Almeria, 7-10 November, 2007.
- Çağatay, M. N., Eris, K., Ryan, W.B.F., Sancar, U., Polonia, A., Akçer, S. Biltekin, D., Gasperini, L., Gorur, N. Lericolais G., Bard, E., 2009.** Late Pleistocene–Holocene evolution of the northern shelf of the Sea of Marmara. *Marine Geology*, 265: 87-100.
- Darby, D.A., 2008.** Arctic perennial ice cover over the last 14 million years. *Paleoceanography* 23, pp. 1–9.

- Davis, O. K. And Ellis, B., 2010.** Early occurrence of sagebrush steppe, Miocene (12 Ma) on the Snake River Plain. *Review of Palaeobotany and Palynology* 160 (2010) 172–180.
- Dean, W.Y., Martin, E., Monod, O., Demir, O., Rickards, A. B., Bultynck, P. And Bozdoğan, N., 1997.** Lower Paleozoic stratigraphy, Karadere- Zirze area, central Pontides, northern Turkey. In: Göncüoğlu, M. C. and DERMAN, A. S. (eds) *Early Paleozoic Evolution in NW Gondwana. Turkish Association of Petroleum Geologists Special Publications*, 3, 32-38.
- de Bruijn, H., Saraç, G., van den Hoek Ostende, L., Roussiakis, S., 1999.** The status of the genus name *Parapodemus* Schaub, 1938; new data bearing on an old controversy. *Deinsea* 7, 95–112.
- DeDeckker, P., 1988.** Biological and sedimentary facies of Australian salt lakes. *Palaeogeog., Palaeoclimatol., Palaeoecol.*, 62:237–270.
- Diniz, F., 1984.** Etude palynologique du bassin pliocène de Rio Maior. *Paléobiologie Continentale* 14, 259-267.
- Dowsett, H.J. and Poore, R.Z., 1991.** Pliocene sea surface temperatures of the North Atlantic Ocean at 3.0 Ma. *Quat. Sci. Rev.*, 10: 189-204.
- Dowsett, H.J., Cronin, T.M., Poore, R.Z., Thompson, R.S., Whatley, R.C. and Wood, A.M., 1992.** Micropaleontological evidence for increased meridional heat transport in the North Atlantic Ocean during the Pliocene. *Science*, 258:1133-1135.
- Dowsett, H.J., Thompson, R.S., Barron, J.A., Cronin, T.M., Fleming, R.F., Ishman, S.E., Poore, R.Z., Willard, D.A., Holtz, T.R., 1994.** Joint investigations of the Middle Pliocene climate: I. PRISM paleoenvironmental reconstructions. *Global and Planetary Change* 9, 169–195.
- Dowsett, H. J., Chandler, M. A., Cronin, T. M. and Dwyer, G. S., 2005.** Middle Pliocene sea surface temperature variability. *Paleoceanography*, 20.
- Draut, A. E., Raymo, M. E., McManus, J. F. and Oppo, D. W., 2003.** Climate stability during the Pliocene warm period. *Paleoceanography* 18, 1078.
- Drivaliari, A., 1993.** Images polliniques et paléoenvironnements au Néogène supérieur en Méditerranée orientale. Aspects climatiques et paléogéographiques d'un transect latitudinal (de la Roumanie au delta du Nil). PhD Thesis, Univ. Montpellier 2, 333 p.
- Ehlers, E., 1960.** Bericht über die bisher im Rahmen der Expertise Ptolemas durchgeführten geologischen und paläontologischen Untersuchungen. Internal report, nr. 66164, Archiv Bundesanst. *Geowiss. and Rohst.*, Hannover.
- Erinç, S., 1996.** *Klimatoloji ve Metodları* (genişletilmiş 4.cü baskı).

- Erinç, S., 1959.** Regional and seasonal distribution of climatic elements in Turkey and its Dynamic-Genetic Background, *İstanbul Üniversitesi Coğrafya Enstitüsü, International Edition*, No:5.
- Eronen, J., Ataabadia, M. M., Micheels A., Karme, A., Bernor, R. L. And Fortelius, M., 2009.** Distribution history and climatic controls of the Late Miocene Pikermian chronofauna. *PNAS*, 2009, vol. 106, no. 29, 11867–11871.
- Fauquette, S., Guiot, J., Suc, J.-P., 1998a.** A method for climatic reconstruction of the Mediterranean Pliocene using pollen data. *Palaeogeogr. Palaeoclimatol, Palaeoecol.* 144, 183– 201.
- Fauquette, S., Suc, J.-P., Guiot, J., Diniz, F., Feddi, N., Zheng, Z., Bessais, E., Drivaliari, A., 1999.** Climate and biomes in the west Mediterranean area during the Pliocene. *Palaeogeogr. Palaeoclimatol. Palaeoecol.* 152, 15– 36.
- Flower, B. and Kennett, J.P. 1993.** Middle Miocene ocean-climate transition: high resolution oxygen and carbon isotopic records from Deep Sea Drilling Project Site 588A, Southwest Pacific. *Paleoceanography*, 8:811-843.
- Flower, B. and Kennett, J.P. 1994.** The middle Miocene climatic transition: East Antarctic ice sheet development, deep ocean circulation and global carbon cycling. *Palaeogeography, Palaeoclimatology, Palaeoecology*, 108:537-555.
- Fornaciari, E., Iaccarino, S., Mazzei, R., Rio, D., Salvatorini, G., Bossio, A. and Monteforti, B., 1997.** Calcareous plankton biostratigraphy of the Langhian historical stratotype. In: A. Montanari, G. S., Odin and R. Coccioni (eds), *Miocene Stratigraphy: an Integrated Approach. Elsevier Science*:89-106.
- Gheorghian, M., 1978.** In D. A. Ross, N. P. , Neprochnov, and Micropaleontological investigations sediments from sites 377, 380 and 381 of Leg42B. *Initial reports of the Deep Sea Drilling*, v.41, p.783-788.
- Gillet, S., Gramann, F., Steffens, P., 1978.** Neue biostratigraphische Ergebnisse aus dem brackischen Neogen an Dardanellen und Marmara–Meer (Türkei). *Newsletters of Stratigraphy* 7, 53–64.
- Gillet H., 2004.** *La stratigraphie tertiaire et la surface d'érosion messinienne sur les marges occidentales de la mer Noire: stratigraphie sismique haute resolution.* Thèse, Univ. Bretagne occidentale, 259 p.
- Gong, W., Chen, C., Dobeš, C., Fu, C.-X., Koch, M., 2008.** Phylogeography of a living fossil: Pleistocene glaciations forced *Ginkgo biloba* L. (Ginkgoaceae) into two refuge areas in China with limited subsequent postglacial expansion. *Molecular Phylogenetics and Evolution* 48 (2008) 1094–1105.
- Görür N., Cagatay M.N., Sakinç M., Sümengen M., Sentürk K., Yaltırak C., Tchapylyga A., 1997.** Origin of the Sea of Marmara as deduced from neogene

to Quaternary paleogeographic evolution of its frame. *Intern. Geol. Rev.*, 39 : 342-352.

Görür N., Cagatay M.N., Sakiç M., Akkök R., Tchapylya A., Natalin B., 2000.

Neogene Paratethyan succession in Turkey and its implications for the palaeogeography of the Eastern Paratethys. In "Tectonics and Magmatism in Turkey and the surrounding area", Bozkurt E., Winchester J.A., Piper J.D.A. éd., *Geol. Soc. London*, spec. publ., 173: 251-29.

Graham, A., 1996. A contribution to the geological history of the Compositae. In: Hind D, Beentje H, eds. *Proceedings of the Kew International Compositae Conference 1994*, Vol. 1. London: Royal Botanic Gardens, Kew, pp. 123–140.

Gray, J., 1960. Fossil chlorophycean algae from the Miocene of Oregon. *J. Paleontol.*, 34:453–463.

Guy-Ohlson, D., 1992. Botryococcus as an aid in the interpretation of palaeoenvironment and depositional processes. *Rev. Palaeobot. and Palynol.*, 71:1–15.

Hably, L., Kvaček Z. and Szakmany, G., 1996. Flora, vegetation and climate of the Pliocene age in Hungary. *Studia Naturalia*, 9: 99–105.

Hajar, L., Francois, L., Khater, C., Jomaa, I., Deque, M., Cheddadi, R., 2010. Cedrus libani (A. Rich) distribution in Lebanon: Past, present and future. *C. R. Biologies* 333 (2010) 622–630.

Haywood, A. M., Dowsett, H. J., Valdes, P. J., Lunt, D. J., Francis, E. J. And Sellwood, B. W., 2009. Introduction. Pliocene climate, processes and problems. *Phil. Trans. R. Soc. A* 2009 367, 3-17.

Herbert, T. D., and Schuffert, J., 1998. Alkenone unsaturation estimates of late Miocene through late Pliocene sea surface temperature changes, ODP Site 958, *Proceedings of the Ocean Drilling Program*, Scientific Results, v. 159T, p. 17-22.

Heusser L. 1988. Pollen distribution in marine sediments on the continental margin of Northern California. *Marine Geology*, 80, 131-147.

Hristovski, K. D., Olson, L., Hild, N., Peterson, D. and Burge, S., 2007. The municipal solid waste system and solid waste characterization at the municipality of Veles, Macedonia. *Waste Management*. Volume 27, Issue 11, 2007, Pages 1680-1689.

Hsü, K. J., 1972. When the Mediterranean dried up: *Sci. Amer.*, v. 227, p. 26-36.

Hsü, K. J., 1974. The Miocene desiccation of the Mediterranean sea and its climatic and Zoogeographic implications: *Naturwissenschaften*, v. 61, p. 137-142.

Hsü, K. J., and Bernoulli, D., 1978. Genesis of the Tethys and the Mediterranean: *Init. Rep. Deep-Sea Drilling Project*, v. XLII, no. 1, p. 943-950.

- Hsü, K.J. and Giovanoli, F., 1979.** Messinian event in the Black Sea. *Palaeogeogr. Palaeoclimatol. Palaeoecol.*, 29(1-2), 75-94.
- Ioakim, C., Rondoyanni, T. and Mettos, A., 2005.** **The Miocene Basins of Greece (Eastern Mediterranean) from a palaeoclimatic perspective,** *Revue de Paléobiologie, Genève* (décembre 2005) 24 (2) : 735-748.
- Ivanov, D., Ashraf, A.R., Mosbrugger, V., Palmarev, E., 2002.** Palynological evidence for Miocene climate change in the Forecarpathian Basin (central Paratethys, NW Bulgaria). *Palaeogeogr. Palaeoclimatol. Palaeoecol.* 178, 19– 37.
- Ivanov, D., Ashraf, R. A., Utescher, T., Mosbrugger, V. And Slavomirova, E., 2007.** Late Miocene vegetation and climate of the Balkan region: palynology of the Beli Breg Coal Basin sediments, *Geologica Carpathica*, August 2007, 58, 4, 367—381.
- Jansen, E., Overpeck, J., Briffa, K.R., Duplessy J.-C., Joos, F., Masson-Delmotte, V., Olago, D., Otto-Bliesner, B., Peltier, W.R., Rahmstorf, S., Ramesh, R., Raynaud, D., Rind, D., Solomina, O., Villalba, R., and Zhang, D., 2007.** Palaeoclimate. In *Climate Change 2007: The Physical Science Basis. Contribution of Working Group I to the Fourth Assessment Report of the Intergovernmental Panel on Climate Change.* S. Solomon, D. Qin, M. Manning, Z. Chen, M. Marquis, K.B. Averyt, M. Tignor, and H.L. Miller, Eds. Cambridge University Press, pp. 433-497.
- Jiménez-Moreno G., 2005.** *Utilización del análisis polínico para la reconstrucción de la vegetación, clima y estimación de paleoaltitudes a lo largo de arco alpino europeo durante el Mioceno (21-8 Ma).* Thèse, Univ. Grenade et C. Bernard – Lyon 1, 311 p.
- Jiménez-Moreno, G., Rodríguez-Tovar, F.-J., Pardo-Igúzquiza, E., Fauquette, S., Suc, J.-P., Müller, P., 2005.** High-resolution palynological analysis in late early-middle Miocene core from the Pannonian Basin, Hungary: Climatic changes, astronomical forcing and eustatic fluctuations in the Central Paratethys. *Palaeogeogr., Palaeoclimatol., Palaeoecol.* 216 (1-2), 73-97.
- Jiménez-Moreno G., 2006.** Progressive substitution of a subtropical forest for a temperate one during the middle Miocene climate cooling in Central Europe according to palynological data from cores Tengelic-2 and Hidas-53 (Pannonian Basin, Hungary). *Review of Palaeobotany and Palynology* 142 (2006) 1–14.
- Jiménez-Moreno, G., and Suc, J.-P., 2007.** Middle Miocene latitudinal climatic gradient in Western Europe: Evidence from pollen records. *Palaeogeography, Palaeoclimatology, Palaeoecology* 253 (2007) 224–241.

- Jiménez-Moreno, G., Fauquette, S., Suc, J.-P., Abdul-Aziz, H., 2007a.** Early Miocene repetitive vegetation and climatic changes in the lacustrine deposits of the Rubielos de Mora Basin (Teruel, NE Spain). *Palaeogeography, Palaeoclimatology, Palaeoecology* 250, 101–113.
- Jiménez-Moreno, G., Abdul-Aziz, H., Rodríguez-Tovar, F.J., Pardo-Igúzquiza, E., Suc, J.-P., 2007b.** Palynological evidence for astronomical forcing in Early–Middle Miocene lacustrine deposits from Rubielos de Mora Basin (NE Spain). *Palaeogeography, Palaeoclimatology, Palaeoecology* 252, 601–616.
- Jiménez-Moreno, G., Fauquette, S., Suc, J.-P., 2008.** Vegetation, climate and paleoaltitude reconstructions of eastern alpine mountain ranges during the Miocene based on pollen records from Austria, Central Europe. *Journal of Biogeography* 35, 1638–1649.
- Jiménez-Moreno, G., Mandić, O., Harzhauser, M., Pavelić, D., Vranjković, A., 2008.** Vegetation and climate dynamics during the early Middle Miocene from Lake Sinj (Dinaride Lake System, SE Croatia). *Review of Palaeobotany and Palynology* 152 (2008) 237–245.
- Jiménez-Moreno G., Fauquette S., Suc J.-P., 2009.** Miocene to Pliocene vegetation and climate estimates in the Iberian Peninsula from pollen data. *Review of Palaeobotany and Palynology*, under press.
- Karabörk M. Ç., E. Kahya, and M. Karaca, 2005.** The influences of the Southern and North Atlantic oscillations on climatic surface variables in Turkey. *Hydrological Processes*, 19, 1185–1211.
- Karaca M., A. Deniz, and M. Tayanç, 2000.** Cyclone Track Variability over Turkey in Association with Region Climate. *International Journal of Climatology*. 20, 1225–1236.
- Kasaplıgil, B., 1977.** A Late-Tertiary conifer-hardwood forest from the vicinity of Güvem village, near Kızılcahamam, Ankara. *Bulletin of the Mineral Research and Exploration Institute of Turkey* 88, 25–33.
- Kloosterboer-van Hoeve, M. L., Steenbrink, J., Brinkhuij, H., 2001.** A short-term cooling event, 4.205 million years ago, in the Ptolemais Basin, northern Greece. *Palaeogeography, Palaeoclimatology, Palaeoecology* 173 (2001) 61–73.
- Kloosterboer-van Hoeve, M. L., Steenbrink, Visscher, H., J., Brinkhuis, H., 2006.** Millennial-scale climatic cycles in the Early Pliocene pollen record of Ptolemais, northern Greece. *Palaeogeography, Palaeoclimatology, Palaeoecology* 229 (2006) 321–334.
- Kovar-Eder, J., Kvacek, Z., Martinetto, E. and Roiron, P., 2006.** Vegetation of southern Europe around the Miocene/Pliocene boundary (7–4 Ma—The High Resolution Interval I) as reflected in the macrofossil record, in Agusti, J., Oms,

- O., and Meulenkamp, J.E., eds., Late Miocene to Early Pliocene Environment and Climate Change in the Mediterranean Area: *Palaeogeography, Palaeoclimatology, Palaeoecology*, v. 238, p. 321–339.
- Kovar-Eder, J., Henriette, Jechorek, H., Kvacek, Z. and Parashiv, V., 2008.** The integrated plant record: an essential tool for reconstructing Neogene zonal vegetation in Europe. *Palaios*, 2008, v. 23, p. 97–111.
- Krijgsman, W., Stoica, M., Vasiliev, I., Popov, V.V., 2010.** Rise and fall of the Paratethys Sea during the Messinian Salinity Crisis. *Earth and Planetary Science Letters* 290 (2010) 183–191.
- Kürschner, W. M., Van der Burgh, J., Visscher, H. And Dilcher, D. L., 1996.** Oak leaves as biosensors of Late Neogene and Early Pleistocene paleoatmospheric CO₂ concentrations. *Mar. Micropaleontol.* 27, 299–312.
- Kürschner, W. M., Kvacek, Z. and Dilcher, D. L., 2008.** The impact of Miocene atmospheric carbon dioxide fluctuations on climate and the evolution of terrestrial ecosystems. *Pnas*, vol. 105, no. 2, 449-453.
- Lawrence, D. M., and Slater, A. G., 2005.** A projection of severe nearsurface permafrost degradation during the 21st century, *Geophys. Res. Lett.*, 32, L2440.
- LePage, B. A., 2003.** A new species of *Thuja* (Cupressaceae) from the Late Cretaceous of Alaska: implications of being evergreen in a polar environment *American Journal of Botany*. 2003;90:167-174.
- Li, J.J., 1991.** The uplift of the Qinghai–Xizang Plateau and its effect on environment. In: Liu, T. (Ed.), *Quaternary Geology and Environment in China*. Science Press, Beijing, pp. 265– 272.
- Li, J. J., Fang, X. M., 1999.** Uplift of the Tibetan Plateau and environmental changes. *Chinese Science Bulletin*, 44, 2117–2124.
- Lisiecki, L.E., Raymo, M.E., 2007.** Plio-Pleistocene climate evolution: trends and transitions in glacial cycle dynamics. *Quaternary Science Reviews*, 26, 56-69.
- Lolis, C. J., Bartzokas, A., Metaxas, D.A., 1999.** Spatial covariability of the climatic parameters in the Greek area. *Int. J. Climatol.* 19:185–196.
- Loukas, L., Vasiliades, Dalezios, N. R. And Domenikiotis, C., 2001.** Rainfall-Frequency Mapping for Greece, *Phys. Chem. Earth (B)*, Vol. 26, No. 9, pp. 669-674.
- Mariolopoulos, E. G., 1938.** Climate of Greece. Athens, Greece.
- Marunteanu, M. and Papaianopol, I., 1998.** Mediterranean calcareous nannoplankton in the Dacic Basin. *Romanian Journal of Stratigraphy*, 78:115-121.
- McKenzie, D. P., 1972.** Active tectonics of the Mediterranean region: *Geophys. Jour. Roy. Astr. Soc.*, v. 30 , p. 109.
- Melinte-Dobrinescu, M. C., Suc, J.-P., Clauzon, G., Popescu, S-M., Armijo, R., Meyer,B., Biltekin, D., Çağatay, M. N., Uçarkuş, G., Jouannic, G.,**

- Fauquette, S., and Çakır, Z., 2009.** The Messinian Salinity Crisis in the Dardanelles region: Chronostratigraphic constraints *Palaeogeography, Palaeoclimatology, Palaeoecology* 278 (2009) 24–39.
- Meulenkamp, J.E., Kovac, M., Cicha, I., 1996.** On Late Oligocene to Pliocene depocentre migrations and the evolution of the Carpathian-Pannonian system. *Tectonophysics* 266, 310-317.
- Meulenkamp, J. E., Sissingh, W., 2003.** Tertiary palaeogeography and tectonostratigraphic evolution of the Northern and Southern Peri-Tethys platforms and the intermediate domains of the African-Eurasian convergent plate boundary zone. *Palaeogeography, Palaeoclimatology, Palaeoecology* 196, 209-228.
- Meulenkamp, J.E., Sissingh, W., Calvo, J. P., Daams, R., Londeix, L., Cahuzac, B., Kovac, M., Marunteanu, M., Ilynia, L. B., Khondkarian, S. O., Scherba, I. G., Roger, J., Platel, J. –P., Hirsch, F., Sadek, A., Abdel-Gawad, G. I., Yeddi, R. S., Yaich, C., Bouaziz, S., 2000b.** Tertiary. In: Crasquin, S. (Coord.), Atlas Peri-Tethys, Palaeogeographical Maps-Explanatory Notes. CCGM/CGMW, Paris, pp.153-208.
- Miller, K. G., Feigenson, M., Wright, J. D. And Clement, B., 1991.** Miocene isotope reference section, Deep Sea Drilling Project Site 608: an evaluation of isotope and biostratigraphic resolution. *Palaeoceanography*, 6, 33–52.
- Müller, P., Geary, D.H., Magyar, I., 1999.** The endemic molluscs of the Late Miocene Lake Pannon: their origin, evolution and family level taxonomy. *Lethaia* 32, 47–60.
- Nakoman, E., 1967.** Microflore des dépôts tertiaires du sud-ouest de l’Anatolie. *Pollen et Spores*, 9, 1, 121–142.
- Neogene system, 1986.** Stratigraphy of the USSR. *Moscow Nedra.*, 1:419p.
- Nix, H., 1982.** Environmental determinants of biogeography and evolution in Terra Australis. In: Barker, W.R., Greenslade, P.J.M. (Eds.), Evolution of the Flora and fauna of Arid Australia. Peacock Publishing, Frewville, 47–66.
- Ogilvie, A. G., 1920.** A contribution to the geography of Macedonia, *Geographical Journal* 55, 1-34.
- Okay, A.I., Satır, M., Maluski, H., Siyako, M., Monie, P., Metzger, R. and Akyüz S., 1996.** Paleo- and Neo-Tethyan events in northwest Turkey: geological and geochronological constraints. In: A. Yin and M. Harrison (eds.), *Tectonics of Asia* (Cambridge University Press, 420-441).
- Okay, A., Tüysüz, O., 1999.** Tethyan sutures of northern Turkey. In. Durand, B., Jolivet, L., Horvath, F., Seranne, M. (eds) The Mediterranean Basins: Tertiary Extension within the Alpine Orogen. Geological Society London, Special publications, 156, 475-515.

- Olteanu, R., 1978.** Ostracoda from DSDP Leg 42B. *Init. Rep. DSDP*, 42,2: 1017-1038.
- Olteanu, R. and Jipa, D. C., 2006.** Dacian Basin environmental evolution during Upper Neogene within the Paratethys domain. *GEO-ECO-MARINA* 12/2006.
- Pagani, M., Arthur, M. A. and Freeman, K. H., 1999.** Miocene evolution of atmospheric carbon dioxide. *Paleoceanography*. Vol. 14, No 3, Pages: 73-292.
- Percival, S.F., 1978.** Indigenous and reworked coccoliths from the Black Sea. In: Ross, D.A., Neprochnov, Y.P., et al. (Eds.), Leg 42. Initial Report of the Deep Sea Drilling Project 42, 2. U.S. Government Printing Office, 773–780.
- Popescu S.-M., 2001.** *Végétation, climat et cyclostratigraphie en Paratéthys centrale au Miocène supérieur et au Pliocène inférieur d'après la palynologie*. Thèse, Univ. C. Bernard- Lyon 1, 223 p.
- Popescu, S.-M., 2006.** Upper Miocene and Lower Pliocene environments in the southwestern Black Sea region from high-resolution palynology of DSDP site 380A (Leg42B). *Palaeogeography, Palaeoclimatology, Palaeoecology* 238, 64–77.
- Popescu, S.-M., Biltekin, D., Winter, H., Suc, J.-P., Melinte-Dobrinescu, M. C., Klotz, S., Rabineau, M., 2010.** Pliocene and Lower Pleistocene vegetation and climate changes at the European scale: Long pollen records and climatostratigraphy. *Quaternary International* 219 (2010) 152-167.
- Popov, S.V., Ilyina, L.B., Paramonova, N.P., Goncharova, I.A., et al., 2004.** Lithological-paleogeographic maps of Paratethys. *Cour. Forsch.Inst. Senckenb.* 250, 1–46 (10 maps).
- Popov, S. V., Shcherb, I. G., Ilyina, L. B., Nevesskaya, L. A., Paramonova, N. P., Khondkarian, S. O., Magyar, I., 2006.** Late Miocene to Pliocene palaeogeography of the Paratethys and its relation to the Mediterranean. *Palaeogeography, Palaeoclimatology, Palaeoecology* 238 (2006) 91–106.
- Quézel, P., 1973.** Contribution à l'étude phytosociologique du massif du Taurus. *Phytocoenologia*, 1, 131–222.
- Quézel, P., Barbero, M., Akman, Y., 1980.** Contribution à l'étude de la végétation forestière d'Anatolie septentrionale. *Phytocoenologia*, 8, 365–519.
- Quézel, P., Barbero, M., 1985.** Carte de la Végétation potentielle de la région Méditerranéenne. Feuille No 1: Méditerranée Orientale. Editions du Centre National de la Recherche Scientifique.
- Quézel, P., 1986.** The forest vegetation of Turkey. In "Plant Life of S-W Asia", *Proceedings of the Royal Society of Edinburgh*, 89B, 123–134.
- Quézel, P., 1995.** La flore du bassin méditerranéen: origine, mise en place endémisme. *Ecologia Mediterranea*, 21, 19–39.
- Quézel, P., 1998.** Cèdres et cédraies du pourtour méditerranéen: signification bioclimatique et phytogéographique. *Forêt méditerranéenne*, 19, 3, 243–260.

- Quézel, P., Médail, F., 2003.** Ecologie et biogéographie des forêts du bassin méditerranéen. Elsevier, Paris, 8–570.
- Raffi, I., Backman, J., Fornaciari, E., Pälike, H., Rio, D., Lourens, L.J., Hilgen, F.J., 2006.** A review of calcareous nannofossil astrobiochronology encompassing the past 25 million years. *Quaternary Science Review* 25, 3113–3137.
- Ravelo, A.C., Billups K., Dekens, P.S., Herbert, T.D. and Lawrence, K.T., 2007.** Onto the Ice Ages: Proxy Evidence for the onset of Northern Hemisphere Glaciation, From: M. Williams, A. M. Haywood, J. Gregory and D. Schmidt (eds), Deep-time perspectives on climate change: marrying the signal from computer models and biological proxies. The Micropaleontological Society, Special Publications. *The Geological Society, London*, 563-573, (2007).
- Raymo, M. E., Grant, B., Horowitz, M. And Rau, G. H., 1996.** Mid-Pliocene warmth: stronger greenhouse and stronger conveyor. *Mar. Micropaleontol.* 27, 313–326.
- Ross, D.A., 1978.** Black Sea stratigraphy. Initial Report of the Deep Sea Drilling Project, 42, 2, U.S. Gov. Print. Off.: 17-26.
- Ross, D.A. et al., 1978.** Site 380. In “Initial Report of the Deep Sea Drilling Project”, Ross, D.A., Neprochnov, Y.P. et al. eds., 42, 2, U.S. Gov. Print. Off.: 119-291.
- Ross D.A. and Degens E.T., 1974.** Recent sedimentgs of Black Sea. In “The Black Sea – geology, chemistry and biology”. Degens E.T. et Ross D.A. édit., *Amer. Ass. Petrol. Geol. Mem.*, 20: 183-199.
- Rowley, D.B., Currie, B.S., 2006.** Palaeoaltimetry of the late Eocene to Miocene Lunpola basin, central Tibet. *Nature* 439, 677–681.
- Rögl F., and Steininger F.F., 1983.** Vom Zerfall der Tethys zu Mediterran und paratethys. Die neogene Paläogeographie und Palinspastik des zirkum-mediterranean Raumes. *Ann. Naturhist. Mus. Wien*, 85, A : 135-163.
- Ruggieri, G., 1967.** The Miocene and later evolution of the Mediterranean Sea. In: Adams, C.G., Ager, D.V. (Eds.), Aspects of Tethyan Biogeography. *Syst. Assoc. Publ.*, vol. 7, pp. 283–290.
- Ryan, W. B. E, and Cita, W. B., 1978.** The nature and distribution of Messinian erosional surfaces. Indicators of several-kilometers-deep Mediterranean in the Miocene: *Marine Geol.*, v. 27, p. 193-230.
- Sabato, L., Bertini, A., Masini, F., Albianelli, A., Napoleone, G., Pieri, P., 2005.** The lower and middle Pleistocene geological record of the San Lorenzo lacustrine succession in the Sant’Arcangelo Basin (Southern Apennines, Italy). *Quaternary International* 131 (2005) 59–69.

- Sakıncı, M., Yaltrak, C., Oktay, F.Y., 1999.** Palaeogeographical evolution of the Thrace Neogene Basin and the Tethys-Paratethys relations at northwestern Turkey (Thrace). *Palaeogeogr. Palaeoclimatol. Palaeoecol.* 153, 17-40.
- Sakıncı, M., and Yaltrak, C., 2005.** Messinian crisis: what happened around the northeastern Aegean?. *Marine Geology* 221, 423–436.
- Sakıncı, M., 2007.** Trakya Tersiyer'inin silisleşmiş ağaçları. Proje no: 103Y137. İstanbul Technical University.
- Salzmann, U., Haywood, A. M., Lunt, D. J., Valdes, P. J., and Hill, D. J., 2008.** A new global biome reconstruction and data-model comparison for the Middle Pliocene. *Global Ecology and Biogeography*, (2008) 17, 432–447.
- Sayar, C., 1987.** İstanbul ve çevresi Neojen çökelleri ve Paratetis içindeki konumu. *Maden fak. 40. yıl Bülteni*, pp. 250–266.
- Schrader, H.-J., 1978.** Quaternary through Neogene history of the Black Sea, deduced from the paleoecology of diatoms, silicoflagellates, ebridians, and chrysomonads. In "Initial Report of the Deep Sea Drilling Project", Ross, D.A., Neprochnov, Y.P. *et al.* eds., 42, 2, U.S. Gov. Print. Off.: 789-901.
- Shackleton, N.J., Berger, A., Peltier, W.R., 1990.** An alternative astronomical calibration of the lower Pleistocene timescale based on ODP Site 677. *Transactions of the Royal Society of Edinburgh: Earth Sciences* 81, 251-261.
- Shackleton, N.J., Hall, M.A., Pate, D., 1995.** Pliocene stable isotope stratigraphy of Site 846. In: Pisias, N.G., Mayer, L.A., Janecek, T.R., Palmer-Julson, A., van Andel, T.H. (Eds.), Leg 138. *Proceedings of the Ocean Drilling Program, Scientific Results*, 138, pp. 337-355.
- Siyako, M., Bürkan, A.K., Okay, A.I., 1989.** Biga ve Gelibolu yarımadaı'nın Tersiyer jeolojisi ve hidrokarbon olanakları. *TPJD Bülteni*, cilt 1 (3), 183–199.
- Smith, A. D., Taymaz, T., Oktay, E., Yuce, H., Alpar, B., Basaran, H., Jackson, A. J., Kara, S., and Simsek, M., 1995.** High-resolution seismic profiling in the Sea of Marmara (northwest Turkey): Late Quaternary sedimentation and sea-level changes: *Geol. Soc. Amer. Bull.*, v. 107, p. 923-936.
- Stanley, D. J., and Blanpied, C., 1980.** Late Quaternary water exchange between the eastern Mediterranean and the Black Sea: *Nature*, v. 285, p. 537-541.
- Steenbrink, J., Van Vugt, N., Hilgen, F.J., Wijbrans, J.R., Meulenkamp, J.E., 1999.** Sedimentary cycles and volcanic ash beds in the lower Pliocene lacustrine succession of Ptolemais (NW Greece): discrepancy between $^{40}\text{Ar}/^{39}\text{Ar}$ and astronomical ages. *Palaeogeogr. Palaeoclimatol. Palaeoecol.* 152, 283–303.
- Steenbrink, J., Van Vugt, N., Kloosterboer-van Hoeve, M.L., Hilgen, F.J., 2000.** Refinement of the Messinian APTS from sedimentary cycle patterns in the

lacustrine Lava section (Servia Basin, NW Greece). *Earth Planet. Sci. Lett.* 181 (3–4), 161–173.

- Steenbrink, J., F.J. Hilgen, F. J., Krijgsman, W., Wijbrans, J. R., Meulenkamp, J. E., 2006.** Late Miocene to Early Pliocene depositional history of the intramontane Florina–Ptolemais–Servia Basin, NW Greece: Interplay between orbital forcing and tectonics. *Palaeogeography, Palaeoclimatology, Palaeoecology* 238 (2006) 151–178.
- Stoffers P. et Müller G., 1978.** Mineralogy and lithofacies of Black Sea sediments DSDP Project, *Init. Rep. Deep Sea Drill. Proj.*, Ross D.A., Neprochnov Y.P. *et al.* édit., 42, 2, U. S. Gov. Print. Off. : 373-413.
- Suc J.-P., 1980.** Contribution à la connaissance du Pliocène et du Pléistocène inférieur des régions méditerranéennes d'Europe occidentale par l'analyse palynologique des dépôts du Languedoc-Roussillon (sud de la France) et de la Catalogne (nord-est de l'Espagne). Thèse, Univ. Montpellier 2 : 198 p.
- Suc J.-P., 1984.** Origin and evolution of the Mediterranean vegetation and climate in Europe. *Nature*, 307, 5950 : 429-432.
- Suc, J.-P. & Drivaliari, A. 1991.** Transport of bisaccate coniferous fossil pollen grains to coastal sediments: an example from the earliest Pliocene Orb Ria (Languedoc, Southern France). *Review of Palaeobotany and Palynology*, 70, 247-253.
- Suc, J.-P., Diniz, F., Leroy, S. et al., 1995a.** Zanclean (~Brunssumian) to early Piacenzian (~early-middle Reuverian) climate from 48 to 548 north latitude (West Africa, West Europe and West Mediterranean areas). *Mededelingen Rijks Geologische Dienst*, 52, 43–56.
- Suc, J.-P., Bertini, A., Combourieu-Nebout, N., Diniz, F., Leroy, S., Russo-Eromolli, E., Zheng, Z., Bessais, E., Ferrier, J., 1995b.** Structure of West Mediterranean vegetation and climate since 5.3 ma. *Acta zoologica Cracoviensia* 38 (1), 3–16.
- Suc, J.-P., Fauquette, S., Bessedik, M., Bertini, A., Zheng, Z., Clauzon, G., Suballyova, D., Diniz, F., Quézel, P., Feddi, N., Clet, M., Bessais, E., Bachiri Taoufiq, N., Méon, H., Combourieu-Nebout, N., 1999.** Neogene vegetation changes in West European and West circum-Mediterranean areas. In “Hominid Evolution and Climate in Europe”, 1 “Climatic and Environmental Change in the Neogene of Europe”, Agusti, J., Rook, L., and Andrews, P. eds., Cambridge University Press: 370-385. 14.
- Suc, J.-P., Fauquette, S., Popescu, S.-M., 2004.** L'investigation palynologique du Cénozoïque passe par les herbiers. Actes du Colloque “Les herbiers: un outil d'avenir. Tradition et modernité”, Villeurbanne. Edit. Association française pour la Conservation des Espèces Végétales, Nancy, pp. 67–87.

- Suc, J.-P., and Popescu, S.-M., 2005.** Pollen records and climatic cycles in the North Mediterranean region since 2.7 Ma. In: Head, M.J., Gibbard, P.L. (Eds.), Early-Middle Pleistocene Transitions: The Land-Ocean Evidence, *Geological Society of London*, Special Publication 247, 147–158.
- Sümengen, M., Terlemez, I., Şentürk, K., Karasöse, C., Erkan, E.N., Ünay, E., Gürbüz, M., Atalay, Z., 1987.** Gelibolu Yarımadası ve Güneybatı Tersiyer havzasının stratigrafisi, sedimentolojisi ve Tektoniği. MTA Jeoloji Etüdüleri Dairesi Raporu 8128 (245 pp.).
- Szabo, C., Harangi, Sz., and Csontos, L., 1992.** Review of Neogene and Quaternary volcanism of the Carpathian-Pannonian region. *Tectonophysics*, 208:243-256.
- Şengör, A. M. C., 1979.** The North Anatolian transform fault, its age, offset and tectonic significance: *Journal of the Geological Society, London*, 136, 269-282.
- Şengör, A. M. C, Görür, N., and Saroglu, F., 1985.** Strike-slip faulting and related basin formation in zones of tectonic escape: Turkey as a case study, in Biddle, K. T., and Christie-Blick, N., eds., Strike-slip deformation, basin formation, and sedimentation: Tulsa, OK, Soc. Econ. Paleontol. Mineral., Spec. Publ., v. 37, p. 227-264.
- Tappan, H., 1980.** *The Paleobiology of Plant Protists*: San Francisco (W.H. Freeman).
- Thompson and Wallace, 2001.** Thompson, D.W.J. and Wallace, J.M., 2001. Regional climate impact on Northern Hemisphere annular mode. *Science*, 292, 85-89.
- Türkecan, A. and Yurtsever, A., 2002.** Türkiye Jeoloji Haritası, İstanbul Paftası, 1/500,000.MTA Genel Müdürlüğü, Ankara.
- Türkeş M., 1996.** Meteorological drought in Turkey: a historical perspective, 1930-93. *Drought Network News* 8: 17-21.
- Türkeş , M., and Erlat, E., 2003.** Precipitation changes and variability in Turkey linked to the North Atlantic Oscillation during the period 1930-2000. *International Journal Climatology*, 23, 1771–1796.
- Tüysüz, O., 1993.** A geo-traverse from the Black Sea to the central Anatolia: Tectonic evolution of northern Neo-Tethys. *Türkiye Petrol Jeologları Derneği Bülteni*, 5, 1-33 [in Turkish].
- Van der Burgh, J., Visscher, H., Dilcher, D. L. and Kürschner, W. M., 1993.** Paleotatmospheric signatures in Neogene fossil leaves. *Science* 260, 1788–1790.
- Van Vugt, N., Steenbrink, J., Langereis, C.G., Hilgen, F.J., Meulenkamp, J.E., 1998.** Magnetostratigraphy-based astronomical tuning of the early Pliocene lacustrine sediments of Ptolemais (NW Greece) and bed-to-bed correlation with the marine record. *Earth Planet. Sci. Lett.* 164 (3–4), 535–551.

- Vasiliev, I., Krijgsman, W., Langereis, C.G., Panaiotu, C.E., Matenco, L., Bertotti, G., 2004.** Towards an astrochronological framework for the eastern Paratethys Mio-Pliocene sedimentary sequences of the Focsani basin (Romania). *Earth Planet. Sci. Lett.* 227, 231–247.
- Velitzelos, E and Gregor, H.-J., 1990.** Some aspects of the Neogene floral history in Greece. *Review of Palaeobotany and Palynology* 62 (1990): 291-307.
- Wang, C.W., 1961.** The forests of China with a survey of grassland and desert vegetation. Maria Moors Cabot Foundation, vol. 5. Harvard University, Cambridge, Massachusetts.
- Wang, X.M., Zhang, X.L., Wang, M.Z., Li, C.S., 2003.** The palynoflora from Paleogene of Fanshi, Shanxi and discussion on their geological age. *Journal of Shandong University of Science and Technology (Natural Science)* 22 (3), 26–31.
- Wara M.W., Ravelo A.C., Delaney M.L., 2005.** Permanent El Niño-like conditions during the Pliocene warm period. *Science*, v. 309, p. 758– 761.
- Wake, L.V., and Hillen, L.W., 1980.** Study of a “bloom” of the oil-rich alga *Botryococcus braunii* in the Darwin River Reservoir. *Biotechnol. Bioeng.*, 22:1637–1656.
- Wan, S.M., Li, A.C., Peter, D., Clift, J., Stuut, W., 2007.** Development of the East Asian monsoon: Mineralogical and sedimentologic records in the northern South China Sea since 20 Ma. *Palaeogeography Palaeoclimatology Palaeoecology* 152, 37–47.
- Wigley, T. M. And Farmer, G., 1982.** Climate of the Eastern Mediterranean and Near East.- In: Paleoclimates, Paleoenvironments and Human Communities in the Eastern Mediterranean Region in Later Prehistory, J. L. Bintliff and W. Van Zeist (Ed.) B.A.R. *International Series*, 133:3-37; Oxford.
- Xoplaki, E., Luterbacher, J., Burkard, R., Patrikas, I., Maheras, P., 2000.** Connection between the large-scale 500 hPa geopotential height fields and precipitation over Greece during wintertime. *Clim. Res.* 14:129–146.
- Yaltirak, C., Sakiñ, M., Oktay, F. Y., 2000.** Westward propagation of North Anatolian fault into the northern Aegean: Timing and kinematics: Comment and Reply: Comment, *Geology*, February, 2000, v. 28, p. 187-188.
- Yavuz Işık, N., 2007.** Pollen analysis of coal-bearing Miocene sedimentary rocks from the Seyitömer Basin (Kütahya), Western Anatolia, *Geobios*, Volume 40, Issue 5, September-October 2007, Pages 701-708.
- Yavuz Işık, N., 2008.** Vegetational and climatic investigations in the Early Miocene lacustrine deposits of the Güvem Basin (Galatean Volcanic Province), NW Central Anatolia, Turkey, *Review of Palaeobotany and Palynology* 150 (2008) 130–139.

- Yavuz Işık, N. and Demirci, C., 2009.** Miocene spores and pollen from Pelitci,ik Basin, Turkey—environmental and climatic implications. *C. R. Palevol* 8 (2009) 437–446.
- Yunfa, M., Qingquan, M., XiaominFan, X., Fuli, W., Chunhui, S., 2010.** Origin and development of *Artemisia* (Asteraceae) in Asia and its implications for the uplift history of the Tibetan Plateau: a review *Quaternary International*, Article in Press.
- Yurttaş-Özdemir., Ü, 1973.** Biostratigraphy and macrofauna of the Tepek6y Triassic of theKocaeli Peninsula. *Maden Tetkik ve Arama Dergisi*, 77, 57-98 [in Turkish].
- Zachos, J., Pagani, M., Sloan, L. And Billups, K., 2001.** Trends, rhythms, and aberrations in global climate 65 Ma to present. *Science*, 292, 686–693.
- Zachos, J.C., Dickens, G.R., Zeebe, R.E., 2008.** An early Cenozoic perspective on greenhouse warming and carbon-cycle dynamics. *Nature* 451(17), 279–283.
- Zagwijn, W.H., 1960.** Aspects of the Pliocene and early Pleistocene vegetation in the Netherlands. *Mededelingen Geologische Stichting Serie C* 3 (5), 78 pp.
- Zagwijn,W.H., 1963.** Pollen-analytic investigations in the Tiglian of the Netherlands. *Mededelingen Geologische Stichting New Serie* 16, 49-71.
- Zagwijn, W.H.. 1975.** Variations in climate as shown by pollen analysis, especially in the Lower Pleistocene of Europe. In: Wright, A.E. & Moseley, F. (Eds.), *Ice Ages: Ancient and Modern. Geological Journal special issue* 6, pp. 137-152.
- Zagwijn, W.H., and Suc, J.-P., 1984.** Palynostratigraphie du Plio-Pléistocène d'Europe et de Méditerranée nord-occidentales: correlations chronostratigraphiques, histoire de la végétation et du climat. *Paléobiologie Continentale* 14 (2), 475-483.
- Zhang, Y. G., Ji, J., Balsam, W., Liu, L. and Chen, J., 2009.** Mid-Pliocene Asian monsoon intensification and the onset of Northern Hemisphere glaciations. *Geology*, **July 2009** v. 37 no. 7 p. **599-602.**
- Zheng, Z., Cravatte, J., 1986.** Etude palynologique du Pliocène de la Côte d'Azur (France)et du littoral ligure (Italie). *Geobios* 19 (6), 815-823.
- Zhisheng, A., Kutzbach, J.E., Prell, W.L, Porter, C., 2001.** Evolution of Asia Monsoonsand phased uplift of the Himalaya-Tibetan plateau since Late Miocene times.*Nature*, vol. 411, 62-66.
- Zohary, M., 1973.** Geobotanical foundations of the Middle East. Fischer éd., Stuttgart, 2 vol., 739 p.

Contents lists available at ScienceDirect

Palaeogeography, Palaeoclimatology, Palaeoecology

journal homepage: www.elsevier.com/locate/palaeo
The Messinian Salinity Crisis in the Dardanelles region: Chronostratigraphic constraints

Mihaela Carmen Melinte-Dobrinescu^a, Jean-Pierre Suc^{b,*}, Georges Clauzon^c, Speranta-Maria Popescu^b, Rolando Armijo^d, Bertrand Meyer^e, Demet Biltekin^f, M. Namık Çağatay^f, Gülsen Ucarkus^f, Gwénaél Jouannic^e, Séverine Fauquette^g, Ziyadin Çakir^f

^a National Institute of Marine Geology and Geo-ecology (GEOECOMAR), 23-25 Dimitrie Onciul Street, RO-024053 Bucharest, Romania

^b Laboratoire Paléoenvironnements et Paléobiosphère, UMR 5125, CNRS, France; Université Lyon 1, Campus de La Doua, Bâtiment Géode, 69622 Villeurbanne Cedex, France

^c C.E.R.E.G.E. (UMR 6635 CNRS), Université Paul Cézanne, 13545 Aix-en-Provence Cedex, France

^d Laboratoire de Tectonique, Institut de Physique du Globe de Paris (UMR 7154 CNRS), 75252 Paris Cedex, France

^e Université Pierre et Marie Curie, IStEP, UMR 7193, 75252 Paris Cedex 05, France

^f Istanbul Technical University, School of Mines and Eurasia Institute of Earth Sciences, Maslak, 34469 Istanbul, Turkey

^g Institut des Sciences de l'Evolution (UMR 5554 CNRS), Equipe Paléoenvironnements et Paléoclimats, CC 061, Université Montpellier 2, Place Eugène Bataillon, 34095 Montpellier Cedex 05, France

ARTICLE INFO

Article history:

Received 31 August 2008

Received in revised form 29 March 2009

Accepted 7 April 2009

Keywords:

Calcareous nannoplankton
Chronostratigraphy
Messinian Salinity Crisis
Palaeogeography
NE Aegean region

ABSTRACT

An intense controversy on chronostratigraphy of upper Miocene–lower Pliocene deposits and the Messinian Salinity Crisis in the Dardanelles area led to a systematic investigation of calcareous nannoplankton content of 10 key-sections representative of the most relevant regional Kirazlı and Alçitepe formations. Our study shows clearly that the Kirazlı Formation deposits predate the Messinian Salinity Crisis while those of the Alçitepe Formation postdate this outstanding event, which severely impacted the region as widely known around the Mediterranean Basin. Fluvial canyon cutting or gap in sedimentation linked to the peak of the Messinian Salinity Crisis separates the two formations. Detailed palaeoenvironmental investigations (based on the fluctuation and distribution pattern of dinoflagellate cysts, pollen grains and calcareous nannoplankton) allow us to reconstruct the regional palaeogeography before, during and after the Messinian Salinity Crisis. The gathered data do not indicate any marine corridor between the Eastern Mediterranean Sea and Eastern Paratethys through the Marmara Sea region at the time of the Messinian Salinity Crisis.

© 2009 Elsevier B.V. All rights reserved.

1. Introduction

During the Late Neogene, very significant paleobiogeographical changes have taken place in the Mediterranean region *s.l.*, fundamentally because the connection of the Atlantic Ocean with the Mediterranean Sea and the Paratethys (a residual continental sea which covered large areas of the Central and Eastern Europe) Sea was restricted and sometimes completely interrupted (Seneš, 1973; Rögl and Steininger, 1983; Marinescu, 1992; Mărunțeanu and Papaianopol, 1995; Rögl, 1998; Sprovieri et al., 2003; Clauzon et al., 2005; Popov et al., 2006; Popescu, 2006; Melinte, 2006). The latest Miocene is characterized by an exceptional event, which was the severe sea-level

drop of the Mediterranean Sea, leading to the Messinian Salinity Crisis (MSC). This event is characterized throughout the Mediterranean basin by both, deposition of thick evaporites in its deep basins and cutting of huge fluvial canyons across its margins (Hsü et al., 1973; Clauzon, 1973; Cita et al., 1978; Clauzon et al., 1996). Such a prominent geological event produced significant changes in the palaeobiological assemblages, mirrored especially by the marine planktonic organisms (such as planktonic foraminifers, calcareous nannoplankton and dinoflagellates), which are very sensitive to environmental changes.

Consequently, the Messinian Salinity Crisis is now recognized as the most prominent event marking palaeobiogeographical change in the Mediterranean. Examples of case studies of spectacular features associated with the MSC are widespread over the entire Mediterranean area (for an overview, see: Agusti et al., 2006; Rouchy et al., 2006; Suc et al., 2007; CIESM, 2008). The present study is devoted to improving our knowledge of the MSC in the Dardanelles region (Fig. 1A), which is crucial for our understanding of the evolution of the linkage between the Aegean Sea and the Black Sea. The existence of a gateway at the time of the MSC is subject of controversy (in support of such a gateway, see: Esu, 2007; Faranda et al., 2007; Gliozzi et al., 2007; Stoica et al., 2007; in opposition of such a gateway, see: Clauzon et al., 2005;

* Corresponding author.

E-mail addresses: melinte@geoecomar.ro (M.C. Melinte-Dobrinescu), jean-pierre.suc@univ-lyon1.fr (J.-P. Suc), clauzon@cerge.fr (G. Clauzon), speranta.popescu@univ-lyon1.fr (S.-M. Popescu), armijo@ipgp.jussieu.fr (R. Armijo), bertrand.meyer@upmc.fr (B. Meyer), biltekin@itu.edu.tr (D. Biltekin), cagatay@itu.edu.tr (M.N. Çağatay), ucarkus1@itu.edu.tr (G. Ucarkus), gwenael.jouannic@gmail.com (G. Jouannic), severine.fauquette@univ-montp2.fr (S. Fauquette), ziyadin.cakir@itu.edu.tr (Z. Çakir).

Fig. 1. Location and geological maps. A, Study area. B, Geological map of the studied area according to Türkecan and Yurtsever (2002). C, Geological map of the area surrounding the Dardanelles Strait according to Türkecan and Yurtsever (2002).

Popescu, 2006; Gillet et al., 2007; Popescu et al., 2009). However, most of the regional stratigraphic works underestimated the impact of the MSC in the Dardanelles (see, for examples: Görür et al., 1997; Çağatay et al., 1998; Görür et al., 2000; Türkecan and Yurtsever, 2002; Sakıncı and Yaltrak, 2005; Çağatay et al., 2006). The consensus among regional stratigraphers is that the rocks outcropping in the Dardanelles are chiefly sediments of Eocene to Middle–Late Miocene Age and that the region is basically devoid of marine sediments of Pliocene Age (Fig. 1B, C). The typical strong marginal signature of the MSC is present all around the Mediterranean (including the Aegean Sea), with an erosional surface and deeply incised fluvial canyons, subsequently covered and filled with marine Pliocene sediments (examples from the Western Mediterranean: Clauzon, 1973, 1978, 1980a,b, 1982, 1990; Gautier et al., 1994; Clauzon, 1999; Guennoc et al., 2000; Lofi et al., 2003, 2005; Sage et al., 2005; Cornée et al., 2006; Maillard and Mauffret, 2006 – example from the Central Mediterranean: El Euch-El Koundi et al., 2009 – examples from the Eastern Mediterranean: Chumakov, 1973; Delrieu et al., 1993; Poisson et al., 2003). Even if these erosional effects of the MSC were suspected in the Dardanelles region (Çağatay et al., 2006), they were not clearly evidenced until today, mapped and recognized as such using conventional stratigraphy.

Defying the stratigraphical consensus, the study by Armijo et al. (1999) presented evidence for a widespread erosion surface and a prominent canyon that parallels the present-day Dardanelles Strait. These authors interpreted explicitly these features as possibly resulting from the MSC, given the large uncertainties in the ages of the formations mapped in the area (Ternek, 1964). Both the erosion surface and the canyon appear carved into the sediments of the Kirazlı Formation and filled by sediments of the Alçıtepe Formation (Armijo et al., 1999). Consequently, Armijo et al. (1999) deduced for these two

formations a possible Messinian and an early Pliocene Age, respectively (Fig. 2). The study by Armijo et al. (1999) includes a geological map that shows strong folding affecting layers of the Kirazlı Formation, but not those of the overlying Alçıtepe Formation. Therefore, this important unconformity would also correlate roughly with the MSC, providing an invaluable constraint for the age of the propagation of the North Anatolian Fault across the region (Armijo et al., 1999). The work by Armijo et al. (1999) has been strongly criticized (e.g., Yaltrak et al., 2000 vs. Armijo et al., 2000) and its main stratigraphic inferences dismissed (e.g., Sakıncı and Yaltrak, 2005; Çağatay et al., 2006).

In order to clarify the significance of the Messinian Salinity Crisis and its geological imprint in the Dardanelles region, we revised the stratigraphy by a systematic sampling of the most critical sedimentary units and by dating those using calcareous nannofossils. We studied ten key-sections (Enez and Yaylaköy in the Gulf of Saros; Burhanlı, Eceabat, Poyraztepe, Kilitbahir, Seddülbahir, Intepe, Yenimahalle and Truva in the Dardanelles Strait) where Upper Neogene deposits are well exposed (Fig. 1B, C). Our objectives are: (1) dating these sections with respect to global bio- chronostratigraphy using calcareous nannofossils, (2) reconstructing marine, brackish and continental palaeoenvironments using dinoflagellate cyst, calcareous nannoplankton and pollen grain fluctuations, and (3) setting up a reliable chronostratigraphic basis for current and future studies of the Messinian Salinity Crisis in the Dardanelles region, as well as for studies using the MSC as a chronometer for deformation associated with the North Anatolian Fault.

The age of the Alçıtepe Formation is the crucial question for the ongoing controversy (Fig. 2). The Alçıtepe Formation is described as being composed of brackish- to freshwater carbonates, interbedded with marine sandstones and siltstones (Görür et al., 1997; Çağatay et al.,

Fig. 2. Controversed age of the Neogene formations in the Dardanelles Strait area. Time-interval corresponding to the Messinian Salinity Crisis (MSC) is indicated by a red band. Defining the stratigraphic position of the Alçıtepe Formation with respect to the MSC is critical. The hypothesis represented to the left (Armijo et al., 1999) assigns a Pliocene Age (post-MSC) to the Alçıtepe Formation. The hypothesis that is currently proposed in the conventional stratigraphy (e.g., Görür et al., 1997; Sakıncı et al., 1999; Türkecan and Yurtsever, 2002; Çağatay et al., 2006) assigns a Miocene Age (pre-MSC and partly coeval with it) to the Alçıtepe Formation. The undulating line represents an important erosion and tectonic unconformity in the hypothesis of Armijo et al. (1999), only a slight unconformity in the conventional stratigraphy. (For interpretation of the references to colour in this figure legend, the reader is referred to the web version of this article.)

1998) as more or less conformably overlying the Kirazlı Formation (Çağatay et al., 2006). It is overlain by shallow fluvio-marine siliciclastic rocks (Göztepe Formation in the North Marmara region, Truva Formation in the South Marmara region: Görür et al., 2000; Çağatay et al., 2006). A brackish Paratethyan fauna (i.e. *Macra* sp., *Paradacna abichi*, *Dreissena* sp., *Cardium* sp., and *C. edulis*) was reported from the Alçitepe Formation, suggesting that these deposits belong to the Pontian Paratethyan Stage (Gillet et al., 1978; Taner, 1979; Çağatay et al., 2006) and, as a consequence, to the so-called “late Messinian Lago Mare” (Sakıncı and Yaltrak, 2005; Çağatay et al., 2006). Some intercalations of layers with Mediterranean marine faunas (*Ostrea*, *Pecten*) were also reported (Sakıncı and Yaltrak, 2005), that emphasizes the dual, Paratethyan and Mediterranean, influence in the area. To characterize the MSC in the region, we focussed our study on the relationships between the Alçitepe Formation and the underlying Kirazlı Formation.

2. Materials and methods

Calcareous nannoplankton investigations were performed for the sections Enez, Yaylaköy, Buhanlı, Eceabat, Poyraztepe, Kilitbahır, Seddülbahır, Intepe, Yenimahalle and Truva. To retain the original sample composition, smear slides were prepared directly from the untreated samples. The calcareous nannofloral analyses were performed using a light polarizing microscope at ×1600 magnification. The nannofloral taxonomic identification follows Perch-Nielsen (1985) and Young (1998).

Palynological samples were processed from some of these sections: Burhanlı (1 sample), Eceabat (1 sample), Seddülbahır (2 samples), and Intepe (10 samples). Each sample (20 g of dry sediment) was processed using standard method (Cour, 1974): acid digestion, concentration using ZnCl₂ (at density 2.0) and sieving at 10 µm. A 50 µl volume of residue was mounted between the coverslip and microscope slide using glycerine in order to allow rotation of palynomorphs for their complete examination resulting in their proper identification. Counting of palynomorphs was performed using a light microscope, their identification was done at ×1000 magnification. Pollen grains were identified with a botanical

approach. 150 pollen grains except those of *Pinus* were identified and counted per sample. For the slides showing very poor concentrations in dinoflagellate cysts, a new sieving at 20 µm was performed on the pollen residue that permitted to concentrate the dinoflagellate cysts and a new slide was obtained using 50 µl from the new residue. Only the ten samples from the Intepe sections were analysed: all the specimens present on a slide were identified and counted. The freshwater algae *Pediastrum* and *Botryococcus* were considered as transported by rivers, and were also counted. Their vertical distribution documents duration and intensity of freshwater inputs.

3. Results

3.1. Bio- and chronostratigraphy

The calcareous nannoplankton offers an accurate way of biostratigraphic dating based on successive calibrated datum events within the time-window 8–4 Ma, which is of interest to this study. The calibration of the FAD (First Appearance Datum) and the LAD (Last Appearance Datum) follows Berggren et al. (1995), Backman and Raffi (1997), Lourens et al. (2004) and Raffi et al. (2003, 2006); we also took into account the available data for nannofloral distribution within the Messinian–Zanclean interval from the Eastern Mediterranean (i.e., Castradori, 1998; Snel et al., 2006; Wade and Bown, 2006). The distribution of six nannofossils was particularly useful for biostratigraphical purposes, as follows (Fig. 3): *Amaurolithus primus* (FAD: 7.424 Ma; LAD: 4.50 Ma; Plate I, Fig. 5), *Reticulofenestra rotaria* (FAD: ca. 7.41 Ma; LAD: imprecise, up to ca. 6 Ma; Plate I, Fig. 1), *Nicklithus (= Amaurolithus) amplificus* (FAD: 6.909 Ma; LAD: 5.978 Ma; Plate I, Fig. 4), *Triquetrorhabdulus rugosus* (FAD: 12.671 Ma; LAD: 5.279 Ma; Plate I, Fig. 6), *Ceratolithus acutus* (FAD: 5.345 Ma; LAD: 5.040 Ma; Plate I, Figs. 2–3), and *Reticulofenestra pseudoumbilicus* (FAD: 8.761 Ma; LAD: 3.839 Ma). The absolute age of the FADs and LADs (after Raffi et al., 2003, 2006) is indicated as a rough guide; noticeably, when different ages are given for the same event, we accepted those proposing the largest range especially for the Eastern Mediterranean, when available. The nannofloral

Fig. 3. Chronostratigraphy, calcareous nannoplankton biostratigraphy of the Late Miocene and Early Pliocene and inferred age of the studied sections. Chronology refers to Lourens et al. (2004), calcareous nannoplankton events according to Berggren et al. (1995) and Raffi et al. (2006). The grey strips correspond to two steps of the MSC (Clauzon et al., 1996) accepted by a representative community working on the Messinian Salinity Crisis (CIESM, 2008). Locality numbers, see Fig. 1B, C. The double line in the “Formations” column illustrates the lack of sedimentation during the peak of the MSC. It is to notice that the Yenimahalle and Truva sections are not located according to nannoplankton biostratigraphy but with respect to geomorphology and the reverse palaeomagnetism of the Yenimahalle section.

Plate I. Microphotographs of the significant biostratigraphic nannoplankton species (light microscope, crossed nicols, scale bar = 4 μm).

- Fig. 1. *Reticulofenestra rotaria* Theodoridis; Eceabat (sample 1).
 Fig. 2. *Ceratolithus acutus* Gartner and Bukry; Intepe (sample 24).
 Fig. 3. *Ceratolithus acutus* Gartner and Bukry; Kilitbahir (sample 4).
 Fig. 4. *Nicklithus amplificus* (Bukry and Percival) Gartner and Bukry; Intepe (sample 1).
 Fig. 5. *Amaurolithus primus* (Bukry and Percival) Gartner and Bukry; Intepe (sample 35).
 Fig. 6. *Triquetrorhabdulus rugosus* Bramlette and Wilcoxon; Kilitbahir (sample 4).

events (i.e., occurrence and extinction of the above-mentioned nannofossils) allow us to identify the NN11 and NN12 Zones of Martini (1971), including the subzones of Berggren et al. (1995). As done by Bukry (1975) and more recently used by Backman and Raffi (1997) for the global ocean, it appeared necessary to subdivide Zone NN12. We subdivided it into Subzones a and b, defined by the LAD of *Discoaster quinqueramus* and the FAD of *Ceratolithus acutus*, respectively. *D. quinqueramus* is rarely recorded in the Mediterranean, but its extinction occurred at the beginning of the peak of the MSC. The Messinian Erosional Surface may be considered as coeval of this bioevent, making the subdivision into Subzones NN12a and NN12b very useful in the Mediterranean region. Hence, the chronostratigraphic resolution in the Mediterranean can be improved in distinguishing post-Salinity Crisis sediments below the FAD of *C. acutus* from those above it (Fig. 3).

Concerning the nannofloral preservation observed in the studied samples, this is generally moderate, which means that dissolution and/or overgrowth of the nannofossils hindered the specific identification up to 25%. Some samples (i.e., from the lower part of the Intepe section) showed a good preservation; hence, the nannofloral

specimens could be identified up to 90% to species level. A few samples (i.e., 14–21 from Intepe), contain nannofloral assemblages poorly preserved, showing severe dissolution, fragmentation and/or overgrowth; the specific identification being hindered up to 70%.

In general, the investigated samples contain nannofloral reworkings mainly from Cretaceous, Paleogene and Lower Miocene. The reworked taxa represent between 25 and 60% of the total recorded nannofloras. Hence, we may address questions about the possibility that some Miocene taxa, such as *Triquetrorhabdulus rugosus* and *Reticulofenestra pseudoumbilicus*, are *in situ* or reworked, impeding the accuracy of the biostratigraphical interpretation. *T. rugosus* is generally rare in the studied samples throughout its range (2–4 specimens/sample) and thus the possibility of this taxon being reworked into younger deposit is very low. *Reticulofenestra pseudoumbilicus* (>7 μm) is common in all the studied samples, being probably also common in the underlying Miocene deposits. But the fact that the recorded specimens of *R. pseudoumbilicus* are well-preserved, being found in assemblages yielding a moderate to good preservation, indicate that the nannofloral compositions are primary.

Fig. 4. Distribution and biostratigraphy of the calcareous nannofloras from the Enez locality. Nannoplankton biostratigraphic markers are in bold characters. a, Clay.

3.1.1. Gulf of Saros

The studied section near Enez is located on the eastern shoreline of the Enez lagoon (40°46'24" N latitude, 26°04' E longitude; Fig. 1B), an area described as covered by deltaic deposits of the Meriç River overlying the Kirazlı Formation (Çağatay et al., 1998, 2006) while Sakinç et al. (1999) indicates some bioclastic carbonate rocks attributed to the Alçitepe Fm., the upper part of which could belong to Zanclean. Here, brownish clays with a thickness of about 8 m have provided a poorly to moderately preserved nannoflora in 7 samples (Fig. 4). The co-occurrence of *Triquetrorhabdulus rugosus* and *Ceratolithus acutus* throughout the whole section is indicative for the NN12b nannofossil Subzone (Fig. 4), representing the extreme end of Messinian (after the MSC) to early Zanclean (Fig. 3).

Fig. 6. Distribution and biostratigraphy of the calcareous nannofloras from the Eceabat locality. Nannoplankton biostratigraphic markers are in bold characters. a, Clay; b, Sand; c, Limestone.

At Yaylaköy, blue clays are exposed along the beach (40°36'22" N latitude, 26°21'28" longitude E; Fig. 1B) below coquina beds with *Ostrea* and interbedded with *Cardium*-bearing sands of Zanclean Age (Çağatay et al., 1998, 2006). This section probably corresponds to the upper third of the Erikli section of Sakinç et al. (1999) and Sakinç and Yaltrak (2005) that the authors refer to the Alçitepe Formation. The studied sample of the blue clays includes *Amaurolithus delicatus*, *Braarudosphaera bigelowii*, *Calcidiscus leptoporus*, *Ceratolithus acutus*, *Coccolithus pelagicus*, *Florisphaera profunda*, *Thoracosphaera* sp., *Reticulofenestra pseudoumbilicus*, *Sphenolithus abies* and *Triquetrorhabdulus rugosus*. Reworked specimens are rare. According to the co-occurrence of *C. acutus* and *T. rugosus*, this moderately preserved nannoflora also belongs to the beginning of the nannofossil Subzone NN12b covering an interval from the topmost of the Messinian (after the MSC) up to the early Zanclean (Fig. 3).

3.1.2. Dardanelles Strait

At Burhanlı (40°18'17" N latitude, 26°33'08" E longitude; Fig. 1C), six samples from the variegated clays alternating with sands of the Kirazlı Formation provided a moderately preserved nannoflora along 10 m with a continuous record of *Reticulofenestra pseudoumbilicus*, *R. rotaria* and *Triquetrorhabdulus rugosus* (Fig. 5). *Nicklithus amplificus* was recorded in the two uppermost samples. Reworkings from Cretaceous, Eocene, Oligocene and Miocene have been observed. Such an assemblage is considered to represent the upper part of the nannofossil Subzone NN11b and the lowermost part of the nannofossil Subzone NN11c (Fig. 5), i.e. latest Tortonian to early Messinian in age (Fig. 3).

Northward Eceabat (40°11'30" latitude N, 26°21'18" E longitude; Fig. 1C), four samples have been studied from the whitish clayey base (20 m thick) belonging to the Kirazlı Formation, and one sample (10 m higher) from a clayey intercalation within calcareous tabular deposits of the Alçitepe Formation (Sakinç et al., 1999). All of them display a nannoflora characterized by a poor to moderate preservation and few reworked specimens. It includes *Amaurolithus primus*, *Reticulofenestra pseudoumbilicus*, *R. rotaria* (samples 1, 2, 4, 5), *Nicklithus amplificus* (samples 1 and 3) and *Triquetrorhabdulus rugosus* (Fig. 6). Such an

Fig. 5. Nannoflora of the Burhanlı locality with indication of nannoplankton subzones. Nannoplankton biostratigraphic markers are in bold characters. a, Clay; b, Sand.

assemblage belongs to the NN11c nannofossil Subzone (Fig. 6), early Messinian in age (Fig. 3).

In the nearby Poyraztepe hill (40°12'27.6" N latitude, 26°21'59.9" E longitude; Fig. 1B), the upper part of the section belonging to the Alçitepe Formation and with a stratigraphic position higher than the top of the Eceabat section, provided a nannoflora from its uppermost clayey intercalations (two samples) just underlying the topmost limestone. These samples contain, among an important amount of reworked specimens from Eocene to Lower Miocene, *Triquetrorhabdulus rugosus* and *Reticulofenestra pseudoumbilicus*, without any specimen of *Reticulofenestra rotaria* or *Nicklithus amplifiscus* (Fig. 7). Taking into account their stratigraphic position (see above) and the absence of the latter species, we consider that these samples belong to the NN11d Subzone (Fig. 7), i.e. the late Messinian, and more precisely to the first step of the MSC with respect to the importance of reworking (Fig. 3).

Near Kilitbahir, two sections were investigated: the one on the seashore (40°10'08.8" E longitude, 26°22'16" N latitude) belongs to the Kirazlı Fm. and that along a path above the castle (40°08'35" E longitude, 26°22'10" N latitude) to the Alçitepe Fm. (Sakıncı et al., 1999; Sakıncı and Yaltrak, 2005) (Fig. 1C) where we took one sample. The coastal locality is made of 1.50 m of blue clays rich in mollusc shells where three samples were taken overlain by about 20 m of sands including, 4 m over their base, a blue clayey bed (1 m thick) which provided one sample. The three lowermost samples contain *Triquetrorhabdulus rugosus* and *Reticulofenestra pseudoumbilicus* (Fig. 8); additionally, huge Eocene to Lower Miocene reworked nannofossils are present. The sample 4 (from the intercalated clays of the Kilitbahir seashore section) contains, besides the above-mentioned taxa, *Ceratolithus acutus*; a similar nannofloral assemblage with *C. acutus* and *T. rugosus* was identified in the sample Kilitbahir castle (Fig. 8). The two samples yielding *C. acutus* are characterized by a weak reworking. They belong to Subzone NN12b (Fig. 8), i.e. the extreme end of Messinian to earliest Zanclean (Fig. 3). The three coastal samples belong to Subzone NN12a, i.e. to the latest Messinian (Fig. 3). As it has been demonstrated that the Mediterranean reflooding anticipated the base of the Zanclean Stage (Popescu et al.,

Fig. 8. Distribution and biostratigraphy of the calcareous nannofloras from the Kilitbahir seashore and castle localities. Nannoplankton biostratigraphic markers are in bold characters. a, Clay; b, Sand.

2007, 2009) as defined by its GSSP (Global Stratotype and Point Section; Van Couvering et al., 2000) and in the absence of any unconformity between samples of Subzones NN12b and NN12a, we consider that all these samples immediately postdate the MSC (Fig. 3).

At the beach of Seddülbahir, there are two distinct sections both referring to the Alçitepe Fm. (Sakıncı and Yaltrak, 2005), of which the second section constitutes the reference exposure: (1) to the East (40°02'30" N latitude, 26°11'12.1" E longitude; Fig. 1C), a thin section

Fig. 7. Distribution and biostratigraphy of the calcareous nannofloras from the Poyraztepe locality. Nannoplankton biostratigraphic markers are in bold characters. a, Clay; b, Limestone.

Fig. 9. Nannoflora of the East Seddülbahir locality with indication of nannoplankton subzone. Nannoplankton biostratigraphic markers are in bold characters. a, Sand; b, Sandstone; c, Clay.

(5 m thick) mostly made of sands, clays and marls, yielded within five samples nannofloral assemblages with *Triquetrorhandulus rugosus*, *Reticulofenestra pseudoumbilicus* and *R. rotaria* (Fig. 9), that we refer to the Subzone NN11b (Fig. 9), i.e. latest Tortonian–earliest Messinian in age; notably, *Nicklithus amplificus* does not occur (Fig. 3); (2) to the West (40°02'38" N latitude, 26°10'55" E longitude; Fig. 1C), the second section, made of thick clays (about 30 m) rich in mollusc shells, provided nannofloras containing as significant biostratigraphical taxa *Triquetrorhandulus rugosus* and *Reticulofenestra pseudoumbilicus* and, additionally, *Ceratolithus acutus* in the topmost 11 m (samples 8–12; Fig. 10). Reworking is weak. As there is no unconformity within the West Seddülbahir section, we may suppose that the nannofloral assemblages evidence a continuous passing from Subzone NN12a to NN12b (Fig. 10), i.e. from the latest Messinian (after the peak of the MSC, as at the Kilitbahir seashore section) to the earliest Zanclean (Fig. 3). However, the West Seddülbahir section is obviously discordant over the East Seddülbahir one.

The Intepe section (40°1'27" N latitude, 26°20'33" E longitude; Fig. 1C) (Gillet et al., 1978; Sakıncı and Yaltrak, 2005) is about 77 m thick, of which we studied the 36 m thick central part (Fig. 11A, C). The studied section is made up of sands, clays, calcareous sandstones and thin limestones. It is topped by yellow sands and pebbly sandstones and a thick calcareous flagstone which constitutes a local reference surface.

Fig. 10. Distribution and biostratigraphy of the calcareous nannofloras from the West Seddülbahir locality. Nannoplankton biostratigraphic markers are in bold characters. a, Dark-light clay; b, Coquina; c, Sand; d, Silt; e, Sandstone; f, Limestone.

The section is rich in *Maetra* shells and displays also gastropod shells such as *Melanopsis*. In its middle part, the section displays a thin lignite (5 cm thick) (Gillet et al., 1978) overlain by a sand (2 cm thick) and a *Maetra* coquina (17 cm thick) (Fig. 11B–E). Thirty five samples were studied: all of them provided a relatively well-preserved nannoflora (Fig. 12). From the biostratigraphic point of view, *Amaurolithus primus* was discontinuously present all along the section, *Reticulofenestra rotaria* was found in samples 14–18, *Nicklithus amplificus* in samples 1 and 7, *Triquetrorhandulus rugosus* almost continuously recorded from sample 1 to sample 31, *Ceratolithus acutus* regularly from sample 24 to sample 35 (Fig. 12). Hence, the lower part of the studied section (samples 1–18) belongs to the NN11c calcareous nannofossil Subzone according to the joint occurrence of *R. rotaria* and *T. rugosus* and the occasional presence of *N. amplificus*, while its upper part (i.e. from sample 24 where *C. acutus* appears) represents a large part of Subzone NN12b including the disappearance level of *T. rugosus* (Fig. 12). As a consequence, the interval between samples 18 and 24 belongs to the NN11c Subzone and maybe to NN12a Subzone for the uppermost layers just preceding sample 24 (Fig. 12). Distinction between Messinian and Zanclean can be placed around samples 23–24 according to the first appearance level of *Ceratolithus acutus*. If some changes in sea-level have been recorded in the section with respect to an impact of the MSC, they concern the layers immediately underlying the appearance of *C. acutus* (Fig. 12). Our research focused on the lignite corresponding to our sample 21 (Fig. 11C–D) as a first candidate because of its deposition under few centimetres of water. The lignite has an erosional contact with the overlying fine sand (Fig. 11E). But the expected gap in sedimentation is obvious in a lateral outcrop where the lignite is directly overlain on a long distance by reddish clays indicative of an emersion phase (Fig. 11F), because in present-day lignite quarries, lignites catch fire readily upon exposure, firing the overlying clays, thus producing ‘porcellanite’ (Fig. 11G). This is the obvious signature of an emersion event that occurred just below the appearance of *C. acutus*. Accordingly, we may consider that (1) the lignite corresponds to the first step of the MSC (5.960–5.760 Ma; Fig. 3) characterized by a weak fall in sea-level (ca. 150 m), (2) the immediately overlying clays correlate with the sea-level rise separating the two steps of the MSC (5.76–5.60 Ma; Fig. 3), (3) the erosion of these clays or their firing (when being partly preserved during the duration of exposure) signs the huge drop in sea-level (ca. 1500 m) of the peak of the MSC (5.60–5.46 Ma; Fig. 3) (Clauzon et al., 1996).

About 34 m thick, the Yenimahalle section (39°57'51" N latitude, 26°17'46" E longitude; Fig. 1C), is mostly composed of sands, sandstones and limestones including some clayey intercalations. It was described by Çağatay et al. (2006) who included it into the Alçıtepe and Truva formations. Palaeomagnetic measurements performed here by W. Krijgsman (Çağatay et al., 2006) revealed a continuous reverse signal and the section was assigned to Chron C3r, and more precisely to its Messinian part. However, the Yenimahalle section should be relatively younger than the Intepe section because it overlies an obvious morphological surface atop the Intepe section, an assumption also supported by the absence of any calcareous coccolith at Yenimahalle. *Ceratolithus acutus* is recorded without *Triquetrorhandulus rugosus* in the upper part of the Intepe section (Fig. 12) which could belong to the uppermost part of Chron C3r, and possibly Chron C3n.4n (see Fig. 3). As a consequence, the Yenimahalle section should be assigned at least to the next reverse chron, i.e. Chron C3n.3r (Fig. 3).

The Truva section (39°57'30" N latitude, 26°14'47" E longitude; Fig. 1C), with a thickness of only 5 m, and composed of limestones and intercalated clays, overlies the Yenimahalle section (Fig. 3) and tops the Truva Fm. No calcareous nannoplankton was found in the two studied samples.

The above-presented results, based on calcareous nannoplankton investigations, allow us to date in detail several Upper Miocene–Lower Pliocene reference sections from NW Turkey, and also to identify in detail a succession of nannofloral events the time-interval encompassing the MSC (Fig. 3). Particularly, the first record of *Ceratolithus acutus* in

Fig. 11. Lithostratigraphy of the central part of the Intepe section. A, General view of the Intepe section where red lines indicate the studied (i.e. central) part of the section; B, Upper part of the studied section; C, Lithological log of the studied section, location of samples; 1, Dark-light clay; 2, Sand; 3, Calcareous sandstone; 4, Mollusc shells; 5, Lignite; 6, Sandstone; 7, Limestone; 8, Silt. D, Lignite-coquina interval including the discontinuity caused by the peak of the MSC; E, Detail of Fig. 4D; F, Fired clays (porcellanite) overlying the lignite and marking the discontinuity caused by the peak of the MSC; G, Present-day analogue from the S Romania lignite Lupoaia quarry showing clays transformed into porcellanite after being naturally fired by the underlying lignite. (For interpretation of the references to colour in this figure legend, the reader is referred to the web version of this article.)

the Kilitbahir, West Seddülbahir and Intepe sections may reliably be considered as corresponding practically to its first appearance datum. This event is well-calibrated in the global ocean at 5.345 Ma (Raffi et al., 2006), i.e. slightly after the end of the peak of the MSC, estimated at 5.46 Ma by Clauzon et al. (2008). At Kilitbahir and West Seddülbahir, *C. acutus* was recorded for the first time few metres above the Messinian Erosional Surface (Figs. 8 and 10), i.e. in a consistent chronostratigraphic position with its global first appearance datum. The same situation concerns the Intepe section, where the first occurrence of *C. acutus* has been found slightly above the Messinian discontinuity which corresponds to the Messinian Erosional Surface in an embayment context (Fig. 12). The nannofloral scarcity of some layers (i.e., samples 24 and 25; Fig. 12) does not constitute an impediment for biostratigraphical interpretations; probably these layers correspond to a restrictive environment, such as a lagoonar one, as it is suggested in the next section of “Palaeoenvironment reconstructions”.

3.2. Palaeoenvironmental reconstructions

We particularly considered palynomorphs, mainly those from the Intepe section where we focussed on the interval encompassing the MSC: (1) the dinoflagellate cysts for the coastal marine environment reconstruction, and (2) the pollen grains for the vegetation, climate and palaeoaltitude reconstruction. We also took into account the calcareous nannoplankton, a group of marine algae which live in marine surface waters (0–200 m), being therefore affected by changes in the surface water environment, particularly salinity, temperature and nutrient availability.

3.2.1. Coastal marine palaeoenvironments

The dinoflagellate cyst flora from the Intepe section comprises only 12 taxa, to which can be added freshwater algae *Pediastrum* and *Botryococcus* colonies that are interpreted as markers of freshwater input (Fig. 13). Preservation of dinoflagellate cysts is poor to moderate. The dinoflagellate cyst assemblage is constituted by: (1) oceanic species and other neritic species as *Spiniferites mirabilis* and *S. membranaceus*, (2) marine autotrophic cosmopolitan species such as *Lingulodinium machaerophorum*, *Operculodinium centrocarpum* sensu Wall and Dale, *Spiniferites bentorii*, *S. bentorii* subsp. *truncates*, *S. hyperacanthus*, *S. ramosus*, *S. bulloideus*, *Spiniferites* spp., and (3) Paratethyan brackish species as *Galeacysta etrusca* and *Impagidinium globosum*. Despite dinoflagellate cyst scarcity, their assemblage distribution, completed by the relative frequency of freshwater algae documents palaeoenvironmental changes.

During the time-interval covered by the samples 15–19, the dinoflagellate cyst flora was dominated by marine euryhaline species as *O. centrocarpum*, *L. machaerophorum*, *Spiniferites bulloideus*, *S. ramosus*, *S. bentorii*, *S. bentorii* subsp. *truncates* and *S. hyperacanthus*. The high relative abundance of *Spiniferites* spp. (more than 50%), characterized by short processes, denotes the deterioration of marine conditions, in relation with a drop in salinity (Kokinis and Anderson, 1995; Hallett, 1999; Ellegaard, 2000; Sorrel et al., 2006; Popescu et al., 2009). Notably is the presence of *S. mirabilis* within sample 15 (one specimen), and that the presence of the Paratethyan brackish dinoflagellate cysts *Galeacysta etrusca* in sample 17 (one specimen) and *Impagidinium globosum* in samples 17 and 19 (one specimen per sample). Morphology of *G. etrusca* denotes that this specimen belongs to the morphological group “C” (small endocyst/exocyst ratio) of Popescu et al. (2009).

Samples 20 to 23 are characterized by the absence of dinoflagellate cysts and a huge increase (acme) of *Pediastrum* that indicates a new change in the local environments marked now by fresh- to brackish-water conditions. Salinity at that time is probably less than 4.6 pps, according to minimum of salinity tolerance of *S. ramosus* (Marret and Zonneveld, 2003), identified as the most tolerant marine eurihaline species within the Intepe assemblage. This interpretation is consistent

with the absence of calcareous nannoplankton and deposition of a lignite layer (sample 21) when shallow water and freshwater conditions prevailed. *Pediastrum* was also found in sample 24, with a high relative abundance indicating continuous fresh- to brackish-water conditions, sometimes interrupted by incursions of marine waters. Sample 25 is characterized by the increase in diversity of marine dinoflagellate cysts and their relative abundance in opposition to a decrease of *Pediastrum*, that indicates increased salinity. Sample 26 is characterized by the absence of marine dinoflagellate cysts and increase of freshwater algae, which marked the return of prevalent freshwater input.

According to the dinoflagellate cyst record, we may propose a relatively detailed evolution of the environmental conditions during the time-interval corresponding to samples 15–26, as summarized:

- from samples 15 to 19, the local environment was characterized by low saline conditions that evolved progressively to brackish ones at the top of the interval; this decrease in salinity of surface waters caused unfavourable conditions for marine plankton;
- samples 20 to 23 indicate an huge input of freshwater resulting in brackish to freshwater conditions in the local environment, sometimes interrupted by some marine incursion revealed by the calcareous nannoplankton;
- samples 24 and 26 display the progressive increase in salinity in relation with the reflooding by marine waters and some freshwater inputs.

The nannoflora is on the whole consistent with the information from the dinoflagellate cyst assemblages. It is to be noticed that the nannoplankton diversity is drastically altered in samples 21–23, while it was high in the lower part of the studied section (especially in the interval covered by the samples 5–13), being again considerably reduced in its upper part (samples 31–35) (Fig. 12). Notably, the nannofossil *Braarudosphaera bigelowii* and the calcareous dinoflagellate *Thoracosphaera* have a more consistent frequency within samples 8–13 than in lower levels, indicating stressful conditions in relation probably with the drop in salinity. At the beginning of the interval covered by the samples 14–21, a significant increase of the calcareous dinoflagellate *Thoracosphaera* was observed. *Thoracosphaera* decreases at the top of the above-mentioned interval; its lowermost abundance corresponds to the highest frequency of *B. bigelowii*. The significant decrease of *Thoracosphaera*, coincident with the *B. bigelowii* bloom, is probably indicative of an important decrease in salinity. The collapse of the calcareous nannofossil assemblage and the progressive decrease in salinity possibly mirrored the transition from a hyposaline (even brackish) environment to a continental one, marked by the deposition of the lignite layer. Between samples 22 and 26, *B. bigelowii* represents a minor component of the nannoflora. *Thoracosphaera*, after a peak recorded at the base of this interval, drops significantly. In the uppermost part of the section (samples 27–35), *B. bigelowii* and *Thoracosphaera* represent minor component of the assemblages.

Finally, considering both the dinoflagellate cysts from samples 15–26 and the nannoplankton all along the section, three main ecostratigraphic intervals have been identified within the studied part of the Intepe section (earlier first):

- (1) samples 1–13, denoting almost normal marine conditions;
- (2) samples 14–21, illustrative of a sea-level fall associated with a decrease in salinity (probably brackish conditions), and some emersion at the top;
- (3) samples 22–35, indicating a return to almost normal marine conditions in spite of some deterioration in the uppermost layers.

3.2.2. Continental palaeoenvironments

Fourteen samples from the Dardanelles Strait area provided pollen grains in sufficient quantity. They inform on the vegetation and climate of the region just before and after the MSC. Herbs (Asteraceae, Poaceae, Amaranthaceae–Chenopodiaceae, Caryophyllaceae, *Artemisia*, etc.)

Fig. 13. Dinoflagellate cyst flora from eight samples of the central part of the Intepe section. Lithological legend, see Fig. 11.

predominated before the MSC (in the sections Burhanlı, Eceabat, Intepe *p.p.*), while trees were mostly composed of warm-temperate elements (*Quercus*, *Carya*, *Zelkova*, etc.) with few subtropical elements (*Taxodiaceae*, *Engelhardia*) (Fig. 14). Sample 21 shows important halophytes plus an over-representation of aquatic plants (*Potamogeton*, *Typha*, *Myriophyllum* and *Sparganium*), the reason for which it was not drafted on Fig. 14. It denotes a coastal freshwater marsh. After the MSC (Intepe *p.p.*, Seddülbahır), subtropical trees (with *Cathaya*) show larger percentages as *Pinus* and altitudinal trees (*Cedrus*, *Abies* and *Picea*) do (Fig. 14). Increase in disaccate pollen grains (*Pinus*, *Cedrus*, *Abies* and *Picea*) could indicate a more distal location of the upper part of the Intepe section with respect to the palaeoshoreline, i.e. in relation with marine water invasion after the MSC. However, the larger representation of halophytes (*Amaranthaceae*–*Chenopodiaceae*) and freshwater plants (*Sparganium*, *Alismataceae*, *Potamogeton*) after the MSC points out nearby coastal environments.

As a consequence, we suggest that the increase in conifer pollen grains (*Cedrus* mainly, with *Abies* and *Picea*) could be significative of some relief uplift resulting in a larger representation of altitudinal conifers in coastal pollen records. Southward Intepe, the Kayacı Mount which is made of Eocene–Oligocene volcanic rocks does not bear evidence for uplift during the latest Miocene. *Cedrus* does not like silice-rich soils but prefers limestones, dolomites and flysch (Quézel, 1998; Quézel and Médail, 2003). Hence, the Kayacı relief, whatever its elevation, is not a likely source of *Cedrus* pollen grains. Armijo et al. (1999) suggested that the Ganos–Gelibolu Mountain, situated to the North of the Dardanelles and mostly constituted by Eocene–Oligocene flysch and volcanics, has been uplifted during the Messinian as a response to the propagation of the North Anatolian Fault (Fig. 1B). This relief appears to be a good candidate for the habitat of *Cedrus* forests as they exist today in southern Turkey (Quézel, 1998; Quézel and Médail, 2003). Uplift occurred during the late Messinian, fact also suggested by the significant percentages in *Cedrus* plus *Abies* and *Picea* recorded at Intepe section after the MSC (samples 22–26) unlike for the older deposits (in the sections Burhanlı, Eceabat and in the section Intepe in samples 15–20) (Fig. 14). Reconstruction of the mean annual temperature by palaeoclimatic transfer function

(Fauquette et al., 1998) in the coastal area using a selection of the pollen assemblage (altitudinal trees are not employed for this calculation) offers a possibility to estimate minimum palaeoaltitude of the nearby massif (Fauquette et al., 1999) as altitudinal elevation of conifer belts is narrowly linked to temperature and, as a consequence, to latitude of the massif (Ozenda, 1989). This estimate was done using the pollen record of Intepe after the MSC. The most probable mean annual temperature calculated for the Intepe samples 22–26 is 16.5 °C (range: 15–18.5 °C). This temperature is recorded today, in Turkey at low altitude, at around 38.5° of North latitude. We applied the method described by Fauquette et al. (1999) where the estimated palaeotemperatures are shifted into present-day latitudes. The obtained latitudes are projected onto the altitudinal elevation gradient of the *Abies*–*Picea* belt with respect to decreasing latitude (100 m in altitude per degree in latitude; Ozenda, 1989) and provide a range from 1800 to 2400 m for the minimum altitude of the Ganos Mountain with a most probable value at 2000 m.

4. Discussion

The new bio- and ecostratigraphic data provide consistent information on the impact of the MSC in the Dardanelles which must be discussed in terms of palaeogeographic inferences and palaeoenvironments.

4.1. Palaeogeographic inferences

This study allows to date precisely some key-sections defining the regional stratigraphic framework, which was previously established by rather loose correlations of lithological units, using mollusc macrofossils which may have a large range of salinity tolerance and duration (Sakinç et al., 1999).

The most important outcome concerns the Alçitepe Fm. which is younger than the Messinian Salinity Crisis according to its two reference sections:

- at Seddülbahır, the Alçitepe Fm. deposits (i.e. the West Seddülbahır section) are separated from the Kirazlı Fm. deposits (i.e. the East Seddülbahır section) by an angular unconformity which hence corresponds to the Messinian Erosional Surface (Fig. 15B, C);
- at Intepe, the Alçitepe Fm. deposits conformably overlie those of the Kirazlı Fm., but we infer an unconformity between the two formations, resulting from emersion and probably some erosion in relation with the peak of the MSC, this is what we called the Messinian discontinuity (Figs. 11 and 15B).

In addition, the Eceabat section plus its uppermost part at Poyraztepe hill, which were conventionally attributed to the Alçitepe Fm. on the basis of lithological similarities, have a late Messinian Age and must belong to the Kirazlı Fm. (Figs. 3 and 15B, D). Beds of the Eceabat and Poyraztepe sections dip 10–15° NW and culminate at altitude 143 m (Fig. 15D). Southward, between Eceabat and Kilitbahır (Fig. 15D), sediments younger than the MSC start from the seashore and relate to the early Zanclean up to the altitude 125 m where calcareous nannoplankton (*Triquetrorhabdulus rugosus* and *Ceratolithus acutus*) was recorded above the Kilitbahır castle (Figs. 3 and 15B, D, G). As they are covered by the Alçitepe calcareous flagstone, they are obviously nested within the Messinian succession of Eceabat–Poyraztepe from which they are necessarily separated by the Messinian Erosional Surface (Fig. 15B, D) that can be observed along the road southward Eceabat. The extremity of the Gelibolu Peninsula obviously corresponds to the sedimentary filling of a fluvial valley incised during the peak of the MSC (Fig. 15A), as suggested by Armijo et al. (1999). Many places in this area, such as at Nuriyamut (Fig. 15A,

Fig. 12. Distribution and biostratigraphy of the calcareous nannofloras from the Intepe section. Nannoplankton biostratigraphic markers are in bold characters. Lithological legend, see Fig. 11.

B), show thick early Zanclean sands with gravels with an organization in foreset beds dipping (25–30° at Nuriyamut) in the direction of the axis of the Messinian valley (Fig. 15F). Such a sedimentary organization is typical of a Gilbert-type fan delta which is the widespread feature of sediments within the Messinian canyons after the Mediterranean reflooding (Clauzon, 1990; Clauzon et al., 1990). Clayey deposits at West Seddülbahir and Kilitbahir seashore constitute the bottomset beds of the Dardanelles Gilbert-type fan delta more or less imbricated with coarser foreset beds (Fig. 15B). The calcareous topmost plateau of Alçitepe constitutes the abandonment surface of the Gilbert-type fan delta.

On the southern coastline of the Dardanelles Strait, the horizontal sediments observed in the Intepe section are suddenly (300 m northward the section) replaced by sands and gravels organized in foreset beds dipping (25°) in direction of the Dardanelles Gilbert-type fan delta (Fig. 15E). The contact is obviously erosional as it has been precisely followed on land back to the Güzelyalı village (Fig. 15A). This Gilbert-type fan delta construction fills a canyon cut by a tributary of the main Dardanelles fluvial drain during the peak of the MSC (Fig. 15A, B). Here, the Messinian Erosional Surface incises deposits of the Kirazlı Formation and laterally evolves into a discontinuity in an interfluvial context (Intepe section) as drawn on Fig. 15B.

Accordingly, the impact of the MSC is obvious in the region and must henceforth be considered in the stratigraphic setting and palaeogeographic reconstructions. The gap in sedimentation because of erosion (Dardanelles and tributary Messinian canyons) or non-deposition (Intepe) is chronostratigraphically located in the extremely latest Messinian, i.e. during the peak of the MSC, and not in the Zanclean as repeatedly assumed as by Sakıncı et al. (1999) and Sakıncı and Yaltrak (2005). Nesting of post-MSC deposits within the pre-MSC ones must allow distinguishing the Alçitepe Formation from the Kirazlı Formation without using the often misleading facies characteristics.

In the Gulf of Saros, nesting of the post-MSC sediments within the older ones is not very pronounced, probably because of a relatively weak fluvial activity. The most intense fluvial activity during the MSC was located in the Dardanelles Strait area; it rapidly diminished when passing to an interfluvial context, as supported by the palaeoenvironmental data.

In spite of its reverse palaeomagnetic signal and the record of Paratethyan organisms, the Yenimahalle section cannot be considered as representative of the MSC in the region. Clearly overlying the topmost flagstone of Intepe, which has a similar significance as the abandonment

surface of the Dardanelles Gilbert-type fan delta, the Yenimahalle section has a younger Zanclean Age and represents the end of the Pliocene sedimentation in the region (Fig. 15B).

4.2. Palaeoenvironments

The present-day southern shoreline of the Dardanelles Strait is a key-area because of the proximity of distinct environments during the peak of the MSC (a fluvial canyon and an interfluvial domain) and after it (a subaquatic delta system and a coastal domain). Such an environmental coexistence is somewhat normal but the accuracy of its record over a short distance is infrequent. At Intepe, the local environment fluctuated between marine and almost freshwater conditions in relation with moderate to intense fluvial freshwater inputs. The blooms of the calcareous dinoflagellate *Thoracosphaera* are situated at the base and towards the top of the interval, and may be associated with the calcareous nannoplankton collapse (Fig. 12). Such blooms may reflect unstable marine conditions, associated with an important nutrient record (possibly linked to a high terrigenous input) and the lack of other marine planktonic competitors. The bloom of the calcareous nannoplankton species *Braarudosphaera bigelowii* at the top of the Messinian layers at Intepe is probably related to the salinity fluctuations. As they are highly sensitive to environmental changes, dinoflagellate cysts require that the Intepe locality was very close to the paleoshoreline. A bay head (Fig. 15A) seems to be the probable palaeogeography before and after the MSC while the varying freshwater inputs depended from the nearby river. Such a relatively isolated context was significantly impacted by the successive fluctuations in Mediterranean sea-level and the resulting steps of the MSC (Clauzon et al., 1996). The moderate sea-level fall of the first step of the MSC is marked by the lignite and development of marsh conditions (5.96–5.76 Ma: Fig. 3). The following sea-level rise is indicated by the overlying clays (5.76–5.60 Ma: Fig. 3), mostly eroded or fired during the episode of emersion corresponding to the huge sea-level drop of the peak of the MSC during which the nearby fluvial canyon was cut (5.60–5.46 Ma). Climatic conditions are warm and relatively dry out of the riverbanks, as it is indicated by the large percentages of herb pollen grains. Some relief uplifted during the MSC, event probably connected to the establishment of an important relief in the Ganos–Gelibolu Massif.

According to these results, we may address questions about the gateway between the Black Sea (i.e. the Eastern Paratethys domain) and Aegean Sea (the Eastern Mediterranean domain) through the

Fig. 14. Synthetic pollen diagram from some localities before and after the peak of the MSC. 1, Subtropical trees; 2, *Cathaya*; 3, Warm-temperate trees; 4, Cupressaceae; 5, Cool-temperate trees (*Cedrus* mainly); 6, Boreal trees (*Abies* and *Picea*); 7, Elements without signification; 8, Mediterranean xerophytes; 9, Herbs; 10, Steppe elements (*Artemisia* mainly).

Fig. 15. The Messinian fluvial canyon in the western part of the Dardanelles Strait. A, Map showing the main Messinian fluvial canyon and a tributary (grey surface), the Early Zanclean coastline (dotted grey line), and location of photographs C-F. Sections: 4, Eceabat; 5, Poyraztepe; 6, Kilitbahir (seashore); 7, Kilitbahir (castle); 8, East Seddülbahir; 9, West Seddülbahir; 10, Intepe; 11, Yenimahalle; 12, Truva. B, Reconstructed composite cross-section (a-b, c-d) in the Dardanelles Strait and the extremity of the Gelibolu Peninsula through the Zanclean Gilbert-type fan delta with calcareous nannoplankton events and location of photographs C-F and Fig. 4. 1, Clay; 2, Limestone; 3a, Messinian Erosional Surface; 3b, Messinian discontinuity; 4, Sandy and gravelly foreset beds. C, Photograph of the beach of Seddülbahir showing the Messinian Erosional Surface cutting the clays and sands of the East Seddülbahir section and overlain by the clayey bottomset beds of the West Seddülbahir section. Nannoplankton markers are indicated. D, View of the surroundings of Eceabat showing the Alçıtepe Formation deposits nested within the Eceabat-Poyraztepe sections now reported to the Kirazlı Formation. Nannoplankton markers and the Messinian Erosional Surface are indicated. E, Sandy and gravelly foreset beds near the Intepe section. F, Sandy and gravelly foreset beds at Nuriyamut. G, Kilitbahir (seashore) section with calcareous nannoplankton markers.

Marmara realm during the MSC (Görür et al., 1997; Çağatay et al., 2006) cannot be perpetuated. Indeed, a connection between these domains is incompatible with coeval subaerial erosion both on the Black Sea side (Gillet et al., 2007) and the Dardanelles area (this work) during the peak of the MSC. Connection would have been only possible during times of high sea-level, i.e. before and after the peak of the MSC (Clauzon et al., 2005) or before its first step. However, the extreme scarcity of Paratethyan dinoflagellate cysts in the Intepe sediments does not support such a connection; as these organisms live on the surface waters, we may suppose that a sea-level high-stand should be expressed in their significant increase. For the same reason, a one-way passage resulting in influx of Paratethyan organisms only into the Mediterranean realm (see discussion in: CIESM, 2008) is also questionable. Still, the almost continuous presence of the Paratethyan bivalves and ostracods before and after the MSC in the region (Çağatay et al., 2006) needs to be explained. It has been suggested that their arrival was caused by an episode of high sea-level just before the peak of the MSC (Clauzon et al., 2005; Popescu et al., 2009) and that their persistence was favoured in some relatively isolated environments (lagoons, bays) at the frontier between marine, brackish and fresh-water conditions (Çağatay et al., 2006). But, as we mentioned above, the Paratethyan dinoflagellate cysts (which should be the earliest immigrants as planktonic organisms) are absent or extremely rare in our samples, and therefore do not support the hypothesis of a connection allowing the penetration of the Paratethyan macrofaunas throughout a Marmara marine corridor, connecting the Black Sea (i.e. the Eastern Paratethys domain) and the Aegean Sea (the Eastern Mediterranean domain) during the MSC.

5. Conclusion

The chronological constraints presented in this work provide a sound background for new prospects on the late Miocene–Pliocene stratigraphy and palaeogeography in the Dardanelles region. Evidence of the strong impact of the MSC in the region, as elsewhere around the Mediterranean, indicates that the Dardanelles Strait was a fluvial collector eroding the older rocks during the peak of the MSC. The systematic use of the calcareous nannoplankton biostratigraphical results in a clarified regional stratigraphy, unbiased by lithological similarities. In this region as everywhere around the Mediterranean, the peak of the MSC and the resulting erosional surface or discontinuity may be regarded as a robust chronological indication. The most important results lie in the fact that the Alçıtepe formation is a Pliocene unit postdating the MSC and that the Kırızlı formation is a Late Miocene unit predating the MSC. The study in progress aims at reconstructing the Messinian fluvial network in the Marmara region and deciphering the role of the MSC on the North Anatolian Fault propagation across the region. Finally, the Dardanelles–Marmara region is to be discarded as a connection between the Black Sea (i.e., the Eastern Paratethys domain) and Mediterranean Sea, during the Messinian Salinity Crisis.

Acknowledgements

This research started within the frame of the CNRS-INSU ECLIPSE II Programme and was mostly developed thanks to the ANR EGO Project. One of us (D. Biltekin) was granted by the French Embassy in Turkey for her PhD thesis. This paper is a contribution to the Projects IDEI of the Romanian National Programme of Research, Development and Innovation (Projects No. 144/2007 CNCSIS Code 816 and 364/2007, CNCSIS Code 815 – M.C. Melinte-Dobrinescu). We are indebted to Marie-Pierre Aubry, who reviewed this paper and made valuable suggestions and comments, which significantly increase the chronostratigraphic and ecostratigraphic inferences of the manuscript. We also appreciate the improvements suggested by an anonymous referee. The English language was improved by Oliver Bazely. This paper is the ISEM contribution no 2009-029.

References

- Agusti, J., Oms, O., Meulenkamp, J.E., 2006. Introduction to the Late Miocene to Early Pliocene environment and climate change in the Mediterranean area. *Palaeogeography, Palaeoclimatology, Palaeoecology* 238, 1–4.
- Armijo, R., Meyer, B., Hubert, A., Barka, A., 1999. Westward propagation of the North Anatolian fault into the northern Aegean: timing and kinematics. *Geology* 27, 267–270.
- Armijo, R., Meyer, B., Hubert, A., Barka, A., 2000. Westward propagation of North Anatolian fault into the northern Aegean: timing and kinematics: reply. *Geology* 28, 188–189.
- Backman, J., Raffi, I., 1997. Calibration of Miocene nannofossil events to orbitally tuned cyclostratigraphies from Ceara Rise. *Proceedings of the Ocean Drilling Program, Scientific Results* 154, 83–89.
- Berggren, W.A., Kent, D.V., Swisher III, C.C., Aubry, M.-P., 1995. A revised Cenozoic geochronology and chronostratigraphy. In: Berggren, W.A., Kent, D.V., Aubry, M.-P., Hardenbol, J. (Eds.), *Geochronology, Time Scales and Global Stratigraphic Correlation: A Unified Temporal Framework for an Historical Geology*. Special Publication – Society of Economic Paleontologists and Mineralogists, vol. 54, pp. 141–212.
- Bukry, D., 1975. Coccolith and silicoflagellate stratigraphy, Northwestern Pacific ocean, Deep Sea Drilling Project Leg 32. *Initial Reports of the Deep Sea Drilling Project* 32, 677–701.
- Çağatay, N.M., Görür, N., Alpar, B., Saatçılar, R., Akkök, R., Sakıncı, M., Yüce, H., Yaltrak, C., Kuşçu, I., 1998. Geological evolution of the Gulf of Saros, NE Aegean Sea. *Geo Marine Letter* 18, 1–9.
- Çağatay, N.M., Görür, N., Flecker, R., Sakıncı, M., Tünoğlu, C., Ellam, R., Krijgsman, W., Vincent, S., Dikbaş, A., 2006. Paratethyan–Mediterranean connectivity in the Sea of Marmara region (NW Turkey) during the Messinian. *Sedimentary Geology* 188–189, 171–187.
- Castradori, D., 1998. Calcareous nannofossils in the basal Zanclean of the Eastern Mediterranean Sea: remarks on paleoceanography and sapropel formation. *Proceedings of the Ocean Drilling Program, Scientific Results* 160, 113–123.
- Chumakov, I.S., 1973. Geological history of the Mediterranean at the end of the Miocene – the beginning of the Pliocene according to new data. *Initial Reports of the Deep Sea Drilling Project* 13 (2), 1241–1242.
- CIESM, 2008. Executive summary. In: Briand, F. (Ed.), *The Messinian Salinity Crisis from mega-deposits to microbiology – A consensus report*. CIESM Workshop Monographs, vol. 33, pp. 7–28.
- Cita, M.B., Ryan, W.B.F., Kidd, R.B., 1978. Sedimentation rates in Neogene deep sea sediments from the Mediterranean and geodynamic implications of their changes. *Initial Reports of the Deep Sea Drilling Project* 42A, 991–1002.
- Clauzon, G., 1973. The eustatic hypothesis and the pre-Pliocene cutting of the Rhône Valley. *Initial Reports of the Deep Sea Drilling Project* 13 (2), 1251–1256.
- Clauzon, G., 1978. The Messinian Var canyon (Provence, Southern France) – paleogeographic implications. *Marine Geology* 27, 231–246.
- Clauzon, G., 1980a. Le canyon messinien de la Durance (Provence, France): une preuve paléogéographique du bassin profond de dessiccation. *Palaeogeography, Palaeoclimatology, Palaeoecology* 29, 15–40.
- Clauzon, G., 1980b. Révision de l'interprétation géodynamique du passage Miocène–Pliocène dans le bassin de Vera (Espagne méridionale): les coupes d'Antas et de Cuevas del Almanzora. *Rivista Italiana di Paleontologia* 86 (1), 203–214.
- Clauzon, G., 1982. Le canyon messinien du Rhône: une preuve décisive du "desiccated deep-basin model" (Hsü, Cita et Ryan, 1973). *Bulletin de la Société Géologique de France Série* 7 24 (3), 597–610.
- Clauzon, G., 1990. Restitution de l'évolution géodynamique néogène du bassin du Roussillon et de l'unité adjacente des Corbières d'après les données écostratigraphiques et paléogéographiques. *Paléobiologie Continentale* 17, 125–155.
- Clauzon, G., 1999. L'impact des variations eustatiques du bassin de Méditerranée occidentale sur l'orogène alpin depuis 20 Ma. *Etudes de Géographie Physique* 28, 1–8.
- Clauzon, G., Suc, J.-P., Aguilar, J.-P., Ambert, P., Cappetta, H., Cravatte, J., Drivaliari, A., Doménech, R., Dubar, M., Leroy, S., Martinelli, J., Michaux, J., Roiron, P., Rubino, J.-L., Savoye, B., Vernet, J.-L., 1990. Pliocene geodynamic and climatic evolutions in the French Mediterranean region. *Paleontologia i Evolucio. Memòria Especial* 2, 132–186.
- Clauzon, G., Suc, J.-P., Gautier, F., Berger, A., Loutre, M.-F., 1996. Alternate interpretation of the Messinian salinity crisis: controversy resolved? *Geology* 24, 363–366.
- Clauzon, G., Suc, J.-P., Popescu, S.-M., Mărunțeanu, M., Rubino, J.-L., Marinescu, F., Melinte, M.C., 2005. Influence of the Mediterranean sea-level changes over the Dacic Basin (Eastern Paratethys) in the Late Neogene. *The Mediterranean Lago Mare facies deciphered*. *Basin Research* 17, 437–462.
- Clauzon, G., Suc, J.-P., Popescu, S.-M., Melinte-Dobrinescu, M.C., Quillévéré, F., Warny, S.A., Fauquette, S., Armijo, R., Rubino, J.-L., Lericois, G., Gillet, H., Çağatay, M.N., Ucaruk, G., Escarguel, G., Jouannic, G., Dalesme, F., 2008. Chronology of the Messinian events and paleogeography of the Mediterranean region *s.l.* CIESM Workshop Monographs 33, 31–37.
- Cornée, J.-J., Ferrandini, M., Saint Martin, J.-P., Münch, Ph., Moullade, M., Ribaud-Laurenti, A., Roger, S., Saint Martin, S., Ferrandini, J., 2006. The late Messinian erosional surface and the subsequent reflooding in the Mediterranean: new insights from the Melilla–Nador basin (Morocco). *Palaeogeography, Palaeoclimatology, Palaeoecology* 230 (1–2), 129–154.
- Cour, P., 1974. Nouvelles techniques de détection des flux et de retombées polliniques: étude de la sédimentation des pollens et des spores à la surface du sol. *Pollen et Spores* 16 (1), 103–141.
- Delrieu, B., Rouchy, J.M., Foucault, A., 1993. La surface d'érosion finmessinienne en Crète centrale (Grèce) et sur le pourtour méditerranéen: Rapport avec la crise de salinité méditerranéenne. *Comptes-Rendus de l'Académie des Sciences de Paris Série* 2 318, 1103–1109.
- El Euch-El Koundi, N., Ferry, S., Suc, J.-P., Clauzon, G., Melinte-Dobrinescu, M.C., Gorini, C., Safra, A., El Koundi, M., Zargouni, F., 2009. Messinian deposits and erosion in

- northern Tunisia: inferences on the Sicily Strait during the Messinian Salinity Crisis. *Terra Nova* 21, 41–48.
- Ellegaard, M., 2000. Variations in dinoflagellate cyst morphology under conditions of changing salinity during the last 2000 years in the Limfjord, Denmark. *Review of Palaeobotany and Palynology* 109, 65–81.
- Esu, D., 2007. Latest Messinian “Lago-Mare” Lymnocythidinae from Italy: close relations with the Pontian fauna from the Dacic Basin. *Geobios* 40, 291–302.
- Faranda, C., Gliozzi, E., Ligios, S., 2007. Late Miocene brackish Loxoconchidae (Crustacea, Ostracoda) from Italy. *Geobios* 40, 303–324.
- Fauquette, S., Guiot, J., Suc, J.-P., 1998. A method for climatic reconstruction of the Mediterranean Pliocene using pollen data. *Palaeogeography, Palaeoclimatology, Palaeoecology* 144, 183–201.
- Fauquette, S., Clauzon, G., Suc, J.-P., Zheng, Z., 1999. A new approach for paleoaltitude estimates based on pollen records: example of the Mercantour Massif (south-eastern France) at the earliest Pliocene. *Earth and Planetary Science Letters* 170, 35–47.
- Gautier, F., Clauzon, G., Suc, J.-P., Cravatte, J., Violanti, D., 1994. Age et durée de la crise de salinité messinienne. *Comptes-Rendus de l'Académie des Sciences de Paris Série 2* 318, 1103–1109.
- Gillet, S., Gramann, F., Steffens, P., 1978. Neue biostratigraphische Ergebnisse aus dem brackischen Neogen an Dardanellen und Marmara-Meer (Türkei). *Newsletters of Stratigraphy* 7, 53–64.
- Gillet, H., Lericolais, G., Réhault, J.-P., 2007. Messinian event in the black sea: evidence of a Messinian erosional surface. *Marine Geology* 244, 142–165.
- Gliozzi, E., Ceci, M.A., Grossi, F., Ligios, S., 2007. Paratethyan Ostracods immigrants in Italy during the Late Miocene. *Geobios* 40, 325–337.
- Görür, L., Çağatay, M.N., Sakiç, M., Sumengen, M., Senturk, K., Yaltrak, C., Tchapylyga, A., 1997. Origin of the Sea of Marmara deduced from Neogene to Quaternary paleogeographic evolution of its frame. *International Geological Review* 39, 342–352.
- Görür, L., Çağatay, M.N., Sakiç, M., Tchapylyga, A., Akkök, R., Natalin, B., 2000. Neogene Paratethyan succession in Turkey and its implications for paleogeographic evolution of the Eastern Paratethys. In: Bozkurt, E., Winchester, J.A., Piper, J.A.D. (Eds.), *Tectonics and Magmatism in Turkey and Surrounding Area*. Geological Society of London Special Publication, vol. 173, pp. 251–269.
- Guenoc, P., Gorini, C., Mauffret, A., 2000. Histoire géologique du Golfe du Lion et cartographie du rift oligo-aquitainien et de la surface messinienne. *Géologie de la France* 3, 67–97.
- Hallett, R.I., 1999. Consequences of environmental change on the growth and morphology of *Lingulodinium polyedrum* (Dinophyceae) in culture. Thesis, Westminster Univ. London, 109 pp.
- Hsü, K.J., Cita, M.B., Ryan, W.B.F., 1973. The origin of the Mediterranean evaporites. *Initial Reports of the Deep Sea Drilling Project* 42, 1203–1231.
- Kokinos, J.P., Anderson, D.M., 1995. Morphological development of resting cysts in cultures of the marine dinoflagellate *Lingulodinium polyedrum* (= *L. machaerophorum*). *Palynology* 19, 143–166.
- Lofi, J., Rabineau, M., Gorini, C., Berné, S., Clauzon, G., De Clarens, P., Dos Reis, T., Mountain, G.S., Ryan, W.B.F., Steckler, M., Fouchet, C., 2003. Plio-Quaternary prograding clinoform wedges of the Western Gulf of Lions continental margin (NW Mediterranean) after the Messinian Salinity Crisis. *Marine Geology* 198, 289–317.
- Lofi, J., Gorini, C., Berné, S., Clauzon, G., Dos Reis, A.T., Ryan, W.B.F., Steckler, M.S., 2005. Erosional processes and paleo-environmental changes in the western gulf of Lion (SW France) during the Messinian salinity crisis. *Marine Geology* 217, 1–30.
- Lourens, L.J., Hilgen, F.J., Laskar, J., Shackleton, N.J., Wilson, D., 2004. The Neogene period. In: Gradstein, F.M., Ogg, J.G., Smith, A.G. (Eds.), *A geological Time Scale 2004*. Cambridge University Press, Cambridge, pp. 409–440.
- Maillard, A., Mauffret, A., 2006. Relationship between erosion surfaces and Late Miocene Salinity Crisis deposits in the Valencia Basin (northwestern Mediterranean): evidence for an early sea-level fall. *Terra Nova* 18, 321–329.
- Marinescu, F., 1992. Les bioprovinces de la Paratéthys et leurs relations. *Paleontologia i Evolució* 24–25, 445–453.
- Marret, F., Zonneveld, K.A.F., 2003. Atlas of modern organic-walled dinoflagellate cyst distribution. *Review of Palaeobotany and Palynology* 125, 1–200.
- Martini, E., 1971. Standard Tertiary and Quaternary calcareous nannoplankton zonation. In: Farinacci, A. (Ed.), *Proceedings of the 2nd International Conference on Planktonic Microfossils*, Roma 1970, vol. 2. editura Tecnoscienza, Rome, pp. 739–785.
- Măruntăanu, M., Papaianopol, I., 1995. The connection between the Dacic and Mediterranean Basins based on calcareous nannoplankton assemblages. *Romanian Journal of Stratigraphy* 76, 169–170.
- Melinte, M.C., 2006. Cretaceous–Cenozoic paleobiogeography of the southern Romanian Black Sea onshore and offshore areas. *Geo-Eco-Marina* 12, 79–90.
- Ozenda, P., 1989. Le déplacement vertical des étages de végétation en fonction de la latitude: un modèle simple et ses limites. *Bulletin de la Société Géologique de France Série 8* 5 (83), 535–540.
- Perch-Nielsen, K., 1985. Cenozoic calcareous nannofossils. In: Bolli, H.M., Saunders, J.B., Perch-Nielsen, K. (Eds.), *Plankton Stratigraphy*. Cambridge University Press, Cambridge, pp. 427–554.
- Poisson, A., Wernli, R., Sağular, E.K., Temiz, H., 2003. New data concerning the age of the Aksu Thrust in the south of the Aksu valley, Isparta Angle (SW Turkey): consequences for the Antalya Basin and the Eastern Mediterranean. *Geological Journal* 38, 311–327.
- Popescu, S.-M., 2006. Upper Miocene and Lower Pliocene environments in the southwestern Black Sea region from high-resolution palynology of DSDP site 380A (Leg 42B). *Palaeogeography, Palaeoclimatology, Palaeoecology* 238, 64–77.
- Popescu, S.-M., Suc, J.-P., Melinte, M., Clauzon, G., Quillévère, F., Sütö-Szentai, M., 2007. Earliest Zanclean age for the peak of the Messinian Salinity Crisis. *Palynology* 33 (1), “latest Messinian” northern Apennines: new palaeoenvironmental data from the Maccarone section (Marche Province, Italy). *Geobios* 40 (3), 359–373.
- Popescu, S.-M., Dalesme, F., Jouannic, G., Escarguel, G., Head, M.J., Melinte-Dobrinescu, M.C., Sütö-Szentai, M., Bakrac, K., Clauzon, G., Suc, J.-P., 2009. *Galeacysta etrusca* complex, dinoflagellate cyst marker of Paratethyan influges into the Mediterranean Sea before and after the peak of the Messinian Salinity Crisis. *Palynology* 33 (1).
- Popov, S.V., Shcherba, I.G., Ilyina, L.B., Nevesskaya, L.A., Paramonova, N.P., Khondkarian, S.O., Magyar, I., 2006. Late Miocene to Pliocene palaeogeography of the Paratethys and its relation to the Mediterranean. *Palaeogeography, Palaeoclimatology, Palaeoecology* 238 (1–4), 91–106.
- Quézel, P., 1998. Cèdres et cédraies du pourtour méditerranéen: signification bioclimatique et phytogéographique. *Forêt méditerranéenne* 19 (3), 243–260.
- Quézel, P., Médail, F., 2003. *Ecologie et biogéographie des forêts du bassin méditerranéen*. Elsevier, Paris, 570 pp.
- Raffi, I., Backman, J., Fornaciari, E., Pälke, H., Rio, D., Lourens, L.J., Hilgen, F.J., 2006. A review of calcareous nannofossil astrobiochronology encompassing the past 25 million years. *Quaternary Science Reviews* 25, 3113–3137.
- Raffi, I., Mozzato, C., Fornaciari, E., Hilgen, F.J., Rio, D., 2003. Late Miocene calcareous nannofossil biostratigraphy and astrobiochronology for the Mediterranean region. *Micropaleontology* 49 (1), 1–26.
- Rögl, F., 1998. Palaeogeographic considerations for Mediterranean and Paratethys seaways (Oligocene to Miocene). *Annales des Naturhistorischen Museums in Wien* 99 (A), 279–310.
- Rögl, F., Steininger, F.F., 1983. Vom Zerfall der Tethys zu Mediterran und Paratethys. Die neogene Paläogeographie und Palinspastik des zirkum-mediterranean Raumes. *Annales des Naturhistorischen Museums in Wien* 85 (A), 135–163.
- Rouchy, J.-M., Suc, J.-P., Ferrandini, J., Ferrandini, M., 2006. The Messinian Salinity Crisis revisited. *Sedimentary Geology* 188–189, 1–8.
- Sage, F., Von Grönfeld, G., Déverchère, J., Gaullier, V., Maillard, A., Gorini, C., 2005. Seismic evidence for Messinian detrital deposits at the western Sardinia margin, northwestern Mediterranean. *Marine and Petroleum Geology* 22, 757–773.
- Sakiç, M., Yaltrak, C., 2005. Messinian crisis: what happened around the northeastern Aegean? *Marine Geology* 221, 423–436.
- Sakiç, M., Yaltrak, C., Oktay, F.Y., 1999. Palaeogeographical evolution of the Thrace Neogene Basin and the Tethys–Paratethys relations at northwestern Turkey (Thrace). *Palaeogeography, Palaeoclimatology, Palaeoecology* 153, 17–40.
- Seneš, J., 1973. Correlation hypotheses of the Neogene Tethys and Paratethys. *Giornale di Geologia* 39, 271–286.
- Snel, E., Măruntăanu, M., Meulenkamp, J.E., 2006. Calcareous nannofossil biostratigraphy and magnetostratigraphy of the Upper Miocene and Lower Pliocene of the Northern Aegean (Orphanic Gulf–Strimon Basin areas), Greece. *Palaeogeography, Palaeoclimatology, Palaeoecology* 238, 107–124.
- Sorrel, P., Popescu, S.-M., Head, M.J., Suc, J.-P., Klotz, S., Oberhänsli, H., 2006. Hydrographic development of the Aral Sea during the last 2000 years based on a quantitative analysis of dinoflagellate cysts. *Palaeogeography, Palaeoclimatology, Palaeoecology* 234, 304–327.
- Sprovieri, M., Sacchi, M., Rohling, E.J., 2003. Climatically influenced interactions between the Mediterranean and the Paratethys during the Tortonian. *Paleoceanography* 18 (2), 12–1–12–10.
- Stoica, M., Lazăr, I., Vasiliev, I., Krijgsman, W., 2007. Mollusc assemblages of the Pontian and Dacian deposits from the Topolog–Arges area (southern Carpathian foredeep—Romania). *Geobios* 40, 391–405.
- Suc, J.-P., Rouchy, J.M., Ferrandini, M., Ferrandini, J., 2007. Editorial. The Messinian Salinity Crisis revisited. *Geobios* 40 (3), 231–232.
- Taner, G., 1979. Die Molluskenfauna der Neogenen Formationen der Halbinsel – Gelibolu. *Annales Géologiques des Pays Helléniques* out of Series, 3, 1189–1194.
- Ternek, Z., 1964. Geological map of Turkey at 1/500,000: Istanbul. Maden Tetkik ve Arama Enstitüsü Hazırlamış ve Yayınlamıştır, Ankara.
- Türkecan, A., Yurtsever, A., 2002. Geological map of Turkey at 1/500,000: Istanbul. Maden Tetkik ve Arama Genel Müdürlüğü, Ankara.
- Van Couvering, J.A., Castradori, D., Cita, M.B., Hilgen, F.J., Rio, D., 2000. The base of the Zanclean Stage and of the Pliocene Series. *Episodes* 23 (3), 179–187.
- Wade, B.S., Bown, P.R., 2006. Calcareous nannofossils in extreme environments: the Messinian Salinity Crisis, Pölemi Basin, Cyprus. *Palaeogeography, Palaeoclimatology, Palaeoecology* 233, 271–286.
- Yaltrak, C., Sakiç, M., Oktay, F.Y., 2000. Westward propagation of North Anatolian fault into the northern Aegean: timing and kinematics: comment. *Geology* 28, 187–188.
- Young, J.R., 1998. Chapter 9: Neogene. In: Bown, P.R. (Ed.), *Calcareous Nannofossils Biostratigraphy*. British Micropaleontological Society Publications Series. Kluwer Academic Press, Dordrecht, pp. 225–265.

Contents lists available at ScienceDirect

Quaternary International

journal homepage: www.elsevier.com/locate/quaint
Pliocene and Lower Pleistocene vegetation and climate changes at the European scale: Long pollen records and climatostratigraphy

Speranta-Maria Popescu^{a,*}, Demet Biltekin^b, Hanna Winter^c, Jean-Pierre Suc^{d,1},
Mihaela Carmen Melinte-Dobrinescu^e, Stefan Klotz^f, Marina Rabineau^a,
Nathalie Combourieu-Nebout^g, Georges Clauzon^h, Florina Deaconuⁱ

^a IUEM, Domaines océaniques (UMR 6538 CNRS), 1 place Nicolas Copernic, 29280 Plouzané, France

^b Istanbul Technical University, School of Mines and Eurasia Institute of Earth Sciences, Istanbul, Turkey

^c Polish Geological Institute, 4 Rakowiecka street, 00-975 Warsaw, Poland

^d Laboratoire PaléoEnvironnements et PaléobioSphère (UMR 5125 CNRS), Université Lyon, 1, Campus de La Doua, bâtiment Géode, 69622 Villeurbanne Cedex, France

^e National Institute of Marine Geology and Geoecology, 23–25 Dimitrie Onciul street, P.O. Box 34–51, 70318 Bucharest, Romania

^f Geographical Institute, University of Tübingen, Rümelinstrasse 19–23 72070 Tübingen, Germany

^g Laboratoire des Sciences du Climat et de l'Environnement (UMR 1572), Orme des Merisiers, Centre de Saclay, 91191 Gif sur Yvette Cedex, France

^h C.E.R.E.G.E. (UMR 6635 CNRS), Université Paul Cézanne, 13545 Aix-en-Provence Cedex, France

ⁱ 39 rue Pierre Bourgeois, 69300 Caluire et Cuire, France

ARTICLE INFO

Article history:

Available online 19 March 2010

ABSTRACT

The biostratigraphically calibrated long pollen record at DSDP Site 380 (southwestern Black Sea) displays a high-resolution continuous and contrasted evolution of the vegetation in the region for the entire Pliocene and Lower Pleistocene. An accurate correlation is established with the reference global oxygen isotopic curve and with the Northwestern European climatostratigraphy. Climatostratigraphic relationships are evident at the latitudinal and longitudinal scale of Europe, confirming the extensive strength of pollen analysis as a tool for correlations over large distances.

© 2010 Elsevier Ltd and INQUA. All rights reserved.

1. Introduction

This paper is dedicated to Waldo H. Zagwijn, in recognition of his impressive contribution to European climatostratigraphy.

The quest for long pollen sequences that provide a continuous record of vegetation and climate changes since the Early Pliocene started with Lona (1950) and Zagwijn (1960, 1975), and was continued by Menke (1975) and Wijmstra and Groenhardt (1983) among others. Zagwijn was the first to establish a contrasted climatostratigraphy (from Brunssumian to Menapian for the time-interval on which this paper is focused, ca. 5.3–1 Ma). This paleoclimatological classification is still widely used today despite its

* Corresponding author. Tel.: +33298498741; fax: +33298498760.

E-mail addresses: speranta.popescu@univ-brest.fr (S.-M. Popescu), biltekin@itu.edu.tr (D. Biltekin), hanna.winter@pgi.gov.pl (H. Winter), jeanpierre.suc@gmail.com (J.-P. Suc), melinte@geoecomar.ro (M.C. Melinte-Dobrinescu), stefan.klotz@uni-tuebingen.de (S. Klotz), marina.rabineau@univ-brest.fr (M. Rabineau), Nathalie.Nebout@scea.ipsl.fr (N. Combourieu-Nebout), clauzon@cerge.fr (G. Clauzon), siminadeaconu@yahoo.fr (F. Deaconu).

¹ Present address: IUEM, Domaines océaniques (UMR 6538 CNRS), 1 place Nicolas Copernic, 29280 Plouzané, France.

weakness in independent chronological calibration (biostratigraphy, magnetostratigraphy). The Northwestern Mediterranean pollen diagrams (Autan 1, Garraf 1: Suc, 1984), biostratigraphically calibrated using foraminifera, established a reliable climatostratigraphic relationship between the Mediterranean and Northwestern Europe for the Pliocene and early Pleistocene (Suc and Zagwijn, 1983). However, insufficient sampling resolution of these pollen sequences did not result in an accurate correlation with the oxygen isotope stratigraphy. Recently, long high-resolution pollen records benefiting from an accurate time-control have been obtained for the Early Pliocene in southern Romania (Popescu, 2001; Popescu et al., 2006). Climatostratigraphic relationships were proposed with the corresponding interval of the deep Black Sea pollen record from the DSDP Site 380 (Popescu, 2006) a site that was lacking chronological constraints.

This paper presents new long pollen records from two boreholes, DSDP Site 380 (Black Sea) and Wólka Ligezowska (south Poland), separated by 9°30' latitude (Fig. 1), continuously covering the entire Pliocene and more or less completely the Lower Pleistocene. The new Pliocene–Pleistocene boundary recently lowered to 2.588 Ma, which moves the Gelasian Stage at

Fig. 1. Location map of site with pollen data used in this paper.

the base of the Pleistocene (Gibbard et al., 2010), and reduces the Pliocene to two stages only, the Zanclean and Piacenzian, is used. In addition, climatostatigraphy of these new pollen records is compared to that of other standard long pollen sequences (Fig. 1; Susteren 752.72 in The Netherlands: Zagwijn, 1960; Garraf 1 in the Northwestern Mediterranean: Suc, 1984; Rio Maior F16: Diniz, 1984). The comparison is extended to some other published Pliocene (Fig. 1; Lupoiaia: Popescu, 2001; Popescu et al., 2006) and Early Pleistocene long pollen sequences (Fig. 1; Semaforo, Vrica and Santa Lucia sections from the Crotona area: Combourieu-Nebout, 1990, 1993; Joannin et al., 2007; Suc et al., 2010; and Montalbano Ionico: Joannin et al., 2008).

2. DSDP Site 380

2.1. Setting and previous chronostratigraphy

At Site 380 drilled at 2107 m water depth (Fig. 1; 42°05.94'N, 29°36.82'E), Pliocene deposits are considered to start at a depth of 864.50 m, i.e. just above the pebbly breccia (Hsü, 1978) which correlates to the Messinian Erosional Surface in the deep Black Sea (Gillet et al., 2007). Site 380 is very poor in Mediterranean microplankton (foraminifera: Gheorghian, 1978; nannofossils: Percival, 1978; diatoms: Schrader, 1978; dinoflagellate cysts: Popescu, 2006). However, a climatic subdivision of this sedimentary long record has been proposed by Hsü (1978) using the "Steppe Index" established by Traverse (1978) using pollen grains. A climatostatigraphic approach was detailed by Popescu (2006) for the lowermost 316 m of the series.

2.2. Towards an independent biostratigraphic calibration

The presence of marine microorganisms in some intervals of the series (diatoms: Schrader, 1978; calcareous coccoliths: Percival, 1978; dinoflagellate cysts: Traverse, 1978) required particular attention to the presence of marine dinoflagellate cysts during pollen counting process in order to optimize the selection of new samples which would possibly contain nannofossils. Seventeen levels were chosen (at metre depths: 219, 223.02, 326.14, 334.50, 368.43, 461.53, 471.50, 476.46, 504.35, 509.35, 518, 548.50, 586.49, 682.95, 708.20, 748.45, and 840.07). They correspond to warm phases in the pollen record, i.e. they correspond to global high sea-levels and, as a consequence might be good candidates for indicating temporary connections between the Mediterranean and Black seas. Eight of these samples (at 219, 223.02, 326.14, 368.43,

476.46, 504.35, 748.45, and 840.07 m depth) effectively provided nannofossil markers (Table 1).

Analysis of the new samples gave new chronologic constraints within Site 380 sediments using both (1) nannoplankton zonation (Table 1; Martini, 1971; Okada and Bukry, 1980) and (2) ages of the lowest occurrence (LO), highest occurrence (HO), lower consistent occurrence (LCO) and highest consistent occurrence (HCO) of the species indicated by Raffi et al. (2006):

- *Triquetrorhabdulus rugosus* and *Ceratolithus acutus* have been both recorded at 840.07 m depth. Accordingly this sample is between 5.345 Ma (*C. acutus* LO) and 5.279 Ma (*T. rugosus* HO) (early Zanclean);
- at 748.45 m depth, the presence of *Reticulofenestra pseudoumbilicus* indicates an age older than 3.839–3.79 Ma (*R. pseudoumbilicus* HO) (late Zanclean);
- at 476.46 m and 504.35 m depth, the presence of *Discoaster brouweri* indicates an age older than 2.06–1.926 Ma (*D. brouweri* HO) (late Gelasian);
- at 368.43 m depth, the presence of medium-sized *Gephyrocapsa* indicates an age younger than 1.73–1.67 Ma (medium-sized *Gephyrocapsa* spp. LO) (Calabrian);
- at 326.14 m depth, the presence of *Helicosphaera sellii* indicates an age older than 1.34–1.256 Ma (*H. sellii* HO) (Calabrian);
- the record of *Reticulofenestra asanoi* at depths 223.02 and 219 m indicates that these samples are between 1.136 Ma (*R. asanoi* LCO) and 0.901 Ma (*R. asanoi* HCO) (Calabrian).

2.3. Pollen record and climatostratigraphy

2.3.1. Materials and methods

Samples for pollen analyses were taken every 50 cm when available between 861.65 and 198.95 m depth. Twenty grams were processed following a standard method (Cour, 1974): acid digestion (HCl, HF), concentration techniques (ZnCl₂ at density 2.00, sieving at 10 µm), and mounting in glycerol for allowing rotation of pollen grains that improves their examination and hence their identification. At least 150 pollen grains (not including *Pinus*) were counted per sample. The analysis involves 691 samples which were rich enough in pollen grains. Taxa have been grouped according to their ecological significance (Suc, 1984; Popescu, 2006) and the results are first presented in a semi-detailed pollen diagram where curves are independent (Fig. 2), then in a synthetic pollen diagram where curves are combined (Fig. 3). Percentages have been calculated on the total of pollen grains for both pollen diagrams. Considering the advantage of the ratio between two climatically opposed plants or

Table 1
Nannoplankton content of the DSDP Site 380 studied samples with related nannoplankton zones (NN12 to NN19: Martini, 1971; CN10a to CN14a: Okada and Bukry, 1980) and stratigraphic ages. Crosses indicating abundant reworked specimens (from Cretaceous, Paleogene and Neogene) are shown in bold. Preservation of nannofossils is generally good except for depth 219 m (all material reworked?).

Age	Depth (m)	Nannoplankton Zones (Martini, 1971; Okada and Bukry, 1980)	Amaurolithus primus	Braarudosphaera bigelowii	Calcidiscus leptoporus	Calcidiscus macintyreii	Ceratolithus acutus	Coccolithus pelagicus	Discoaster brouweri	Discoaster pentaradiatus	Discoaster triadiatus	Gephyrocapsa caribbeanica (medium)	Gephyrocapsa oceanica	Gephyrocapsa parallela (large)	Helicosphaera carteri	Helicosphaera sellii	Pontosphaera multipora	Pseudoaemiliana lacunosa	Reticulofenestra asanoi	Reticulofenestra pseudoumbilicus	Rhabdosphaera clavigera	Small reticulofenestrids	Spherolithus abies	Syracosphaera pulchra	Thoracosphaera sp.	Triquetrorhabdulus rugosus	Umbilicosphaera sibogae	Reworked specimens
Calabrian	219.00	NN19 CN14a		X	X			X				X		X	X		X	X	X			X			X			X
	223.02			X	X			X				X	X		X			X	X			X	X					X
	326.14	NN19 CN13b			X			X				X	X		X	X	X	X				X	X			X		X
	368.43				X			X				X	X		X	X	X					X			X			X
Late Gelasian	476.46	NN18 CN12d			X	X		X	X		X				X	X	X					X	X			X		X
	504.35				X	X		X	X		X				X		X					X	X		X	X	X	X
Late Zanclean	748.45	NN13 CN10b-c			X	X		X	X	X					X		X				X		X	X	X	X	X	X
Early Zanclean	840.07	NN12 CN10a	X		X	X	X	X							X		X				X	X	X	X	X	X	X	X

groups of plants shown by Cour and Duzer (1978), the pollen ratio “thermophilous elements/steppe elements” has been calculated (Fig. 3): values <1 can be interpreted as indicating cooling phases, those >1 warming phases (see Suc et al., 2010).

2.3.2. Pollen flora

Forest components are, except for some rare megathermic trees, represented by mega-mesothermic elements (Taxodiaceae mainly with probable *Glyptostrobus*, *Engelhardia*, Sapotaceae, *Cathaya*, *Nyssa*, etc.) and mesothermic elements (*Quercus*, *Carya*, *Pterocarya*, *Liquidambar*, *Zelkova*, *Carpinus*, *Ulmus*, *Parrotia persica*, etc.). In contrast, herbs are dominated by Poaceae, Amaranthaceae-Chenopodiaceae and Asteraceae, among which *Artemisia* is abundant and is the main contributor to steppe vegetation. Microthermic (*Abies*, *Picea*) and meso-microthermic (*Cedrus*, *Tsuga*) trees are not very frequent, similarly to *Pinus* and the Mediterranean xerophytes (*Olea*, *Quercus ilex* type, etc.).

Taxodiaceae needs to be emphasized, taking into account the important place of this taxon along the section, particularly in the Early Pleistocene layers. From 599.30 m depth up to the top of the studied sediments, Taxodiaceae pollen grains show the same morphology (Fig. 4) and may be considered to correspond to the same taxon, probably the last surviving Taxodiaceae species in the region. A detailed comparison was performed at

the scanning electronic microscope with pollen grains of the living genera and species of the family (Reyre, 1968; Jinxiang and Yuxi, 2000). Taxodiaceae pollen grains of the upper part of the Site 380 are from a morphological point of view identical to those of the extant species *Glyptostrobus lineatus* (= *Glyptostrobus pensilis*) (Fig. 4). This is in agreement with the common evidence of *Glyptostrobus europaeus* macroremains in the Upper Neogene of the region (Kasapligil, 1977; Gemici et al., 1991; Ţicleanu, 1992; İnci, 2002). The last extant species *G. lineatus* grows in lowland swamps and riparian forests from northern Vietnam to southern China, where it is common in the river deltas in Guangdong and Fujian (Wu and Raven, 1999). The last Taxodiaceae living in the Black Sea region during the Late Pliocene and Early Pleistocene, i.e. *Glyptostrobus*, occupied deltaic coastal areas suitable for ensuring pollen grain transport in large quantities down to the deep sea basin. As a consequence, the *Glyptostrobus* pollen grains recorded in interglacial deposits between 599.30 and 304.80 m depth may be over-represented, and the high-variability of tree and herb relative abundance somewhat exaggerated (Figs. 2 and 3).

2.3.3. Subdivisions of the pollen diagrams

This long pollen record shows fluctuations of variable amplitude that depict the continuous competition between forest

Fig. 2. Semi-detailed pollen diagram of DSDP Site 380 with the main pollen zones.

environments and open vegetation, as it is today in the area between forested coastal lands of Turkey and Bulgaria (relatively warm and humid climate) and the inner Anatolia steppes (cold and very dry climate) (Quézel and Barbero, 1985; Quézel and Médail, 2003). Seven major vegetation phases are recognized from the semi-detailed pollen diagram (Fig. 2) and are described hereafter. In addition, the establishment of the synthetic pollen diagram, in which relative fluctuations of the pollen groups are more easily decipherable, combined with the pollen ratio “thermophilous elements/steppe elements”, confirms the seven major subdivisions and allows their subdivision (Fig. 3), as described below:

- Pollen zone 1 (861.65–702.80 m). Trees are prevalent with abundant mega-mesothermic and mesothermic elements, with Cupressaceae, *Cathaya* being almost continuously recorded (Fig. 2). This interval corresponds to extended forest environments on the Black Sea coastal plains due to warm and humid climate, while *Artemisia* steppe probably developed on the Anatolian Plateau in relation with drier conditions. Zone 1 is subdivided into three subzones (Fig. 3) as discussed by Popescu (2006):
 - a (861.65–814.40 m). Mega-mesothermic and mesothermic elements are prevalent, herbs and steppe elements show low percentages despite some isolated peaks. This episode corresponds to the first maximum extension of the forest

with warm-moist climate temporarily affected by short cooling (and drying up) interval at 827–832 m.

- b (814.40–727.60 m). Abundance of mega-mesothermic elements is lower, while that of herbs and steppe elements is higher. Forests are reduced and steppe is strengthened during this interval in relation with fluctuating less warm and humid conditions.
- c (727.60–702.80 m). Mega-mesothermic elements, herbs and steppe elements show almost the same abundance as in subzone a. This phase displays the maximum development of forests and the minimum development of steppe. It is the warmest and most humid climatic phase recorded in the pollen diagram.
- Pollen zone 2 (702.80–624 m). Herbs with relatively low amounts of steppe elements are strongly dominant, while several mega-mesothermic elements and Cupressaceae are rare (*Cathaya*, *Arecaceae*, etc.) (Fig. 2). Open environments are developed except for the Anatolian *Artemisia* steppe. This denotes cooler but not very dry climatic conditions. Two subzones can be distinguished (Fig. 3):
 - a (702.80–655.20 m). Abundance of herbs is at maximum, that of trees minimum, except for some very brief peaks. *Artemisia* steppe shows very low percentages. Such a pollen flora indicates very open environments, as herbs are significantly under-represented in pollen records (Favre et al.,

Fig. 3. Synthetic pollen diagram of DSDP Site 380 and pollen ratio "thermophilous elements/steppe elements" (plotted on a semi-logarithmic scale) with the main and secondary pollen zones. When the pollen ratio is >1 : warming phase, when the pollen ratio is <1 : cooling phase. Pollen groups (thermic classification: Nix, 1982): 1, Mega-thermic elements (*Buxus colporate* grains, *Canthium*, *Acanthaceae*, *Sapindaceae*, *Sapotaceae*, *Bombax*, etc.); 2, Mega-mesothermic elements (*Taxodiaceae*, *Arecaceae*, *Engelhardia*, *Platycarya*, *Distylium*, *Hamamelis*, *Microtropis fallax*, etc.); 3, *Cathaya*; 4, Mesothermic elements (*Quercus*, *Carya*, *Pterocarya*, *Liquidambar*, *Carpinus*, *Ulmus*, *Zelkova*, *Alnus*, *Buxus sempervirens*, etc.); 5, *Pinus*; 6, Meso-microthermic trees (*Cedrus*, *Tsuga*, *Keteleeria*); 7, Microthermic trees (*Abies*, *Picea*); 8, Elements without significant (*Ranunculaceae*, *Rosaceae*, indeterminate pollen grains); 9, *Cupressaceae*; 10, Mediterranean xerophytes (*Olea*, *Pistacia*, *Ceratonia*, *Quercus ilex* type, *Rhus cf. cotinus*, *Rhamnus*, *Cistus*, etc.); 11, Herbs (*Amaranthaceae*-*Chenopodiaceae*, *Asteraceae*, *Poaceae*, *Apiaceae*, *Rumex*, *Borraginaceae*, *Convolvulus*, *Cyperaceae*, *Helianthemum*, *Euphorbia*, *Caryophyllaceae*, etc.) including subdesertic elements (*Nolina*, *Lygeum*, *Neurada*, *Prosopis*, *Calligonum*, etc.) and water plants (*Alisma*, *Typha*, *Potamogeton*, *Myriophyllum*, etc.); 12, Steppe elements (*Artemisia*, *Ephedra*, *Hippophae*).

2008). Cooler and not very dry climatic conditions characterize this episode.

- b (655.20–624 m). Mega-mesothermic and mesothermic trees show almost continuously higher percentages as well as steppe elements, indicating less open vegetation and relatively warmer conditions.
- Pollen zone 3 (624–603 m). Very low percentages of mega-mesothermic and mesothermic elements characterize this interval in contrast to high percentages of herbs and the first high percentage of steppe elements ($>25\%$) (Fig. 2). Very open vegetation and enlargement of the Anatolian *Artemisia* steppe probably was a response to a cooler and drier climate.
- Pollen zone 4 (603–460.30 m) is dominated by mega-mesothermic elements, especially *Taxodiaceae* which show their highest amount in the entire pollen diagram. They are probably almost completely composed of pollen grains from *Glyptostrobus*, a coastal tree (Section 2.3.2), significantly over-represented. Large amplitude fluctuations are observed and contrast thermophilous elements and herbs (without a significant steppe component, except for two peaks at the base). Phases dominated by herbs appear somewhat shorter than those dominated by thermophilous trees, which reach their maximum abundance in the entire pollen diagram (Fig. 2). Coastal forests strongly competed with inland open vegetation, probably a response to rapid climate fluctuations (warmer-moister phases opposed to cooler-drier ones), probably corresponding to glacial–interglacial cycles. Three subzones have been designated:
 - a (603–531.70 m). Maxima of thermophilous elements alternate with peaks of herbs with two peaks of steppe elements (Fig. 3). Competition is relatively balanced between coastal forests and inland open environments, illustrating the steady alternation of interglacial–glacial cycles.
 - b (531.70–519.95 m). This short episode is marked by very low values of thermophilous elements as opposed to the high values of herbs with little steppe elements (Fig. 3). Such a sudden opening of vegetation is related to a pronounced cooling without high dryness.
 - c (519.95–460.30 m). The largest repeated maxima of mega-mesothermic elements alternating with shorter maxima of herbs with moderate abundance of steppe elements (Fig. 3) document repeated closure–opening of the vegetation and large amplitude fluctuations in temperature and dryness.
- Pollen zone 5 (460.30–394.50 m) contrasts with the preceding zone by: (1) lower amounts of mega-mesothermic elements (except some peaks) and higher amounts of mesothermic elements, (2) longer temporal development of herbs with a higher frequency in *Artemisia* pollen at the end of the interval (Fig. 2). This illustrates the competition between weakly developed coastal and inner forests and open vegetation, characterized by a significant spread of the Anatolian *Artemisia* steppe. This evolution demonstrates shorter and cooler interglacials in opposition to longer, colder and drier glacials. Three subdivisions are proposed:
 - a (460.60–418 m). The development of *Pinus* weakens herb maxima which display moderate peaks of steppe elements (Fig. 3). Forest development is very weak during interglacials, *Pinus* and *Taxodiaceae* being probably over-represented. Open vegetation was prevalent during glacials but still important during interglacials. Temperature during interglacials was moderate in contrast to cold glacials, both characterized by not very dry conditions.
 - b (418–412.50 m). A brief but well-marked maximum in thermophilous elements characterizes this episode (Fig. 3)

Fig. 4. Taxodiaceae pollen grains. 1. Pollen grain from Site 380 (448 m depth): a, General view showing the papilla ($\times 2000$); b, Detail of the sculpture in the papilla area ($\times 6000$). 2. Pollen grain from Site 380 (448 m depth): a, General view ($\times 2000$); b, Detail of the sculpture ($\times 12000$). 3 and 4. Two modern pollen grains of *Glyptostrobus pensilis* (Staunton ex D. Don) K. Koch, originating from China (Herbarium of the Sun Yat-sen University, Guangzhou; sample number: 28781; originating from the Herbarium of the Lingnam University, China): 3a, General view ($\times 2000$); 3b, Detail of the sculpture ($\times 12000$); 4a, General view of another grain ($\times 2000$); 4b, Detail of its sculpture ($\times 12000$).

which corresponds to a forest strengthening during a fluctuating interglacial.

- c (412.50–394.50 m). This interval shows the return to conditions of subzone 5a (Fig. 3) with more steppe elements and, as a consequence, drier conditions.
- Pollen zone 6 (394.50–302.40 m) shows increased thermophilous forest elements among which mesothermic trees became more important with strong repeated fluctuations between forest and open (with important steppe) environments (Fig. 2). This indicates large amplitude climatic fluctuations between interglacials and glacials. In detail, 7 subzones have been identified (Fig. 3):
 - a (394.50–380.07 m). Thermophilous elements, particularly the mega-mesothermic ones, are very abundant, indicating a pronounced interglacial episode.
 - b (380.07–359.30 m). Herbs and steppe plants prevail despite several moderate forest developments especially concerning mesothermic elements, denoting glacial–interglacial cycles characterized by lower temperatures.
 - c (359.30–351.60 m). Thermophilous elements are again dominant, indicative of an interglacial including some secondary coolings. Conditions appear less warm than during subzone a.
 - d (351.60–329.60 m). This is a well-marked steppe phase, i.e. cool and very dry conditions corresponding to a glacial event.
 - e (329.60–322.10 m). This is another forest interglacial phase almost similar to subzone 6a.
 - f (322.10–313.40 m). A maximum in herbs with abundant steppe elements at beginning characterizes this glacial phase.
 - g (313.40–302.40 m). Thermophilous trees increase again with a more important contribution of mesothermic elements during this distinctive interglacial.

- Pollen zone 7 (302.40–198.95 m). This zone is greatly dominated by herbs and steppe elements, the remaining thermophilous trees being henceforth mostly composed by the mesothermic elements (Fig. 2). Climatically, this zone corresponds to long severe climatic conditions interrupted by some short and weak improvements, allowing its subdivision in 4 subzones (Fig. 3):

- a (302.40–294.40 m) represents a first intense maximum in herbs and steppe elements although two brief modest recoveries of thermophilous elements, as a response to a strong glacial.
- b (294.40–269.90 m) corresponds to a low increase in mesothermic trees accompanied by *Pinus*. Herbs and steppe elements weakened, indicating a not very well pronounced interglacial.
- c (269.90–252.40 m). Two important peaks of herbs and steppe elements alternate with a weak maximum in thermophilous elements, indicating improving conditions during a dominant glacial phase.
- d (252.40–198.95 m). This last subzone is characterized by a large prevalence of herbs and steppe elements, punctuated by several small increases in thermophilous trees. It corresponds to dominant glacial phases interrupted by short interglacials.

2.3.4. Relationships with the oxygen isotope stratigraphy

Using the chronologic constraints provided by nannoplankton, these major phases and their subdivisions are correlated with Marine Isotope Stages (MIS) chosen on the basis of age control, general trends and amplitude of the respective fluctuations (Fig. 5).

Fig. 5. Synthetic pollen diagram of DSDP Site 380 with detailed pollen zonation, localized nannoplankton evidences and proposed climatostratigraphic relationships with (1) the reference oxygen isotope curve (some Marine isotope Stages are plotted) (Shackleton et al., 1990, 1995) correlated with nannofossil biohorizons (Raffi et al., 2006), and (2) the NW European climatostratigraphic succession (Zagwijn, 1960, 1998). Nannofossil biohorizons: LO, Lowest Occurrence; HO, Highest Occurrence; LCO, Lowest Consistent occurrence; HCO, Highest Consistent Occurrence (Raffi et al., 2006). Chronostratigraphy (ATNTS 2004: magnetic reversals, stage boundaries) is from Lourens et al. (2004). Pollen groups: see Fig. 3.

This integrated approach led to an accurate age model of Site 380 (Fig. 5).

- Pollen zone 1. This warm phase was already referred to the Zanclean Stage by Popescu (2006). However, the evidence of nannofossils at 840.07 m implies a readjustment of the correlation to the oxygen isotope curve initially proposed by Popescu (2006). The oldest cooling event at 828.02 m does not correlate with MIS T4 but with MIS T8. Similarly, correlation of the preceding tree maxima with the $\delta^{18}O$ minima suggests that the oldest sample from aragonite is ca. 5.45 Ma old (i.e. older

than the 5.327 Ma age previously proposed; Popescu, 2006). The end of pollen subzone 1a, i.e. pollen zone 4 in Popescu (2006), can be correlated with MIS T5, the end of pollen subzone 1b, i.e. pollen zone 5 in Popescu (2006), with MIS Gi19–Gi17.

- Pollen zone 2. The beginning and the end of this interval obviously correlate with MIS MG 2 (well-marked cooling) and MIS G1 just preceding the earliest glacial (MIS 104), respectively.

- Pollen zone 3. The three recorded peaks in herbs and *Artemisia* steppe are hence correlated to MIS cluster 104–100–98–96.

Fig. 6. Compared deep Black Sea and global ocean sedimentation rates. A. Sedimentation rate at Site 380 between 861.50 and 198.95 m depth. Coordinates of the inflection points are those indicated in the last column of Table 2 (thick dotted lines delimit intervals discussed in the text, with indication of the mean sedimentation rate of which). B. Global ocean sedimentation rate (Lisiecki and Raymo, 2005). The dotted line indicates the mean value for the three time-intervals 5–3.1 Ma, 2.5–2 Ma and 1.7–0.9 Ma.

One peak is missing in the pollen record probably because of an insufficient sampling resolution.

- Pollen zone 4. These fluctuations reflect short-term interglacial–glacial cycles (~41 ky) within a relatively warm global context. It correlates to the MIS 95–63 interval. Fourteen of the seventeen known interglacials have been recorded in the pollen diagram despite the relatively low sample resolution in the lower half of the interval. Correlations can also be refined by identifying the brief cooling at 531.70–519.95 m (pollen subzone 4b) as the well-marked MIS 82.
- Pollen zone 5 is correlated to the MIS 62–50 interval, with the pollen subzone 5b corresponding to MIS 55.
- Pollen zone 6. This interval runs from MIS 49 to MIS 37, the limits between the main subdivisions (pollen subzones 6a–b and 6d–e) being located at transitions between MIS 47–46 and MIS 38–37, respectively. In addition, the pollen diagram displays almost the same number of fluctuations as the oxygen isotope curve for this interval.
- Pollen zone 7. This pollen zone is correlated to the oxygen isotope curve from MIS 36 to MIS 22, its fluctuations being possibly correlated to detailed isotopic stages (see for example MIS 35, very well-marked in the pollen diagram = pollen subzone 7b).

The age model of the 861.65–198.95 m depth interval of Site 380 allows reconstruction of the sedimentation rate in the deep Black Sea (Fig. 6A). The curve shows a first break at 3.37 Ma, i.e. 702.40 m in depth, which coincides with the observed opening of the vegetation (Fig. 3). Below this point, the slope of the curve is weaker

(mean sedimentation rate: ca 8 cm/ky); above it, the sedimentation rate is almost two times higher (average: 19 cm/ky). It therefore records a significant increase in terrigenous material input around 3.4 Ma that might correspond to enhanced erosion related to the abrupt change from dominating trees to dominating herbs. The continuing open vegetation episode after 3.1 Ma up to 2.43 Ma (pollen subzone 2b and pollen zone 3) corresponds to a decrease in the sedimentation rate (Fig. 6A). The study of Pliocene – Pleistocene sedimentation rates from 57 globally spread ODP sites (Lisiecki and Raymo, 2005) showed an evolution of sedimentation rates from 2.5 to 3 cm/ky between 5 and 3.1 Ma, increasing to 4–4.5 cm/ky between 2.5 and 2 Ma and finally about 4.5 cm/ky between 1.7 and 0.9 Ma (Fig. 6B). The global oceanic sedimentation rate increased from Pliocene to Pleistocene. The same overall evolution is present in the Black Sea, but the initial increase seems to start a little earlier (3.4 Ma instead of 3.1 Ma in this case). The second increase is about the same date: 2.4 Ma in the Black Sea and 2.5 Ma in the global ocean, which is also the most commonly recognized timing for the initiation of Northern Hemisphere major glaciations. Values of average sedimentation rates are much higher in this case, because of the importance of fluvial inputs in the Black Sea.

The previous chronostratigraphy of Pliocene and Pleistocene sediments cored at Site 380, based on a low-resolution pollen record (Traverse, 1978), was already significantly modified for its lower part (861.65–704.34 m depth) by the high-resolution pollen record of Popescu (2006). The present results covering the entire Pliocene and Lower Pleistocene of this section significantly change the previous chronostratigraphy (Hsü, 1978; Muratov et al., 1978; Ross, 1978) and demonstrate the interest of high-resolution pollen analyses in the Black Sea.

Evidence of climate cycles (i.e. glacial–interglacial successions) in the Site 380 pollen record is apparent from 624 m depth (prevalent herbs and trees in turn) (Figs. 3 and 4). Despite an insufficient chronologic resolution in some parts of the pollen record through lack of available sediment, a detailed correlation with the reference isotope curve has been established. At the base (624 m depth), climate cycles clearly report the obliquity forcing (~41 ky) resulting in fast and intense replacements in the vegetation and an almost equal duration of glacial and interglacial phases. Above 303 m depth (i.e. at about 1.22 Ma), the ~41 ky signal attenuates, and the duration of cycles increases with longer glacial which is coherent with the installation of dominant ~100 ky climatic cycles, i.e. the onset of the Mid-Pleistocene Revolution (Maslin and Ridgwell, 2005; Lisiecki and Raymo, 2007).

2.3.5. Relationships with the Northern Europe climatostratigraphy

The long–distance correlation reliability of Zagwijn's Pliocene and Early Pleistocene climatostratigraphy has been severely questioned. This climatostratigraphy effectively suffered from several handicaps, such as the lack in sedimentary continuity especially for the Pleistocene, the weakness of an independent chronostratigraphic calibration, and the low number of pollen samples (Donders et al., 2007; Kemna and Westerhoff, 2007; Westerhoff et al., 2008; Westerhoff, 2009).

However, the high-quality of pollen record of Site 380 and the great variability of the prevalent pollen groups undoubtedly allow its use as a reliable climatostratigraphic reference. Detailed comparison with the pre-existing Northern Europe climatostratigraphy (Zagwijn, 1960, 1963, 1975, 1985, 1998; Zagwijn and Suc, 1984) can be attempted (Fig. 5):

- Pollen zone 1. As already argued by Popescu (2006), this warm phase clearly corresponds to the Brunsumian of the NW

European climatostratigraphy, with two warm phases surrounding a less warm phase, respectively correlated with the three Brunsumian subdivisions.

- Pollen zone 2. This well-delimited episode is related to the Reuverian of the NW European climatostratigraphy, as it is a transitional interval between the last warm Pliocene event and the earliest glacials. It is made of two subzones:
 - subzone 2a, which reports the same temperature sequence as Reuverian A;
 - subzone 2b, which resembles Reuverian B by the same argument.

The transitional Reuverian C was not identifiable at Site 380 because there is an insufficient quantity of pollen grains between 638.20 and 625.35 m.

- Pollen zone 3. This first global cold interval corresponds to the Praetiglian, which is less homogenous than previously considered (cf. Suc and Zagwijn, 1983).
- Pollen zone 4. This zone immediately follows the earliest glacials. Accordingly, it is correlated with the “warm” Tiglian. Considering that the number of climatic fluctuations within the Tiglian was not definitely established as they have been regularly increased by new pollen data (see the increasing complexity of Tiglian A and, especially, Tiglian C in the successive Zagwijn papers), some correlations with the Tiglian three main subdivisions can also be suggested. On the basis of a rather robust correlation between pollen subzone 4b (well-marked cooling) with the “cold” Tiglian B, (1) the Site 380 subzone 4a can be correlated with Tiglian A, while (2) subzone 4c is correlated with Tiglian C, although the former is more subdivided.
- Pollen zone 5. This dominantly well-delimited “cold” phase is considered to correspond to the Eburonian. On the whole, subzone 5 displays numerous similarities with the Eburonian seven subdivisions (such as, for example, beginning and ending with a strong cooling). However, the significantly higher number of its fluctuations makes risky a detailed correlation with Eburonian. Only the pronounced interglacial of subzone 5b could be referred to Eburonian VI because of its location in the upper part of zone 5.
- Pollen zone 6. This zone contrasts with the previous one in indicating warmer conditions. Accordingly, it can be correlated with the Waalian, which is regarded as a dominantly “warm” phase. The two warmest episodes of zone 6 (subzone a on the one hand, subzones e–g on the second hand) sandwiching the complex subzones b–d can be compared with Waalian A and C, respectively.
- Pollen zone 7. Similar reasoning refers this zone to the Menapian, with two “cold” phases (subzones a and c–d) bordering a warmer one (subzone b regarded as equivalent of Menapian II).

This climatostratigraphic interpretation of the Site 380 pollen diagram not only provides a detailed age model for the considered section (Fig. 5) but also supports the large geographic scale reliability of the Zagwijn climatostratigraphy, at least concerning the main phases. In addition, the Site 380 climatostratigraphy clarifies the relationships of the Northern Europe climatostratigraphy and the oxygen isotope stratigraphy. Several authors, including Zagwijn himself (Zagwijn, 1975, 1998; Zagwijn and Suc, 1984; Kukla and Cilek, 1996; Leroy, 2007) have published calibrations and/or estimates of the age of some of the climatostratigraphic unit boundaries (Table 2). On the whole, taking also into account the numerous modifications of the Global Polarity Time Scale over the last decades, most of these ages are consistent with the ages provided

by correlation of pollen fluctuations documented here with the reference oxygen isotope curve. More precisely, the ages indicated by Zagwijn (1998) and Leroy (2007) are very close to these ages (Fig. 5; Table 2).

In addition, the ratio between “thermophilous elements/steppe elements” from Site 380 (Fig. 3) constitutes a temperature curve which resembles that published by Zagwijn (1998) for The Netherlands, especially concerning the climate megaphases (pollen zones 1 to 7), especially those encompassing glacial–interglacial cycles. Phases corresponding to a global cool to cold context (pollen zones 3, 5 and 7; i.e. Praetiglian, Eburonian, Menapian) have glacials (prevalent herbs) more marked than interglacials (dominant trees). Phases corresponding to warm contexts (pollen zones 4 and 6; i.e., Tiglian and Waalian) have interglacials (prevalent trees) more marked than glacials (dominant herbs).

2.3.6. Spectral analysis

The relationship with astronomical cycles was used as a base for a spectral analysis after grouping the pollen zones into three parts (Fig. 7):

- lower part: pollen zones 1–2 (861.65 m–624 m; 5.45–2.62 Ma) prior to the onset of glacial–interglacial cycles;
- middle part: pollen zones 3–6 (624 m–302.40 m; 2.62–1.22 Ma) when short-term glacial–interglacial cycles (~41 ky) dominated;
- upper part: pollen zone 7 (302.40–198.95 m depth; 1.22–0.85 Ma) when long-term glacial–interglacial cycles (~100 ky) started.

Generally, due to differences in the number of samples considered (upper part: 129; middle part: 560; lower part: 1132), the significance of the spectral analysis is higher in the lower part of the section, a fact that can also be read from the width of the respective confidence intervals – the smaller the confidence interval, the higher the significance of the analysis.

Analysis of the lower part relies on the highest number of assemblages and is related to the highest significance, i.e. to the smallest confidence intervals. Accordingly, although a weak ~41 ky obliquity cycle exists, the spectral bands are dominated by multiple ~21–23 ky precession cycles. The ~100 ky cycle signal is represented only by 6 out of 34 plant taxa, i.e. Asteraceae Asteroideae, Cupressaceae, Cyperaceae, Poaceae, *Carya*, and *Quercus ilex*.

Significant information is provided for the middle part, yielding a stronger influence of the ~100 ky cycle and the ~41 ky obliquity cycle (especially when considering the representative taxa). In addition, the multiple ~19–21 ky and ~21–23 ky precession periodicities can be observed across all taxa analysed. A more distinguished view on the ~100 ky cycle is given by comparing those taxa which respond to this orbital signal with those, due to their physiology or environmental requirements, which are not influenced. Ranunculaceae, *Artemisia* and Asteraceae Asteroideae clearly respond to the ~100 ky cycle, whereas *Pinus*, Asteraceae Cichorioideae, Caryophyllaceae, Brassicaceae, and deciduous *Quercus* have no affinity (number of non-zero entries are equivalent).

Although the significance is comparatively low, the upper part reveals that vegetation largely responded to ~100 ky climatic cycles as a multiple ~19–23 ky precession periodicity. The ~100 ky cycle is best reflected by Amaranthaceae–Chenopodiaceae, Asteraceae Cichorioideae, and *Carpinus betulus*, whereas Rosaceae and *Betula* do not show any affinity to the long climatic cycles. Interestingly, Ranunculaceae is the only taxon which indicates the occurrence of a weak ~41 ky cycle.

Table 2

Proposed ages in Ma (Zagwijn, 1960, 1998; Zagwijn and Suc, 1984; Kukla and Cilek, 1996; Leroy, 2007) for some limit boundaries of the NW Europe climatostratigraphic subdivisions. Some of them (base of Praetiglian, Eburonian and Menapian) have been calibrated on the basis of paleomagnetic reversals (Van Montfrans, 1971; Zagwijn, 1975). For most of the limit boundaries of the NW Europe climatostratigraphic subdivisions, the proposed ages in this paper are extrapolated from the climatostratigraphic interpretation of the Site 380 pollen record.

NW EUROPE CLIMATOSTRATIGRAPHY	Zagwijn (1975)	Zagwijn and Suc (1984)	Kukla and Cilek (1996)	Zagwijn (1998)	Leroy (2007)	This paper	
BAVELIAN	~0.72	~1.0		~1.07	~1.05	0.85	
M E N A P I A N							
III ----- II ----- I							
W A A L I A N	0.9	1.2	~1.12	~1.25	~1.25	1.22	
		c ----- b ----- a				~1.25	1.28
		~1.3				1.42	
		~1.4				~1.5	~1.45
E B U R O N I A N	1.7	1.6	~1.56	1.75	1.7	1.76	
		VII ----- VI ----- I-V					1.59
							1.63
T I G L I A N	~2.2	~1.8		~2.0		2.14	
		~1.9		~2.12		2.18	
		A ----- B ----- C					
PRAETIGLIAN	2.6	2.3	~2.19	2.5	2.45	2.62	
R E U V E R I A N		~2.45		~2.6			
		~2.7		~3.0		3.12	
		~3.2				3.37	
B R U N S S U M I A N		~3.75				3.94	
		~4.05					5.08
		~5.2					5.45
SUSTERIAN							

Fig. 7. Power spectra calculated for Site 380 using fast Fourier transformation (5% tapered, 7 band Tukey-window, confidence intervals for the upper, middle and lower parts of the pollen record. Solid black lines representing the average power spectra calculated on taxa with more than 40% non-zero entries in the respective section. Dotted grey lines representing the average power spectra calculated on three taxa representative of the pollen zonation: these are for the lower part *Amaranthaceae*, *Acer*, *Picea*; for the middle part *Asteraceae* Cichorioideae, Ranunculaceae, *Carpinus orientalis*; and for the upper part *Amaranthaceae*, *Asteraceae* Cichorioideae, *Carpinus orientalis* (for better representation spectral density is multiplied with factor 3 for all parts). Vertical dashed lines indicate the major cycles, vertical double dotted lines represent multiples of the major cycles.

3. Wólka Ligezowska

The Wólka Ligezowska (Poland: 51°35'N, 20°35'E) pollen record comes from cored sediments (39.50 m in depth) through a paleo-lake in the southern part of the Mazovian Lowland. The studied sediments (from 34.50 to 10.15 m depth) consist of silts, clays, gyttjas and sandy layers interrupted by peat layers (Jakubowicz et al., 1994). These sediments belong to the Polish “pre-glacial

complex” the age of which is now accepted to be Pliocene and Early Pleistocene on the basis of palynostratigraphy (Baraniecka, 1991; Stuchlik, 1994).

3.1. Material and methods

A total of 113 samples were processed in 10% HCl and then boiled in 10% KOH. They were treated with heavy liquid in order to

remove the mineral fraction, and finally the Erdtman (1969) acetolysis method was applied.

At least 250 pollen grains were counted per sample, but usually 1000 (sometimes 500 pollen grains of trees and shrubs). Results are presented in a semi-detailed pollen diagram (with independent curves) where taxa have been grouped according to their ecological significance, and in a synthetic pollen diagram (combined curves) (Fig. 8). Calculation of pollen percentages was based on the total number of pollen grains.

3.2. Vegetation changes and subdivisions of the pollen diagrams

The relative distribution of the main groups in the semi-detailed pollen diagram results in identification of 3 pollen zones. The synthetic pollen diagram allows further subdivision of pollen zones 1 and 2 in three subzones, respectively (Fig. 8).

- Pollen zone 1 (34.50–25.90 m). This interval suggests that plant ecosystems were connected with wet habitats:
 - swamp forest with *Alnus*, *Nyssa*, Taxodiaceae, Cupressaceae, *Carya* and *Pterocarya*;
 - marshes inhabited by shrubby *Salix* and *Myrica*, dwarf shrubs represented by Ericaceae, accompanied by Poaceae and Cyperaceae as main components of herbs;
 - meso-hygrophilous forest dominated by *Ulmus*, *Pterocarya*, *Liquidambar*, *Fraxinus* *Carya* and *Juglans* in the drier areas.

The deciduous forest community was composed of mesothermic taxa such as *Quercus*, *Castanea*, *Ilex* (represented by pollen grains of *Ilex aquifolium* and *Ilex margaritatus* types), *Eucommia*, *Aesculus*, *Liriodendron*, *Magnolia*, *Parrotia persica*, *Carpinus*, *Tilia*, *Fagus*, and *Acer*. *Pinus*, *Picea*, *Abies* and *Tsuga* are in low frequencies and suggest that colder conditions existed in the region. Variability of *Sequoia* type expresses changes both in humidity and temperature. Such arboreal pollen assemblages indicate a warm-temperate and humid climate at low altitude. However, the regular occurrence of *Itea*, a mega-mesothermic element, is notable.

Three subzones have been identified:

- a (34.50–30.15 m). It is characterized by abundant mega-mesothermic elements, and is the oldest warm phase in the pollen record.
- b (30.15–27.34 m). This phase indicates some cooling as expressed by a decrease in mega-mesothermic elements and an increase in mesothermic elements over *Pinus sylvestris* type and herbs.
- c (27.34–25.90 m). Mega-mesothermic are abundant, making this phase the youngest warm one in the pollen record.
- Pollen zone 2 (25.90–17.60 m). The gradual decrease in thermophilous elements and disappearance of those characterizing humid climate and their replacement by *Quercus* might indicate more continental climatic conditions (increasing dryness and decreasing temperature). Larger percentages of

Fig. 8. Semi-detailed and synthetic pollen diagrams of Wólka Ligezowska borehole with pollen zonation. Pollen groups (thermic classification: Nix, 1982): 1, Megathermic elements; 2, Mega-mesothermic elements; 3, *Pinus haploxylon* type (= *Cathaya*); 4, Mesothermic elements; 5, *Pinus sylvestris* type; 6, Meso-microthermic trees; 7, Microthermic trees; 8, Elements without signification; 9, Cupressaceae; 10, Mediterranean xerophytes; 11, Herbs; 12, Steppe elements; 13, Ericaceae.

herbaceous pollen and their higher diversity (Poaceae, Cyperaceae, Asteraceae, Oenotheraceae, Brassicaceae, *Lythrum*, *Rumex acetosa* type, *Polygonum persicaria* type, etc.) show the development of varied communities in open habitats. *Phlomis*, *Cistus*, *Medicago* and Malvaceae (*Lavatera*) pollen grains occur in small amounts.

This zone has been subdivided into three subzones:

- a (25.90–23.02 m). It displays several fluctuations with a first development of *Alnus* (mesothermic tree) followed by a well-marked strengthening of *Betula* (meso-microthermic tree) alternating with the last peaks of mega-mesothermic trees (*Nyssa* and *Sequoia* type), and finally an increase in *Quercus* (mesothermic tree), *Pinus sylvestris* type and herbs. Climatologically, this evolution expresses a weak cooling, then alternating warmer and cooler conditions, and finally less humid ones.
- b (23.02–20.07 m). Large amounts of Poaceae (herbs) contrast with those of *Quercus* (mesothermic tree) according to variability in *Pinus sylvestris* type, illustrating temperate conditions with fluctuating humidity.
- c (20.07–17.60 m). *P. sylvestris* type frequency increases, suggesting the development of an open pine forest, while herbs (Poaceae and Cyperaceae mostly) remain abundant and *Quercus* becomes rare. Probably, the climate conditions weakened with a drop in winter and summer temperatures and decreased precipitation.
- Pollen zone 3 (17.60–10.15 m). The strong expansion of herbs and dwarf shrubs documents the onset of severe cooling (earliest glacials). The vegetation was dominated by Poaceae, Cyperaceae, various species of *Artemisia*, and Ericaceae including *Bruckenthalia* and *Calluna vulgaris*. Typical heliophytes such as *Ephedra*, *Helianthemum*, *Thalictrum*, *Plantago* and *Gypsophila* were commonly recorded. The rich taxonomic composition reflects a considerable variety of plant communities in a humid tundra-like vegetation (dominant Cyperaceae) including dwarf shrubs (Ericaceae, *Betula nana*) and some drier assemblages (Poaceae, *Artemisia*, Chenopodiaceae). Some improvements in temperature and decrease in moisture are indicated by *Pinus sylvestris* type indicating some reappearance of forest areas (tundra-park) in opposition to typical tundra phases (Ericaceae, Poaceae, Cyperaceae and *Artemisia* growths).

3.3. Relationships with the Northwestern European climatostratigraphy

The earliest development of tundra-like vegetation in the upper part of the Wólka Ligezowska pollen record establishes the Pliocene age of the underlying sediments. As the changes in vegetation pointed out by the Wólka Ligezowska pollen diagrams are apparent and significant, a direct climatostratigraphic relationship can be reliably proposed with the Northwestern European chart (Zagwijn, 1998) as shown on Fig. 9. Pollen zone 1, the forest lower phase rich in thermophilous trees, correlates with the Brunssumian with consistent subdivisions (a slightly cooler phase, subzone b, sandwiched between the two warmest phases of the pollen record, pollen subzones a and c). The overlying pollen zone 2, dominated by mesothermic elements, relates to the Reuverian, including the respective subdivisions: two cyclic climatic phases, pollen subzones a and b (at a lesser thermic level), preceded the transitional episode of pollen subzone c which is accordingly related to Reuverian C. The tundra-like following pollen zone 3 is connected with the Praetiglian and displays fluctuations.

4. Synthesis at the European scale

Concerning the Pliocene and earliest Pleistocene, climatostratigraphic relationships may be established between DSDP Site 380 and the Northwestern European climatostratigraphy (Zagwijn, 1998) as well as for the Wólka Ligezowska pollen record for the Pliocene and earliest Pleistocene (Brunssumian to Praetiglian) (Fig. 9). Similar reliable relationships have already been proposed for the Early Pliocene (Zanclean and early Piacenzian) between Site 380, Garraf 1 (Suc, 1984), Susteren 752.72 (Zagwijn, 1960) and Rio Maior F16 (Diniz, 1984) by Popescu (2006) (Fig. 9). Such climatostratigraphic relationships can be also tentatively extended to the Late Pliocene (Piacenzian) and Early Pleistocene (earliest Gelasian) (Fig. 9). The base of the Praetiglian (MIS 104) has been plotted on each pollen diagram where the most important break in vegetation is recorded (development of tundra-like environment in The Netherlands and Poland, *Artemisia* steppe in the Mediterranean region *s.l.*, Cupressaceae woodland in Portugal). Each bioclimatic province specifically responded to the successive changes in climate according to latitude, longitude and regional features (Suc et al., 1995; Suc and

Fig. 9. Climatostratigraphic correlations for the Pliocene and earliest Pleistocene (1) between the synthetic pollen diagram of Site 380, Lupoia, Garraf 1, Susteren 752.72, Rio Maior F16, and Wólka Ligezowska, (2) with the NW European climatostratigraphic succession (Zagwijn, 1998), and (3) the reference oxygen isotope curve (some Marine Isotope Stages are plotted) (Shackleton et al., 1990, 1995). Chronostratigraphy (ATNTS 2004: magnetic reversals, stage boundaries) is from Lourens et al. (2004). Magnetostratigraphy performed at Lupoia is indicated (Popescu, 2001). Pollen groups (thermic classification: Nix, 1982): 1, Megathermic elements; 2, Mega-mesothermic elements; 3, *Cathaya* (= *Pinus haploxylon* type at Wólka Ligezowska); 4, Mesothermic elements; 5, *Pinus*; 6, Meso-microthermic trees; 7, Microthermic trees; 8, Elements without signification; 9, Cupressaceae; 10, Mediterranean xerophytes; 11, Herbs; 12, Steppe elements; 13, Ericaceae.

Fig. 10. Climatostratigraphic correlations for the Lower Pleistocene (1) between the synthetic pollen diagram of Site 380, Semaforo, Vrica, Santa Lucia and Montalbano Ionico, (2) with the NW European climatostratigraphic succession (Zagwijn, 1998), and (3) the reference oxygen isotope curve (some Marine Isotope Stages are plotted) (Shackleton et al., 1990, 1995). Chronostratigraphy (ATNTS 2004: magnetic reversals, stage boundaries) is from Lourens et al. (2004). Magnetostratigraphy performed at Vrica is indicated (Zijderveld et al., 1991). Pollen groups: see Fig. 3.

Popescu, 2005). The most intense regional differentiation in vegetation characterizes the Praetiglian phase whereas the Brunssumian phase is much more homogeneous. The climatostratigraphic correlation of the Lupoaia section with the Brunssumian subdivisions has been slightly modified (Fig. 5) from the earlier interpretation (cf. Popescu, 2006).

A comprehensive climatostratigraphic calibration of the Mediterranean pollen records was established by Suc and Popescu (2005) with a precise correlation to the oxygen isotope stratigraphy of several pollen localities. The results from the calibrated Site 380 offers the opportunity to correlate this record with the southern Italian long pollen records representative of the almost entire Gelasian and Calabrian stages (Croton: Semaforo, Vrica, Santa Lucia; Combourieu–Nebout, 1990, 1993; Joannin et al., 2007; Montalbano Ionico: Joannin et al., 2008) giving an extended Northwestern European climatostratigraphy framework (Fig. 10). As *Pinus* is very abundant in Semaforo, Vrica, Santa Lucia and Montalbano Ionico pollen floras, the corresponding synthetic pollen diagrams have been established without this taxon. This allows a more distinct display of climatic fluctuations (percentages are calculated on the total of pollen grains minus those of *Pinus*). The major NW European climatostratigraphic subdivisions (Tiglian, Eburonian, Waalian, Menapian) can be recognized in the southern Mediterranean sections. The secondary climatic phases of NW Europe are more difficult to recognize, because the large amplitude of the ~41 ky forced “glacial–interglacial” fluctuations somewhat hides them. However, they can be delimited using the relevant calibration of the South Mediterranean sections with the marine isotopic stratigraphy (Fig. 10). As does the uppermost part of Site 380, the pollen diagram of Montalbano Ionico shows the replacement of ~41 ky climate cycles by ~100 ky ones (Joannin et al., 2008).

5. Conclusion

Thanks to the contrasted images of the vegetation that it displays, the high-resolution pollen record of the entire Pliocene

and Lower Pleistocene at DSDP Site 380, biostratigraphically well-calibrated through nannoplankton, allows a continuous, accurate and robust relationship with the reference oxygen isotope curve and the Northwestern European climatostratigraphy. Pollen records at Site 380 and Wólka Ligezowska, obtained in very different paleo-vegetation contexts, both corroborate the reliability of the Northwestern European climatostratigraphy, supporting its value as a direct response to past climatic changes. All the main subdivisions of the Northwestern European climatostratigraphy and most of the secondary ones can be identified in the Northern Hemisphere long pollen records, confirming the strength of this tool.

Acknowledgments

The IODP curatorial staff provided samples of DSDP Site 380. The Polish Geological Survey performed the Wólka Ligezowska borehole. The French Embassy in Turkey assisted one of the authors (D. B.). This paper is a contribution to the French ANR EGEO Project. This work also has been supported by the Programme type EUROCORES-ESF, Grant N° 2/2008 of CNCSIS (National Council of Scientific Research of Romania). Two anonymous reviewers are acknowledged for their constructive comments. We particularly thank M.B. Cita and B. Pillans for their invitation to publish in this special volume and N. Catto, Editor-in-Chief of Quaternary International for his help and consideration. We want to warmly honour Waldo H. Zagwijn's perceptive pioneer contribution.

References

- Baraniecka, M.D., 1991. Profil Róžce na tle podstawowych profili osadów preglacjalnych na południowym Mazowszu. *Przegląd Geologiczny* 5-6, 254–257.
- Comboureu–Nebout, N., 1990. Les cycles glaciaire-interglaciaire en région méditerranéenne de 2,4 à 1,1 Ma: Analyse pollinique de la série de Croton (Italie méridionale). *Paléobiologie Continentale* 17, 35–59.
- Comboureu–Nebout, N., 1993. Vegetation response to upper Pliocene glacial/interglacial cyclicity in the central Mediterranean. *Quaternary Research* 40, 228–236.

- Cour, P., 1974. Nouvelles techniques de détection des flux et de retombées polliniques: étude de la sédimentation des pollens et des spores à la surface du sol. *Pollen et Spores* 16 (1), 103–141.
- Cour, P., Duzer, D., 1978. La signification climatique, édaphique et sédimentologique des rapports entre taxons en analyse pollinique. *Annales des Mines de Belgique* 7/8, 155–164.
- Diniz, F., 1984. Etude palynologique du bassin pliocène de Rio Maior. *Paléobiologie Continentale* 14, 259–267.
- Donders, T.H., Kloosterboer-van Hoeve, M.L., Westerhoff, W., Verreussel, R.M.C.H., Lotter, A.F., 2007. Late Neogene continental stages in NW Europe revisited. *Earth-Science Reviews* 85, 161–186.
- Erdtman, G., 1969. *Handbook of Palynology, Morphology, Taxonomy, Ecology*. Munksgaard, Copenhagen, 486 pp.
- Favre, E., Escarguel, G., Suc, J.-P., Vidal, G., Thévenod, L., 2008. A contribution to deciphering the meaning of AP/NAP with respect to vegetation cover. *Review of Palaeobotany and Palynology* 148, 13–35.
- Gemici, Y., Akyol, E., Akgün, F., Şeçmen, Ö., 1991. Soma kömür havzası fosil makro ve mikroflorası. *Maden Tetkik ve Arama Dergisi* 11, 161–178.
- Gheorghian, M., 1978. Micropaleontological investigations of sediments from sites 379, 380, and 381 of leg 42B. In: Ross, D.A., Neprochnov, Y.P., et al. (Eds.), *Leg 42. Initial Report of the Deep Sea Drilling Project 42, 2. U.S. Government Printing Office*, pp. 783–787.
- Gibbard, P.L., Head, M.J., Walker, M.J.C., The Subcommission on Quaternary Stratigraphy, 2010. Formal ratification of the Quaternary System/Period and the Pleistocene Series/Epoch with a base at 2.58 Ma. *Journal of Quaternary Science* 25 (2), 96–102.
- Gillet, H., Lericolais, G., Réhault, J.-P., 2007. Messinian events in the Black Sea: evidence of a Messinian erosional surface. *Marine Geology* 244, 142–165.
- Hsü, K.J., 1978. Correlation of Black Sea sequences. In: Ross, D.A., Neprochnov, Y.P. (Eds.), *Leg 42. Initial Report of the Deep Sea Drilling Project 42, 2. U.S. Government Printing Office*, pp. 489–497.
- İnci, U., 2002. Depositional evolution of Miocene coal successions in the Soma coalfield, Western Turkey. *International Journal of Coal Geology* 51, 1–29.
- Jakubowicz, B., Makowska, A., Skompski, S., 1994. *Szczegółowa Mapa Geologiczna Polski w skali 1:50,000, arkusz Nowe Miasto nad Pilicą*. CAG PIG, Warsaw.
- Jinxing, L., Yuxi, H., 2000. *Atlas of Structure of Gymnosperms*. Science Press, Beijing, China, 244 pp.
- Joannin, S., Quillévéré, F., Suc, J.-P., Lécuyer, C., Martineau, F., 2007. Early Pleistocene climate changes in the central Mediterranean region as inferred from integrated pollen and planktonic foraminiferal stable isotope analyses. *Quaternary Research* 67, 364–374.
- Joannin, S., Ciaranfi, N., Stefanelli, S., 2008. Vegetation changes during the late early Pleistocene at Montalbano Jonico (Province of Matera, southern Italy) based on pollen analysis. *Palaeogeography, Palaeoclimatology, Palaeoecology* 270, 92–101.
- Kasapgil, B., 1977. A Late-Tertiary conifer-hardwood forest from the vicinity of Güvem village, near Kızılcahamam, Ankara. *Bulletin of the Mineral Research and Exploration Institute of Turkey* 88, 25–33.
- Kemna, H.A., Westerhoff, W.E., 2007. Remarks on the palynology-based chronostratigraphical subdivision of Pliocene terrestrial deposits in NW-Europe. *Quaternary International* 164–165, 184–196.
- Kukla, G., Cilek, V., 1996. Plio-Pleistocene megacycles: record of climate and tectonics. *Palaeogeography, Palaeoclimatology, Palaeoecology* 120, 171–194.
- Leroy, S.A.G., 2007. Progress in palynology of the Gelasian–Calabrian stages in Europe: ten messages. *Revue de micropaleontologie* 50, 293–308.
- Lisiecki, L.E., Raymo, M.E., 2005. A Pliocene–Pleistocene stack of 57 globally distributed benthic $\delta^{18}\text{O}$ records. *Paleoceanography* 20, 1–17.
- Lisiecki, L.E., Raymo, M.E., 2007. Plio-Pleistocene climate evolution: trends and transitions in glacial cycle dynamics. *Quaternary Science Reviews* 26, 56–69.
- Lona, F., 1950. Contributi alla storia delle vegetazione e del clima nella Val Padana. *Analisi pollinica del giacimento villafranchiano di Lefte (Bergamo)*. Atti della Società Italiana di Scienze Naturali 89, 120–178.
- Lourens, L.J., Hilgen, F.J., Laskar, J., Shackleton, N.J., Wilson, D., 2004. The Neogene period. In: Gradstein, F.M., Ogg, J.G., Smith, A.G. (Eds.), *A Geological Time Scale 2004*. Cambridge University Press, Cambridge, pp. 409–440.
- Martini, E., 1971. Standard Tertiary and Quaternary calcareous nannoplankton zonation. In: Farinacci, A. (Ed.), *Proceedings, Second International Conference on Planktonic Microfossils, vol. 2. Edition Tecnoscienza, Rome*, pp. 739–785.
- Maslin, M.A., Ridgwell, A.J., 2005. Mid-Pleistocene revolution and the ‘eccentricity myth’. In: Head, M.J., Gibbard, P.L. (Eds.), *Early–Middle Pleistocene transitions: The Land–Ocean Evidence*. Geological Society, London, Special Publication 247 pp. 19–34.
- Menke, B., 1975. *Vegetationsgeschichte und Florenstratigraphie Nordwest-Deutschlands im Pliozän und Frühquartär. Mit einem Beitrag zur Biostratigraphie des Weichselfrühglazials*. Jahrbuch für Geologie, ser. A 26, 3–151.
- Van Montfrans, H.M., 1971. Palaeomagnetic dating in the North Sea Basin. *Earth and Planetary Science Letters* 11, 226–235.
- Muratov, M.V., Neprochnov, Y.P., Ross, D.A., Trimonis, E.S., 1978. Basic features of the Black Sea Late Cenozoic history based on the results of deep-sea drilling, Leg 42B. In: Ross, D.A., Neprochnov, Y.P., et al. (Eds.), *Leg 42. Initial Report of the Deep Sea Drilling Project 42, 2. U.S. Government Printing Office*, pp. 1141–1148.
- Nix, H., 1982. Environmental determinants of biogeography and evolution in Terra Australis. In: Barker, W.R., Greenlade, P.J.M. (Eds.), *Evolution of the Flora and Fauna of Arid Australia*. Peacock Publ, Frewville, pp. 47–66.
- Okada, H., Bukry, D., 1980. Supplementary Modification and introduction of code numbers to the Low-Latitude coccolith biostratigraphic zonation (Bukry 1973–1975). *Marine Micropaleontology* 5, 321–325.
- Percival, S.F., 1978. Indigenous and reworked coccoliths from the Black Sea. In: Ross, D.A., Neprochnov, Y.P., et al. (Eds.), *Leg 42. Initial Report of the Deep Sea Drilling Project 42, 2. U.S. Government Printing Office*, pp. 773–780.
- Popescu, S.-M., 2001. Repetitive changes in Early Pliocene vegetation revealed by high-resolution pollen analysis: revised cyclostratigraphy of southwestern Romania. *Review of Palaeobotany and Palynology* 120 (3–4), 181–202.
- Popescu, S.-M., 2006. Late Miocene and Early Pliocene vegetation in the southwestern Black Sea region from high-resolution palynology of DSDP site 380A (Leg 42B). *Palaeogeography, Palaeoclimatology, Palaeoecology* 238, 64–77.
- Popescu S.-M., Suc J.-P., Loutre M.-F., 2006. Early Pliocene vegetation changes forced by eccentricity-precession. Example from Southwestern Romania. In Agusti J., Oms O., Meulenkaamp J.E. (Eds.), Late Miocene to Early Pliocene Environment and Climate Change in the Mediterranean Area. *Palaeogeography, Palaeoclimatology, Palaeoecology* 238, 1–4, pp. 340–348.
- Quézel, P., Barbero, M., 1985. Carte de la végétation potentielle de la région méditerranéenne. Feuille n 1: Méditerranée orientale. Editions du Centre National de la Recherche Scientifique, Paris, p. 66.
- Quézel, P., Médail, F., 2003. *Ecologie et biogéographie des forêts du bassin méditerranéen*. Elsevier, Paris, p. 571.
- Raffi, I., Backman, J., Fornaciari, E., Päläke, H., Rio, D., Lourens, L.J., Hilgen, F.J., 2006. A review of calcareous nannofossil astrobiocronology encompassing the past 25 million years. *Quaternary Science Reviews* 25, 3113–3137.
- Reyre, Y., 1968. La sculpture de l'exine des pollens des Gymnospermes et des Chlamydozpermes et son utilisation dans l'identification des pollens fossiles. *Pollen et Spores* 10 (2), 198–220.
- Ross, D.A., 1978. Summary of results of Black Sea drilling. In: Ross, D.A., Neprochnov, Y.P., et al. (Eds.), *Leg 42. Initial Report of the Deep Sea Drilling Project 42, 2. U.S. Government Printing Office*, pp. 1149–1178.
- Schrader, H.-J., 1978. Quaternary through Neogene history of the Black Sea, deduced from the paleoecology of diatoms, silicoflagellates, ebridians, and chrysomonads. In: Ross, D.A., Neprochnov, Y.P., et al. (Eds.), *Leg 42. Initial Report of the Deep Sea Drilling Project 42, 2. U.S. Government Printing Office*, pp. 789–901.
- Shackleton, N.J., Berger, A., Peltier, W.R., 1990. An alternative astronomical calibration of the lower Pleistocene timescale based on ODP Site 677. *Transactions of the Royal Society of Edinburgh: Earth Sciences* 81, 251–261.
- Shackleton, N.J., Hall, M.A., Pate, D., 1995. Pliocene stable isotope stratigraphy of Site 846. In: Pisias, N.G., Mayer, L.A., Janecek, T.R., Palmer-Julson, A., van Andel, T.H. (Eds.), *Leg 138. Proceedings of the Ocean Drilling Program, Scientific Results*, 138, pp. 337–355.
- Stuchlik, L., 1994. Some late Pliocene and early Pleistocene pollen profiles from Poland. In: Boulter, M.C., Fisher, H.C. (Eds.), *Cenozoic Plants and Climates of the Arctic*. NATO Asi Series I, vol. 27. Springer-Verlag, pp. 371–382.
- Suc, J.-P., 1984. Origin and evolution of the Mediterranean vegetation and climate in Europe. *Nature* 307 (5950), 429–432.
- Suc, J.-P., Diniz, F., Leroy, S., Poumot, C., Bertini, A., Dupont, L., Clet, M., Bessais, E., Zheng, Z., Fauquette, S., Ferrier, J., 1995. Zanclean (~ Brunsumian) to early Piacenzian (~ early-middle Reuverian) climate from 4° to 54° north latitude (West Africa, West Europe and West Mediterranean areas). *Mededelingen Rijks Geologische Dienst* 52, 43–56.
- Suc, J.-P., Combourieu-Nebout, N., Seret, G., Klotz, S., Popescu, S.-M., Gautier, F., Clauzon, G., Westgate, J., Sandhu, A.S., 2010. The Crotona series: a synthesis and new data. *Quaternary International* 219, 121–133.
- Suc, J.-P., Popescu, S.-M., 2005. Pollen records and climatic cycles in the North Mediterranean region since 2.7 Ma. In: Head, M.J., Gibbard, P.L. (Eds.), *Early–Middle Pleistocene Transitions: The Land–Ocean Evidence*. Geological Society of London, Special Publication 247 pp. 147–158.
- Suc, J.-P., Zagwijn, W.H., 1983. Plio-Pleistocene correlations between the northwestern Mediterranean region and northwestern Europe according to recent biostratigraphic and paleoclimatic data. *Boreas* 12, 153–166.
- Traverse, A., 1978. Palynological analysis of DSDP Leg 42B (1975) cores from the Black Sea. In: Ross, D.A., Neprochnov, Y.P., et al. (Eds.), *Leg 42. Initial Report of the Deep Sea Drilling Project 42, 2. U.S. Government Printing Office*, pp. 993–1015.
- Țicleanu, N., 1992. Main coal-generating paleophytocoenoses in the Pliocene of Oltenia. *Romanian Journal of Paleontology* 75, 75–80.
- Westerhoff, W., 2009. The Early Pleistocene fluvial system of key reference sites in the Tegelen-Maalbeek area (The Netherlands). SEQS Conference, Orce and Lucena. In: Martínez-Navarro, B., Toro Moyano, I., Palmqvist, P., Agustí, J. (Eds.), *The Quaternary of Southern Spain: a Bridge between Africa and the Alpine Domain*, p. 21.
- Westerhoff, W.E., Kemna, H.A., Boenigk, W., 2008. The confluence area of Rhine, Meuse, and Belgium rivers: Late Pliocene and Early Pleistocene fluvial history of the northern lower Rhine Embayment. *Geologie en Mijnbouw* 87 (1), 107–126.
- Wijmstra, T.A., Groenhart, M.C., 1983. Record of 700,000 years vegetational history in Eastern Macedonia (Greece). *Revista de la Academia Colombiana Ciencias Exactas, Fisicas y Naturales* 15, 87–98.
- Wu, Z.-Y., Raven, P.H. (Eds.), 1999. *Taxodiaceae. Flora of China 4. Cycadaceae through Fagaceae*. Science Press, Beijing, Missouri Botanical Garden Press, St. Louis, 453 pp.
- Zagwijn, W.H., 1960. Aspects of the Pliocene and early Pleistocene vegetation in the Netherlands. *Mededelingen Geologische Stichting Serie C* 3 (5), 78 pp.

- Zagwijn, W.H., 1963. Pollen-analytic investigations in the Tiglian of the Netherlands. *Mededelingen Geologische Stichting New Serie* 16, 49–71.
- Zagwijn, W.H., 1975. Variations in climate as shown by pollen analysis, especially in the Lower Pleistocene of Europe. In: Wright, A.E. & Moseley, F. (Eds.), *Ice Ages: Ancient and Modern*. Geological Journal special issue 6, pp. 137–152.
- Zagwijn, W.H., 1985. An outline of the Quaternary stratigraphy of the Netherlands. *Geologie en Mijnbouw* 64, 17–24.
- Zagwijn, W.H., 1998. Borders and boundaries: a century of stratigraphical research in the Tegelen – Reuver area of Limburg (The Netherlands). In: van Kolfschoten, Th., Gibbard, P.L. (Eds.), *The Dawn of the Quaternary*. Proceedings of the SEQS-EuroMan Symposium 1996, 60. *Mededelingen Nederlands Instituut voor Toegepaste Geowetenschappen TNO*, pp. 19–34.
- Zagwijn, W.H., Suc, J.-P., 1984. Palynostratigraphie du Plio-Pléistocène d'Europe et de Méditerranée nord-occidentales: correlations chronostratigraphiques, histoire de la végétation et du climat. *Paléobiologie Continentale* 14 (2), 475–483.
- Zijderveld, J.D.A., Hilgen, F.J., Langereis, C.G., Verhallen, P.J.J.M., Zachariasse, W.J., 1991. Integrated magnetostratigraphy and biostratigraphy of the upper Pliocene–lower Pleistocene from the Monte Singa and Crotona areas in Calabria, Italy. *Earth and Planetary Science Letters* 107, 697–714.

ABSTRACT

This study concerns a long marine section (DSDP Site 380: Late Miocene to Present) and onshore exposed sections from the Late Miocene and/or Early Pliocene. The main target of this study is to reconstruct vegetation and climate in the North Anatolia and North Aegean region for the last 7 Ma. Two vegetation types were alternately dominant: thermophilous forests and open vegetations including *Artemisia* steppes. During the Late Miocene, most of the tropical and subtropical plants declined because of the climatic deterioration. However, some of them survived during the Late Pliocene, such as those which constituted coastal swamp forests (*Glyptostrobus*, *Engelhardia*, Sapotaceae, *Nyssa*) or composed deciduous mixed forests with mesothermic trees. Simultaneously, herbaceous assemblages became a prevalent vegetation component despite steppe elements (*Artemisia*, *Ephedra*, *Hippophae rhamnoides*) did not significantly develop. At 2.6 Ma, as a response to the onset of Arctic glaciations, subtropical elements rarefied despite some taxa persisted (*Glyptostrobus*, *Engelhardia*, Sapotaceae, *Nyssa*). In parallel, deciduous mixed forest assemblages composed of mesothermic trees (deciduous *Quercus*, *Betula*, *Alnus*, *Liquidambar*, *Fagus*, *Carpinus*, *Tilia*, *Acer*, *Ulmus*, *Zelkova*, *Carya*, *Pterocarya*) almost disappeared too while steppe environments strongly enlarged. Then, *Artemisia* steppic phases developed during longer temporal intervals than mesophilous tree phases all along the glacial-interglacial cycles (first with a period of 41 kyrs, then 100 kyrs). Since 1.8 Ma, herbaceous ecosystems including *Artemisia* steppes still continuously enlarged up today. Such an expansion of *Artemisia* steppes in the Ponto-Euxinian region was observed at the earliest Pliocene but their earliest settlement in Anatolia seems to have occurred in the Early Miocene. The development of the *Artemisia* steppes in Anatolia might result from the uplift of the Tibetan Plateau. Relictuous plants such as *Carpinus orientalis*, *Pterocarya*, *Liquidambar orientalis*, *Zelkova* persisted up today. This story can be explained by some influence of the Asian monsoon which reinforced as a result from the uplifted Tibetan Plateau.

TITRE

Végétation et climat des régions nord-anatolienne et nord-égéenne depuis 7 Ma d'après l'analyse pollinique

RESUME

Cette étude concerne un long enregistrement sédimentaire marin (Site DSDP 380 : Miocène supérieur à Présent) et des affleurements à terre de dépôts marins ou lacustres du Miocène supérieur et/ou du Pliocène inférieur. L'objectif principal de cette recherche est de reconstruire la végétation et le climat des régions nord-anatolienne et nord-égéenne des 7 derniers Ma. Deux types de végétation y furent alternativement : les forêts de plantes thermophiles et les formations ouvertes incluant les steppes à *Artemisia*. A la fin du Miocène, la plupart des éléments mégathermes (tropicaux) et méga-mésothermes (subtropicaux) avaient régressé en raison des détériorations climatiques. Cependant, certains d'entre eux ont survécu pendant le Pliocène supérieur, notamment ceux qui constituaient des forêts littorales marécageuses (*Glyptostrobus*, *Engelhardia*, Sapotaceae, *Nyssa*) ou participaient à des forêts mixtes avec des arbres décidus mésothermes. Pendant ce temps, les formations ouvertes à herbes sont devenues prédominantes dans la végétation sans que les éléments steppiques (*Artemisia*, *Ephedra*, *Hippophae rhamnoides*) soient très abondants. A 2,6 Ma, sous l'effet des premières glaciations arctiques, les éléments méga-mésothermes se sont très raréfiés malgré la persistance de quelques reliques (Taxodiaceae : probablement *Glyptostrobus*, *Engelhardia*, Sapotaceae, *Nyssa*). Simultanément, les forêts mixtes à éléments mésothermes (*Quercus* décidus, *Betula*, *Alnus*, *Liquidambar*, *Fagus*, *Carpinus*, *Tilia*, *Acer*, *Ulmus*, *Zelkova*, *Carya*, *Pterocarya*, etc) ont aussi quasiment disparu tandis que les environnements steppiques se développaient fortement. Désormais, tout au long des cycles glaciaire-interglaciaire (d'abord de 41 ka de périodicité puis de 100 ka), les steppes à *Artemisia* occuperont plus d'espace temporel que les phases arborées. Depuis 1,8 Ma, les environnements à herbes et les steppes à *Artemisia* n'ont cessé de s'étendre jusqu'à aujourd'hui. Cette expansion des steppes à *Artemisia* dans la région du Pont-Euxin a été observée au tout début du Pliocène mais leur premier enregistrement en Anatolie date du Miocène inférieur. Le développement de la steppe à *Artemisia* en Anatolie pourrait résulter du soulèvement du Plateau tibétain. Le maintien dans cette région de plantes thermophiles reliques en situation de refuges (*Carpinus orientalis*, *Pterocarya*, *Liquidambar orientalis*, *Zelkova*) peut être expliqué par l'influence grandissante de la mousson asiatique dont le renforcement aurait aussi résulté du soulèvement du Plateau tibétain.

DISCIPLINE Palynologie, Géologie

MOTS-CLES Anatolie, Nord Egée, Néogène supérieur, Quaternaire, Palynologie, Végétation, Climat

INTITULE ET ADRESSE DES LABORATOIRES :

Laboratoire PaléoEnvironnements et PaléobioSphère, UMR CNRS 5125, Université Claude Bernard – Lyon 1, 27-43 boulevard du 11 Novembre, 69622 Villeurbanne, France

School of Mines and Eurasia Institute of Earth Sciences, Istanbul Technical University, Maslak, 34469 Istanbul, Turkey

Appendix D

Neogene flora, vegetation and climate dynamics in southeastern Europe and the northeastern Mediterranean

G. JIMENEZ-MORENO^{1,2,3}, S.-M. POPESCU¹, D. IVANOV⁴ & J.-P. SUC¹

¹*Laboratoire PaléoEnvironnement et PaléobioSphère (UMR CNRS 5125), Université Claude Bernard - Lyon 1, 27-43 boulevard du 11 Novembre, 69622 Villeurbanne, France (e-mail: gonzaloj@ugr.es; popescu@univ-lyon1.fr; jean-pierre.suc@univ-lyon1.fr)*

²*Departamento de Estratigrafía y Paleontología, Universidad de Granada, Avda. Fuente Nueva S/N 18002, Granada, Spain*

³*Center for Environmental Sciences & Education, Box 5694, Northern Arizona University, Flagstaff, AZ 86011USA. (present address) (e-mail: Gonzalo.Jimenez-Moreno@NAU.EDU)*

⁴*Institute of Botany, Bulgarian Academy of Sciences, Acad. G. Bonchev Str., 23, 1113 Sofia, Bulgaria (e-mail: dimiter@bio.bas.bg)*

Abstract: Pollen analysis of Miocene and Pliocene sediments from southeastern Europe and the northeastern Mediterranean is represented in pollen synthetic diagrams based on ecological criteria in order to clearly visualize changes in the composition and structure of the vegetation through time. New pollen data, together with abundant existing palynological information from this area, show a progressive reduction in plant diversity caused by a decrease in the most thermophilous and high-water requirement plants and, on the contrary, an increase in warm-temperate (mesothermic) and seasonal-adapted taxa during the Middle–Late Miocene and Pliocene. At the same time, an increase in high-elevation trees and herbs has been recorded, with a strong augmentation in *Artemisia*, first in the eastern Mediterranean and later on in the western Mediterranean area. This has been interpreted as a response of the vegetation to global and regional processes, including climate cooling related to the development of the East Antarctic Ice Sheet (EAIS), uplift of regional mountains during Alpine orogenesis and progressive movement of Eurasia towards northern latitudes as a result of the northwards collision of Africa.

Pollen analyses dealing with Miocene–Pliocene sediments from the Paratethys are rare. Studies have focused on the Miocene and Pliocene palynology of the Central Paratethys (Petrescu *et al.* 1989a, b; Planderová 1990; Nagy 1991, 1992, 1999; Petrescu & Malan 1992) and Turkey (Benda 1971; Benda *et al.* 1975; Akgün & Akyol 1999), but the lack of any quantitative information render these analyses limited. However, palynological data, with reliable botanical identification, are available for the Miocene (Ivanov 1995; Ivanov & Koleva-Rekalova 1999; Palamarev & Ivanov 2001; Ivanov *et al.* 2002; Jiménez-Moreno *et al.* 2005) and the Pliocene (Drivaliari 1993; Drivaliari *et al.* 1999; Popescu 2001, 2002, 2006; Popescu *et al.* 2006a, b) of the same region. In these studies, pollen was not used for biostratigraphy but for climatic information, as independent biostratigraphic dating was available (see below). Pollen counts and a statistical treatment of the data were made to obtain reliable information about floral diversity, organization of the vegetation and to better visualize vegetation and climate change.

The geographical position of the studied area, between Africa and Eurasia and between a Mediterranean and temperate climate, makes this region of great interest for palaeobotanic studies. Today, the southeastern part of the area is mainly occupied by steppe vegetation rich in *Artemisia* (i.e. the central Anatolian steppes), that is the main refuge area of thermophilous plants (mostly along the Turkish coastlines: Zohary 1973; Quézel & Médail 2003). Alpine tectonics were active during the Neogene, producing uplift of the Carpathians, Dinarides, Balkan, Rhodope and Taurides mountains. Then, important palaeogeographical changes occurred (see below; Rögl 1998; Meulenkamp & Sissingh 2003; Popov *et al.* 2004) that may have contributed to the pattern of vegetation distribution seen today.

In this paper, we present a synthesis of palynological data, interpreted vegetation and climate dynamics based on Miocene and Pliocene deposits from Eastern Europe. New sections of Middle and Late Miocene age from this area have been analysed, adding new information to the already

published data. Changes in vegetation have been observed from the Langhian to the early Pliocene (16.3–3 Ma). These are mainly related to global climatic changes, in temperature and precipitation, that are linked to atmospheric and palaeogeographic changes that were of significant importance during the Neogene.

Regional setting

The studied area comprises Neogene basins formed within the Central–Eastern Paratethys Sea. They were generated during the Neogene, like the rest of the basins belonging to the Paratethys, as a product of the collision of the African plate and Eurasia. These basins are delimited by the Carpathians, Balkan, Dinarides and Taurides, occupying parts of Hungary, Romania, Bulgaria, Serbia,

Greece and Turkey (Fig. 1; Kojumdjieva & Popov 1989; Rögl 1998; Meulenkamp & Sissingh 2003; Goncharova *et al.* 2004; Ilyina *et al.* 2004; Paramonova *et al.* 2004; Khondkarian *et al.* 2004a, b). During the Neogene, the Paratethys displayed a long-term trend of decreasing marine influence and a correlative reduction in size with regard to the marine depositional domains. Marine deposition lasted throughout the Early and Middle Miocene up to approximately 12 Ma, when uplift caused the sea to retreat from the Pannonian basin complex where a brackish lake formed instead (Rögl 1998). However, during the Early and Middle Miocene, connection between the Mediterranean Sea and the Paratethys existed that allowed for a free marine faunal exchange (Harzhauser *et al.* 2003). The first impairment of marine connections is evident in the Late Badenian (Early Serravallian)

Fig. 1. Geographic map of the studied area and location of the sites. 1 Nireas-1; 2 Valea Morilor; 3 Ruzhintsi; 4 Catakbagyaka; 5 Hinova; Husnicioara and Valea Visenilor; 6 Lupoia; 7 Ticleni-1; 8 Ravno Pole and Lozenec; 9 Sandanski; 10 Lion of Amphipoli; 11 Nestos-2; 12 Site 380 A; 13 Aghios Vlassios; 14 Avadan; 15 Lataquie; 16 Drenovets C-1; 17 Deleina C-12; 18 Makrilia; 19 Tengelic-2.

when dysaerobic bottom conditions and a stratified water column characterized the Paratethyan realm (Kovac *et al.* 2004). With the onset of the Sarmatian, marine connection to the Mediterranean almost completely ceased, and was reflected by the development of a highly endemic molluscan fauna (Harzhauser & Piller 2004). Finally, at the Sarmatian/Pannonian boundary (Serravallian/Tortonian boundary), the Central Paratethys became entirely restricted and the brackish Lake Pannon was established. Sporadic brief connections occurred during the Late Miocene and Pliocene between the Eastern Paratethys (Dacic and Euxinian basins) and the Mediterranean Sea as documented by nannoplankton influxes (Mărușeanu & Papaianopol 1998; Semenenko & Olejnik 1995). One of these short connections also concerned the southeastern Pannonian Basin during the so-called Portaferrian regional Stage (Pontian). Some of these connections occurred just before and just after the Messinian salinity crisis (Clauzon *et al.* 2005; Snel *et al.* 2006), resulting in the same responses (i.e. an intense erosion, then the construction of Gilbert-type fan deltas) to the Messinian desiccation and Zanclean flooding as in the Mediterranean Basin itself (Clauzon *et al.* 2005). However, during the late Neogene, most of the Paratethyan basins were disconnected and evolved as isolated lakes, some of them being temporarily connected with the Mediterranean Sea (Mărușeanu & Papaianopol 1995).

The independent evolution of the different sub-domains of the Paratethys led to the construction of several regional stratigraphies, constituted by stages based on diverse groups of organisms, mainly bivalves and ostracods, and benthic and planktonic foraminifera etc. (Marinescu 1978; Papaianopol & Motas 1978; Papaianopol & Marinescu 1995; Rögl 1998; Fig. 2). Reliable correlations are established between the Eastern Paratethys regional stratigraphy and the Mediterranean standard stratigraphy using nannoplankton (Papaianopol & Mărușeanu 1993; Mărușeanu & Papaianopol 1995, 1998; Drivaliari *et al.* 1999; Clauzon *et al.* 2005; Snel *et al.* 2006).

Chronological background

A total of 19 sections and a total of 680 samples have been studied for pollen. Of those 19 sections, 12 (or a part) belong to the Miocene and 14 (or a part) to the Pliocene (Fig. 2). As far as possible, an independent age control has been obtained; it is indicated in Table 1 with the authors of the pollen analyses. The timescale of Gradstein *et al.* (2004) has been used.

Methods

Identification was performed comparing the Neogene pollen grains with those of the living relative plants

using databanks of modern pollen grains and modern and past pollen grain photographs. Based on the results of the pollen spectra, standard synthetic diagrams (Suc 1984) with *Pinus* and Pinaceae have been constructed. In these pollen diagrams, taxa have been arranged into 12 different groups based on ecological criteria in order to obtain some visualization of the vegetation (see below) and more easily compare with reference oxygen isotopic curves. This method has been proven to be a very efficient tool for high-resolution climatic studies characterizing warm–cold alternations related to Milankovitch cycles for both the Miocene (Jiménez-Moreno *et al.* 2005) and the Pliocene (Popescu 2001, 2006; Popescu *et al.* 2006a, b).

Pollen data will be available, after publication, on the web from the ‘Cenozoic pollen and climatic values’ database (CPC) (<http://medias.obs-mip.fr/cpc>).

Results

Plant diversity and vegetation

Even if some parts of the studied region are characterized today by a very diverse flora and are main refuge areas of thermophilous plants (i.e. the Ponto-Euxinian area) (Quézel & Médail 2003), a richer and more diverse flora has been identified for the Mio-Pliocene that consisted of elements found presently in different geographic areas:

- (1) Tropical and subtropical Africa, America and Asia (*Avicennia*, *Bombax*, Caesalpinaceae, *Engelhardia*, *Platycarya*, Taxodiaceae, Hamamelidaceae, *Myrica*, Sapotaceae, etc.).
- (2) Warm-temperate latitudes of the Northern Hemisphere (*Acer*, *Alnus*, *Betula*, Cupressaceae, *Fagus*, *Populus*, deciduous *Quercus*, *Salix*, etc.).
- (3) Mediterranean region (*Olea*, *Phillyrea*, *Ceratonia*, evergreen *Quercus*, etc.).

All of these taxa grew in the Eastern European area during the Miocene.

We use the Chinese flora as a present-day comparison for the southeastern Europe and Middle East flora during the Neogene as it is the closest living example of this floral inventory (Suc 1984). Flora of the broad-leaved evergreen forest was represented by 45 typical tropical and subtropical taxa (i.e. megathermic and mega-mesothermic elements, respectively) in the studied region during the middle Miocene’s warmest phase; only 21 of them persisted until the early Pliocene and have presently disappeared from the area. Flora of the evergreen and deciduous mixed forest was represented by 21 subtropical and warm-temperate taxa (i.e. mega-mesothermic and mesothermic

Fig. 2. Miocene and Pliocene chronostratigraphy and temporal situation of the studied sites. Correlations between standard stages and Paratethys stages by Harzhauser & Piller (in press) after data of Steininger (1999), Sprovieri (1992), Sprovieri *et al.* (2002), Fornaciari & Rio (1996) and Fornaciari *et al.* (1996). Oxygen isotope curve after Zachos *et al.* (2001); all stages recalibrated according to Gradstein & Ogg (2004), Gradstein *et al.* (2004) and Lourens *et al.* (2004).

elements, respectively) in the studied region during the Middle Miocene's warmest phase; they persisted here during the Early Pliocene and 17 among them are still living in the area.

The vegetation was characterized by a complex mosaic due to its dependency on several factors (water availability, characteristics of the soils, orientation of relief slopes, etc.) which superimposed its latitudinal–altitudinal organization. The most important factor, similar to

present day, would be altitude, controlling both temperature and precipitation. Therefore, the vegetation would be organized in altitude belts, which have been compared with those found today in subtropical to temperate southeastern China, the most reliable model. The following have been distinguished:

(1) a coastal marine environment characterized by the presence of an impoverished mangrove composed of *Avicennia* which is mainly

Table 1. Age control of the 19 considered pollen localities indicating the authors of the pollen analyses

#	Section	Location	Datation	Age	Pollen analysis by
Nireas 1	Greece	* Drivaliari 1993	Aquitanian-Burdigalian	Drivaliari, A.
Valea Morilor	Romania	*) Papatanopol et al. 1995	Zanclean	Jiménez-Moreno, G.
Ruzhintsi	Bulgaria) Kojumdjieva 1976; Palmarev & Ivanov 2001	Sarmatian	Jiménez-Moreno, G.
Catakbagyaka	Turkey	i Sickenberg et al. 1975; Heissig 1976	Langhian	Jiménez-Moreno, G.
Hinova	Romania) Marinescu 1978	Early Zanclean	Popescu, S.-M.
			* Clauzon et al. 2005	Bosphorlian	
5	Husnicioara	Romania	* § ♣ Popescu et al. 2006a	Early Zanclean	Popescu, S.-M.
			¶ Ticleanu & Diaconita 1997		
5	Valea Visenilor	Romania	§ ♣ Popescu et al. 2006b	Early Zanclean	Popescu, S.-M.
			¶ Ticleanu & Diaconita 1997		
6	Lupoaia	Romania	♣ Popescu et al. 2006b	Zanclean	Popescu, S.-M.
			i Radulescu et al. 1997; Apostol & Enache 1979		
			§ Radan & Radan 1998; Van Vugt 2001		
			¶ Ticleanu & Diaconita 1997		
			♣ Popescu S.-M. 2002; Popescu et al. 2006b		
7	Ticleni 1	Romania	*) ♣ Drivaliari et al. 1999	Zanclean	Drivaliari, A. 1993
8	Ravno Pole	Bulgaria) Drivaliari 1993	Pontian-Dacian	Drivaliari, A.
Lozenc	Bulgaria	i Gromolard & Guerin 1980; Thomas et al. 1986	Dacian	Drivaliari, A.
Sandanski	Bulgaria	i Kojumdjieva et al. 1982; Spassov N. (personal information)	Maeotian	Ivanov, D.
Lion of Amphipoli	Bulgaria	* Melinte, M.C. (personal information)	Early Zanclean	Suc, J.-P.
Nestos 2	Greece	* + Drivaliari 1993	Messinian-Zanclean	Drivaliari, A.
Site 380 A	Black Sea	¶ Hsü 1978; Hsü & Giovanoli 1979; Letouzey et al. 1978	Late Miocene-Early Pliocene	Popescu, S.-M.
			♣ Popescu 2006		
Aghios Vlassios	Greece	+ Spaak 1983; Drivaliari 1993	Early Pliocene	Drivaliari, A. 1993
Avadam	Turkey	¶ Robertson A.H.F. (personal information)	Early Zanclean	Suc, J.-P.
Latakie	Syria	¶ Rubino J.-L. (personal information)	Early Zanclean	Suc J.-P.
Drenovets C-1	Bulgaria	+) ^ Kojumdjieva et al. 1989	Sarmatian to Pontian	Ivanov, D.
Deleina C-12	Bulgaria	+) ^ Kojumdjieva & Popov 1989	Badenian to Pannonian	Ivanov, D.
Makrilia	Greece	* Sachse et al. 1999	Tortonian	Sachse, M.
Tengelic-2	Hungary	* +) Nagymarosi 1982; Bohn-Havas 1982; Korecz-Laky 1982	Burdigalian Ottnangian to Sarmatian	Jiménez-Moreno, G.

+ Foraminifera ^ Ostracods i Mammals ♣ Climatostratigraphy * Nanoplankton) Bivalves § Palaeomagnetism ¶ Lithostratigraphy

accompanied by halophytes (Amaranthaceae–Chenopodiaceae, *Armeria*, etc.);

(2) a broad-leaved evergreen forest, from sea level to around 700 m altitude characterized by *Taxodium* or *Glyptostrobus*, *Myrica*, *Rhus*, Theaceae, Cyrillaceae–Clethraceae, *Bombax*, Euphorbiaceae, *Distylium*, *Castanopsis*, Sapotaceae, Rutaceae, *Mussaenda*, *Ilex*, *Hedera*, *Ligustrum*, *Jasminum*, Hamamelidaceae, *Engelhardia*, *Rhoiptelea*, etc.;

(3) an evergreen and deciduous mixed forest, above 700 m altitude; characterized by deciduous *Quercus*, *Engelhardia*, *Platycarya*, *Carya*, *Pterocarya*, *Fagus*, *Liquidambar*, *Parrotia*, *Carpinus*, *Celtis*, *Acer*, etc. Within this vegetation belt, riparian vegetation has been identified, composed of *Salix*, *Alnus*, *Carya*, *Carpinus*, *Zelkova*, *Ulmus*, *Liquidambar*, etc. The shrub level was dominated by Ericaceae, *Ilex*, Caprifoliaceae, etc.;

(4) above 1000 m, a mid-altitude deciduous and coniferous mixed forest with *Betula*, *Fagus*, *Cathaya*, *Cedrus*, *Tsuga*.

(5) above 1800 m altitude, a coniferous forest with *Abies* and *Picea*.

Vegetation dynamics

The following description of the Miocene and Pliocene vegetation dynamics in the southern Forecarpathian Basin and Greece–Turkey is a brief summary of the pollen analysis of Drivaliari (1993), Ivanov (1995), Drivaliari *et al.* (1999), Popescu (2001, 2006), Popescu *et al.* (2006a, b), Jiménez-Moreno *et al.* (2005) and Jiménez-Moreno (2005).

Burdigalian–Langhian (20.4–13.6 Ma). The regular occurrence and abundance of thermophilous species typical of the lowest altitudinal belts described above and the relative scarcity of altitudinal elements (Fig. 3) are characteristic for vegetation of this time. The coastal marine environment was then occupied by an impoverished *Avicennia* mangrove and several halophytes (Nagy & Kóky 1991; Nagy 1999; Plaziat *et al.* 2001; Jiménez-Moreno 2005). In the hinterland, lowlands were populated by a broad-leaved evergreen forest, characterized by *Alchornea*, Passifloraceae, *Pandanus*, *Rhus*, Theaceae, Cyrillaceae–Clethraceae, *Bombax*, Rubiaceae, Chloranthaceae, *Reevesia*, Euphorbiaceae, *Distylium*, *Castanopsis*, Sapotaceae, Rutaceae, *Mussaenda*, *Ilex*, *Hedera*, *Itea*, *Alangium*, cf. Mastixiaceae, *Ligustrum*, *Jasminum*, Hamamelidaceae, *Engelhardia*, *Rhoiptelea*, Schizaeaceae, Gleicheniaceae, etc. Within this vegetation belt, swamp forests were also well developed during this time period. Its components, such as *Taxodium* or *Glyptostrobus*, *Nyssa*, *Myrica*, *Planera*, show

comparatively high values in the pollen spectra. Probably the low elevation palaeogeography and very humid conditions at that time in the studied area favoured the wide distribution of swamp forests and of ecologically related riparian forests with *Platanus*, *Liquidambar*, *Zelkova*, *Carya*, *Pterocarya* and *Salix*.

An evergreen and deciduous mixed forest mainly composed of mesothermic elements such as *Quercus*, *Carya*, *Pterocarya*, *Fagus*, Ericaceae, *Ilex*, Caprifoliaceae, *Liquidambar*, *Parrotia*, *Carpinus*, *Celtis*, *Acer*, but also *Engelhardia*, *Platycarya*, etc., characterized areas of higher altitude. Within this vegetation belt, riparian vegetation has been identified, composed of *Salix*, *Alnus*, *Carya*, *Carpinus*, *Zelkova*, *Ulmus*, etc.

It should also be mentioned that conifer pollen, mainly *Pinus* and indeterminate Pinaceae, can be particularly abundant, presumably because of the capacity of saccate pollen for long-distance transport (Heusser 1988; Suc & Drivaliari 1991; Cambon *et al.* 1997; Beaudouin 2003): during the Badenian, the basin developed its largest extension so that the studied sections had the maximum distance from the coastline (Fig. 3). Mid- and high-altitude elements (*Tsuga*, *Cedrus*, *Abies* and *Picea*) and *Cathaya* seem not to vary significantly in sections of this age (Fig. 3).

Serravallian–Tortonian–Messinian (13.6–5.3 Ma). During this time-interval, important changes in the vegetation are observed: *Avicennia*, which populated the coastal areas in previous times, is not found commonly and several megathermic elements (*Buxus bahamensis* group, *Alchornea*, *Bombax*, Iacacinaceae, *Croton*, Melastomataceae, etc.), typical from the broad-leaved evergreen forest, became rare and most of them disappeared (Fig. 3). The evergreen–deciduous mixed forest suffered a great transformation due to the loss and decrease in the abundance of several megamesothermic evergreen plants. This kind of vegetation was progressively enriched by deciduous mesothermic plants, such as deciduous *Quercus*, and *Fagus*, *Alnus*, *Acer*, *Eucommia*, *Betula*, *Alnus*, *Carpinus*, *Ulmus*, *Zelkova*, *Tilia*, etc. Thus, the vegetation shows a tendency towards increasing proportions of mesothermic deciduous elements coming from higher altitudes.

Even if the thermophilous elements decreased during this period, the swamp forest continued to be well developed. At the same time, the vegetation from mid- (*Cathaya*, *Tsuga* and *Cedrus*) and high-altitude (*Picea* and *Abies*) belts clearly strengthened. For instance, *Tsuga* (mid-altitude indicator) is absent in the Badenian (Langhian and Early Serravallian) or very rare, it is still rare at the base of the Volhynian (approx. 12.7 Ma), but reaches

Fig. 3. Synthetic pollen diagrams of the sections spanning the Miocene until 6 Ma. Taxa have been grouped according to their ecological significance as follows: 1 Megathermic (= tropical) elements (*Avicennia*, *Amanoa*, *Alchornea*, *Fothergilla*, *Exbucklandia*, Euphorbiaceae, Sapindaceae, Loranthaceae, Arecaceae, Acanthaceae, *Canthium* type, Passifloraceae, etc.). 2 Mega-mesothermic (= subtropical) elements (Taxodiaceae, *Engelhardia*, *Platycarya*, *Myrica*, Sapotaceae, *Microtropis fallax*, *Symplocos*, *Rhoiptelea*, *Distylium* cf. *sinensis*, *Embolanthera*, *Hamamelis*, Cyrillaceae–Clethraceae, Araliaceae, *Nyssa*, *Liriodendron*, etc.). 3 *Cathaya*, an altitudinal conifer living today in Southern China. 4 Mesothermic (= warm-temperate) elements (deciduous *Quercus*, *Carya*, *Pterocarya*, *Carpinus*, *Juglans*, *Celtis*, *Zekkova*, *Ulmus*, *Tilia*, *Acer*, *Parrotia* cf. *persica*, *Liquidambar*, *Alnus*, *Salix*, *Populus*, *Fraxinus*, *Buxus sempervirens* type, *Betula*, *Fagus*, *Ostrya*, *Parthenocissus* cf. *henryana*, *Hedera*, *Lonicera*, *Elaeagnus*, *Ilex*, *Tilia*, etc.). 5 *Pinus* and poorly preserved Pinaceae pollen grains. 6 Meso-microthermic (= mid-altitude) trees (*Tsuga*, *Cedrus*). 7 Microthermic (= high-altitude) trees (*Abies*, *Picea*). 8 Non-significant pollen grains (undetermined ones, poorly preserved pollen grains, some cosmopolitan or widely distributed elements such as Rosaceae and Ranunculaceae). 9 Cupressaceae. 10 Mediterranean xerophytes (*Quercus ilex* type, *Carpinus* cf. *orientalis*, *Olea*, *Phillyrea*, *Ligustrum*, *Pistacia*, *Ziziphus*, *Cistus*, etc.). 11 Herbs (Poaceae, *Erodium*, *Geranium*, *Convolvulus*, Asteraceae Asteroideae, Asteraceae Cichorioideae, Lamiaceae, *Plantago*, *Euphorbia*, Brassicaceae, Apiaceae, *Knautia*, *Helianthemum*, *Rumex*, *Polygonum*, *Asphodelus*, Campanulaceae, Ericaceae, Amaranthaceae–Chenopodiaceae, Caryophyllaceae, Plumbaginaceae, Cyperaceae, *Potamogeton*, *Sparganium*, *Typha*, Nymphaeaceae, etc.) including some subdesertic elements (*Lygeum*, *Neurada*, *Nitraria*, *Calligonum*). 12 Steppe elements (*Artemisia*, *Ephedra*).

up to 10% in the middle and upper part of the Volhynian (Fig. 3). This palaeofloristic change occurs slowly and gradually without major fluctuations. A similar vegetation change is observed during the same time-interval in other areas of Europe (e.g. Spain, southern France, Switzerland and Austria: Bessedik 1985; Jiménez-Moreno 2005).

The herbs (mainly Poaceae, Amaranthaceae–Chenopodiaceae, *Artemisia*, Caryophyllaceae, Polygalaceae, Lamiaceae, Asteraceae Asteroideae and Asteraceae Cichorioideae) also became more abundant (Fig. 3). This may be due to a somewhat drier climate during that time as is also indicated by macrofloras of the same area (Palamarev 1991; Palamarev &

Ivanov 2004) and confirmed by sedimentological data (Koleva-Rekalova 1994; Ivanov & Koleva-Rekalova 1999): in Bessarabian to Chersonian sediments (12–9.1 Ma) of northeast Bulgaria, aragonite sediments occur which are assumed to have been formed under a seasonally dry climate. This trend continued during the Late Miocene. Presumably, open landscapes covered by more xerophytic herbaceous communities existed during that time.

Pliocene (5.3–c. 3.2 Ma). The vegetation was then characterized by a mosaic of different plant associations inherited from the Miocene. The same vegetation dynamics marked by disappearance of thermophilous plants and increase in mesothermic and micro-mesothermic plants continued. Some of the coastal areas of this region were still inhabited by *Avicennia* mangrove (*Avicennia* pollen at Site 380A at 781.63 m) and several megathermic elements typical from the broad-leaved evergreen forest occupying the lowlands, such as *Amanoa*, *Pachysandra*, *Entada*, Meliaceae, Mimosaceae, Sapindaceae, Tiliaceae, Euphorbiaceae, Acanthaceae and *Fothergilla*, are sporadically present. They disappeared during the early Pliocene (between 4–3.5 Ma) (Popescu 2001). The mega-mesothermic plants, belonging to these plant associations, such as *Engelhardtia*, *Microtropis*, *Distylium*, *Parthenocissus*, Sapotaceae, Arecaceae, etc., are still abundant and persisted through the Pliocene (Fig. 4). Swampy (mainly *Taxodium* or *Glyptostrobus*, *Nyssa*, *Myrica*) and marshy (Cyperaceae, Poaceae, Cyrtaceae–Clethraceae, *Myrica*) elements, populating deltaic areas, were very abundant. Trees from the family Taxodiaceae did not disappear from this area until the middle Pleistocene (Mamatsashvili 1975).

The mixed deciduous forest (mainly made up of conifers like *Pinus*, and several deciduous trees such as *Quercus*, *Acer*, *Carpinus*, *Parrotia*, *Carya*, *Pterocarya*, *Liquidambar*, *Platanus*, *Tilia*, *Ulmus*, *Zelkova*, etc.), situated at higher altitude, as well as the trees belonging to the highest altitudinal belts, become more abundant during this period (*Cathaya*, *Cedrus*, *Tsuga*, *Picea* and *Abies*) (increasing percentages of these elements are compared on Fig 3 and 4).

Another important fact that makes a difference between the Pliocene and the Miocene is the strong development of the steppe with *Artemisia* in the Ponto-Euxinian region since the early Pliocene (Site 380A, Fig. 4).

Climatic evolution: regional vs. global climatic change

The high presence of mega- and mega-mesothermic elements during the Early and early Mid-Miocene suggests the existence of a warm, subtropical

climate and a tendency towards slightly cooler conditions in the late Mid-Miocene. Climate was also quite humid, to support the development of such a large association of thermic elements (of present-day 'Asiatic' affiliation and climate) which require very humid conditions all year (Wang 1961). The major change is the impoverishment in plant diversity produced by the disappearance of the most thermophilous plants and the consequent enrichment in mesothermic plants (mainly deciduous *Quercus*, *Alnus*, etc.) and high-elevation conifers, from the Serravallian to the Pliocene.

The floral assemblages during the Early and early Mid-Miocene clearly reflect the Miocene Climatic Optimum (MCO: Zachos *et al.* 2001; Shevenell *et al.* 2004) well-recorded at Tengelic-2 (Jiménez-Moreno *et al.* 2005). The major change registered in plant diversity is related to a gradual decrease in temperature and precipitation after the MCO (Ivanov *et al.* 2002; Jiménez-Moreno *et al.* 2005). This fact is well documented on a worldwide scale and has been correlated with the general decrease in temperature observed by several authors as a gradual increase in the isotopic $\delta^{18}\text{O}$ values of foraminifera from deep-sea sediments (DSDP Sites 608: Miller *et al.* (1991) and 588: Zachos *et al.* (2001)) during this timespan and related to an increase in the size of the EAIS (East Antarctic Ice Sheet) (Zachos *et al.* 2001) (Fig. 2). The isotopic values also indicate that this cooling continued during the Late Miocene and Pliocene (Zachos *et al.* 2001) (Fig. 2).

High-elevation conifers seem not to vary along the sections of early and early Mid-Miocene; however, these elements are abundant in the samples and indicate that the surrounding mountains were already significantly uplifted. Mid- (including *Cathaya*) and high-elevation conifers clearly increase during the late Mid-Miocene and Late Miocene. This can be observed in the boreholes Deleina C-12 and Drenovets C-1 (Fig. 3).

In addition, an augmentation in herbs, mainly *Artemisia*, Amaranthaceae–Chenopodiaceae, Poaceae, Asteraceae, etc., during the Late Miocene and Pliocene, indicates more open vegetation, and drier conditions. Supporting this interpretation is the substitution of thermophilous elements with high humidity requirements all year (Asiatic-like vegetation) by mesothermic (mainly deciduous) elements which can survive under seasonal climate with respect to the precipitation (Popescu 2001; Ivanov *et al.* 2002; Jiménez-Moreno *et al.* 2005).

The noticeable increase in mesothermic plants and high-elevation conifers can be interpreted as a result of climate cooling, or by uplift of surrounding mountains (Kuhlemann & Kempf 2002). In both situations, altitudinal elements would increase.

Fig. 4. Synthetic pollen diagrams of the studied sections spanning the Late Miocene (from 6 Ma) and Pliocene. For legend of plant groups, see Figure 3.

It is quite difficult to separate one process from another (global climatic forcing vs. the regional one), due to the tectonic situation of the studied area and the fact that they may have interfered. However, the vanishing of several thermophilous plants, which lived at low elevations and thus were not affected by the regional uplift, and the climate reconstructions using mainly taxa growing at low to middle–low altitude confirm a decrease in mean annual temperatures (Ivanov *et al.* 2002; Jiménez-Moreno *et al.* 2005; Mosbrugger *et al.* 2005). Then, it is clear that even if the uplift of the surrounding mountains may have influenced the regional climate, the evolution of the vegetation during both the Miocene and Pliocene was very dependent on the global climatic signal as shown in previous studies (Popescu 2001, 2002; Ivanov *et al.* 2002; Jiménez-Moreno *et al.* 2005). Hence, according also to the rapid nature of the recorded change in vegetation, we consider that global cooling was the most efficient forcing.

The origin of the steppe with *Artemisia*

Open herbaceous formations in the southern Mediterranean area are known since the Burdigalian (Suc *et al.* 1995a, b; Bachiri Taoufiq *et al.* 2001; Jiménez-Moreno 2005; Jiménez-Moreno & Suc in press). They were already well-developed during the Zanclean in other regions of the Mediterranean area (Suc *et al.* 1999) but were relatively poor in *Artemisia*. It is at the end of the Pliocene, as the climate got cooler and glacial–interglacial cycles appeared in the Northern Hemisphere, when the steppes with *Artemisia* became of significant importance (Suc *et al.* 1995b) during the glacial periods (Suc & Cravatte 1982; Combourieu-Nebout & Vergnaud Grazzini 1991; Beaudouin 2003) and even during interglacials (Subally *et al.* 1999) because of the ambivalent significance of *Artemisia* from the temperature viewpoint (cold vs. warm species: Subally & Quézel 2002).

The presence of steppe vegetation with *Artemisia* in the Ponto-Euxinian region (i.e. in Anatolia according to Site 380A pollen record; Popescu 2001, 2006) in the Late Miocene and their significant strengthening in the Early Pliocene is very informative. Their early presence and development in this region, contrary to the extreme scarcity of *Artemisia* in the Moroccan steppes in the Late Miocene and Early to Middle Pliocene (Bachiri Taoufiq 2000; Suc *et al.* 1999), indicates that Anatolia and neighbouring areas could have been the source area of this kind of vegetation for the rest of the Mediterranean region, a style of vegetation that became very abundant during the cold periods of the Quaternary (Popescu 2001; Suc & Popescu

2005). The early settlement and then development of *Artemisia* steppe vegetation in Anatolia may have resulted from migration from the east of this genus as a consequence of uplift of the Tibetan Plateau (where *Artemisia* species are still abundant today) and the succeeding reinforced Asiatic monsoon (Zhisheng *et al.* 2001).

Conclusions

Pollen data show a progressive reduction in the most thermophilous and high-water requirement plants typical of a broad-leaved evergreen forest and, in contrast, an increase in seasonal-adapted plants coming from higher altitude belts, including mesothermic (mainly deciduous) elements, altitudinal trees and herbs, during the Middle–Late Miocene and Pliocene. This has been interpreted as the response of the vegetation to global climate cooling, accentuated by the regional uplift of the surrounding mountains during Alpine tectonics. This process may also have been favoured by progressive movement of Eurasia towards northern latitudes.

The appearance of steppe vegetation with *Artemisia* on the Anatolian Plateau since the Late Miocene and its development in the Early Pliocene, significantly earlier than in the rest of Southern Europe, is informative. This suggests that the Anatolian *Artemisia*-rich steppes could have been the source area of this kind of open vegetal formation for the rest of the Mediterranean area during the Quaternary.

This paper is a contribution to the French Programme 'Environnement, Vie et Sociétés' (Institut Français de la Biodiversité). The authors thank J. Agustí and M. Harzhauser for their helpful reviews, and the EEDEN Programme (ESF) for invitations to participate in international workshops about the subject. Nurdan Yavuz-Isik is thanked for providing the Miocene samples from Turkey.

References

- AKGÜN, F. & AKYOL, E. 1999. Palynostratigraphy of the coal-bearing Neogene deposits graben in Büyük Menderes Western Anatolia. *Géobios*, **32**, 367–383.
- APOSTOL, L. & ENACHE, C. 1979. Etude de l'espèce *Dicerorhinus megarhinus* (de Christol) du bassin carbonifère de Motru. *Travaux du Musée d'Histoire Naturelle "Grigore Antipa"*, Bucarest, **20**, 533–540.
- BACHIRI TAOUFIQ, N. 2000. Les environnements marins et continentaux du corridor rifain au Miocène supérieur d'après la palynologie. PhD thesis, Université Hassan II – Mohammedia, Casablanca (Morocco).
- BACHIRI TAOUFIQ, N., BARHOUN, N., SUC, J.-P., MEON, H., ELAOUAD, Z. & BENBOUZIANE, A.

2001. Environment, végétation et climat du Messinien au Maroc. *Paleontologia i Evolució*, **32–33**, 127–138.
- BEAUDOUIN, C. 2003. Effets du dernier cycle climatique sur la végétation de la basse vallée du Rhône et sur la sédimentation de la plate-forme du golfe du Lion d'après la palynologie. PhD thesis, Université Claude Bernard Lyon-1, France, 403pp.
- BENDA, L. 1971. Grundzüge einer pollenanalytischen Gliederung des türkischen Jungtertiärs. *Beihefte zum Geologischen Jahrbuch*, **113**, 46pp.
- BENDA, L., HEISSIG, K. & STEFFENS, P. 1975. Die Stellung der vertebraten-faunengruppen der Türkei innerhalb der chronostratigraphischen systeme von Tethys und Paratethys. *Geologische Jahrbuch*, **B**, **15**, 109–116.
- BESSEDIK, M. 1985. Reconstitution des environnements Miocènes de régions nord-ouest méditerranéennes à partir de la palynologie. PhD thesis, Université de Montpellier, France, 162pp.
- BOHN-HAVAS, M. 1982. Mollusca fauna of Badenian and Sarmatian stage from the borehole Tengelic 2. In: NAGY, E., BODOR, E., HAGYAMAROSI, A. ET AL. (eds) *Palaeontological examination of the geological log of the borehole Tengelic 2*. Annales Instituti Geologici Publici Hungarici, **65**, 200–203.
- CAMBON, G., SUC, J.-P., ALOISI, J.-C. ET AL. 1997. Modern pollen deposition in the Rhône delta area (lagoonal and marine sediments) France. *Grana*, **36**, 105–113.
- CLAUZON, G., SUC, J.-P., POPESCU, S.-M., MARUNTEANU, M., RUBINO, J.-L., MARINESCU, F. & MELINTE, M. C. 2005. Influence of the Mediterranean sea-level changes over the Dacic Basin (Eastern Paratethys) in the Late Neogene. *Basin Research*, **17**, 437–462.
- COMBOURIEU-NEBOUT, N. & VERGNAUD GRAZZINI, C. 1991. Late Pliocene Northern hemisphere glaciation: the continental and marine responses in Central Mediterranean. *Quaternary Science Reviews*, **10**, 319–334.
- DRIVALIARI, A. 1993. Images polliniques et paléoenvironnement au Néogène supérieur en Méditerranée orientale. Aspects climatiques et paléogéographiques d'un transect latitudinal (de la Roumanie au Delta du Nil). PhD thesis, Université Montpellier-2, France, 333pp.
- DRIVALIARI, A., ȚICLEANU, N., MARINESCU, F., MĂRUNȚEANU, M. & SUC, J.-P. 1999. A Pliocene climatic record at Ticleni (Southwestern Romania). In: WRENN, J. H., SUC, J.-P. & LEROY, S. A. G. (eds) *The Pliocene: Time of Change*. American Association of Stratigraphic Palynologists Foundation, Dallas, 103–108.
- FORNACIARI, E. & RIO, D. 1996. Latest Oligocene to early Middle Miocene quantitative calcareous nannofossil biostratigraphy in the Mediterranean region. *Micropaleontology*, **42**, 1–36.
- FORNACIARI, E., DI STEFANO, A., RIO, D. & NEGRI, A. 1996. Middle Miocene quantitative calcareous nannofossil biostratigraphy in the Mediterranean region. *Micropaleontology*, **42**, 37–63.
- GONCHAROVA, I. A., SHCHERBA, I. G., KHONDKARIAN, S. O. ET AL. 2004. Lithological-Paleogeographic maps of Paratethys. Map 5: Early Middle Miocene. *Courier Forschungsinstitut Senckenberg*, **250**, 19–21.
- GRADSTEIN, F. M. & OGG, J. G. 2004. Geologic time scale 2004 – why, how, and where next! *Lethaia*, **37**, 175–181.
- GRADSTEIN, F. M., OGG, J. G., SMITH, A. G., BLEEKER, W. & LOURENS, L. J. 2004. A new geologic time scale with special reference to Precambrian and Neogene. *Episodes*, **27**, 83–100.
- GROMOLARD, C. & GUERIN, C. 1980. Mise au point sur *Parabos cordieri* (de Crystol), un Bovidé (Mammalia, Artiodactyla) du Pliocène d'Europe Occidentale. *Géobios*, **13**, 741–755.
- HARZHAUSER, M., MANDIC, O. & ZUSCHIN, M. 2003. Changes in Paratethyan marine molluscs at the Early/Middle Miocene transition: diversity, palaeogeography and palaeoclimate. *Acta Geologica Polonica*, **53**, 323–339.
- HARZHAUSER, M. & PILLER, W. E. 2004. The Early Sarmatian – hidden seesaw changes. *Courier Forschungsinstitut Senckenberg*, **246**, 89–112.
- HARZHAUSER, M. & PILLER, W. E. in press. Benchmark data of a changing sea – palaeogeography, palaeobiography and events in the Central Paratethys during the Miocene. *Palaeogeography, Palaeoclimatology, Palaeoecology*
- HEISSIG, K. 1976. Rhinocerotidae (Mammalia) aus der Anchiitherium-Fauna Anatoliens. *Geologisches Jahrbuch Reihe B*, **19**, 121pp.
- HEUSSER, L. 1988. Pollen distribution in marine sediments on the continental margin of Northern California. *Marine Geology*, **80**, 131–147.
- HSÜ, K. J. 1978. Correlation of Black Sea sequences. In: ROSS, D. A., NEPROCHNOV, Y. P. ET AL. (eds) *Initial Reports of the Deep Sea Drilling Project*, US Government Printing Office, **42**, 489–497.
- HSÜ, K. J. & GIOVANOLI, F. 1979. Messinian event in the Black Sea. *Palaeogeography, Palaeoclimatology, Palaeoecology*, **29**, 75–94.
- ILYINA, L. B., SHCHERBA, I. G., KHONDKARIAN, S. O. ET AL. 2004. Lithological-Paleogeographic maps of Paratethys. Map 6: Mid Middle Miocene. *Courier Forschungsinstitut Senckenberg*, **250**, 23–25.
- IVANOV, D. 1995. Palynological investigations of Miocene sediments from North-West Bulgaria (in Bulgarian, English abstract). PhD thesis, Institute of Botany BAS, Sofia, 45pp.
- IVANOV, D. A. & KOLEVA-REKALOVA, E. 1999. Palynological and sedimentological data about Lake Sarmatian palaeoclimatic changes in the Forecarpathian and Euxinian basins (Northern Bulgaria). *Acta Paleobotanica, Supplement*, **2**, 307–313.
- IVANOV, D., ASHRAF, A. R., MOSBRUGGER, V. & PALAMAREV, E. 2002. Palynological evidence for Miocene climate change in the Forecarpathian Basin (Central Paratethys, NW Bulgaria). *Palaeogeography, Palaeoclimatology, Palaeoecology*, **178**, 19–37.
- JIMÉNEZ-MORENO, G. 2005. Utilización del análisis polínico para la reconstrucción de la vegetación, clima y estimación de paleoaltitudes a lo largo de arco alpino europeo durante el Mioceno (21-8 m.a.). PhD thesis, Univ. Granada and Univ. C. Bernard – Lyon 1, 318pp.
- JIMÉNEZ-MORENO, G., RODRÍGUEZ-TOVAR, F. J., PARDO-IGÚZQUIZA, E., FAUQUETTE, S., SUC, J.-P.

- & MÜLLER, P. 2005. High resolution palynological analysis in the late early-middle Miocene core from the Pannonian Basin, Hungary: climatic changes, astronomical forcing and eustatic fluctuations in the Central Paratethys. *Palaeogeography, Palaeoclimatology, Palaeoecology*, **216**, 73–97.
- JIMÉNEZ-MORENO, G. & SUC, J.-P. in press. Middle Miocene Latitudinal Climatic Gradient in Western Europe: Evidence from Pollen Records. *Palaeogeography, Palaeoclimatology, Palaeoecology*.
- KOJUMDIEVA, E. 1976. Paléocologie des communautés des mollusques du Miocène en Bulgarie du Nord-Ouest. III. Communautés des mollusques du Volhynien (Sarmatien inférieur). *Geologica Balcanica*, **6**, 53–63.
- KOJUMDIEVA, E. & POPOV, N. 1989. Paléogéographie et évolution géodynamique de la Bulgarie Septentrionale au Néogène. *Geologica Balcanica*, **19**, 73–92.
- KOJUMDIEVA, E., NIKOLOV, I., NEDJALKOV, P. & BUSEV, A. 1982. Stratigraphy of the Neogene in the Sandanski Graben. *Geologica Balcanica*, **12**, 69–81.
- KOJUMDIEVA, E., POPOV, N., STANCHEVA, M. & DARAKCHIEVA, S. 1989. Correlation of the biostratigraphic subdivision of the Neogene in Bulgaria after molluscs, foraminifers and ostracods. *Geologica Balcanica*, **19**, 9–22.
- KOLEVA-REKALOVA, E. 1994. Sarmatian aragonite sediments in North-eastern Bulgaria – origin and diagenesis. *Geologica Balcanica*, **25**, 47–64.
- KORECZ-LAKY, I. 1982. Miocene foraminifera fauna from the borehole Tengelic 2. In: NAGY, E., BODOR, E., HAGYAMAROSI, A. ET AL. (eds) *Palaeontological examination of the geological log of the borehole Tengelic 2*, Annales Instituti Geologici Publici Hungarici, **65**, 186–187.
- KOVAC, M., BARATH, I., HARZHAUSER, M., HLAVATY, I. & HUDACKOVA, N. 2004. Miocene depositional systems and sequence stratigraphy of the Vienna Basin. *Courier Forschungsinstitut Senckenberg*, **246**, 187–212.
- KUHLEMANN, J. & KEMPF, O. 2002. Post-Eocene evolution of the North Alpine Foreland Basin and its response to Alpine tectonics. *Sedimentary Geology*, **152**, 45–78.
- LETOUZEY, J., GONNARD, R., MONTADERT, L., KRISTCHEV, K. & DORKEL, A. 1978. Black Sea: Geological setting and recent deposits distribution from seismic reflection data. In: ROSS, D. A., NEPROCHNOV, Y. P. ET AL. (eds) *Initial Reports of the Deep Sea Drilling Project*, US Government Printing Office, **42**, 1077–1084.
- LOURENS, L., HILGEN, F., SHACKLETON, N. J., LASKAR, J. & WILSON, D. 2004. The Neogene Period. In: GRADSTEIN, F., OGG, J. & SMITH, A. (eds) *Geologic Time Scale 2004*. Cambridge University Press.
- MAMATSASHVILI, N. S. 1975. The palynological characteristics of the Kolkhida Quaternary continental deposits (The Georgian SSR). *Metsniereba*, Tbilisi, 114pp.
- MARINESCU, F. 1978. Stratigrafia Neogenului superior din sectorul vestic al Bazinului Dacic. Editura Academiei Republicii Socialista România (in Romanian), 155pp.
- MĂRUNȚEANU, M. & PAPAIAANOPOL, I. 1995. L'association de nannoplankton dans les dépôts romaniens situés entre les vallées de Cosmina et de Cricovu Dulce (Munténie, bassin dacique, Roumanie). *Romanian Journal of Paleontology*, **76**, 169–170.
- MĂRUNȚEANU, M. & PAPAIAANOPOL, I. 1998. Mediterranean calcareous nannoplankton in the Dacic Basin. *Romanian Journal of Stratigraphy*, **78**, 115–121.
- MEULENKAMP, J. E. & SINGH, W. 2003. Tertiary palaeogeography and tectonostratigraphic evolution of the Northern and Southern Peri-Tethys platforms and the intermediate domains of the African-Eurasian convergent plate boundary zone. *Palaeogeography, Palaeoclimatology, Palaeoecology*, **196**, 209–228.
- MILLER, K. G., FEIGENSON, M., WRIGHT, J. D. & CLEMENT, B. 1991. Miocene isotope reference section, Deep Sea Drilling Project Site 608: an evaluation of isotope and biostratigraphic resolution. *Palaeoceanography*, **6**, 33–52.
- MOSBRUGGER, V., UTESCHER, T. & DILCHER, D. L. 2005. Cenozoic continental climatic evolution of Central Europe. *PNAS*, **102**, 14964–14969.
- NAGY, E. 1991. Climatic changes in the Hungarian Neogene. *Review of Palaeobotany and Palynology*, **65**, 71–74.
- NAGY, E. 1992. Magyarorszag Neogen sporomorphainak ertekelese. *Geologica Hungarica*, **53**, 1–379.
- NAGY, E. 1999. *Palynological correlation of the Neogene of the Central Paratethys*. Geological Institute of Hungary, Budapest, 149pp.
- NAGY, E. & KÓKAY, J. 1991. Middle Miocene mangrove vegetation in Hungary. *Acta geologica Hungarica*, **34**, 45–52.
- NAGYMAROSI, A. 1982. Badenian-Sarmatian nannoflora from the borehole Tengelic 2. In: NAGY, E., BODOR, E., HAGYAMAROSI, A. ET AL. (eds) *Palaeontological examination of the geological log of the borehole Tengelic 2*, Annales Instituti Geologici Publici Hungarici, **65**, 145–149.
- PALAMAREV, E. 1991. Composition, structure and main stages in the evolution of Miocene paleoflora in Bulgaria. Dsc thesis, (in Bulgarian). BAS, Sofia, 60pp.
- PALAMAREV, E. & IVANOV, D. 2001. Charakterzüge der vegetation des Sarmatien (Mittel- bis Obermiozän im südlichen Teil des Dazischen Beckens (Südost Europa)). *Palaeontographica*, **B259**, 209–220.
- PALAMAREV, E. & IVANOV, D. 2004. Badenian vegetation of Bulgaria: biodiversity, palaeoecology and palaeoclimate. *Courier Forschungsinstitut Senckenberg*, **249**, 63–69.
- PAPAIAANOPOL, I. & MĂRUNȚEANU, M. 1993. Biostratigraphy (molluscs and calcareous nannoplankton) of the Sarmatian and Meotian in eastern Muntenia (dacic basin-Rumania). *Zemni plyn a nafta*, **38**, 9–15.
- PAPAIAANOPOL, I. & MARINESCU, F. 1995. Lithostratigraphy and age of Neogene deposits on the Moesian Platform, between Olt and Danube Rivers. *Romanian Journal of Stratigraphy*, **76**, 67–70.
- PAPAIAANOPOL, I. & MOTAS, I. C. 1978. Marqueurs biostratigraphiques pour dépôt post-chersoniens du Bassin Dacique. *Dari de Seama ale Institutului de Geologie si Geofizica*, Stratigrafie, **64**, 283–294.
- PAPAIAANOPOL, I., JIPA, D., MARINESCU, F., ȚICLEANU, N. & MACALET, R. 1995. Upper Neogene from the

- Dacic Basin – Guide to excursion B2 (post-congress) X congress RCMNS, Bucuresti. *Romanian Journal of Stratigraphy*, **76**, 1–43.
- PARAMONOVA, N. P., SHCHERBA, I. G. & KHONDAKARIAN, S. O. 2004. Lithological-Paleogeographic maps of Paratethys. Map 7: Late Middle Miocene (Late Serravallian, Sarmatian s.s., Middle Sarmatian s.l.). *Courier Forschungsinstitut Senckenberg*, **250**, 27–31.
- PETRESCU, I. & MALAN, L. 1992. *Contributions to the knowledge of Upper Neogene microflora East of Turnu-Severin (Summary)*. Univ. Babes-Bolyai, Cluj-Napoca Gradina Botanica, Contributii Botanice 1991–1992: 135–143.
- PETRESCU, I., CERNITA, P., MEILESCU, C. *ET AL.* 1989a. Preliminary approaches to the palynology of the Lower Pliocene (Dacian) deposits in the Husnicioara area (Mehedinti county, SW Romania). *Studia Universitatis Babes-Bolyai Geologia-Geografia*, **34**, 67–74.
- PETRESCU, I., NICA, T., FILIPESCU, S. *ET AL.* 1989b. Paleoclimatical significance of the palynological approach to the Pliocene deposits of Lupoiaia (Gorj county). *Studia Universitatis Babes-Bolyai Geologia-Geografia*, **34**, 75–81.
- PLANDEROVÁ, E. 1990. *Miocene microflora of slovak Central Paratethys and its biostratigraphical significance*. Dionyz Stur Institute of Geology, Bratislava (Slovakia), 143pp.
- PLAZIAT, J.-C., CAVAGNETTO, C., KOENIGUER, J.-C. & BALTZER, F. 2001. History and biogeography of the mangrove ecosystem, based on a critical reassessment of the paleontological record. *Wetlands Ecology and Management*, **9**, 161–179.
- POPESCU, S.-M. 2001. *Végétation, climat et cyclostratigraphie en Paratéthys centrale au Miocène supérieur et au Pliocène inférieur d'après la palynologie*. PhD thesis. Université Claude Bernard Lyon-1, Lyon, France.
- POPESCU, S.-M. 2002. Repetitive changes in Early Pliocene vegetation revealed by high-resolution pollen analysis: revised cyclostratigraphy of southwestern Romania. *Review of Palaeobotany and Palynology*, **120**, 181–202.
- POPESCU, S.-M. 2006. Upper Miocene and Lower Pliocene environments in the southwestern Black Sea region from high-resolution palynology of DSDP site 380A (Leg 42B). *Palaeogeography, Palaeoclimatology, Palaeoecology*, **238**, 64–77.
- POPESCU, S.-M., KRIJGSMAN, W., SUC, J.-P., CLAUZON, G., MARUNTEANU, M. & NICA, T. 2006a. Pollen record and integrated high-resolution chronology of the early-Pliocene Dacic Basin (Southwestern Romania). *Palaeogeography, Palaeoclimatology, Palaeoecology*, **238**, 78–90.
- POPESCU, S.-M., SUC, J.-P. & LOUTRE, M.-F. 2006b. Early Pliocene vegetation changes forced by eccentricity-precession. Example from Southwestern Romania. *Palaeogeography, Palaeoclimatology, Palaeoecology*, **238**, 340–348.
- POPOV, S. V., RÖGL, F., ROZANOV, A. Y., STEININGER, F. F., SHCHERBA, I. G. & KOVAC, M. (eds) 2004. Lithological-Paleogeographic maps of Paratethys. 10 maps Late Eocene to Pliocene. *Courier Forschungsinstitut Senckenberg*, **250**, 1–46.
- QUÉZEL, P. & MÉDAIL, F. 2003. *Ecologie et biogéographie des forêts du bassin méditerranéen*, Elsevier France, 571pp.
- RADAN, S. C. & RADAN, M. 1998. Study of the geomagnetic field structure in the Tertiary in the context of magnetostratigraphic scale elaboration. I – The Pliocene. *An. Inst. Geol. Rom.*, **70**, 215–231.
- RADULESCU, C., SAMSON, P.-M., SEN, S., STIUCA, E. & HOROI, V. 1997. Les micromammifères pliocènes de Deanic (bassin Dacique, Roumanie). In: AGUILAR, J.-P., LEGENDRE, S. & MICHAUX, J. (eds) *Biochrom'97*, Mémoires Travaux E.P.H.E., Institute Montpellier, **21**, 635–647.
- RÖGL, V. F. 1998. Palaeogeographic considerations for Mediterranean and Paratethys seaways (Oligocene to Miocene), *Annalen des Naturhistorischen Museums in Wien*, **99A**, 279–310.
- SACHSE, M., MOHR, B. & SUC, J.-P. 1999. The Makrilaflora (Crete, Greece) – a contribution to the Neogene history of the climate and vegetation of the Eastern Mediterranean. *Acta palaeobotanica*, **Supplement 2**, 365–372.
- SEMENENKO, V. N. & OLEJNIK, E. S. 1995. Stratigraphic correlation of the Eastern Paratethys Kimmerian and Dacian stages by molluscs, dinocyst and nannoplankton data. *Rom. J. Stratigraphy*, **76**, 113–114.
- SHEVENELL, A. E., KENNETT, J. P. & LEA, D. W. 2004. Middle Miocene southern cooling and Antarctic cryosphere expansion. *Science*, **305**, 1766–1770.
- SICKENBERG, O., BECKER-PLATEN, J. D., BENDA, L. *ET AL.* 1975. Die Gliederung des höheren Jungtertiärs und Altquartärs in der Türkei nach Vertebraten und ihre Bedeutung für die internationale Neogen-Stratigraphie. *Geologisches Jahrbuch Reihe B*, **15**, 167pp.
- SNEL, E., MĂRUNTEANU, M., MACALET, R., MEULENKAMP, J. E. & VAN VUGT, N. 2006. Late Miocene to Early Pliocene chronostratigraphic framework for the Dacic Basin, Romania. *Palaeogeography, Palaeoclimatology, Palaeoecology*, **238**, 107–124.
- SPAACK, P. 1983. Accuracy in correlation and ecological aspects of the planktonic foraminiferal zonation of the Mediterranean Pliocene. *Utrecht Micropalaeontological Bulletin*, **28**, 160pp.
- SPROVIERI, R. 1992. Mediterranean Pliocene biochronology: a high resolution record based on quantitative planktonic foraminifera distribution. *Rivista Italiana di Paleontologia e Stratigrafia*, **98**, 61–100.
- SPROVIERI, R., BONOMO, S., CARUSO, A. *ET AL.* 2002. An Integrated calcareous plankton biostratigraphic scheme and biochronology of the Mediterranean Middle Miocene. *Rivista Italiana di Paleontologia e Stratigrafia*, **108**, 337–353.
- STEININGER, F. F. 1999. Chronostratigraphy, Geochronology and Biochronology of the “European Land Mammal Mega-Zones” (ELMMZ) and the Miocene “Mammal-Zones” (MN-Zones). In: RÖSSNER, G. E. & HEISSIG, K. (eds) *The Miocene Land Mammals of Europe*. Dr. Friedrich Pfeil, München, Germany, 9–24.
- SUBALLY, D., BILLODEAU, G., TAMRAT, E., FERRY, S., DEBARD, E. & HILLAIRE-MARCEL, C. 1999. Cyclic climatic records during the Olduvai subchron (uppermost Pliocene) on Zakynthos Island (Ionian Sea). *Geobios*, **32**, 793–803.

- SUBALLY, D. & QUÉZEL, P. 2002. Glacial or interglacial: *Artemisia* a plant indicator with dual responses. *Review of Palaeobotany and Palynology*, **120**, 123–130.
- SUC, J.-P. 1984. Origin and evolution of the Mediterranean vegetation and climate in Europe. *Nature*, **307**, 429–432.
- SUC, J.-P. & CRAVATTE, J. 1982. Etude palynologique du Pliocène de Catalogne (nord-est de l'Espagne). *Paléobiologie Continentale*, **13**, 1–31.
- SUC, J.-P. & DRIVALIARI, A. 1991. Transport of bisaccate coniferous fossil pollen grains to coastal sediments: an example from the earliest Pliocene Orbria (Languedoc, Southern France). *Review of Palaeobotany and Palynology*, **70**, 247–253.
- SUC, J.-P. & POPESCU, S.-M. 2005. Pollen records and climatic cycles in the North Mediterranean region since 2.7 Ma. In: HEAD, M. J. & GIBBARD, P. L. (eds) *Early–Middle Pleistocene Transitions: The Land–Ocean Evidence*. Geological Society of London, Special Publication, **247**, 147–158.
- SUC, J.-P., DINIZ, F., LEROY, S. *ET AL.* 1995a. Zanclean (~ Brunsumian) to early Piacenzian (~ early-middle Reuverian) climate from 4° to 54° north latitude (West Africa, West Europe and West Mediterranean areas). *Mededelingen Rijks Geologische Dienst*, **52**, 43–56.
- SUC, J.-P., BERTINI, A., COMBORIEU-NEBOUT, N. *ET AL.* 1995b. Structure of West Mediterranean vegetation and climate since 5.3 Ma. *Acta zoologica Cracoviense*, **38**, 3–16.
- SUC, J.-P., FAUQUETTE, S., BESEDIK, M. *ET AL.* 1999. Neogene vegetation changes in West European and West circum-Mediterranean areas. In: AGUSTÍ, J., ROOK, L. & ANDREWS, P. (eds) *The Evolution of Neogene Terrestrial Ecosystems in Europe*. Cambridge University Press, Cambridge, 378–388.
- THOMAS, H., SPASSOV, N., KODJUMJIEVA, E. *ET AL.* 1986. Résultats préliminaires de la première mission paléontologique franco-bulgare à Dorkovo (arrondissement de Pazardjik, Bulgarie). *Comptes Rendus de l'Académie des Sciences, Paris*, **302**, 1037–1042.
- TICLEANU, N. & DIACONITA, D. 1997. The main coal facies and lithotypes of the Pliocene coal basin, Oltenia, Romania. In: GAYER, R. & PESEK, J. (eds) *European Coal Geology and Technology*, Geological Society, London, Special Publication, **125**, 131–139.
- VAN VUGT, N., LANGEREIS, C. G. & HILGEN, F. J. 2001. Orbital forcing in Pliocene-Pleistocene Mediterranean lacustrine deposits: dominant expression of eccentricity versus precession. *Palaeogeography, Palaeoclimatology, Palaeoecology*, **172**, 193–205.
- WANG, C. W. 1961. The forests of China with a survey of grassland and desert vegetation. Maria Moors Cabot Foundation, **5**, Harvard University Cambridge, Massachusetts, 313pp.
- ZACHOS, J., PAGANI, M., SLOAN, L. & BILLUPS, K., 2001. Trends, rhythms, and aberrations in global climate 65 Ma to present. *Science*, **292**, 686–693.
- ZHISHENG, A., KUTZBACH, J., PRELL, W. L. & PORTER, S. C. 2001. Evolution of Asian monsoons and phased uplift of the Himalaya-Tibetan plateau since Late Miocene times. *Nature*, **411**, 62–66.
- ZOHARY, M. 1973. Geobotanical foundations of the Middle East. Fischer ed., Stuttgart, **2 vol.**, 739pp.

Appendix E**ROYAL SOCIETY
OPEN SCIENCE****Messinian vegetation and climate of the intermontane
Florina-Ptolemais-Servia Basin, NW Greece: How well do
plant fossils reflect past environments?**

Journal:	Royal Society Open Science
Manuscript ID	RSOS-192067
Article Type:	Research
Date Submitted by the Author:	28-Nov-2019
Complete List of Authors:	Bouchal, Johannes; Swedish Museum of Natural History, Palaeobiology Güner, Tuncay H.; Istanbul University Cerrahpaşa, Faculty of Forestry, Department of Forest Botany Velitzelos, Dimitrios; National and Kapodistrian University of Athens Faculty of Geology and Geoenvironment, Section of Historical Geology and Palaeontology Velitzelos, Evangelos; National and Kapodistrian University of Athens Faculty of Geology and Geoenvironment, Section of Historical Geology and Palaeontology Denk, Thomas; Swedish Museum of Natural History, Dep. of Palaeobotany
Subject:	Palaeontology < EARTH SCIENCES, ecology < BIOLOGY, plant science < BIOLOGY
Keywords:	Biome reconstruction, proxy biases, climate reconstruction, plant macrofossils, dispersed pollen, light and scanning electron microscopy
Subject Category:	Earth science

Author-supplied statements

Relevant information will appear here if provided.

Ethics

Does your article include research that required ethical approval or permits?:

This article does not present research with ethical considerations

Statement (if applicable):

CUST_IF_YES_ETHICS :No data available.

Data

It is a condition of publication that data, code and materials supporting your paper are made publicly available. Does your paper present new data?:

Yes

Statement (if applicable):

This article and all data used in this article are made available in bioRxiv.

<https://www.biorxiv.org/content/10.1101/848747v1>

Supplementary Material are available within the figshare repository:

<https://doi.org/10.6084/m9.figshare.10327646.v1>

Conflict of interest

I/We declare we have no competing interests

Statement (if applicable):

CUST_STATE_CONFLICT :No data available.

Authors' contributions

This paper has multiple authors and our individual contributions were as below

Statement (if applicable):

[revised manuscript text omitted]

1) Velitzelos & Schneider, 1979; 2) Velitzelos & Petrescu, 1981; 3) Velitzelos et al., 1983; 4) Velitzelos & Gregor, 1985; 5) Mai & Velitzelos, 1992; 6) Mai & Velitzelos, 1997;

7) Velitzelos & Kvaček, 1999; 8) Kvaček et al., 2002; 9) Denk & Velitzelos, 2002; 10) Velitzelos & Denk, 2002; 11) this study

Climate parameter	CLAMP Physg3arcAZ	CLAMP PhysgAsia1	CA modified	CA modified 10-90%iles
MAT (°C)	10–13.5	8.7–11.5	8.6–21.2	9.9–18.4
CMMT (°C)	1–5	-2.7–2.3	≥ 1.2	-
WMMT (°C)	19.2–22.8	19–22.6	-	-
GROWSEAS (months)	6–8	5.5–7	-	-
MMGSP (mm)	110–160	100–160	-	-
Three_WET (mm)	500–780	400–750	-	-
Three_DRY (mm)	180–260	80–220	-	-
3 WET/3 DRY	< 4	< 5.5	-	-

MAT = mean annual temperature, CMMT = coldest month mean temperature, WMMT = warmest month mean temperature, GROWSEAS = duration of growing season, MMGSP = mean month growing season precipitation, Three_WET = precipitation of three consecutive wettest months, Three_DRY = precipitation of three consecutive driest months.

Table 3. Vegetation types recognised for the pre-evaporitic Messinian of the Florina–Ptolemais–Servia Basin.

Vegetation type	Main (and accessory) taxon/taxa	Biome ^a	Vegetation unit(s) ^b	Modern (Neogene) analogue	References
Swamp forest	Taxodium, Glyptostrobus	NLD	VU3	Taxodium swamp forests SE USA; (Taxodium/Glyptostrobus swamp forests widespread in N Hemisphere Neogene)	1, 2
Swamp forest	Alnus, (Sassafras)	BLD	VU3	Alnus swamp forest	3, 4
Riparian forest	Pterocarya, Zelkova, Ulmus, (Sassafras)	BLD	VU4	Riparian and alluvial forest of Georgia and Iran	1, 3, 4, 5, 6
Well-drained forest	Quercus kubinyi, Q. pseudocastanea, (Carpinus, Tilia etc.)	BLD	VU5b	Lowland oak-hornbeam forests; ("Quercetum mixtum")	4, 7
Well-drained forest	Fagus, (Quercus pseudocastanea)	BLD	VU5b	Lowland beech forests of N Turkey, Georgia, N Iran; ("Fagetum gussonii")	3, 4, 7
Well-drained forest	Fagus, Abies, Cedrus, Cathaya	MIXED	VU6b	Montane Fagus-Abies forest, montane Fagus-Cedrus-Pinus forest; Abant Gölü; Erbaa-Çatalan	8, 9, 10
Well-drained laurophyllous forest	Quercus drymeja, (Q. sosnowsky)	BLE	VU6a	Quercus dilatata association (with Taxus, Pinus, Acer etc.)	11
Well-drained sclerophyllous forests/shrublands	Quercus mediterranea, Chamaerops, Olea	BLE/ SHRUBLAND	VU0	Mediterranean sclerophyllous forest/shrublands	10
[?] ^c Grassland-steppe forest	Poaceae	GRASSLAND/ SHRUBLAND	VU0	Forest-steppe of SE Europa to Afghanistan	8, 10, 11, 12

1) Mai, 1995; 2) Dolezych & Schneider, 2007; 3) Denk et al., 2001; 4) Akhiani et al., 2010; 5) Maharramova, 2015; 6) Kozłowski et al., 2018; 7) Kvaček et al., 2002; 8) Mayer & Aksoy, 1986; 9) Akkemik, 2003; 10) van Zeist & Bottema, 1991; 11) Freitag, 1971; 12) Erdős et al., 2018

^aBiome classification follows the phsiognomic approach of Woodward et al., 2004. ^bVegetation units as in Table 1.

^c[?] expresses the uncertainty around a possible extra-regional signal in the Vegora pollen record. According to Erdős et al. (2018) steppe forest with *Stipa* and other grasses and different species of *Quercus* (forest-steppes of the type 'Region A - SE Europe') is characterized by MAP of 420-600 mm; this would be much drier than the inferred MAP for the FPS.

Figure 1. Fossil localities and lithological map of the Florina-Ptolemais-Servia Basin. Map redrawn after Steenbrink et al. (1999, 2006), Ognjanova-Rumenova (2005), Ivanov (2001) and Koufos (2006). Fossil localities: (1) Bitola Basin, Republic of North Macedonia, PF. (2) Vegora Basin, MF and PF (3) Dytiko, VF. (4) Prosilio, MF. (5) Lava, MF. (6) Likoudi, MF. (7) Serres Basin. (2–7) Greece. (8) Sandanski Graben, Bulgaria, PF. Abbreviations: Plant macrofossils (MF), palynoflora (PF), vertebrate fossils (VF).

71x52mm (300 x 300 DPI)

Figure 2. Lithology and polarity zones of the Vegora section (redrawn after Steenbrink et al., 2006). Position of fossil bearing strata following Velitzelos and Schneider (1979) and Kvaček et al. (2002).

119x284mm (300 x 300 DPI)

Figure 3. Light microscopy (LM) and scanning electron microscopy (SEM) micrographs of algae, fern and fern allies, and gymnosperm palynomorphs.

[revised manuscript text omitted]

35 91 36 92 **2. Material and Methods**

93 *2.1. Geological setting*

94 The old open-pit lignite quarry of Vegora is located in western Macedonia, NW Greece, ca. 2
95 km E of the town of Amyntaio and is part of the Neogene Florina-Ptolemais-Servia
intermontane basin (FPS). The FPS is part of the Pelagonian basin that extends to the north
into North Macedonia (Fig. 1). The NNW–SSE trending FPS is ca. 120 km long and presently
at elevations between 400 and 700 m a.s.l. and is flanked by mountain ranges to the east and
the west. Main ranges include Baba Planina (2,601 m), Verno (2,128 m), and Askio (2,111 m)
to the east of the basin and Voras (2,528 m), Vermio (2,065 m), Olympus (2,917 m) to the
west (Fig. 1). These ranges are mainly comprised of Mesozoic limestones, Upper
Carboniferous granites and Paleozoic schists.

[revised manuscript text omitted]
5–1.86 (NOW database, <http://pantodon.science.helsinki.fi/now/locality.php?p=ecometrics>) corresponds to the diet types “mixed-closed habitats”, “regular browsers”, and “selective browsers” according to Janis (1988) and hence provides an excellent match with the environments inferred for the FPS.

From Lava (Fig. 1), Steenbrink et al. (2000) investigated two sequences covering two
sedimentary cycles each. Based on palaeomagnetic correlation these sequences are dated as c.
6.8–6.7 Ma and c. 6.3 Ma. The pollen assemblages are comparable to the Vegora assemblage
but differ in some respects. First, the Lava sections have a continuous high amount of *Pinus*
pollen (20 to >60%) suggesting that the fossil site was located closer to pine forests than was
the Vegora lake. Second, *Cedrus* pollen is abundant with values between 10 and >30%. Third,
Steenbrink et al. (2000) did not report evergreen oak pollen, although evergreen *Quercus* is
known from Lava based on leaf fossils (Velitzelos et al., 2014). Steenbrink et al. (2000)
inferred a humid temperate climate without dry season for the investigated sedimentary
cycles. In addition, they suggested that expansions of *Fagus* accompanied by a decrease of
*Abies* might reflect subtle increases in montane humidity. Overall, they suggested
continuously wet and warm-temperate climate conditions for the investigated period for Lava.
Velitzelos et al. (2014) provided revised taxon lists for the roughly coeval macrofossil (leaves
and fruits/seeds) localities Prosilio and Lava (age based on palaeomagnetic correlation, 6.7–
6.4 Ma; Steenbrink et al., 2006). The macroflora is very similar to the one from Vegora in
terms of composition. However, whereas *Quercus sosnowskyi* is among the most abundant
elements in Vegora, only a few leaves represent this species in Prosilio; also *Glyptostrobus* is
much less abundant. *Pinus* is represented by cones, leaf fascicles, and leafy branches; this is
in accordance with the high amount of pine pollen documented in the palynological record.
*Fagus* is a frequent element as well, while *Abies* is not recorded in the macroflora.
Likoudi, 20 km S of Lava, is located in a small basin south of the main FPS (Fig. 1). The
macroflora (leaves, fruits and seeds) is very rich (see revised and updated floral list in
Velitzelos et al., 2014). The precise age of the Messinian diatomaceous marls is not clear
(Knobloch & Velitzelos, 1986) although it unambiguously is pre-evaporitic. The flora is
characterised by the high diversity of conifers (11 genera of Cupressaceae and Pinaceae,
including *Torreya* – as *Egeria* sp. in Velitzelos et al., 2014). As in Vegora, *Fagus* is a
dominating element. Other taxa (*Cercis*, *Laria*, *ef. Nerium*) are not known from other FBS
floras. Well-preserved cones of *Cedrus* and cones and leafy twigs of *Cathaya* and *Taiwania*
suggest that these genera were not growing at high elevations but nearby the area of
deposition (lake). If coeval with the Lava deposits, this would explain the relatively high
amounts of *Cedrus* pollen in the palynological section of Lava.
Ivanov & Slavomirova (2002) investigated a 70 m succession of lacustrine sediments in the
Bitola Basin (Northern Macedonia; borehole V-466; 1 in Fig. 1) about 10 km E of Bitola and
40 km NNW of Vegora. Based on a vertebrate fauna on top of these sediments the plant-
bearing sediments are assigned a late Miocene age (Dumurdžanov et al., 2002; Ognjanova-
Rumeno, 2005). From these sediments, abundant leaves of *Quercus sosnowskyi* have been
reported (Dumurdžanov et al., 2002). The pollen assemblage is similar to the one from

[revised manuscript text omitted]
 Erdős L, Ambarli D, Anenkhonov OA, Bátori Z, Cserhalmi D, Kiss M, Kröel-Dulay G, Liu H,
Magnes M, Molnár Z, et al. 2018 The edge of two worlds: A new synthesis on Eurasian
forest-steppes. *Appl. Vegetation Sci.* 21, 345–362.
- Favre E, Escarguel G, Suc J-P, Vidal G, Thévenod L. 2008 A contribution to deciphering the
meaning of AP/NAP with respect to vegetation cover. *Rev. Palaeobot. Palynol.* 148,
13–35.
- Fang J, Wang Z, Tang Z. 2009 Atlas of Woody Plants in China. Volumes 1 to 3 and index.
Beijing: Higher Education Press.
- Fauquette S, Suc J-P, Bertini A, Popescu S-M, Warny S, Bachiri Taoufiq N, Perez Villa, M-J,
Chikhi H, Subally D, Feddi N, Clauzon G, Ferrier J. 2006 How much did climate force
the Messinian salinity crisis? Quantified climatic conditions from pollen records in the
Mediterranean region. *Palaeogeogr., Palaeoclimatol., Palaeoecol.* 238, 281–301.
- Ferguson DK, Hofmann C-C, Denk T. 1999 Taphonomy: field techniques in modern
environments. In: Jones TP, Rowe NP (eds) *Fossil Plants and Spores: modern
techniques*. London: Geological Society, pp 210–213.
- Freitag, H. 1971 Die natürliche Vegetation Afghanistans. *Vegetatio* 22, 285–344.
- Gersonde R, Velitzelos E. 1978 Diatomeenpaläoökologie im Neogen-becken von Vegora N-
W Mazedonien (vorläufige Mitteilung). *Ann. Géol. Pays Hellèn.* 30, 373–382.
- Grimm GW, Bouchal JM, Denk T, Potts A. 2016 Fables and foibles: A critical analysis of the
Palaeoflora database and the Coexistence Approach for palaeoclimate reconstruction.
*Rev. Palaeobot. Palyn.* 233, 611–622.
- Grimm GW, Potts A. 2016 Fallacies and fantasies: the theoretical underpinnings of the
Coexistence Approach for palaeoclimate reconstruction. *Clim. Past* 12, 611–622.
- Halbritter H, Ulrich S, Grimsson F, Weber M, Zetter R, Hesse M, Buchner R, Svojtka M,
Frosch-Radivo A. 2018 *Illustrated Pollen Terminology*, 2nd ed. Cham (Switzerland):
Springer Nature.
- Herbert TD, Lawrence KT, Tzanova A, Peterson LC, Caballero-Gill R, Kelly CS. 2016 Late
Miocene global cooling and the rise of modern ecosystems. *Nature Geosci.* 9, 843–847.
- Ivanov DA. 2001 Palaeoecological interpretation of a pollen diagram from the Sandanski
graben (Southwest Bulgaria). *Comptes Rend. Acad. Bulg. Sci.* 54, 65–68.
- Ivanov DA, Slavomirova E. 2002 Preliminary palynological data on Neogene flora from
Bitola Basin (F.Y.R.O.M.). *Comptes Rend. Acad. Bulg. Sci.* 55, 81–86.
- Janis CM. 1986 An estimation of tooth volume and hypsodonty indices in ungulate mammals,
and the correlation of these factors with dietary preference. In: DE Russell, JP Santoro,
D Sigogneau-Russell (eds) *Teeth revisited*. Proc. 7th Int. Symp. Dental Morph. Mém.
Mus. Nat. Hist. naturelle, Sér. C 53, 367–387.
- Jiang X-L, Hipp AL, Deng M, Su T, Zhou Z-K, Yan M-X. 2019 East Asian origins of
European holly oaks (*Quercus* section *Ilex* Loudon) via the Tibet-Himalaya. *J.*
*Biogeogr.* 46, 2188–2202. doi.org/10.1111/jbi.13654
- Karistinos N, Ioakim C. 1989 Palaeoenvironmental and palaeoclimatic evolution of the
Serres Basin (N. Greece) during the Miocene. *Palaeogeogr., Palaeoclimatol.,*
*Palaeoecol.* 70, 275–285.
- Knobloch E, Velitzelos E. 1986a Die obermiozäne Flora von Likudi bei Ellassona (Thessalien,
Griechenland). *Doc. Nat.* 29, 5–20.
- Knobloch E, Velitzelos E. 1986b Die obermiozäne Flora von Prosilion bei Kozani (Süd-
Mazedonien, Griechenland). *Doc. Nat.* 29, 29–33.

- Kottek M, Grieser J, Beck C, Rudolf B, Rubel F. 2006 World map of the Köppen-Geiger
climate classification updated. *Meteorol. Z.* 15, 259–263.
- Koufos GD. 1982 *Hipparion crassum* Gervais, 1859 from the lignites of Ptolemais
(Macedonia-Greece). *Proc. Kon. Ned. Akad. Wetten. B.* 85, 229–239.
- Koufos GD. 2006 The Neogene mammal localities of Greece: Faunas, chronology and
biostratigraphy. *Hellen. J. Geosci.* 41, 183–214
- Koufos GD, Kostopoulos DS, Koliadimou KK. 1991 Un nouveau gisement de mammifères
dans le Villafranchien de Macédoine occidentale (Grèce). *C. R. Acad. Sci. Paris, ser. II*
313, 831–836.
- Kovar-Eder J, Jechorek H, Kvaček Z, Parashiv V. 2008 The Integrated Plant Record: An
essential tool for reconstructing Neogene zonal vegetation in Europe. *Palaios* 23, 97–
111.
- Kovar-Eder J, Kvaček Z, Martinetto E, Roiron P. 2006 Late Miocene to Early Pliocene
vegetation of southern Europe (7–4 Ma) as reflected in the megafossil plant record.
*Palaeogeogr., Palaeoclimatol., Palaeoecol.* 238, 321–339.
- Kozłowski G, Bétrisey S, Song Y. 2018 Wingnuts (*Pterocarya*) and walnut family. Relict
trees: linking the past, present and future. *Natural History Museum Fribourg:*
*Switzerland.*
- Kvaček Z, Velitzelos D, Velitzelos E. 2002 Late Miocene Flora of Vegora Macedonia N.
Greece. Athens: Koralis.
- Maharramova E. 2015 Genetic diversity and population structure of the relict forest trees
*Zelkova carpinifolia* (Ulmaceae) and *Pterocarya fraxinifolia* (Juglandaceae) in the
South Caucasus. PhD Dissertation. Freie Universität Berlin.
- Mai HD. 1995 Tertiäre Vegetationsgeschichte Europas. Stuttgart: Gustav Fischer.
- Mai DH, Velitzelos E. 1992 Über fossile Pinaceen-Reste im Jungtertiär von Griechenland.
*Feddes Repert.* 103, 1–18.
- Mai DH, Velitzelos E. 1997 Paläokarpologische Beiträge zur jungtertiären Flora von Vegora
(Nordgriechenland). *Feddes Repert.* 108, 507–526.
- Marinova E, Harrison SP, Bragg F, Connor S, de Laet V, Leroy SAG, Mudie P, Atanassova J,
Bozilova E, Caner H, Cordova C, Djamali M, Filipova-Marinova M, Gerasimenko N,
Jahns S, Kouli K, Kotthoff U, Kvavadze E, Lazarova M, Novenko E, Ramezani E,
Röpke A, Shumilovskikh L, Tanțău I, Tonkov S. 2018 Pollen-derived biomes in the
Eastern Mediterranean–Black Sea–Caspian–Corridor. *J. Biogeogr.* 45, 484–499.
- Mayer H, Aksoy, H. 1986 Wälder der Türkei. Stuttgart: Gustav Fischer.
- Mosbrugger V, Utescher T. 1997 The coexistence approach — a method for quantitative
reconstructions of Tertiary terrestrial palaeoclimate data using plant fossils.
*Palaeogeogr., Palaeoclimatol., Palaeoecol.* 134, 61–86
- Ognjanova-Rumenova N. 2005 Upper Neogene siliceous microfossils from Pelagonia Basin
(Balkan Peninsula). *Geol. Carpathica* 56 (4), 347–358.
- Pavlides SB, Mountrakis DM. 1986 Neotectonics of the Florina–Vegorit–Ptolemais
Neogene Basin (NW Greece): an example of extensional tectonics of the greater Aegean
area. *Ann. Géol. Pays Hellen.* 33, 311–327.
- Peel MC, Finlayson BL, McMahon TA. 2007 Updated world map of the Köppen-Geiger
climate classification. *Hydrol. Earth Syst. Sci.*, 11, 1633–1644.
- Psilovikos A, Karistinos N. 1986 A depositional sedimentary model for the Neogene
uraniferous lignites of the Serres graben, Greece. *Palaeogeogr., Palaeoclimatol.,*
*Palaeoecol.* 56, 1–16.

Punt W, Hoen PP, Blackmore S, Nilsson RH, Le Thomas A. 2007 Glossary of pollen and
spore terminology. *Rev. Palaeobot. Palynol.* 143, 1–81.
- Ratnam J, Bond WJ, Fensham RJ, Hoffmann WA, Archibald S, Lehmann CER, Anderson
MT, Higgins SI, Sankaran M. 2011 When is a ‘forest’ a savanna, and why does it
matter? *Glob. Ecol. Biogeogr.* 20, 653–660.
- Roberts S, Rassios A, Wright L, Vacondios I, Vrachatis G, Grivas E, Nesbitt RW, Neary CR,
Moat T, Kostantopoulou L. 1988 Structural controls on the location and form of the
Vourinos chromite deposits. In: Boissonnas J, Omenetto P (eds) *Mineral deposits within*
*the European Community*. Springer, Berlin Heidelberg New York, pp 249–266.
- Rubel F, Brugger K, Haslinger K, Auer I. 2017 The climate of the European Alps: Shift of
very high resolution Köppen-Geiger climate zones 1800–2100. *Meteorol. Z.*, 26, 115–
125. <https://doi.org/10.1127/metz/2016/0816>.
- Schneider W. 1992 Floral successions in the Miocene swamps and bogs of Central. Europe.
*Z. Geol. Wiss.* 20, 555–570.
- Spicer RA. 2008 CLAMP. In: Gornitz V. Ed. *Encyclopedia of Paleoclimatology and Ancient*
*Environments*. Dordrecht: Springer.
- Steenbrink J, Van Vugt N, Hilgen FJ, Wijbrans JR, Meulenkamp JE. 1999 Sedimentary
cycles and volcanic ash beds in the lower Pliocene lacustrine succession of Ptolemais
(NW Greece): discrepancy between $^{40}\text{Ar}/^{39}\text{Ar}$ and astronomical ages. *Palaeogeogr.*,
*Palaeoclimatol.*, *Palaeoecol.* 152, 283–303.
- Steenbrink J, Van Vugt N, Kloosterboer-van Hoeve ML, Hilgen FJ. 2000 Refinement of the
Messinian APTS from sedimentary cycle patterns in the lacustrine Lava section (Serbia
Basin, NW Greece). *Earth Planet. Sci. Lett.* 181, 161–173.
- Steenbrink J, Hilgen FJ, Krijgsman W, Wijbrans JR, Meulenkamp JE. 2006 Late Miocene to
Early Pliocene depositional history of the intramontane Florina-Ptolemais-Serbia Basin,
NW Greece: Interplay between orbital forcing and tectonics. *Palaeogeogr.*,
*Palaeoclimatol.*, *Palaeoecol.* 238, 151–178.
- Suc J-P. 1984 Origin and evolution of the Mediterranean vegetation and climate in Europe.
*Nature* 307, 429–432.
- Suc J-P, Popescu S-M, Do Couto D, Clauzon G, Rubino J-L, Melinte-Dobrinescu MC,
Quillévéré F, Brun J-P, Dumurdžanov N, Zagorchev I, Lesdić V, Tomić D, Sokoutis D,
Meyer B, Macaleț R, Jelen B, Rihelj H. 2015 Marine gateway vs. fluvial stream within
the Balkans from 6 to 5 Ma. *Mar. Pet. Geol.* 66, 231–245.
- Suc J-P, Popescu SM, Fauquette S, Bessedik M, Jiménez-Moreno G, Taoufiq B, Zheng Z,
Medail F, Klotz S. 2018 Reconstruction of Mediterranean flora, vegetation and climate
for the last 23 million years based on an extensive pollen dataset. *Ecol. mediterr.* 44,
53–85.
- Traverse A. 2007 *Paleopalynology*. Topics in Geobiology 28. Dordrecht, Springer.
- Utescher T, Bruch AA, Erdei B, François I, Ivanov D, Jacques FMB, Kern AK, Liu Y-SC,
Mosbrugger V, Spicer RA. 2014 The Coexistence Approach—Theoretical background
and practical considerations of using plant fossils for climate quantification.
*Palaeogeogr.*, *Palaeoclimatol.*, *Palaeoecol.* 410, 58–73.
- Van de Weerd A. 1979 Early Ruscinian rodents and lagomorphs (Mammalia) from the
lignites near Ptolemais (Macedonia, Greece). *Proc. Kon. Nederl. Akad. Wet.*, B. 82,
127–170.
- Van Zeist W, Bottema S. 1991 Late Quaternary vegetation of the Near East. *Beih. Tübinger*
*Atlas Vord. Orient, Reihe A (Naturwiss.)* 18, 1–156.

Van Zeist W, Woldring H, Stapert, D. 1975 Late Quaternary vegetation and climate of
southwestern Turkey. *Palaeohist.* 14, 35–143.
- Velitzelos D, Denk T. 2002 Leaf epidermal characteristics of late Tertiary conifers from
Greece: taxonomic significance and limitations. 6th Europ. Paleobot. Palynol. Conf.
Athens, Greece, Abstracts, 183–184.
- Velitzelos D, Bouchal JM, Denk T. 2014 Review of the Cenozoic floras of Greece. *Rev.*
*Palaeobot. Palynol.* 204, 1–15.
- Velitzelos E, Gregor H-J. 1985 Neue paläofloristische Befunde im Neogen Griechenlands.
*Doc. Nat.* 25, 1–4.
- Velitzelos E, Krach JE, Gregor H-J, Geissert F. 1983 *Bolboschoenus vegorae* – ein Vergleich
fossiler und rezenter Rhizomknollen der Strandbinse. *Doc. Nat.* 5, 1–57.
- Velitzelos E, Kvaček Z. 1999 Review of the late Miocene flora of Vegora western
Macedonia, Greece. *Acta Palaeobotanica, Suppl.* 2 (Proceed. 5th EPPC) 419–427.
- Velitzelos E, Petrescu I. 1981 Seltene pflanzliche Fossilien aus dem Braunkohlebecken von
Vegora. *Ann. géol. Pays hellén.* 30, 767–777.
- Velitzelos E, Schneider HE. 1979 Jungtertiäre Pflanzenfunde aus dem Becken von Vegora in
West-Mazedonien. 3. Mitteilung: Eine Fächerpalme (*Chamaerops humulis* L.). *Ann.*
*géol. Pays hellén.* 29, 796–799.
- Woodward FI, Lomas MR, Kelly CK. 2004 Global climate and the distribution of plant
biomes. *Phil. Trans. R. Soc. Lond. B.* 359, 1465–1476.
- Yang J, Spicer RA, Spicer TEV, Li C-S. 2011 'CLAMP Online': a new web-based
palaeoclimate tool and its application to the terrestrial Paleogene and Neogene of North
America. *Palaeobiodiv. Palaeoenviro.* 91, 163–183.
- Zetter R. 1989 Methodik und Bedeutung einer routinemäßigen kombinierten
lichtmikroskopischen und rasterelektronenmikroskopischen Untersuchung fossiler
Mikrofloren. *Cour. Forschungsinst. Senck.* 109, 41–50.

**Table and Figure Captions**

**Table 1.** Plant taxa recorded from unit 1 (lignite seam) and unit 2 (blue marls) of the Vegora
section.

**Table 2.** Estimated climate parameters for the pre-evaporitic Messinian of Vegora from two
CLAMP calibration datasets and from CA.

**Table 3.** Vegetation types recognised for the pre-evaporitic Messinian of the Florina–
Ptolemais–Servia Basin.

**Figure 1.** Fossil localities and lithological map of the Florina–Ptolemais–Servia Basin.
Map redrawn after Steenbrink et al. (1999, 2006), Ognjanova-Rumenova (2005), Ivanov
(2001) and Koufos (2006). Fossil localities: (1) Bitola Basin, Republic of North Macedonia,
PF. (2) Vegora Basin, MF and PF (3) Dytiko, VF. (4) Prosilio, MF. (5) Lava, MF. (6)
Likoudi, MF. (7) Serres Basin. (2–7) Greece. (8) Sandanski Graben, Bulgaria, PF.
Abbreviations: Plant macrofossils (MF), palynoflora (PF), vertebrate fossils (VF).

**Figure 2.** Lithology and polarity zones of the Vegora section (redrawn after Steenbrink et al.,
2006). Position of fossil bearing strata following Velitzelos and Schneider (1979) and Kvaček
et al. (2002).

**Figure 3.** Light microscopy (LM) and scanning electron microscopy (SEM) micrographs of
algae, fern and fern allies, and gymnosperm palynomorphs.

(a) *Botryococcus* sp. cf. *B. kurzii*. (b) *Spirogyra* sp. 1/ *Ovoidites elongatus*. (c) *Spirogyra* sp.
2/*Cycloovoidites cyclus*. (d–e) *Osmunda* sp., (d) EV, (e) PV. (f) *Cryptogramma* vel
*Cheilanthes* sp, PV. (g–h) *Pteris* sp., (g) PV, (h) DV. (i) Davalliaceae vel Polypodiaceae sp./
*Verrucatosporites alienus* (R.Potonié) P.W.Thomson et Pflug, 1953, EV. (j)
*Leavigatosporites haardti*, EV. (k–l) *Inaperturopollenites hiatus*. (m) *Abies* sp., EV. (n–o)
*Cathaya* sp., (n) PV, (o) SEM detail, nanoechinolate sculpturing of cappa (PRV). (p) *Cedrus*
sp., EV. (q) *Pinus* subgenus *Pinus* sp., EV. (r) *Pinus* subgenus *Strobilus* sp., EV. (s–t) *Tsuga*
sp. 1, (s) PV, (t) monosaccus and corpus detail, PRV. (u–v) *Tsuga* sp. 2, (u) PV, (v)
monosaccus and corpus detail, PRV.

Abbreviations: equatorial view (EV), polar view (PV), distal view (DV), proximal view
(PRV). Scale bars 10 µm (LM, h, t, v), 1 µm (o).

**Figure 4.** LM and SEM micrographs of Poales, Vitales, Rosales, Fagales, Malpighiales, and
Geraniales.

(a) *Typha* sp, tetrad, PV. (b–c) Poaceae gen. indet., EV, (c) exine detail, PRV. (d–e)
Monocotyledone indet., (d) PV, (e) PRV. (f–g) *Parthenocissus* sp., EV. (h) *Ulmus* vel *Zelkova*
sp., PV. (I) *Fagus* sp., EV. (j–k) *Quercus* sect. *Cerris* sp., EV, (k) SEM detail, mesocolpium
exine sculpturing. (l–m) *Quercus* sect. *Ilex* sp., EV, (m) SEM detail, mesocolpium exine
sculpturing. (n–o) *Quercus* sect. *Quercus* sp., PV, (o) SEM detail, apocolpium exine
sculpturing. (p–q) Castanoideae gen. indet. sp., EV, (q) SEM detail, mesocolpium exine
sculpturing. (r) *Carya* sp., PV. (s) *Platycarya* sp., PV. (t) Engehardioideae gen. indet., PV. (u)
*Alnus* sp., PV. (v) *Betula* sp., PV. (w) *Carpinus* sp., PV. (x) *Corylus* sp., PV. (y) *Salix* sp.,
EV. (z–aa) *Geranium* sp., (z) PV, (aa) clavae detail.

Abbreviations: equatorial view (EV), polar view (PV), proximal view (PRV). Scale bars 10
973 µm (LM, e, g), 1 µm (c, k, m, o, q, aa).

**Figure 5.** LM and SEM micrographs of Sapindales, Malvales, Caryophyllales, Cornales,
Asterales, Dipsacales, and Apiales.

(a–b) *Cotinus* sp, EV. (c–d) *Pistacia* sp., (c) PV, (d) exine SEM detail. (e–f) *Acer* sp. 1, (e)
PV, (f) mesocolpium SEM detail. (g–h) *Acer* sp. 2, (g) PV, (h) mesocolpium SEM detail. (i–j)
*Craigia* sp., (i) PV, (j) apocolpium SEM detail. (k) Amaranthaceae gen. indet. sp. 1. (l)
Amaranthaceae gen. indet. sp. 2. (m–n) Caryophyllaceae gen. indet. sp. (o–p) *Nyssa* sp., (o)
PV, (p) exine sculpturing and aperture SEM detail. (q–r) *Fraxinus* sp., (q) EV, (r)
mesocolpium SEM detail. (s–t) *Olea* sp., EV. (u) Cichorioideae gen. indet. sp., PV. (v)
Asteroideae gen indet. sp. 1, PV. (w) Asteroideae gen indet. sp. 2, PV. (x–z) *Valeria* sp., (x–
y) PV, (z) aperture SEM detail. (aa–bb) Apiaceae gen. indet. sp. 1, EV. (cc–dd) Apiaceae gen.
indet. sp. 2, EV. (ee–ff) Angiosperm pollen fam. et gen. indet. sp., (ee) EV, (ff) mesocolpium
SEM detail.

Abbreviations: equatorial view (EV), polar view (PV). Scale bars 10 µm (LM, b, n, t, y, z, bb,
dd), 1 µm (d, f, h, j, p, r, ff).

**Figure 6.** Coexistence-Approach diagram showing coexistence intervals for MAT and
CMMT. MAT and CMMT climate ranges of relict taxa *a priori* excluded from the analysis
are shown on the left side of the diagram.

Blue bars, coldest month mean temperature; red bars, 10–90 percentile climatic range; dark
red extensions, full climatic range.

**Figure 7.** Köppen signal diagram for the macrofossil and pollen floras of Vegora.

To test and illustrate the stability of the climatic signal, gymnosperms (common alpine
elements) and azonal elements (e.g. riparian or swamp vegetation) were excluded in some
runs.

Supplementary Material

Supplementary Material Tables S1 - Climatic parameters of NLR

Table S1 (1). Fossil species and climatic parameters of the corresponding NLR (depending on the fossil-species and their botanical affinities, climate parameters of species, sections, subgenera, genera, or subfamilies are used as NLR).

Table S1 (2). Climatic parameters of NLR.

Supplementary Material Tables S2 - Köppen-Geiger climate type signatures.

Table S2 (1). Scored Köppen-Geiger signatures of all NLR species of the macrofossil and pollen flora of Vegora.

Tables S2 (2). Köppen-Geiger signature values and diagram of the macrofossil and pollen flora of Vegora.

Supplementary Material S3 - Systematic palaeobotany and descriptions of palynomorphs from the plant fossil bearing strata of Vegora (sample S115992).

Supplementary Material Tables S4 - Palynomorph abundance of sample S115992.

Supplementary Material S5 - Coding of leaf physiognomic characters for morphotypes from the Vegora lignite mine macroflora. Output PDF files from online CLAMP analysis (<http://clamp.ibcas.ac.cn>).

Supplementary Material S6 - Köppen-Geiger categories

Appendix F

Response letter:

Associate Editor Comments to Author (Dr Emily Lindsey):

This manuscript represents an important contribution to paleobotany, the investigation of the relationship between pollen/spore and plant macrofossil proxies. The article has been reviewed by three reviewers, two of whom recommend accepting it with only minor revisions, and one of whom recommended rejection of the article. However, it seems that the concerns of the third reviewer can be addressed by various revisions that were suggested by the three reviewers -- restructuring the article as necessary for clarity and general editing; making sure all relevant data is clearly presented; moving some figures from the supplemental data into the main manuscript; expanding discussion of/comparisons with other sites/paleoclimate studies; and more explicitly acknowledging potential shortcomings.

Response: Thank you very much for clear instructions how to improve this work. We followed all of the reviewers' recommendations. In a few cases, we did not follow the reviewers' suggestions. For every such instance, we provide explanations why we did so.

In addition, in the revised Supplementary Information 3 we added additional pollen images to support determinations such as *Pistacia*.

Reviewer comments to Author:

Reviewer: 1

Comments to the Author(s)

The manuscript submitted by Bouchal et al. is an excellent case-study of parallel investigation of a macroflora and a microflora provided by the same stratigraphic level (there are so few!).

I consider that the text is very well constructed and the figures and tables clear and very useful.

In intermontane small basins such as the Florina-Ptolemais-Servia Basin, it is difficult to determine if the recorded pollen grains come from a short-distance fluvial transport or from a long-distance air transport in contrast to the macroremains which represent the local palaeovegetation. The authors rightly underline the differences in pollen floras from Vegora and Lava although macrofloras are very similar. This is also true for the Prosilio locality, which provided a pollen flora somewhat different from the previous ones (see: Biltekin, 2010 pp. 76–78 – attached) coming from a pre-evaporitic Messinian layer (just underlying the Messinian Erosional Surface shown in Suc et al., 2015).

Response: Thank you. We elaborate on the Prosilio locality in the revised manuscript and clarified the position of the thin layer investigated by Biltekin within the context of the larger section studied by Steenbrink (2000).

I have only very few comments on this paper: some minor ones are directly indicated on the annotated manuscript (attached) and the annotated Supplementary Material Table S4 (attached).

Response: Thank you. We followed most of the recommendations in the annotated manuscript. One exception is the case of Maccarone, a section in Italy dated to 5.5–5.4 Ma that has been discussed a lot. We appreciate that the northward excursion of a subdesertic grass during this period when at the same time humid subtropical woody vegetation prevailed and the conflicting explanations for this are very interesting and provide another example how difficult the interpretation of dispersed pollen data is. However, since the age of this section is much younger, and the discussion of it would need a lot more text, we think this should be done elsewhere.

Maybe, some suggestions expressed below could be followed by the authors:

- to introduce (in lines 495–513 or 515–531) a brief allusion to the pre-evaporitic Messinian pollen flora from Intepe (Dardanelles Strait, Turkey – Melinte-Dobrinescu et al., 2009, *Palaeogeogr. Palaeoclimatol. Palaeoecol.*, 278, 24–39) which displays a very open palaeovegetation (see also for details: Biltekin, 2010 pp. 61–64 – attached);
- to use for comparison (Subsect. 4.2) some of the pollen data yielded by Jiménez-Moreno et al. (2007) – attached;
- to compare the palaeoclimatic estimates provided by the 'Coexistence Approach' (Subsect. 4.3) with some palaeoclimatic estimates given by the 'Climatic Amplitude Method' on coeval localities from the region s.l. (Site 380: Fauquette et al., 2006);
- to pay some caution to strong differences in palaeoclimate reconstructions between the CLAMP method and transferred pollen records (as evidenced on an early Pleistocene flora by Girard et al., 2019, *Rev. Palaeobot. Palynol.*, 267, 54–61).

Response: Thank you for these interesting additions. We considered all of them in the revised manuscript.

As a conclusion, the Bouchal et al.'s manuscript deserves to be rapidly published after a minor revision. My name can be indicated to the authors.

Dr. Jean-Pierre Suc.

Reviewer: 2

Comments to the Author(s)

Another very informative paper from Bouchal et al. on the Neogene floras and vegetations of Eastern Mediterranean area. The only pity is that many useful pieces of information are somehow hidden in Supplementary material...

Response: Thank you. We moved Tables explaining the meaning of Köppen climate types and pollen abundancies from the supplement into the main text.

I did some remarks / corrections / suggestions directly in the pdf attached to this review. But definitely, well done !

Response: Thank you for some very important remarks. We considered most of them when relating to facts, but not all of them when relating to language.

A single recommendation that we did not follow needs some clarification: Line 355, 3.4. *Inferring past climate with CA.* "You should surely compare your results with those presented earlier by Kvaček et al., 2004, see here: <https://doi.pangaea.de/10.1594/PANGAEA.141806> "

This analysis is problematic and we decided not to cite it: the NLR taxa used for the CA in the mentioned Pangea record do not at all correspond to the NLR taxa determined as most appropriate in the text (Kvaček et al., 2002) and in the Supplement of the text (Kvaček et al., 2002). One limit of the MAT coexistence interval is made by *Taxodium*, which according to the rules of CA (see Utescher et al., 2014) must not be included in a CA analysis because it is a narrow endemic relic species whose modern climatic range is not representative for the genus' climatic amplitude. Other limiting taxa, such as *Ulmus alata*, are simply wrong NLRs that perhaps were used because they were available from the Palaeoflora database.

Jakub Sakala, March 13, 2020

Reviewer: 3

Comments to the Author(s)

Here are some major concerns.

1. Age control. The manuscript failed to show the details of the chronology data.

Response: Thank you for pointing this out. As far as we know, however, age control for the leaf and pollen/spore assemblage studied here is one of the best ones available for the entire Mediterranean region. As stated in the manuscript, we are confident about the time span during which the investigated sedimentary rocks were deposited. This time span being 400 ka is more accurate than vertebrate based MN zonation, which are – rightly – considered very good secondary age constraints for fossil plant assemblages.

2. Pollen sample and treatment. Only one pollen sample was analyzed, which makes it difficult to make climate estimates and biome reconstruction for a certain interval.

Response: Thank you. The purpose of our study was to directly compare environmental signal from pollen/spores and leaves from a single layer.

How much sample was used for treatment? How many pollen grains were counted?

Response: Please refer to the original submission where all this information is provided.

3. Pollen data presentation. A diagram of major pollen taxa should be added in the main text rather than in the supplementary data, as this is the fundamental data.

Response: Thank you. In accordance with Jakub Sakala's advice we moved the pollen counting into the main text.

4. Manuscript structure. The manuscript could be more well-organized. As for now, it is not easy to clearly follow.

Response: Thank you. We tried our best to make the text more easily to follow.